# Low-dose metformin targets the lysosomal AMPK pathway through PEN2

Teng Ma[1,6], Xiao Tian[1,6], Baoding Zhang[1,6], Mengqi Li[1], Yu Wang[1], Chunyan Yang[1], Jianfeng Wu[2], Xiaoyan Wei[1], Qi Qu[1], Yaxin Yu[1], Shating Long[1], Jin-Wei Feng[1], Chun Li[1], Cixiong Zhang[1], Changchuan Xie[1], Yaying Wu[1], Zheni Xu[1], Junjie Chen[3], Yong Yu[1], Xi Huang[1], Ying He[1], Luming Yao[1], Lei Zhang[1], Mingxia Zhu[1], Wen Wang[4], Zhi-Chao Wang[4], Mingliang Zhang[5], Yuqian Bao[5], Weiping Jia[5], Shu-Yong Lin[1], Zhiyun Ye[1], Hai-Long Piao[4], Xianming Deng[1✉], Chen-Song Zhang[1✉] & Sheng-Cai Lin[1✉]

Metformin, the most prescribed antidiabetic medicine, has shown other benefits such as anti-ageing and anticancer effects[1–4]. For clinical doses of metformin, AMP-activated protein kinase (AMPK) has a major role in its mechanism of action[4,5]; however, the direct molecular target of metformin remains unknown. Here we show that clinically relevant concentrations of metformin inhibit the lysosomal proton pump v-ATPase, which is a central node for AMPK activation following glucose starvation[6]. We synthesize a photoactive metformin probe and identify PEN2, a subunit of γ-secretase[7], as a binding partner of metformin with a dissociation constant at micromolar levels. Metformin-bound PEN2 forms a complex with ATP6AP1, a subunit of the v-ATPase[8], which leads to the inhibition of v-ATPase and the activation of AMPK without effects on cellular AMP levels. Knockout of *PEN2* or re-introduction of a PEN2 mutant that does not bind ATP6AP1 blunts AMPK activation. In vivo, liver-specific knockout of *Pen2* abolishes metformin-mediated reduction of hepatic fat content, whereas intestine-specific knockout of *Pen2* impairs its glucose-lowering effects. Furthermore, knockdown of *pen-2* in *Caenorhabditis elegans* abrogates metformin-induced extension of lifespan. Together, these findings reveal that metformin binds PEN2 and initiates a signalling route that intersects, through ATP6AP1, the lysosomal glucose-sensing pathway for AMPK activation. This ensures that metformin exerts its therapeutic benefits in patients without substantial adverse effects.

Metformin is the usual first-line drug of choice to reduce blood glucose levels in patients with type 2 diabetes mellitus. It also has other clinically beneficial effects such as reductions in body weight and hepatic fat content, and decreased cancer incidence in patients with diabetes who take the drug[1,3]. Administration of metformin to various organisms, including nematodes (*C. elegans*) and mice, can also extend lifespan and health span[9,10]. Metformin requires transporters of the OCT family to enter cells, which restricts its primary target organs to the liver, the kidney and the intestine[5,11]. Various mechanisms of action for metformin to exert its roles have been proposed. Metformin can inhibit complex I of the mitochondrial electron transport chain in hepatocytes[12,13], which leads to decreases in ATP and increases in AMP levels and in turn activates AMPK through the canonical adenine-nucleotide-dependent mechanism[14]. Increased AMP also inhibits fructose-1,6-bisphosphatase-1 and adenylate cyclase to block gluconeogenesis[15,16]. Metformin has also been proposed to alter cellular redox status, which increases $NAD^+/NADH$ ratios and leads to the suppression of the utilization of gluconeogenic substrates. Metformin may also exert its glucose-lowering effects in the gut by promoting the secretion of glucagon-like peptide 1 (GLP-1)[1].

Among the various potential effectors of metformin identified, AMPK, a master controller of metabolic homeostasis, has been placed at centre stage[17,18]. AMPK, through phosphorylating acetyl-CoA carboxylase 1 (ACC1) and ACC2, is indispensable for the attenuation of hepatic steatosis and atherosclerosis in diabetic mice that have been given chronic metformin treatment[19,20]. Duodenal activation of AMPK is essential for GLP-1 secretion in L cells, and is required for the acute glucose-lowering effect of metformin when orally administered[21]. Furthermore, the metformin-mediated retardation of ageing in *C. elegans* is through an AMPK-dependent mechanism[9,18].

It has been widely accepted that metformin activates AMPK by inhibiting complex I of the mitochondrial electron transport chain, which impairs ATP synthesis and in turn increases AMP/ATP and ADP/ATP ratios[12–14]. However, the decrease in energy levels could only be observed at peak concentrations after high doses of metformin in mice (≥ 250 mg kg$^{-1}$ orally, which yields peak plasma concentrations of

[1]State Key Laboratory for Cellular Stress Biology, Innovation Centre for Cell Signalling Network, School of Life Sciences, Xiamen University, Fujian, China. [2]Laboratory Animal Research Centre, Xiamen University, Fujian, China. [3]Analysis and Measurement Centre, School of Pharmaceutical Sciences, Xiamen University, Fujian, China. [4]CAS Key Laboratory of Separation Science for Analytical Chemistry, Dalian Institute of Chemical Physics, Chinese Academy of Sciences, Liaoning, China. [5]Department of Endocrinology and Metabolism, Shanghai Clinical Centre for Diabetes, Shanghai Diabetes Institute, Shanghai Key Laboratory of Diabetes Mellitus, Shanghai Jiao Tong University Affiliated Sixth People's Hospital, Shanghai, China. [6]These authors contributed equally: Teng Ma, Xiao Tian, Baoding Zhang. ✉e-mail: xmdeng@xmu.edu.cn; cszhang@xmu.edu.cn; linsc@xmu.edu.cn

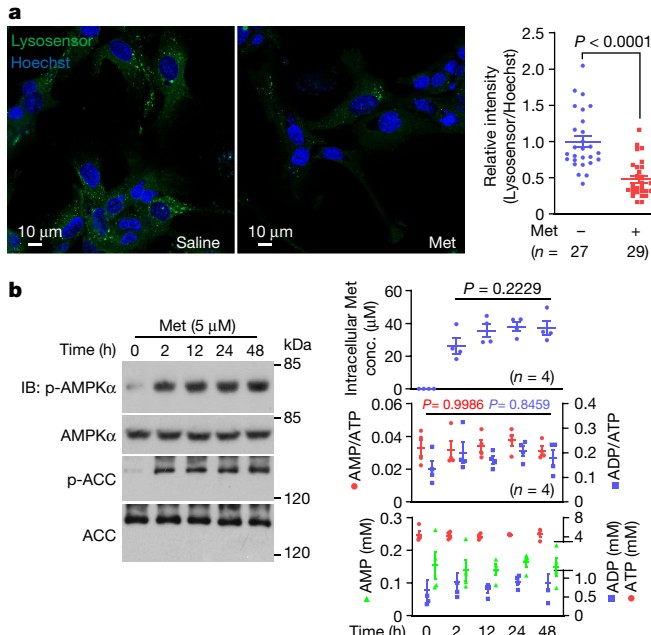

**Fig. 1 | Metformin activates AMPK without increasing AMP/ADP levels.**
**a**, Low-dose metformin deacidifies lysosomes in mouse primary hepatocytes
(left). Cells were treated with 5 µM metformin (Met) for 2 h, and the relative
fluorescence intensities of Lysosensor are shown (right). **b**, Metformin does
not increase AMP/ADP levels in mouse primary hepatocytes. Cells were treated
with 5 µM metformin for the indicated time periods followed by analysis of
phosphorylated (p)-AMPKα and p-ACC by immunoblotting (IB; left), AMP/ATP
and ADP/ATP ratios, and the absolute concentrations of AMP, ADP and ATP by
mass spectrometry (bottom right). After washing three times with PBS, the
intracellular metformin concentrations (conc.) were measured by mass
spectrometry (top right). For gel source data, see Supplementary Fig. 1. Data
are the mean ± s.e.m., $n$ values are labelled on each panel. $P$ values were
calculated using two-sided Mann–Whitney test (**a**) or one-way analysis of
variance (ANOVA) followed by Tukey's (**b**, bottom right) or Sidak's test (**b**, top
right). Experiments in **a** were performed three times and experiments in **b** were
performed five times.

125–150 µM after 1–2 h and rapidly decreases thereafter[16]). By compari-
son, the plasma metformin concentrations in patients taking standard
clinical doses of 1.5–2 g per day (Glucophage, 0.5 g three times a day
or four times a day) have been reported to be only 5–30 µM (ref. [11])
(Extended Data Fig. 1a), which may not be sufficient to increase AMP/
ATP and ADP/ATP ratios[22,23]. Therefore, it is necessary to explore how
clinically relevant doses of metformin activates AMPK.

## PEN2 binds to metformin

We found that metformin at clinical doses sufficiently inhibited the
vacuolar H⁺-ATPase (v-ATPase) on the lysosome (Fig. 1a, b and Extended
Data Fig. 1, with detailed discussions in Supplementary Note 1). We
therefore used an affinity-based approach to analyse protein extracts
of purified lysosomes to identify potential direct targets for metformin
(Fig. 2a). Two types of photoactive metformin probes, Met-P1 and
Met-P2, were synthesized (Extended Data Fig. 2a), but only Met-P1
was able to inhibit lysosomal acidification; Met-P2 had no effect and
was therefore discarded (Extended Data Fig. 2b). After incubation with
lysosome lysates, Met-P1 was conjugated to proteins by ultraviolet irra-
diation and then biotinylated (chemical reactions shown in Extended
Data Fig. 2c). NeutrAvidin beads were used to pull down the conjugates
for analysis by mass spectrometry (MS). As listed in Supplementary
Table 1, we engineered expression plasmids for a total of 367 proteins,
and verified that 113 proteins of them could be pulled down by Met-P1

when individually expressed in HEK293T cells (Supplementary Note 2).
Next, we individually knocked down those 113 proteins in mouse embry-
onic fibroblasts (MEFs) through lentivirus-mediated short hairpin RNA
(shRNA) silencing. We observed that depletion of PEN2, but not others,
rendered the cells insensitive to metformin treatment, as assessed by
levels of AMPK activation and inhibition of v-ATPase (Extended Data
Figs. 2d and 3a, b). Consistently, knockout of *PEN2* blocked low-dose
metformin-induced AMPK activation and v-ATPase inhibition in pri-
mary hepatocytes, MEFs and HEK293T cells (Fig. 2b, c and Extended
Data Fig. 3c–i, k, l; note that knockout of *PEN2* did not affect basal lyso-
somal pH levels (Supplementary Note 2)). Of note, depletion of PEN2
in all three cell types did not affect the transport of metformin into
cells (Extended Data Fig. 3j). PEN2 was originally identified as a com-
ponent of γ-secretase[7]. Unlike PEN2, other subunits of γ-secretase did
not directly participate in AMPK activation for low-dose metformin
(Extended Data Figs. 3m–s, 4a, b and 6m–o; detailed discussions on
the relationship between metformin and γ-secretase are provided in
Supplementary Note 3). Imaging by confocal microscopy (Extended
Data Fig. 4c), stochastic optical reconstruction microscopy (STORM;
Fig. 2d) and APEX tag-based transmission electron microscopy (Fig. 2e,
with validation data in Extended Data Fig. 4d) showed that a portion
of PEN2 (approximately 40%; Extended Data Fig. 4c) was localized on
the lysosome. This finding was confirmed in subcellular fractiona-
tion assays (Extended Data Fig. 4e, with detailed discussions on PEN2
localization provided in Supplementary Note 4 and Extended Data
Figs. 4f, g and 5a), and metformin did not alter the subcellular localiza-
tion of PEN2 (Extended Data Fig. 5b, c). These results indicate that the
pool of lysosomally localized PEN2 may have a distinct role, whereby
it participates in metformin-induced AMPK activation (discussed in
Supplementary Note 5). Indeed, constructs of PEN2 fused to other
organelle-specific proteins did not restore AMPK activation by met-
formin when re-introduced into *Pen2*⁻/⁻ MEFs (Extended Data Fig. 5d,
with validation data in Extended Data Fig. 5d, e).

High concentrations of metformin can increase cellular levels
of AMP, which can allosterically activate AMPK; therefore, it was
anticipated that AMPK activation induced by high metformin levels
would be lysosome-independent. Indeed, high concentrations of
metformin, which increased AMP/ATP and ADP/ATP ratios (Extended
Data Fig. 1m–p), bypassed the requirement of PEN2 for AMPK acti-
vation, as did phenformin and buformin (Extended Data Fig. 5f).
Moreover, PEN2 deficiency did not affect glucose-starvation-induced
AMPK activation (Extended Data Fig. 5g) or other agonists (Extended
Data Fig. 5g, h).

We next investigated the biophysical nature that underlies the bind-
ing of PEN2 to metformin. Differential scanning calorimetry assays
showed a shift in the thermal transition midpoint in the presence of
metformin (Extended Data Fig. 6a). Isothermal calorimetry (ITC) and
surface plasmon resonance (SPR) measurements further gave esti-
mated dissociation constant ($K_D$) values of 1.7 µM and 0.15 µM (with
an association rate constant ($k_a$) value of 2,815 M⁻¹s⁻¹), respectively.
These values are within the range of detected intracellular metformin
concentrations in animals or human patients administered with regular
doses (Fig. 2f and Extended Data Fig. 6b, f, with detailed discussions in
Supplementary Note 6). The ITC measurement gave an additional met-
formin binding site, with a much higher $K_D$ of 98 µM, which is beyond
the ranges of clinically relevant intracellular concentrations of met-
formin (Extended Data Fig. 6b). As a control, other γ-secretase subu-
nits did not show apparent binding affinity to metformin (Extended
Data Fig. 6c). We also performed mass spectrometry on purified PEN2
conjugated to Met-P1 to identify the residue(s) responsible for binding
metformin. As a result, the Y47 residue of PEN2 was identified (Extended
Data Fig. 6d), which indicates that metformin may be able to bind the
amino-terminal cytosolic face. In silico modelling further illustrated
that at the N-terminal region of PEN2, metformin forms direct con-
tacts with PEN2 through the F35 and E40 residues on PEN2 (Extended

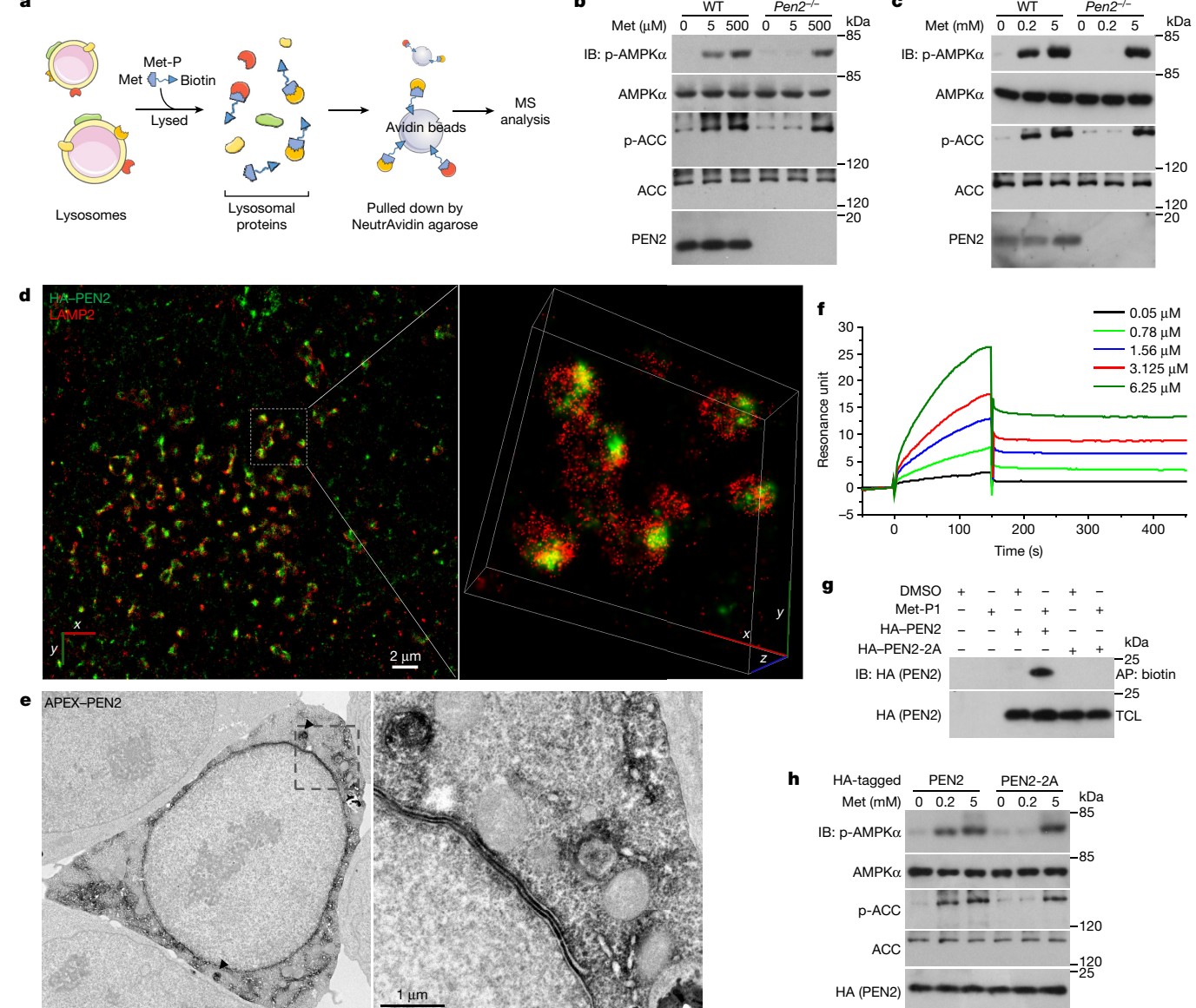

**Fig. 2 | PEN2 binds to metformin and is required for low-dose metformin-induced AMPK activation. a**, A schematic depicting the procedure of the affinity-based approach that used a photoactive metformin probe (Met-P) to identify target(s) of metformin from protein extracts of lysosomes purified from MEFs. MS, mass spectrometry. **b**, **c**, Knockout of *Pen2* blocks the activation of AMPK by low-dose metformin. Mouse primary hepatocytes (**b**) and MEFs (**c**; clone 1, and same hereafter, unless stated otherwise) were treated with 5 μM and 200 μM metformin for 2 h and 12 h, respectively, followed by analysis of p-AMPKα and p-ACC. WT, wild type. **d**, **e**, STORM image of MEFs (**d**) and TEM image of HEK293T cells (**e**) showing that a portion of PEN2 is localized to the lysosome (**e**, black arrowheads) and overlaps with the lysosome marker

LAMP2 (**d**). **f**, **g**, PEN2 is able to bind metformin. **f**, In SPR assays, PEN2 was incubated with metformin at the indicated concentrations. **g**, In Met-P1-binding assays, HEK293T cells transfected with PEN2 or PEN2-2A were lysed, incubated with 10 μM Met-P1 and then biotinylated, and then affinity pull-down (AP) of biotinylated proteins was performed. TCL, total cell lysate. **h**, PEN2-2A does not mediate AMPK activation by metformin. *Pen2*⁻/⁻ MEFs re-introduced with haemagglutinin (HA)-tagged PEN2-2A were treated with 200 μM metformin for 12 h, followed by analysis of p-AMPKα and p-ACC. For gel source data, see Supplementary Fig. 1. Experiments in this figure were performed three times, except those in **b** and **c**, which were performed four times.

Data Fig. 6e). Indeed, mutation of both F35 and E40 to alanine on PEN2 (PEN2-2A) blocked its interaction with metformin (Fig. 2g and Extended Data Fig. 6f). Re-introduction of PEN2-2A into *Pen2*⁻/⁻ MEFs did not restore metformin-induced AMPK activation, or v-ATPase inhibition, even though PEN2-2A shares a similar subcellular localization with wild-type PEN2 (Fig. 2h and Extended Data Fig. 6h, with validation data in Extended Data Fig. 6g, i). The mass spectrometry results also revealed an additional, but much weaker, metformin-binding site at the carboxy-terminal (luminal) face of PEN2. Given that metformin may be transported through endocytosis and may be present in the lumen of lysosomes, we examined possible binding of metformin to the

C terminus of PEN2. We found that mutation of residues at this site did not block metformin binding or dampen AMPK activation (Extended Data Fig. 6d, j–l).

## ATP6AP1 tethers PEN2 to v-ATPase

We next investigated how metformin binding causes PEN2 to intersect with and inhibit v-ATPase. We analysed PEN2 that was immunoprecipitated after incubation with protein extracts of lysosomes by mass spectrometry. A total of 1,881 proteins were detected in the PEN2 prey, among which 889 were changed after metformin treatment. Of these 889 proteins, 123 are

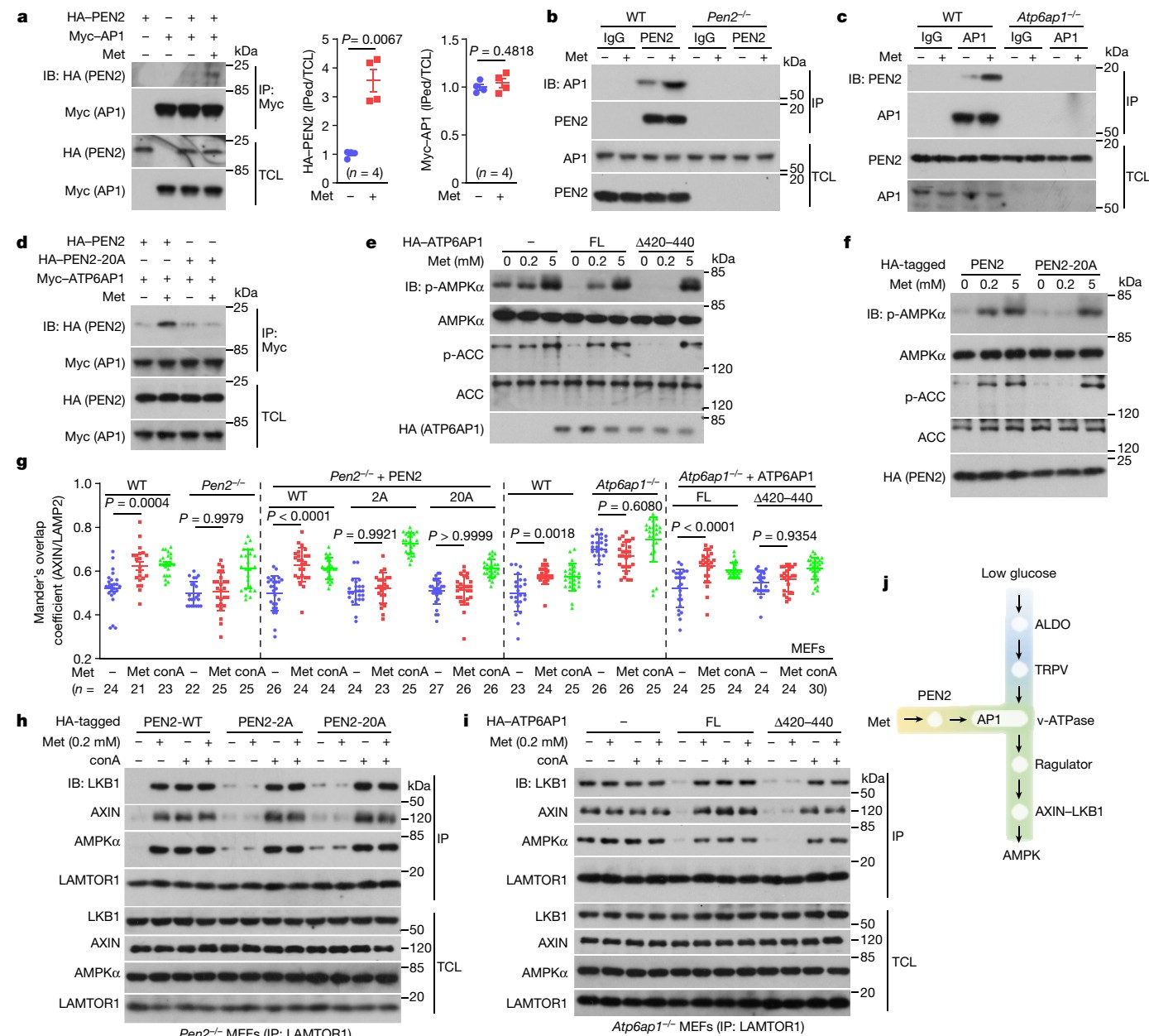

**Fig. 3 | ATP6AP1 tethers PEN2 to v-ATPase for AMPK activation.**
**a**–**c**, Identification of ATP6AP1 as an interacting protein of PEN2. Lysates of HEK293T cells expressing HA–PEN2 or Myc–ATP6AP1 (**a**), and lysates from wild-type MEFs, *Pen2*[−/−] MEFs (**b**) or *Atp6ap1*[−/−] MEFs (**c**) were incubated with 10 μM metformin and immunoprecipitated (IP) for the PEN2 and AP1 proteins. **d**, Metformin does not promote the interaction between ATP6AP1 and PEN2-20A. HEK293T cells transfected with HA-tagged PEN2 or PEN2-20A were lysed and treated as in **a**. The interaction between ATP6AP1 and PEN2 was analysed by IP followed by IB. **e**–**i**, Loss of the PEN2–ATP6AP1 interaction abolishes the effects of metformin on AMPK activation. *Atp6ap1*[−/−] MEFs re-introduced with ATP6AP1[Δ420–440] (**e**) or *Pen2*[−/−] MEFs re-introduced with the PEN2-20A mutant (**f**) were treated with 200 μM metformin for 12 h followed by analysis of p-AMPK

and p-ACC. **g**–**i**, The effects of ATP6AP1 and PEN2 mutants on the lysosomal translocation of AXIN (**g**), and the formation of the AXIN-based complex (**h**, **i**) were analysed. Concanamycin A (conA; 5 μM for 2 h) was used as a control. FL, full length. **j**, A schematic depicting that the metformin–PEN2–ATP6AP1 and the FBP–aldolase axes constitute two incoming shunts that converge at v-ATPase to elicit AMPK activation through the lysosomal pathway. For gel source data, see Supplementary Fig. 1. Data are the mean ± s.e.m., *n* values are labelled on each panel, and *P* values were calculated using two-sided Student's *t*-test (**a**, for Myc–ATP6AP1), two-sided Student's *t*-test with Welch's correction (**a**, for HA–PEN2) or two-way ANOVA, followed by Tukey's test (**g**). Experiments in this figure were performed three times, except for **a** (four times), and **h** and **i** (five times).

lysosome-resident proteins (Supplementary Table 2). Among these 123 candidates, we were particularly interested in ATP6AP1 (also known as Ac45), an accessory factor of v-ATPase[8], because its metformin-dependent interaction with PEN2 could be verified by co-immunoprecipitation assays in cells and in vitro (Fig. 3a–c and Extended Data Fig. 7a, b). Domain-mapping experiments identified that amino-acid residues from 420 to 440, which constitute the transmembrane domain of ATP6AP1,

were responsible for PEN2 binding (Extended Data Fig. 7c). This finding was reinforced by results from experiments that used the chimeric construct LAMP2[TM]–ATP6AP1, which has the ATP6AP1 transmembrane domain replaced by the transmembrane domain of the lysosomal protein LAMP2. This construct did not interact with PEN2 (Extended Data Fig. 7d). In addition, PEN2 mutations on its interface towards ATP6AP1 (based on in silico docking assays; PEN2-20A), impaired the interaction between PEN2

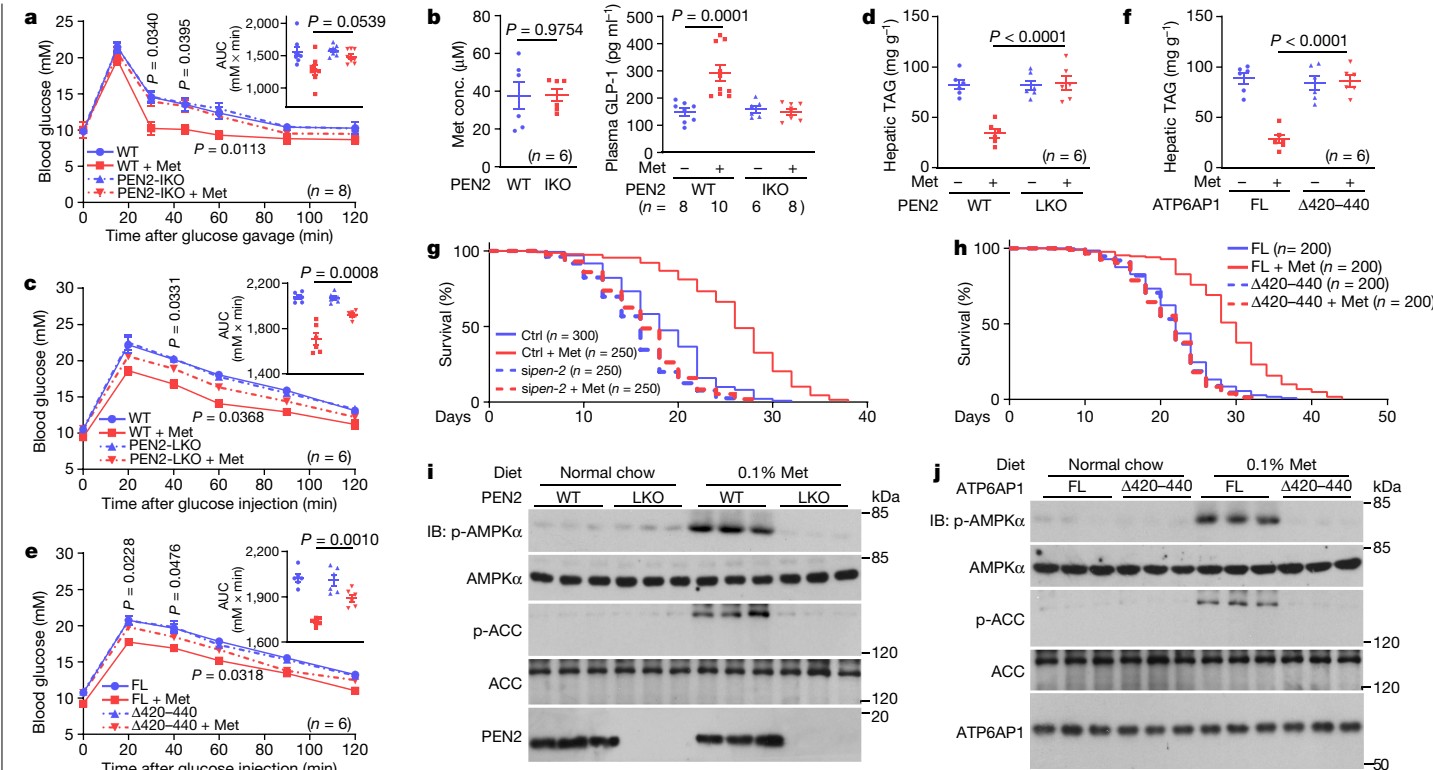

**Fig. 4 | PEN2 and ATP6AP1 are required for the biological effects of metformin. a, b,** Intestinal PEN2 is required for the metformin-induced glucose-lowering effect. PEN2-IKO mice were administered with metformin as depicted in Extended Data Fig. 12d. Oral glucose tolerance test analysis (**a**), measurements of duodenal metformin concentrations (**b**, left) and measurements of plasma GLP-1 levels before and after 15 min of glucose gavaging (**b**, right) were then performed. **c, d,** PEN2 is required for metformin-induced reduction in hepatic fat. Mice in which *Pen2* was specifically knocked out in the liver (LKO) were treated with metformin as depicted in Extended Data Fig. 12f. Intraperitoneal glucose tolerance test results (**c**) and hepatic TAG levels (**d**) in mice after 16 weeks of treatment of metformin are shown. **e, f,** ATP6AP1 is required for metformin-induced reduction in hepatic fat. Mice were treated as depicted in Extended Data Fig. 12m. Intraperitoneal glucose tolerance test results (**e**) and hepatic TAG levels (**f**) in mice after 16 weeks of treatment of metformin are shown. **g, h,** PEN2 and ATP6AP1 are required for metformin-induced lifespan extension in *C. elegans*. WT (N2) nematodes with *pen-2* (T28D6.9) knocked down using siRNA (si*pen-2*) (**g**) or *ATP6AP1*[-/-] (*vha-19*) nematodes with full-length ATP6AP1

or ATP6AP1[Δ420–440] stably expressed (**h**) were treated with 50 mM metformin. Lifespan data are shown as Kaplan–Meier curves (statistical analyses are provided in Supplementary Table 3). Ctrl, control. **i, j,** PEN2 and ATP6AP1 are required for AMPK activation induced by 0.1% metformin in the diet. The 5-week-old PEN2-LKO mice (**i**; tamoxifen was injected at 4 weeks old) or 8-week-old ATP6AP1-LKO mice expressing ATP6AP1[Δ420–440] (**j**; viruses injected at 4 weeks old, and tamoxifen was injected at 5 weeks old), were fed with normal chow diet containing 0.1% metformin for 1 week, as previously described[10]. Hepatic AMPK activation was then analysed by IB. For gel source data, see Supplementary Fig. 1. Data are shown as the mean ± s.e.m., *n* values are labelled on each panel, and *P* values were calculated using two-way repeated-measures ANOVA followed by Tukey's test (**a, c** and **e** compared blood glucose between the WT/ATP6AP1-FL + Met group and the PEN2-IKO/LKO/ATP6AP1[Δ420–440] + Met group at each time point; see also insets of **a, c** and **e** for area under the receiver operator characteristic curve (AUC) values, and *P* values by two-way ANOVA, followed by Tukey's test), two-sided Student's *t*-test (**b**, left), and two-way ANOVA, followed by Tukey's test (right panel of **b**, and **d, f**). Experiments in this figure were performed three times.

and ATP6AP1 (Fig. 3d and Extended Data Fig. 7e–g). Of note, ATP6AP1 itself did not bind Met-P1 (Extended Data Fig. 7h). Together, these results indicate that after binding to metformin, lysosomal PEN2 is recruited to ATP6AP1 of the v-ATPase complex.

We next examined how ATP6AP1 mediates the inhibition of v-ATPase by metformin. First, as an integral member of v-ATPase, knockout of ATP6AP1 led to constitutive activation of AMPK (Extended Data Fig. 7i–k). When we re-introduced the truncated ATP6AP1 mutant (Δ420–440), which lacks the transmembrane domain required for its interaction with PEN2, into *Atp6ap1*[-/-] MEFs, the basal activity of v-ATPase was restored (Extended Data Fig. 8a, with validation data in Extended Data Fig. 7l). Of note, the ATP6AP1[Δ420–440] mutant did not mediate metformin-induced v-ATPase inhibition or AMPK activation (Fig. 3e and Extended Data Fig. 8b, c). Similar restoration of v-ATPase activity, as well as blockade of AMPK activation, was observed when the LAMP2[TM]–ATP6AP1 chimeric construct was re-introduced into *Atp6ap1*[-/-] MEFs (Extended Data Fig. 8a, d, with validation data in Extended Data Fig. 7l). Furthermore, re-introduction of PEN2-20A, which cannot interact with ATP6AP1 even

though it is localized in a similar manner as wild-type PEN2 (Extended Data Fig. 8e), into *Pen2*[-/-] MEFs blocked the activation of AMPK and the inhibition of v-ATPase (Fig. 3f and Extended Data Fig. 8f). These results indicate that metformin-bound PEN2, by gaining affinity to ATP6AP1, inhibits v-ATPase to activate AMPK.

We previously reported that glucose deprivation can activate lysosomal AMPK without increasing AMP/ADP levels through v-ATPase, Ragulator and AXIN[24], which are downstream of the fructose-1,6-bisphosphate sensor aldolase. Knockout of *AXIN*, *LAMTOR1* (a subunit of Ragulator) or the v0c subunit of v-ATPase (*ATP6v0c*) in the liver, MEFs or HEK293T cells blocked the activation of AMPK by low-dose metformin (Extended Data Fig. 9a–h). Re-introduction of the AMPKβ1-G2A mutant, which cannot localize on lysosomes, into MEFs that are deficient in both AMPKβ1 and AMPKβ 2 also blocked the activation of AMPK by low-dose metformin (Extended Data Fig. 9a–h). Moreover, high concentrations of metformin bypassed the requirement for AXIN and LAMTOR1 in AMPK activation (Extended Data Fig. 9a–c, h). PEN2 and ATP6AP1 seem to act as factors upstream of AXIN and

LAMTOR1 through their regulation of v-ATPase. This is based on the fact that the lysosomal translocation of AXIN—and the formation of the AXIN-based complex—was dampened in $Pen2^{-/-}$ MEFs, in $Pen2^{-/-}$ MEFs expressing PEN2-2A or PEN2-20A mutants, and in A$tp6ap1^{-/-}$ MEFs expressing the ATP6AP1$^{\Delta420-440}$ mutant when treated with metformin (Fig. 3g–i, Extended Data Figs. 9i–l and 10a, b). Blockade of v-ATPase by its inhibitor concanamycin A restored these phenotypes (Fig. 3g–i, Extended Data Figs. 9i, k, 10a, b and 11a, b). As additional controls, aldolase and TRPV, which are required for signalling of low glucose to v-ATPase and AMPK[6,25], were dispensable for the PEN2-sensed AMPK activation by metformin. This result was supported by the following lines of evidence: (1) expression of ALDOA-D34S, which mimics a high glucose state and blocks glucose-deprivation-induced AMPK activation in both mouse liver and cultured cells[6] (Extended Data Fig. 11c), did not block metformin-induced AMPK activation (Extended Data Fig. 11d, e); and (2) a quadruple knockout of $Trpv1$–$Trpv4$ in MEFs, or knockdown of $Trpv2$–$Trpv4$ in the liver of $Trpv1^{-/-}$ mice (leaving those cells or tissues with scarce TRPV expression[25]), did not affect the activation of AMPK when treated with metformin (Extended Data Fig. 11f, g). Together, PEN2–ATP6AP1 relays the signal of metformin, as an intersecting shunt, to inhibit v-ATPase, which primes the lysosomal translocation of AXIN and LKB1 to the lysosomal surface for phosphorylation and activation of AMPK (schematically represented in Fig. 3j).

## Phenotypes in animal models

We next explored the functions of PEN2 and ATP6AP1 to mediate the beneficial effects of metformin in animal models. We observed that mice that had PEN2 depleted specifically in the intestine (PEN2-IKO mice; generated as illustrated in Extended Data Fig. 12c, d), had impaired postprandial glucose-lowering effects of metformin, similar to those observed in intestine-specific $Ampka$ knockout (AMPKα-IKO) mice (Extended Data Fig. 12a, b). We also observed impaired promotion of GLP-1 and insulin secretion by metformin (Fig. 4a, b and Extended Data Fig. 12e). Meanwhile, hepatic-specific depletion of PEN2 (PEN2-LKO mice; generated as illustrated in Extended Data Fig. 12f) led to strong impairments in the activation of AMPK in mouse liver. The effects of administration of metformin for 4 months to decrease levels of hepatic triglycerides (TAGs), as well as glucose tolerance in high-fat diet (HFD)-induced obese mice, were also impaired (Fig. 4c, d and Extended Data Fig. 12g–k). Similarly, re-introduction of ATP6AP1$^{\Delta420-440}$ into mouse liver with $Atp6ap1$ knocked out did not rescue the metformin effects on AMPK activation or on TAG level reduction (Fig. 4e, f and Extended Data Fig. 12l–r). Therefore, PEN2 and ATP6AP1 are required for the effect of metformin to reduce hepatic fat by activating the lysosomal AMPK pathway.

We also tested whether lifespan extension induced by metformin depends on PEN2 and ATP6AP1. Consistent with previous reports[9], metformin at 50 mM was able to extend the lifespan of *C. elegans* (Extended Data Fig. 13a), and no increases in AMP/ATP and ADP/ATP ratios were observed (Extended Data Fig. 13b). Knockdown of T28D6.9, the nematode orthologue of *PEN2*, blocked the metformin-induced lifespan extension effect in *C. elegans* (Fig. 4g, Extended Data Fig. 13b, c and Supplementary Table 3, with validation data in Extended Data Fig. 13d). Similarly, expression of mammalian ATP6AP1$^{\Delta420-440}$ in *ATP6AP1$^{-/-}$ C. elegans* impaired the metformin-induced AMPK activation and lifespan extension effects (Fig. 4h, Extended Data Fig. 13f, i and Supplementary Table 3). Of note, genetic manipulation of *pen-2* and *ATP6AP1* or the living bacteria (OP50 and HT115) on the culture plates did not affect the cellular uptake of metformin (Extended Data Fig. 13e, j, k). Finally, we examined the effects of normal chow diet that contained 0.1% metformin—a diet that has been shown to extend the lifespan and retard the ageing of mice through the activation of AMPK[10]—on the activation of AMPK in mice with hepatic depletion of PEN2 or expression of ATP6AP1$^{\Delta420-440}$. The activation of AMPK was strongly dampened in

both of these mouse strains (Fig. 4i, j and Extended Data Fig. 13l). Taken together, PEN2, in conjunction with ATP6AP1, appears to be responsible for the three main biological benefits of metformin: lowering glucose levels, reducing hepatic fat content and extending lifespan.

## Discussion

Here we identified that PEN2 is a target of metformin. After stimulation, PEN2 binds to the ATP6AP1 subunit of and inhibits the activity of v-ATPase without increasing AMP or ADP, which then activates lysosomal AMPK. The PEN2–ATP6AP1 axis therefore constitutes a signalling shunt that intersects the lysosomal v-ATPase–AXIN–AMPK axis, which enables metformin at low concentration to make use of the AMP-independent AMPK-activation pathway, which is can also be triggered by glucose starvation. We also established that the PEN2–ATP6AP1 pathway is not involved in AMPK activation at low glucose levels, which indicates that the PEN2–ATP6AP1 axis is a parallel route to the v-ATPase complex. Therefore, the two axes, PEN2–ATP6AP1 and aldolase–TRPV, sense the presence of metformin and the lowered levels of glucose, respectively, and impinge on v-ATPase to control the activation of AMPK. This finding underscores the important role of v-ATPase as a signalling node for lysosomal AMPK activation (Extended Data Fig. 13m).

We also showed that the PEN2–ATP6AP1 axis is required for the three main beneficial effects of metformin: postprandial glucose reduction, hepatic fat reduction and lifespan extension, all of which strictly depend on AMPK[9,20] (Extended Data Fig. 13g, h, 14a–i and Supplementary Notes 7–9). Our data are also consistent with previous findings that metformin can promote GLP-1 secretion in the intestine to lower blood glucose in an AMPK-dependent manner[21], unless high doses of metformin are administered[26]. However, although it has been shown that metformin can also inhibit hepatic gluconeogenesis[1], we found that low doses of metformin did not do so, as assessed by pyruvate tolerance tests and the quantification of gluconeogenic genes (Extended Data Fig. 14j, k). Moreover, we also found that the PEN2–ATP6AP1 axis is required for the inhibition of mTORC1 signalling by metformin (Extended Data Fig. 14l–n), in addition to the activation of AMPK. However, mTORC1 inhibition did not seem to be involved in the abovementioned beneficial effects mediated by AMPK (Supplementary Note 10).

The intersection of metformin signalling to the lysosomal AMPK pathway, without perturbing AMP/ADP levels, might underlie the reason why metformin exerts many benefits with few side effects. This pathway only allows the activation of a small pool of AMPK[6,27], and this innate pathway is perhaps related to calorie restriction and would be less likely to cause adverse effects compared with global AMPK activation. It has been shown that indiscriminate AMPK activation results in harmful rather than beneficial effects. For example, the AMPK activator MK-8722, which appears to activate all AMPK subunit isoform combinations, can cause cardiac hypertrophy[28]. Moreover, naturally occurring mutations in *PRKAG2* (encoding AMPKγ2), which cause increases in the basal activity of AMPK, are associated with cardiac disorders[29,30]. In summary, we identified that PEN2 is the molecular target for metformin and it intersect the glucose-sensing pathway to activate AMPK, which elicits benefits that resemble those induced by glucose starvation or calorie restriction. The PEN2–ATP6AP1 axis offers potential targets for screening for substitutes for metformin, which may be available to a wider range of tissues, such as muscle, thereby engendering better efficacy in treating diabetes and other metabolic diseases.

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

## Methods

### Data reporting

The following sample sizes were similar to those previously used by us and others in this field: $n$ = 4–6 human participants or mice were used to determine the pharmacokinetics of metformin[16,31,32]; $n$ = 5–7 mice were usually used to determine the effects of metformin on blood glucose[33,34] and fatty liver[20,35]; $n$ = 100–300 worms were used to determine lifespan[36,37]; $n$ = 20–42 cells from 2–6 dishes or experiments were included when conclusions were based on immunofluorescence staining[6,25]; $n$ = 3–5 samples were used for evaluation of the levels of AMP, ADP and ATP in cells and tissues[6,15,25,27]; $n$ = 3 samples to determine the expression levels and phosphorylation levels of a specific protein[24,38]; $n$ = 3 samples to determine the mRNA levels of a specific gene[20,24,39]; and $n$ = 3 samples to determine the activity of v-ATPase in vitro[25]. No statistical methods were used to predetermine sample sizes. All experimental findings were repeated as stated in the figure legends, and all additional replication attempts were successful. For animal experiments, mice or nematodes in each genotype were housed under the same condition or place. For cell experiments, cells of each genotype were cultured using the same condition. Each experiment was designed and performed along with proper controls, and samples for comparison were collected and analysed under the same conditions for the same batch of experiments. Randomization was applied wherever possible. For example, during MS analyses, samples were processed and subjected to the mass spectrometer in random order. For the animal experiments, sex-matched (only for mice), age-matched littermate mice in each genotype were randomly assigned to pharmacological or diet treatments. In cell experiments, cells of each genotype were parallel seeded and randomly assigned to different treatments. Otherwise, randomization was not performed. For example, when performing immunoblotting, samples needed to be loaded in a specific order to generate the final figures. Blinding was applied wherever possible. For example, samples, cages or agar plates during sample collection and processing were labelled as code names that were later revealed by the individual who picked and treated the animals or the cells, but did not participate in sample collection and processing until assessing the outcomes. Similarly, during microscopy data collection and statistical analyses, the fields of view were chosen on a random basis, and were often performed by different operators, thereby preventing potentially biased selection for desired phenotypes. Otherwise, blinding was not performed, such as the measurement of v-ATPase activity and the determination of metformin binding to PEN2 in vitro, for which operators had to know the conditions of each well and added reagents to the well accordingly during the measurement.

### Determination of metformin pharmacokinetics in human participants

Women with obesity and diagnosed with polycystic ovary syndrome (PCOS) between September 2017 and July 2020 at Shanghai Jiao Tong University Affiliated Sixth People's Hospital were recruited. This study was approved by the Ethics Committee of Shanghai Sixth People's Hospital and was in accordance with the Declaration of Helsinki and Good Clinical Practice. All participants signed an informed consent form before enrolment. This study was registered on the Chinese Clinical Trial Registry (ChiCTR-IOR-17013169; http://www.chictr.org.cn/show-proj.aspx?proj=21769).

Participants included in the study fulfilled the following criteria: (1) aged 18–40 years (inclusive); (2) body–mass index (BMI) higher than 27.5 kg m$^{-2}$; and (3) meeting the diagnostic criteria of PCOS. The diagnostic criteria of PCOS were as follows: (1) irregular menstruation over the past year; (2) hyperandrogenism; and/or (3) ultrasound examination of polycystic ovaries. Participants with the following conditions were excluded: (1) after hysterectomy; (2) congenital adrenal hyperplasia, Cushing's syndrome or androgen-secreting tumours within 5 years;

(3) mothers pregnant or lactating; (4) thrombosis-related history or risk factors, such as deep vein thrombosis, pulmonary embolism, myocardial infarction (angina), valvular heart disease, atrial fibrillation or cerebrovascular accident (transient ischaemic attack); (5) abnormal liver function (for example, caused by viral hepatitis); (6) a history of liver malignancy or adenoma, or a history of genital or breast malignancies; (7) a history of severe or frequent migraine attacks; (8) renal insufficiency; and (9) other factors that may affect the efficacy of the drug or cause complications by the drug.

Participants were administered with Diane 35 (Bayer), one tablet per day, plus metformin (Bristol-Myers Squibb) at 2,000 mg day$^{-1}$. Such a combined treatment lasted 6 months, which was then switched to metformin-alone treatment for another 6 months. At the endpoint of the treatment, six participants with the lowest BMI were selected to determine pharmacokinetics. A potential selection bias may be introduced. However, they were characterized as follows (mean ± s.d.), and were therefore able to represent the general population: aged 25.5 ± 6.2 years; weight, 69.0 ± 6.4 kg; BMI, 26.5 ± 1.1 kg m$^{-2}$; waist circumference, 89.8 ± 5.8 cm; hip circumference, 102.2 ± 11.4 cm; waist/hip ratio, 0.89 ± 0.12; fasting plasma glucose, 5.1 ± 0.7 mM; fasting serum insulin 28.2 ± 15.3 μU ml$^{-1}$; HbA1c, 5.1 ± 0.2%; total cholesterol, 5.1 ± 1.0 mM; TAGs, 1.4 ± 0.7 mM; high-density lipoprotein cholesterol, 1.4 ± 0.3 mM; low-density lipoprotein cholesterol, 3.0 ± 0.6 mM. Participants were fasted for 12 h before experiments (starting from 22:00 of the previous day), followed by taking orally 0.5 g of metformin hydrochloride extended-release tablets (Bristol-Myers Squibb) per person. Blood samples were taken at 1, 3, 6 and 12 h after metformin intake, followed by serum preparation.

### Mouse strains

$AXIN^{F/F}$, $LAMTOR1^{F/F}$, $AXIN^{LKO}$ and $LAMTOR1^{LKO}$ mice were generated and maintained as previously described[24]. Wild-type C57BL/6 mice (000664), DBA2 mice (000671) and ROSA26-FLPe mice (016226) were obtained from The Jackson Laboratory. ICR mice (N000294) were obtained from GemPharmatech. $TRPV1^{-/-}$ mice were obtained from The Jackson Laboratory provided by D. Julius (003770). $TRPV1^{-/-}$ mice with knockdown of $TRPV2$–$TRPV4$ or $GFP$ were generated as previously described[25]. $APH1A^{F/F}APH1B^{-/-}APH1C^{F/F}$ mice were obtained from The Jackson Laboratory provided by B. De Strooper (030985). $ATG5^{F/F}$ mice were obtained from RIKEN, provided by N. Mizushima (BRC no. RBRC02975). $AMPKA1^{F/F}$ (014141) and $AMPKA2^{F/F}$ mice (014142) were obtained from Jackson Laboratory, provided by S. Morrison.

The $Pen2^{F/F}$ mouse strain was generated according to an androgenetic haploid embryonic stem cell (AG-haESC)-based, CRISPR–Cas9-mediated genomic editing strategy as previously described[40], but with minor modifications. The AG-haESCs carrying deletions in the differentially DNA methylated regions (DKO-AG-haESCs) were a gift from J.-S. Li (Shanghai Institute of Biochemistry and Cell Biology, CAS). The single guide RNAs (sgRNAs) targeting $Pen2$, 5′-GTGGTACTACTGCGCACGCG-3′ and 5′-GAAAGAATGAGCGAACGCCT-3′ (at intron 1 and intron 2, respectively) were cloned into a pSpCas9(BB)-P2A-mCherry-puromycin-sgRNA vector that was modified from the pSpCas9(BB)-2A-GFP vector (Addgene, 48138). The editing efficiency of each sgRNA was examined by transfecting into L929 cells followed by sequencing the targeted genomic segments before experiments. The template plasmid was constructed by inserting a 0.6-kb genomic fragment (containing exon 1 between the two $loxP$ sites), along with a 1-kb fragment as a 5′-homology arm and a 1-kb fragment as 3′-homology arm into a pBKS vector. Approximately $1 \times 10^6$ DKO-AG-haESCs in a 6-well dish were transfected with 2 μg per well of each sgRNA (0.1 μg μl$^{-1}$) and 4 μg of template plasmids (0.2 μg μl$^{-1}$) with Lipofectamine 2000 transfection reagent. At 24 h after transfection, cells were trypsinized, washed with PBS and stained with 5 mg ml$^{-1}$ Hoechst for 5 min. Those within the haploid 1C peak (haploid cells at G1 phase), mCherry-positive cells, were sorted on a FACSAria III flow cytometer (BD). Approximately 7,000 sorted cells were

then cultured in a well of a gelatin-coated 6-well dish in EmbryoMax DMEM medium containing 10% ES-FBS and non-essential amino acids. Genotypes of individual clones were verified by sequencing, and the *loxP*-flanked *Pen2* clones were selected for generating diploid ESCs. The verified clones were first treated with 0.05 µg ml$^{-1}$ demecolcine for 10 h, digested to single cells and then intracytoplasmically injected into the mature oocyte (cultured in M2 medium) derived from the $F_1$ generation of C57BL/6 × DBA2 female mice at 8–10 weeks old. After 24 h, 2-cell ESCs were transplanted into pseudopregnant ICR female mice (8–10 weeks old, >26 g), and the offspring carrying the *loxP*-flanked *Pen2* allele were further outcrossed 6 times to C57BL/6 mice before experiments.

*Atp6ap1*$^{F/F}$ mice were generated according to the traditional, homologous recombination method[41]. In brief, a targeting vector was generated by inserting a 1-kb genomic fragment (containing exon 3 and exon 4) and an *FRT*-flanked *PGK-Neo-polyA* sequence (as a positive selection marker), along with a 3-kb fragment as 5′-homology arm and a 3-kb fragment, followed by a *MC1-TK-polyA* sequence (as a negative selection marker), as a 3′-homology arm, into a pBKS vector. The targeting vector was then linearized and purified. A total of 10 µg of the linearized vector (1 µg µl$^{-1}$) was electroporated into $1 × 10^{10}$ JM8A3 ESCs, followed by selection with G418, and genotyped by Southern blotting. *Atp6ap1*$^{F/F}$ chimeric mice were obtained by microinjecting *loxP*-flanked *Atp6ap1* ESC clones into C57BL/6 blastocysts (10–15 ESCs to a blastocyst), then transplanted into a pseudopregnant ICR female mice. The *PGK-Neo* allele was removed by crossing chimeric mice with ROSA26-FLPe mice, and was further outcrossed six times to C57BL/6 mice before experiments.

The *Pen2*$^{F/F}$ mice were then crossed with *Alb-CreERT2* or *Villin-CreERT2* mice to generate inducible liver-specific *Pen2* knockout mice (PEN2-LKO) or inducible intestine-specific *Pen2* knockout mice (PEN2-IKO). The *Atp6ap1*$^{F/F}$ mice were crossed with *Alb-CreERT2* to generate inducible liver-specific *Atp6ap1* knockout mice (ATP6AP1-LKO). *Ampka1/2*$^{F/F}$ mice were crossed with *Villin-CreERT2* mice to generate inducible intestine-specific *Ampka* knockout mice (AMPKα-IKO). *Pen2*, *Atp6ap1* and *Ampka* were deleted by intraperitoneally injecting mice with tamoxifen (dissolved in corn oil) at 200 mg kg$^{-1}$, 3 times a week. Knockout efficiencies were analysed 1 week after the last injection by western blotting. ATP6AP1-LKO mice expressing wild-type ATP6AP1 or ATP6AP1$^{Δ420–440}$ were generated by injection into the tail vein different adeno-associated viruses (AAVs) carrying indicated inserts before knockout of the endogenous ATP6AP1 by tamoxifen. Levels of the re-introduced ATP6AP1 proteins were analysed 4 weeks after virus injection.

To generate Tg-ALDOA and Tg-ALDOA-D34S mice strains, human ALDOA or human ALDOA-D34S was cloned into the pLiv-Le6 vector (containing the constitutive human *APOE* gene promoter and its hepatic control region) between *Cla* I and *Xho* I sites. Vectors were then linearized with *Not* I and *Spe* I restriction enzymes. A 6.12-kb fragment was recovered using a QIAquick Gel Extraction kit (28706, Qiagen), followed by removal of endotoxin using an EndoFree Plasmid Maxi kit (12362, Qiagen). Plasmids were diluted to 2.5 ng µl$^{-1}$, and 1 pl was injected intranuclearly into the male pronucleus in zygote at embryonic day 0.5 of C57BL/6 mice. Two-cell embryos were transplanted into the pseudopregnant ICR female mice. The positive $F_0$ offspring were identified by sequencing and crossed with C57BL/6 mice. The $F_1$ mice carrying the transgenic genomic were selected by PCR, and ALDOA or ALDOA-D34S expression was examined by immunoblotting. One verified $F_1$ mouse of each genotype was chosen to set up the transgenic mouse strain, which was then outcrossed six times to C57BL/6 mice before experiments.

## Metformin treatment of mice

The protocols described below for all mouse experiments were approved by the Institutional Animal Care and the Animal Committee

of Xiamen University (XMULAC20180028). Unless stated otherwise, mice were housed with free access to water and standard diet (65% carbohydrate, 11% fat, 24% protein) under specific pathogen-free conditions. The light was on from 8:00 to 20:00, with the temperature kept at 21–24 °C and humidity at 40–70%. Male littermate controls were used throughout the study. Metformin was supplied either in drinking water at desired concentrations or in standard diet at 0.1% (w/w) for 1 week. For creating the diabetic mouse model, mice were fed a HFD (60% calories from fat; D12492, Research Diets) for desired time periods starting at 4 weeks old.

The following ages of mice were used. (1) For isolating primary hepatocytes: normal-chow-diet-fed wild-type and *loxP*-flanked *Ampka* mice aged 4 weeks (Fig. 1a, b, Extended Data Figs. 1c, e–g, 13g, h and 14f–h, n); normal-chow-diet-fed PEN2-LKO mice aged 6 weeks (Fig. 2b, Extended Data Figs. 3c, j and 14l, j; into which tamoxifen was injected at 4 weeks old); HFD-fed wild-type mice aged 38 weeks (Extended Data Fig. 12i; fed with a HFD for 34 weeks starting from 4 weeks old); HFD-fed PEN2-LKO mice aged 38 weeks (Extended Data Fig. 12i; into which tamoxifen was injected at 35 weeks old after 31 weeks of HFD treatment starting from 4 weeks old); normal-chow-diet-fed ATP6AP1-LKO mice expressing full-length ATP6AP1 or ATP6AP1$^{Δ420–440}$ aged 8 weeks (Extended Data Fig. 14m, j; into which an AAV carrying ATP6AP1 was injected at 4 weeks old and tamoxifen at 5 weeks old); and HFD-fed ATP6AP1-LKO mice expressing full-length ATP6AP1 or ATP6AP1$^{Δ420–440}$ aged 38 weeks (Extended Data Fig. 12p; into which an AAV was injected at 34 weeks old and tamoxifen at 35 weeks old, after 34 weeks of HFD treatment starting from 4 weeks old). (2) For glucose tolerance tests (GTTs), insulin tolerance tests (ITTs) and measurements of metformin, GLP-1, insulin and TAG contents: wild-type mice aged 5 weeks (Extended Data Fig. 1h, l, m); PEN2-IKO and AMPKα1/2-IKO mice aged 6 weeks (Fig. 4a, b and Extended Data Fig. 12b, e; mice at 4 weeks old were injected with tamoxifen and were fed a HFD for 1 week, and then treated with metformin for another week); HFD-fed PEN2-LKO mice aged 54 weeks (Fig. 4c, d and Extended Data Fig. 12g, h; mice at 4 weeks old were fed a HFD for 31 weeks, and then injected with tamoxifen; at 38 weeks old, mice were treated with metformin for 16 weeks); HFD-fed ATP6AP1-LKO mice expressing full-length ATP6AP1 or ATP6AP1$^{Δ420–440}$ aged 54 weeks (Fig. 4e, f and Extended Data Fig. 12n, o; mice at 4 weeks old were fed a HFD for 30 weeks, and were injected with an AAV at 34 weeks old; at 35 weeks old, the mice were injected with tamoxifen and then treated with metformin for 16 weeks). (3) For pyruvate tolerance tests (PTTs): PEN2-LKO mice aged 6 weeks (Extended Data Fig. 14k; mice at 4 weeks old were injected with tamoxifen and then treated with metformin for 1 week starting from 5 weeks old). (4) For immunoblotting and measurement of adenylates: wild-type mice aged 5 weeks (Extended Data Figs. 1j, k, m, n and 13l; mice were treated with metformin for 1 week starting from 4 weeks old); PEN2-LKO mice aged 6 weeks (Fig. 4i; mice at 4 weeks old were injected with tamoxifen and then treated with metformin for 1 week starting from 5 weeks old); ATP6AP1-LKO mice expressing full-length ATP6AP1 or ATP6AP1$^{Δ420–440}$ aged 9 weeks (Fig. 4j; mice at 4 weeks old were injected with an AAV; at 5 weeks old, the mice were injected with tamoxifen and then treated with metformin for 1 week starting from 8 weeks old); HFD-fed PEN2-LKO mice aged 39 weeks (Extended Data Fig. 12j; mice at 4 weeks old were fed a HFD for 31 weeks and then injected with tamoxifen; at 38 weeks old, mice were treated with metformin for 1 week); HFD-fed ATP6AP1-LKO mice expressing full-length ATP6AP1 or ATP6AP1$^{Δ420–440}$ aged 39 weeks (Extended Data Fig. 12q; mice at 4 weeks old were fed a HFD for 30 weeks and then injected with an AAV at 34 weeks old; at 35 weeks old, the mice were injected with tamoxifen and then treated with metformin for 1 week starting from 38 weeks old); AXIN-LKO mice, LAMTOR1-LKO mice and Tg-ALDOA-D34S aged 7 weeks (Extended Data Figs. 9d, e and 11e; mice were treated with metformin for 1 week starting from 6 weeks old); Tg-ALDOA-D34S aged 6 weeks (Extended Data Fig. 11c; mice were starved for 16 h); hepatic *ATP6v0c* knockdown mice aged

7 weeks (Extended Data Fig. 9f; mice were treated with metformin 1 week starting from 6 weeks old, into which an AAV carrying a short interfering siRNA (siRNA) against *ATP6v0c* was intravenously injected at 4 weeks old); and *TRPV1*[−/−] and hepatic *TRPV2–TRPV4* knockdown mice aged 8 weeks (Extended Data Fig. 11g; mice were treated with metformin for 1 week starting from 7 weeks old, into which an AAV carrying siRNAs against *TRPV–TRPV4* was intravenously injected at 5 weeks old). (5) For hepatic haematoxylin and eosin (H&E) staining: HFD-fed PEN2-LKO mice aged 54 weeks (Extended Data Fig. 12k; mice at 4 weeks old were fed a HFD for 31 weeks and then injected with tamoxifen; at 38 weeks old, mice were treated with metformin for 16 weeks); HFD-fed ATP6AP1-LKO mice expressing full-length ATP6AP1 or ATP6AP1[Δ420−440] aged 54 weeks (Extended Data Fig. 12r; mice at 4 weeks old were fed a HFD for 30 weeks and then injected with an AAV at 34 weeks old; at 35 weeks old, the mice were injected with tamoxifen and then treated with metformin for 16 weeks). (6) For all the other experiments, mice aged 4 weeks were used.

## Serology, GTTs, ITTs and PTTs

Mice were individually caged for 1 week before each experiment. For GTTs, mice were fasted for 6 h (8:00 to 14:00), then gavaged or intraperitoneally injected with glucose at 1.5 g kg$^{-1}$ (for lean mice) or 1 g kg$^{-1}$ (for HFD-induced diabetic mice). Blood glucose was measured at the indicated time points through tail-vein bleeding using a OneTouch UltraVue automatic glucometer (LifeScan). ITTs were performed as per the GTTs, except that 1 U kg$^{-1}$ insulin was intraperitoneally injected. PTTs were performed as GTTs, except that 1 g kg$^{-1}$ sodium pyruvate was intraperitoneally injected into 16-h fasted (18:00 previous day to 10:00) mice. GTTs, ITTs and PTTs were performed using different batches of mice.

For measuring GLP-1 levels, 300 µl of blood from each mouse was collected in an ice-cold, K$_2$EDTA spray-coated tube (366420, BD P800 blood collection system) containing pre-added 50 µl aprotinin (5 mg ml$^{-1}$) and 50 µl diprotin A (5 mg ml$^{-1}$) as previously described[42,43] (with modifications of concentrations of aprotinin and diprotin A used). Plasma was then prepared by centrifugation at 3,000g for 10 min at 4 °C, and 100 µl was used to determine the levels of GLP-1 using a GLP-1 EIA kit according to the manufacturer's instructions.

For measuring insulin levels, approximately 100 µl of mouse blood was collected at each time point (from the submandibular vein plexus) and was placed at room temperature for 20 min, followed by centrifugation at 3,000g for 10 min at 4 °C. A total of 5 µl of serum (the supernatant) was used to determine the levels of insulin using a Mouse Ultrasensitive Insulin ELISA kit according to the manufacturer's instructions. The five-parameter logistic fitted standard curve for calculating the concentration of insulin was generated from the Arigo Biolaboratories website (https://www.arigobio.cn/ELISA-calculator).

## Histology

For H&E staining, liver tissues were fixed in 4% (v/v) paraformaldehyde for 24 h at room temperature then transferred to embedding cassettes. The cassettes were then washed in running water for 12 h, followed by successive soaking, each for 1 h, in 70% ethanol (v/v in water), 80% ethanol and 95% ethanol. The fixed tissues were further dehydrated in anhydrous ethanol for 1 h twice, followed by immersing in 50% xylene (v/v in ethanol) for 30 min, with two changes of xylene (15 min each) and two changes of paraffin wax (58–60 °C; 1 h each). The dehydrated tissues were embedded in paraffin on a HistoCore Arcadia paraffin embedding machine (Leica). Paraffin blocks were then sectioned at a thickness of 3 µm, dried on an adhesion microscope slide, followed by rehydrating in the following order: two changes of xylene at 70 °C 10 min each; two changes of anhydrous ethanol 5 min each; two changes of 95% ethanol 5 min each; one change of 80% ethanol for 5 min; one change of 70% ethanol for 5 min; one change of 50% ethanol for 5 min; and briefly in water. The sections were then stained in haematoxylin

solution for 8 min, then washed in running water for 5 min, differentiated in 1% hydrochloric acid (in ethanol) for 30 s, washed in running water for 1 min, immersed in 0.2% (v/v in water) ammonium hydroxide solution for 30 s, washed in running water for 1 min and stained in eosin Y solution for 30 s. The stained sections were dehydrated in 70% ethanol for 5 min, twice in 95% ethanol for 5 min each, twice in anhydrous ethanol for 5 min each and two changes of xylene for 15 min each. The stained sections were mounted with Canada balsam and visualized on a Leica DM4 B microscope. Images were processed using LAS X software (v.3.0.2.16120, Leica), and formatted using Photoshop 2021 software (Adobe).

For measuring the TAG content, mice were euthanized, and the livers were immediately removed and rinsed in PBS for three times. Approximately 50 mg tissue was homogenized in 1 ml of PBS containing 5% (v/v) Triton X-100. The homogenates were boiled for 5 min followed by centrifugation at 20,000g at 25 °C for 10 min. The TAG content (from the supernatant) was determined using Labassay triglyceride reagent.

## *C. elegans* strains

Wild-type (N2 Bristol), *aak-2*(*ok524*) and *unc-76*(*e911*) strains were obtained from the *Caenorhabditis* Genetics Center. Unless stated otherwise, worms were maintained on nematode growth medium (NGM) plates (1.7% (w/v) agar, 0.3% (w/v) NaCl, 0.25% (w/v) bacteriological peptone, 1 mM CaCl$_2$, 1 mM MgSO$_4$, 25 mM KH$_2$PO$_4$-K$_2$HPO$_4$, pH 6.0, 0.02% (w/v) streptomycin and 5 µg ml$^{-1}$ cholesterol) spread with *Escherichia coli* OP50 as standard food. Metformin of desired concentrations was added to the autoclaved NGM (cooled down to 50 °C) before pouring onto plates. All worms were cultured at 20 °C.

*pen-2* was knocked down instead of knocked out in *C. elegans* because complete depletion of PEN-2 is lethal to *C. elegans*[7]. To knock down *pen-2*, the growth of nematodes was first synchronized: worms were washed off from agar plates with 15 ml M9 buffer (22.1 mM KH$_2$PO$_4$, 46.9 mM Na$_2$HPO$_4$, 85.5 mM NaCl and 1 mM MgSO$_4$) supplemented with 0.05% (v/v) Triton X-100 per plate, followed by centrifugation at 1,000g for 2 min. The worm sediment was suspended with 6 ml of M9 buffer containing 50% synchronizing bleaching solution (by mixing 25 ml of NaClO solution (5% active chlorine), 8.3 ml of 25% (w/v) NaOH and 66.7 ml of M9 buffer, for a total of 100 ml), followed by vigorous shaking for 2 min and centrifugation for 2 min at 1,000g. The sediment was washed with 12 ml of M9 buffer twice, then suspended with 6 ml of M9 buffer, followed by rotating at 20 °C, 30 r.p.m. for 12 h. Synchronized worms (around the L1 stage) were then placed on RNAi plates (NGM containing 1 mg ml$^{-1}$ IPTG and 50 µg ml$^{-1}$ carbenicillin) spread with HT115 *E. coli* stains containing RNAi against *pen-2* (well A05 on plate 86 from *C. elegans* RNAi Collection (Ahringer)) for 2 days. The knockdown efficiency was examined by determining the levels of *pen-2* mRNA by real-time quantitative PCR (qPCR). Approximately 1,000 worms were washed off a RNAi plate with 15 ml of M9 buffer containing Triton X-100, followed by centrifugation for 2 min at 1,000g. The sediment was then washed with 1 ml of M9 buffer twice, and then lysed with 1 ml of TRIzol. The worms were then frozen in liquid nitrogen, thawed at room temperature and then subjected to repeated freeze–thaw for another two times. The worm lysates were then placed at room temperature for 5 min, then mixed with 0.2 ml of chloroform followed by vigorous shaking for 15 s. After 3 min, lysates were centrifuged at 20,000g at 4 °C for 15 min, and 450 µl of the aqueous phase (upper layer) was transferred to a new RNase-free centrifuge tube, followed by mixing with 450 µl of isopropanol, then centrifuged at 20,000g at 4 °C for 10 min. The sediment was washed with 1 ml of 75% ethanol (v/v) followed by centrifugation at 20,000g for 10 min, and then with 1 ml of anhydrous ethanol followed by centrifugation at 20,000g for 10 min. The sediment was dissolved with 20 µl of RNase-free water after the ethanol was evaporated. The dissolved RNA was then reverse-transcribed to cDNA using ReverTra Ace qPCR RT master mix with a gDNA Remover kit, followed by performing real-time qPCR using Maxima SYBR

Green/ROX qPCR master mix on a CFX96 thermocycler (Bio-Rad). Data were analysed using CFX Manager software (v.3.1, Bio-Rad). Knockdown efficiency was evaluated according to the CT value obtained. The primers for *pen-2* are 5′-TACGTGATCGCCAGCATTGT-3′ and 5′-CGTGTGGACCGATTTCCTGA-3′. The primers for *ama-1* (the internal control) are 5′-GACATTTGGCACTGCTTTGT-3′ and 5′-ACGATTGATTCCATGTCTCG-3′.

The *ATP6AP1*$^{-/-}$ *C. elegans* strains expressing ATP6AP1 or its Δ420–440 mutant were established as follows: ATP6AP1 or its Δ420–440 mutant was introduced to the *unc-76(e911)* *C. elegans* strain, which had been outcrossed six times to the N2 strain; such generated strains were then subjected to knockout of the *ATP6AP1* (*vha-19*) gene, and the uncoordinated phenotype of *unc-76* was rescued. To generate *unc-76* strains expressing ATP6AP1 or the Δ420–440 mutant, cDNA of ATP6AP1 or ATP6AP1$^{Δ420–440}$ was inserted into a pJM1 vector, with GFP as a selection marker, between the *Nhe* I and *Kpn* I sites (expressed under control by a *sur-5* promoter), then injected into the syncytial gonad of the worm. Microinjection was performed using a Leica DMi8 microscope equipped with a M-152 manipulator (Narishige) and a microinjector system (Tritech). The injection pad was prepared by placing 2 drops (approximately 50 μl) of boiling 2% agarose (w/v) onto the centre of a glass coverslip (24 × 50 mm, 0.13–0.15 mm thickness), immediately followed by flattening with another coverslip, then dried at room temperature for 24 h. The injection needle was processed from a glass capillary (Borosil 1.0 × 0.75 mm ID/Fibre with Omega dot fibre, FHC) by a PC-100 Puller (Narishige) using the Step 2 programme (with heater no. 1 at 66 °C, and no. 2 at 75 °C), and was then loaded with 0.5 μl of pJM1-ATP6AP1 or ATP6AP1$^{Δ420–440}$ plasmid (200 ng μl$^{-1}$, centrifuged at 20,000*g* for 15 min before loading), and then connected to the nitrogen gas source at a pressure adjusted to 20 psi. Successful loading of plasmid was checked using a dissection microscope before loading onto the manipulator at a 45° angle in relation to the injection pad. All equipment was adjusted and aligned to make sure that a clear, centred view of the pad and the needle was obtained. The sealed needle tip was then opened by gentle tapping on the lateral side of the coverslip to a size that allowed a five-times broader droplet to be made during each injection. A young adult *unc-76* worm with well distinguishable gonads was then gently anchored on an injection pad covered with a thin layer of microinjection oil (Halocarbon oil 700). The pJM1-ATP6AP1 or ATP6AP1$^{Δ420–440}$ plasmid was then injected into the syncytial arm of the gonad (gonadal sheath) until a slightly visible swelling was achieved. A drop of M9 buffer was then added to the injected worm, and the worm was floated on microinjection oil then picked and recovered on a NGM plate for 2 days. The F$_1$ GFP-expressing hermaphrodite was selected for further culture. The genomic sequence encoding *ATP6AP1* (*vha-19*) was then knocked out from this strain by injecting a mixture of a pDD122 (Peft-3::Cas9 + ttTi5605 sgRNA) vector carrying sgRNAs against *vha-19* (5′-CGTCGAAAAAACCCGATTGTTGG-3′ for intron 2, and 5′-AATGATGTCAGGTTTTTTTCTGG-3′ for intron 3, designed using the CHOPCHOP website http://chopchop.cbu.uib.no/), and the p7616B (*unc-76* (+)) rescue plasmid (100 μg ml$^{-1}$ each) into young adult worms. The F$_1$ hermaphrodite worms with normal postures were individually cultured on a NGM plate. After egg-laying, worms were lysed using Single Worm lysis buffer (50 mM HEPES, pH 7.4, 1 mM EGTA, 1 mM MgCl$_2$, 100 mM KCl, 10% (v/v) glycerol, 0.05% (v/v) NP-40, 0.5 mM DTT and protease inhibitor cocktail), followed by PCR with the primers 5′-AACTGCTTTTGGCTCGAAAATA-3′ and 5′-AAGTAAAAAGGGACAAAAGTCG-3′ for genotyping. The offspring generated from knockout-assured individuals were outcrossed six times to the N2 strain, and the expression levels of ATP6AP1 or ATP6AP1$^{Δ420–440}$ were examined by immunoblotting. Strains expressing ATP6AP1 or ATP6AP1$^{Δ420–440}$ at similar levels were chosen for further experiments.

The ages of the nematodes used in this study were as follows: (1) for lifespan assays, worms at L4 stage were used (Fig. 4g, h, Extended Data Figs. 13a and 14d, e; treated with metformin or *N*-acetylcysteine until death); (2) for analysis of adenylates and pharmacokinetics of metformin, p-AMPKα and reactive oxygen species (ROS) levels (Extended Data Figs. 13b, c, e, f, i–k and 14c, i), worms at L4 stage (after treatment of metformin for 1 day) were used; and (3) for the experiments using *pen-2* knock down worms, worms at L1 stage were used (Extended Data Fig. 13d; fed with HT115 *E. coli* strain containing RNAi against *pen-2*).

### Evaluation of nematode lifespan

Synchronized worms were cultured to the L4 stage before transfer to the desired agar plates. Worms were transferred to new plates every 2 days. Live and dead worms were counted during the transfer. Worms that displayed no movement after gentle touching with a platinum picker were judged as dead. Kaplan–Meier curves were generated using Prism 9 (GraphPad software), whereas the statistical analysis data were analysed using SPSS 27.0 (IBM).

### Reagents

Rabbit polyclonal antibody against LAMTOR1 was raised and validated as previously described[24], and was diluted 1:100 for immunoprecipitation (IP) or 1:500 for immunoblotting (IB). Rabbit polyclonal antibody against ATP6AP1 was raised with bacterially expressed and purified ATP6AP1 (amino acids 440–470, GST-tagged), and was diluted 1:100 for IP. The following antibodies were purchased from Cell Signaling Technology: rabbit anti-phospho-AMPKα-T172 (cat. 2535, 1:1,000 for IB), anti-AMPKα (cat. 2532, 1:1,000 for IB), anti-AMPKβ1/2 (cat. 4150, 1:1,000 for IB), anti-phospho-ACC-Ser79 (cat. 3661, 1:1,000 for IB), anti-ACC (cat. 3662, 1:1,000 for IB), anti-phospho-p70 S6K-S389 (cat. 9234, 1:1,000 for IB), anti-p70 S6K (cat. 2708, 1:1,000 for IB), anti-LKB1 (cat. 3047, 1:1,000 for IB), anti-AXIN1 (cat. 2074, 1:1,000 for IB), anti-presenilin 1 (cat. 3622, 1:1,000 for IB), anti-presenilin 2 (cat. 2192, 1:1,000 for IB), anti-nicastrin (cat. 3632, 1:1,000 for IB), anti-β-tubulin (cat. 2128, 1:1,000 for IB), anti-HA-tag (cat. 3724, 1:1,000 for IB or 1:120 for immunofluorescent (IF) staining), anti-PDI (cat. 3501, 1:1,000 for IB), anti- cytochrome *c* (cat. 4280, 1:1,000 for IB), anti-clathrin (cat. 4796, 1:1,000 for IB), anti-p62 (cat. 23214, 1:1,000 for IB), anti-ATG5 (cat. 12994, 1:1,000 for IB), anti-PDI (Alexa Fluor 488-conjugated, cat. 5051, 1:60 for IF), HRP-conjugated mouse anti-rabbit IgG (conformation-specific, cat. 5127, 1:2,000 for IB), HRP-conjugated goat anti-rat IgG (conformation-specific, cat. 98164, 1:2,000 for IB) and mouse anti-Myc-tag (cat. 2276, 1:500 for IB). Rabbit anti-ATP6v0c (cat. NBP1-59654, 1:1,000 for IB or 1:100 for IP) antibody was purchased from Novus Biologicals. Mouse anti-FLAG M2 (cat. F1804, 1:1,000 for IB), goat anti-rabbit IgG antibody and anti-FLAG M2 affinity gel (cat. A2220, 1:500 for IP) were purchased from Sigma. Rabbit anti-PEN2 (cat. ab154830, 1:1,000 for IB or 1:100 for IP and IF), anti-ATP6AP1 (cat. ab176609, 1:500 for IB), anti-transferrin (cat. ab1223, 1:500 for IB), anti-TGN46 (cat. ab76282, 1:60 for IF) and rat anti-LAMP2 (cat. ab13524; 1:1000 for IB or 1:120 IF) antibodies were purchased from Abcam. Goat anti-AXIN (cat. sc-8567, 1:120 for IF), mouse anti-HA (cat. sc-7392, 1:1,000 for IB, 1:500 for IP or 1:120 for IF), mouse anti-goat IgG-HRP antibody were purchased from Santa Cruz Biotechnology. Normal rabbit control IgG (cat. CR1, 1:100 for IP) was purchased from Sino Biological. Goat anti-mouse IgG (cat. 115-035-003, 1:1,000 for IB) and anti-rabbit (cat. 111-035-003, 1:1,000 for IB) antibodies were purchased from Jackson ImmunoResearch. Donkey anti-goat IgG (cat. A-11055, 1:1,000 for IB), anti-rabbit IgG (cat. A-21206, 1:1,000 for IB), anti-rat IgG (cat. A21209, 1:1,000 for IB), goat anti-rat IgG (cat. A-21247, 1:1,000 for IB), rabbit anti-APH1 (cat. PA1-2010, 1:1,000 for IB), mouse anti-Strep-tag (cat. MA5-17283, 1:1,000 for IB) antibodies were purchased from Thermo Scientific. Rabbit anti-ATP1A1 (cat. 14418-1-AP, 1:60 for IF) and anti-TOMM20 (cat. 11802-1-AP, 1:60 for IF) antibodies were purchased from Proteintech.

Glucose (cat. G7021), HEPES (cat. H4034), lysosome isolation kit (cat. LYSISO1), CaCl$_2$ (cat. C5670), PEP (cat. P7002), β-NADH (cat. N8129), pyruvate kinase (cat. P9136), LDHA (cat. SAE0049), FITC–dextran (cat. FD10S), H$_2$O$_2$ (cat. 323381), metformin (cat. D150959), KCl (cat.

P9333), $MgSO_4$ (cat. M2643), $KH_2PO_4$ (cat. P5655), $NaH_2PO_4$ (cat. S8282), $Na_2HPO_4$ (cat. S7907), DAB (cat. D8001), ethanol (cat. 459836), acetonitrile (cat. 34888), isopropanol (cat. 34863), dichloromethane (cat. 650463), SDS (cat. 436143), sodium acetate (cat. S2889), EmbryoMax DMEM (cat. SLM-220-M), demecolcine (cat. D7385), M2 medium (cat. M7167), corn oil (cat. C8267), insulin (cat. I1882), sodium pyruvate (cat. P2256), halocarbon oil 700 (cat. H8898), agar (cat. A1296), tryptone (cat. T9410), cholesterol (cat. C3045), sodium hypochlorite solution (cat. 239305), IPTG (cat. I6758), carbenicillin (cat. C1613), agarose (cat. A9539), collagenase type IV (cat. C5138), BSA (cat. A2153), $CH_3COOK$ (cat. P1190), magnesium acetate tetrahydrate (cat. M5661), digitonin (cat. D141), oligomycin A (cat. 75351), FCCP (cat. C2920), antimycin A (cat. A8674), rotenone (cat. R8875), ethanolamine (cat. 411000), acetone (cat. 650501), Coomassie Brilliant Blue R-250 (cat. 1.12553), GLP-1 EIA kit (cat. RAB0201), aprotinin (cat. A1153), diprotin A (cat. I9759), trypsin (cat. T1426), 2-mercaptoethanol (cat. M6250), biotin (cat. 14400), $NaN_3$ (cat. S2002), $CH_2Cl_2$ (cat. 270997), pyridine (cat. 270970), L-glutamine (cat. G3126), paraformaldehyde (cat. 158127), haematoxylin solution (cat. 03971), eosin Y solution (cat. 318906), Canada balsam (cat. C1795), xylene (cat. 214736), hydrochloric acid in ethanol (cat. 1.00327), PEG (cat. 89510), phenol red solution (cat. P0290), sucrose (cat. S7903), CsCl (cat. 289329), $Na_2H_2P_2O_7$ (cat. P8135), β-glycerophosphate (cat. 50020), AICAR (cat. A9978), A23187 (cat. C7522), A-769662 (cat. SML2578), tamoxifen (cat. T5648), $NaHCO_3$ (cat. S5761), EGTA (cat. E3889), poly-L-lysine solution (cat. P8920), formaldehyde solution (formalin, cat. F8775), OptiPrep (cat. D1556), $MgCl_2$ (cat. M8266), tetramethylsilane (cat. T24007), Trizma base (Tris, cat. T1503), glycerol (cat. G5516), DMSO (cat. D2650), TCEP (cat. C4706), TBTA (cat. 678937), $CuSO_4$ (cat. C1297), IGEPAL CA-630 (NP-40, cat. I3021), methanol (cat. 646377), $CHCl_3$ (cat. C7559), buformin hydrochloride (cat. SML1496), octyl β-D-glucopyranoside (cat. O8001), Triton X-100 (cat. T9284), concanamycin A (cat. C9705), DTT (cat. 43815), MEA (cat. 30070), glucose oxidase (cat. G2133), catalase (cat. C40), ammonium hydroxide solution (cat. 338818), FLAG peptide (cat. F3290), EDTA (cat. E6758), polybrene (cat. H9268), HCl (cat. 320331), NaCl (cat. S7653), NaOH (cat. S8045), ATP disodium salt (cat. A2383), ATP magnesium salt (cat. A9187), D-mannitol (cat. M4125), Percoll (cat. P4937), imidazole (cat. I5513), chloroquine (cat. C6628), cytochalasin D (cat. C2618), N-acetyl-L-cysteine (cat. A9165), phenformin hydrochloride (cat. P7045), formic acid (cat. 5.43804), ammonium formate (cat. 70221), glucagon (cat. 05-23-2700), DEPC-treated water (cat. 693520), glutaraldehyde solution (cat. G5882), glycine (cat. G8898), streptavidin agarose (cat. 16-126), D-desthiobiotin (cat. 71610-M), myristic-d27 acid (cat. 68698), methoxyamine hydrochloride (cat. 89803), MTBSTFA (with 1% t-BDMCS, cat. M-108), hexane (cat. 34859), fatty acid-free BSA (cat. SRE0098), APS (cat. A3678), TEMED (cat. T9281) and Tween-20 (cat. P9416) were purchased from Sigma. WesternBright ECL and peroxide solutions (cat. 210414-73) were purchased from Advansta. Acrylamide/Bis solution, 30%, 29:1 (cat. 1610156) was purchased from Bio-Rad. TAG (15:0/15:0/15:0, cat. 26962) was purchased from Cayman. Protease inhibitor cocktail (cat. 70221) was purchased from Roche. Hoechst (cat. H1399), LysoSensor Green DND-189 (cat. L7535), ProLong Diamond antifade mountant (cat. P36970), ProLong Live Antifade reagent (cat. P36975), NeutrAvidinTM agarose (cat. 29204), Lipofectamine 2000 (cat. 11668500), DMEM, high glucose (cat. 11965175), DMEM, no glucose (cat. 11966025), DMEM without phenol red (cat. 21063045), MEM amino acids solution (cat. 11130077), MEM non-essential amino acids solution (cat. 11140050), JC-1 (cat. T3168), $CM-H_2DCFDA$ (cat. C6827), CellROX Deep Red (cat. C10422), ESC-qualified fetal bovine serum (FBS) (cat. 30044333), Maxima SYBR Green/ROX qPCR master mix (cat. K0223), Trypan Blue stain (cat. T10282), FBS (cat. 10099141C), penicillin–streptomycin (cat. 15140163), William's E medium, no glutamine (cat. 12551032), liver perfusion medium (cat. 17701), liver digest medium (cat. 17703), GlutaMAX (cat. 35050061), sodium pyruvate (cat. 11360070) and TRIzol (cat. 15596018) were purchased from Thermo

Scientific. Internal Standards 1 (cat. H3304-1002) and Internal Standards 3 (cat. H3304-1104) were purchased from Human Metabolome Technologies. rProtein A Sepharose Fast Flow (cat. 17127904), Protein G Sepharose 4 Fast Flow (cat. 17061806), Series S Sensor Chip CM5 (cat. BR100530), Amine Coupling kit (with EDC and NHS included, cat. BR100050) were purchased from Cytiva. $OsO_4$ (cat. 18465), uranyl acetate (cat. 19481) were purchased from Tedpella. SPI-Pon 812 Embedding kit (cat. 02660-AB) was purchased from SPI. DAPT (cat. S2215), RO4929097 (cat. S1575), bafilomycin A1 (cat. S1413), 3-MA (cat. S2767), Dynasore (cat. S8047), Dyngo-4a (cat. S7163) and nystatin (cat. S1934) were purchased from Selleck. Methyl-β-cyclodextrin (cat. HY-101461) was purchased from MedChemExpress. Labassay triglyceride reagent (cat. 290-63701) was purchased from Wako Pure Chemical Industries. Mouse Ultrasensitive Insulin ELISA kit (cat. 80-INSMSU-E10) was purchased from ALPCO. ReverTra Ace qPCR RT master mix with gDNA remover (cat. FSQ-301) was purchased from Toyobo. 3-(But-3-*yn*-1-*yl*)-3-(2-iodoethyl)-3H-diazirine (cat. BD627886) was purchased from Bide Pharmatech. SMM 293-TII expression medium (cat. M293TII) was purchased from Sino Biological. Seahorse XF base medium (cat. 103334) was purchased from Agilent. Paraplast high melt paraffin (cat. 39601095) was purchased from Leica. [U-$^{13}$C]-glutamine (cat. 184161-19-1), [U-$^{13}$C]-palmitic acid (cat. CLM-409) and [U-$^{13}$C]-glucose (cat. CLM-1396) were purchased from Cambridge Isotope Laboratories.

## Plasmids

Full-length cDNAs used in this study were obtained either by PCR using cDNA from MEFs or by purchasing from Origene, Sino Biological or Genescript. Mutations of *PEN2* and *ATP6AP1* were performed by PCR-based site-directed mutagenesis using PrimeSTAR HS polymerase (Takara). Expression plasmids for various epitope-tagged proteins were constructed in the pcDNA3.3 vector for transfection (ectopic expression) or in the pBOBI vector for lentivirus packaging (stable expression). To express Strep-tagged ATP6AP1, the *ATP6AP1* cDNA was inserted into a modified pcDNA3.3-C-HA vector with the sequence encoding the HA epitope tag replaced by the sequence of the Strep tag[44]. PCR products were verified by sequencing (Invitrogen). The lentivirus-based vector pLV-H1-EF1a-puro was used for expression of siRNA in MEFs and HEK293T cells, and the AAV-based vector pAAV2 for mouse liver. The sequences for siRNAs of mouse *Pen2* were as follows: 5′-GCCTGTGCCGGAAGTACTAT-3′ (1) and 5′-GTTCTTTGGTTAGTC AACATTT-3′ (2). All plasmids, except those used for adenovirus packaging (see below, the 'Packaging and injection of adenovirus and AAV' section), were purified using the caesium chloride density gradient ultracentrifugation method.

## Primary hepatocytes

Human primary hepatocytes were isolated from surgically removed liver tissues. Fresh tissues were minced followed by digesting in 0.25% (w/v) trypsin supplemented with 0.5 mg ml$^{-1}$ collagenase type IV for 10 min at 37 °C. Cells were then immediately plated (at 60–70% confluence) in collagen-coated 6-well plates in William's medium E plus 10% FBS, 100 IU penicillin and 100 mg ml$^{-1}$ streptomycin. After 4 h of attachment, the medium was replaced with fresh William's medium E with 1% (w/v) BSA for another 12 h before further use. This study was approved by the Human Research Ethics Committee of the Shanghai Sixth People's hospital following the principles of the Declaration of Helsinki. Written informed consent was obtained from all participants.

Mouse primary hepatocytes were isolated with a modified two-step perfusion method using liver perfusion medium and liver digest buffer. Before isolation of hepatocytes, mice were first anaesthetized followed by the insertion od a 0.72 × 19 mm intravenous catheter into the postcava. After cutting off the portal vein, mice were perfused with 50 ml of liver perfusion medium at a rate of 5 ml min$^{-1}$, followed by 50 ml of liver digest buffer at a rate of 2.5 ml min$^{-1}$. The digested liver was then briefly rinsed with PBS and then dissected by gently tearing apart the

Glisson's capsule with two sterilized, needle-pointed tweezers on a 6-cm dish containing 3 ml of PBS. The dispersed cells were mixed with 10 ml of ice-cold William's medium E plus 10% FBS and were filtered by passing through a 100-μm cell strainer (BD Falcon). Cells were then centrifuged at 50$g$ at 4 °C for 2 min, followed by washing twice with 10 ml of ice-cold William's medium E plus 10% FBS. Cells were then plated and cultured as for human primary hepatocytes. $Pen2^{-/-}$ hepatocytes were established by infecting $Pen2^{F/F}$ hepatocytes (isolated from $loxP$-flanked $Pen2$ mice) with adenoviruses expressing the Cre recombinase (or GFP as a control) for 6 h, followed by incubating in William's medium E with 1% (w/v) BSA for another 12 h before experiments.

## Cell lines

HEK293T, AD293 (Adeno-X 293) cells, MEFs and L929 cells were maintained in Dulbecco's modified Eagle's medium supplemented with 10% FBS, 100 IU penicillin, 100 mg ml$^{-1}$ streptomycin at 37 °C in a humidified incubator containing 5% $CO_2$. The suspension HEK293T cell line, which was established from HEK293T cells, was a gift from Z. Wang (Xiamen Immocell Biotechnology). Cells were cultured in 400 ml of SMM 293-TII medium in a 2-litre conical glass flask at 37 °C, 160 r.p.m. on a $CO_2$-resistant shaker (cat. 88881101, Thermo Scientific) in a humidified incubator containing 5% $CO_2$. All cell lines were verified to be free of mycoplasma contamination. HEK293T cells were authenticated by STR sequencing. Polyethylenimine (PEI) at a final concentration of 10 μM was used to transfect HEK293T cells. Total DNA to be transfected for each plate was adjusted to the same amount by using a relevant empty vector. Transfected cells were collected 24 h after transfection.

Lentiviruses, including those for knockdown or stable expression, were packaged in HEK293T cells by transfection using Lipofectamine 2000. At 30 h after transfection, medium (approximately 2 ml) was collected and centrifuged at 5,000$g$ for 3 min at room temperature. The supernatant was mixed with 10 μg ml$^{-1}$ (final concentration) polybrene, and this was added to MEFs or HEK293T cells followed by centrifuging at 3,000$g$ for 30 min at room temperature (spinfection). Cells were incubated for another 24 h (MEFs) or 12 h (HEK293T cells) before further treatments.

$LAMTOR1^{F/F}$, $AXIN^{F/F}$ and $ATG5^{F/F}$ MEFs were established by introducing SV40 T antigen using lentivirus into cultured primary embryonic cells from mouse litters. $LAMTOR1^{-/-}$ MEFs were generated by infecting $LAMTOR1^{F/F}$ MEFs with adenoviruses expressing the Cre recombinase for 12 h, as for $AXIN^{-/-}$ MEFs, $ATG5^{-/-}$ MEFs and $APH1A$–$APH1C$ triple knockout MEFs. The infected cells were then incubated in fresh DMEM for another 12 h before further treatments. $TRPV1$–$TRPV4$ quadruple knockout MEFs were generated as previously described[25].

The genes ($PEN2$, $ATP6AP1$, $PSEN1$, $PSEN2$, $NCSTN$, $PRKAB1$ and $PRKAB2$) were deleted from MEFs or HEK293T cells using the CRISPR–Cas9 system. Nucleotides were annealed to their complements containing the cloning tag aaac, and inserted into the back-to-back $Bsm$B I restriction sites of lentiCRISPRv2 vector. The sequence for each sgRNA is as follows: 5′-ATGAGGAGAAGTTGAACCTG-3′ (1) and 5′-CAGATCTACCGGCCCCGCTG-3′ (2) for mouse $Pen2$; 5′-CATCT TCTGGTT CTTCCGAG-3′ (1) and 5′-CCGGAAGTACTACCTGGGTA-3′ (2) for human $PEN2$; 5′-GGTGGCCCGTGATATAACCA-3′ for mouse $Atp6ap1$; 5′-GATGTAGCCG TGGTGGCCGGA-3′ for human $ATP6AP1$; 5′-CTGAGCCAATATCT AATGGG-3′ for mouse $Psen1$; 5′-CACGCT GTGTATGATCG-3′ for mouse $Psen2$; 5′-CTGTGG AATGAACTGGGCAA-3′ for mouse $Ncstn$; 5′-GAGATCCTTACC TTCTCGTG-3′ for mouse $Prkab1$; and 5′-AGCTCGGAGACG TCATGTCG-3′ for mouse $Prkab2$. The constructs were then subjected to lentivirus packaging using HEK293T cells that were transfected with 2 μg of DNA in Lipofectamine 2000 transfection reagent per well of a 6-well plate. At 30 h after transfection, the virus (approximately 2 ml) was collected and used for infecting MEFs or HEK293T cells as described above, except cells cultured to 15% confluence were incubated with the virus for 72 h. In particular, for HEK293T cells, 0.5 ml of fresh DMEM was supplemented to each well

after 36 h of infection. When cells were approaching confluence, they were single-cell-sorted into 96-well dishes. Clones were expanded and evaluated for knockout status by sequencing. For glucose starvation, cells were rinsed twice with PBS and then incubated in glucose-free DMEM supplemented with 10% FBS and 1 mM sodium pyruvate for desired periods of time at 37 °C.

## Packaging and injection of adenovirus and AAV

Adenoviruses (AV) carrying Cre recombinase (Ad-Cre) were packaged using the AdEasy Adenoviral Vector system (240009, Agilent) in AD293 (Adeno-X 293) cells. In brief, pAdEasy vector carrying Cre recombinase was linearized with $Pac$ I for 12 h, and the efficiency was confirmed by subjecting 0.2 μg of each linearized vector to 0.8% agarose gel (w/v, showing an approximately 30-kb band and a 4.5-kb band). The linearized vector was precipitated with two volumes of ethanol then dissolved with 20 μl of sterile water. A total of 5 μg of linearized vector was transfected into $3 × 10^6$ AD293 cells cultured in a 60-mm dish by Lipofectamine 2000, and 3 ml of medium was refreshed after 12 h of transfection. Cells were cultured for another 7 days, with 1.5 ml of fresh medium added (not refreshed) every other day. Cells were collected together with the culture medium followed by three rounds of freeze–thaw cycles. Cell debris was removed by centrifugation at 20,000$g$ for 10 min, and the supernatant was used to infect two 6-cm dishes of AD293 cells followed by 3 rounds of amplification to produce a 10-fold increase in titre. Viral particles were loaded on the top of 5 ml of 15% CsCl (dissolved in TBS (10 mM Tris, 0.9% (w/v) NaCl, 2.5% (w/v) sucrose, pH 8.1)) cushion over 4.5 ml of 40% CsCl cushion (w/v, dissolved in TBS) in an ultracentrifuge tube (cat. 344059, Beckman). The sample wase centrifuged at 30,000 r.p.m. in a SW41 rotor (Beckman) for 16 h at 4 °C. The heavier band was collected followed by dialysis in TBS for 1 h at 4 °C. Purified Ad-Cre viruses were stored at −80 °C.

AAVs were packaged in HEK293T cells using the protocol from Grieger et al.[45]. In brief, cells used for in-house viral production were maintained in 150-mm dishes. A total of 7 μg of pAAV-RC2/9 (AAV2 inverted terminal repeat (ITR) vectors pseudo-typed with AAV9 capsid) plasmid, 21 μg of pAAV-helper plasmid and 7 μg of pAAV2 plasmid (carrying ATP6AP1 or its mutant, or siRNAs against mouse $Trpv2$ to $Trpv4$) were added to 4 ml of DMEM without phenol red, followed by mixing with 175 μl of PEI solution (1 mg ml$^{-1}$, pH 7.5). The mixture was then incubated at room temperature for 20 min and then added to the dishes. At 60 h after transfection, cells were collected by scraping and centrifugation. The viral particles were purified from the pellet using an Optiprep gradient as previously described[45]. The titres of purified AAV were determined by real-time qPCR (see below). Viruses were stored at −80 °C before use, and were delivered to mice intravenously by lateral tail-vein injection. For each mouse, $1 × 10^{11}$ particles of virus, adjusted to 200 μl of final volume (with PBS, pH 7.4), was injected.

## IP and IB assays

For IP endogenous proteins, LAMTOR1, PEN2 and ATP6AP1 were immunoprecipitated and analysed as previously described[24], but with minor modifications. In brief, ten 15-cm dishes of MEFs (grown to 80% confluence) were collected for IP of LAMTOR1, or two 10-cm dishes of MEFs (grown to 80% confluence) were collected for IP of PEN2 and ATP6AP1. Cells were lysed with 750 μl per dish of ice-cold ODG buffer (for LAMTOR1; 50 mM Tris-HCl, pH 8.0, 50 mM NaCl, 1 mM EDTA, 2% (w/v) ODG, 5 mM β-mercaptoethanol with protease inhibitor cocktail) or lysis buffer (for PEN2 and ATP6AP1; 20 mM Tris-HCl, pH 7.5, 150 mM NaCl, 1 mM EDTA, 1 mM EGTA, 1% (v/w) Triton X-100, 2.5 mM sodium pyrophosphate, 1 mM β-glycerophosphate, with protease inhibitor cocktail), followed by sonication and centrifugation at 4 °C for 15 min. Cell lysates were incubated with respective antibodies overnight. Overnight protein aggregates were pre-cleared by centrifugation at 20,000$g$ for 10 min, and protein A/G beads (1:250, balanced with ODG buffer or lysis buffer)

were then added into the lysate–antibody mixture for another 3 h at 4 °C. The beads were centrifuged and washed with 100 times volume of ODG buffer or lysis buffer for 3 times (by centrifuging at 2,000$g$) at 4 °C and then mixed with an equal volume of 2× SDS sample buffer and boiled for 10 min before immunoblotting.

For IP of ectopically expressed PEN2 or ATP6AP1, a 6 cm-dish of HEK293T cells was transfected with different expression plasmids. At 24 h after transfection, cells were collected and lysed in 500 μl of ice-cold lysis buffer, followed by sonication and centrifugation at 4 °C for 15 min. Anti-HA (1:100) or anti-Myc (1:100) antibodies, along with protein A/G beads (1:100), or anti-FLAG M2 Affinity Gel (1:200, pre-balanced in lysis buffer) was added into the supernatant and mixed for 4 h at 4 °C. The beads were washed with 200 times volume of lysis buffer for 3 times at 4 °C and then mixed with an equal volume of 2× SDS sample buffer and boiled for 10 min before immunoblotting. Samples for probing APH1A and OCT1 were not boiled to avoid the formation of insoluble aggregates.

To analyse the levels of p-AMPKα and p-ACC in MEFs, HEK293T cells and primary hepatocytes, cells grown to 70–80% confluence (except for hepatocytes, which were grown to 60–70% confluence) in a well of a 6-well dish were lysed with 250 μl of ice-cold lysis buffer. The lysates were then centrifuged at 20,000$g$ for 10 min at 4 °C and an equal volume of 2× SDS sample buffer was added into the supernatant. Samples were then boiled for 10 min and then directly subjected to IB.

To analyse the levels of p-AMPKα and p-ACC in liver, mice were anaesthetized after indicated treatments. Freshly excised (or freeze-clamped) tissue was lysed with ice-cold lysis buffer (10 μl mg$^{-1}$ liver weight) followed by homogenization and centrifugation as described above. The lysates were then mixed with 2× SDS sample buffer and subjected to IB. To analyse the levels of p-AMPKα and p-ACC in nematodes, about 150 nematodes cultured on a NGM plate were collected for each sample. Worms were quickly washed with ice-cold M9 buffer containing Triton X-100, and were lysed with 150 μl of ice-cold lysis buffer. The lysates were then mixed with 5× SDS sample buffer, followed by homogenization and centrifugation as described above, and then subjected to IB. Analysis of PEN2 expression in mouse intestine was performed as previously described[46]. In brief, duodenal segments of intestine were removed immediately after euthanasia, and washed twice with pre-cold PBS containing inhibitor mix (1 mM PMSF, 5 μg ml$^{-1}$ aprotinin, 1 μg ml$^{-1}$ pepstatin A, 2 μg ml$^{-1}$ leupeptin, 1 mM Na$_3$VO$_4$ and 1 mM NaF). The tissues were then homogenized and sonicated in lysis buffer, followed by homogenization and centrifugation as described above. All samples, as described above, were subjected to IB on the same day of preparation, and any freeze–thaw cycle was avoided.

For IB, the SDS–polyacrylamide gels were prepared in house. In brief, the resolving gel solution (8%, 10 ml) was prepared by mixing 1.9 ml of 30% Acryl/Bis solution, 1 ml of 10× lower buffer (3.5 M Tris, 1% (w/v) SDS, pH 8.8), and 0.48 ml of 65% (w/v) sucrose (dissolved in water) with 6.62 ml of water; and the stacking gel solution (4%, 5 ml) was prepared by mixing 668 μl of 30% Acryl/Bis solution and 1.25 ml of 4× stacking buffer (0.5 M Tris, 0.4% (w/v) SDS, pH 6.8) with 3.08 ml water. For each glass gel plate (with 1.0-mm spacer, cat. 1653308 and 1653311, Bio-Rad), approximately 7 ml of resolving gel solution and 2.5 ml of stacking gel solution were required. APS (to 0.1% (w/v) final concentration) and TEMED (to 0.1% (v/v) final concentration) were added to the resolving gel solution. The resolving gel was overlaid with 2 ml of 75% (v/v) ethanol before acrylamide polymerization. After around 20 min (when a clear line between the resolving gel and ethanol is seen), the overlaid ethanol was poured off, dried using filter paper and then placed at room temperature for another 15 min to let the ethanol evaporate completely. The gel cassette was then filled with APS/TEMED-supplemented stacking gel solution, followed by placing a 15-well comb, and then placed at room temperature for 20 min. After removing the comb, the gel was rinsed with running buffer (25 mM Tris, 192 mM glycine, 1% (w/v) SDS, pH 8.3) before sample loading. Samples

of less than 10 μl were loaded into wells, and the electrophoresis was run at 100 V by a Mini-PROTEAN Tetra Electrophoresis Cell (Bio-Rad). In this study, all samples were resolved on 8% resolving gels, except those for PEN2, APH1A–APH1C, PS1, PS2, LAMTOR1, cytochrome $c$ and ATP6v0c, which were on 15% gels (prepared as those for 8%, except that a final concentration of 15% Acryl/Bis was added to the resolving gel solution), and AMPKβ1/2 and ALDOA, which were on 10% gels. To transfer the resolved proteins, the pre-cut PVDF membrane (0.45 μm, cat. IPVH00010, Merck) was incubated in methanol for 1 min, followed by equilibrating and soaking in pre-cooled transfer buffer (25 mM Tris, 192 mM glycine, 10% (v/v) methanol) for more than 5 min. After preparing the gel/membrane sandwich, the transfer was performed using a voltage at 100 V in a Mini Trans-Blot Cell (Bio-Rad) for 1 h at 4 °C. The blotted PVDF membrane was then incubated in blocking buffer (5% (w/v) BSA or 5% (w/v) non-fat milk (according to the instructions from the antibody suppliers) dissolved in TBST, which is composed of 40 mM Tris, 275 μM NaCl, 0.2% (v/v) Tween-20, pH 7.6) for another 2 h on an orbital shaker at room temperature, followed by rinsing with TBST twice, for 5 min each. The PVDF membrane was incubated with the desired primary antibody (diluted in TBST supplemented with 5% BSA and 0.01% (m/v) NaN$_3$) overnight at 4 °C on an orbital shaker with gentle shaking, followed by rinsing with TBST 3 times, 5 min each at room temperature, and then the secondary antibodies were incubated for 3 h at room temperature with gentle shaking. The secondary antibody (diluted in TBST) was then removed, and the PVDF membrane was further washed with TBST 3 times, 5 min each at room temperature. PVDF membranes were incubated in ECL mixture (by mixing equal volumes of ECL solution and peroxide solution for 5 min), then each membrane was placed onto a plastic wrap and laid with medical X-ray film (Fujifilm) in a light-proof cassette for the desired period of time. The films were then developed with X-OMAT MX developer and replenisher and X-OMAT MX fixer and replenisher solutions (Carestream) on a medical X-ray processor (Model 002, Carestream). The developed films were scanned using a Perfection V850 Pro scanner (Epson) using Epson scan software (v.3.9.3.4), and were cropped using Photoshop 2021 software (Adobe). Levels of total proteins and phosphorylated proteins were analysed on separate gels, and representative immunoblots are shown. The band intensities on developed films were quantified using Image J software (v.1.8.0, National Institutes of Health Freeware).

**Quantification of *G6pc1* and *Pck1* mRNA levels by real-time PCR**

Mouse primary hepatocytes cultured in 10-cm dishes were treated with metformin or with glucagon as a control. Total RNA was then prepared by lysing cells with 1 ml of TRIzol (for each 10-cm dish), followed by the addition of 270 μl of chloroform and vigorous mixing. After centrifugation at 12,000$g$ for 15 min at 4 °C, 510 μl of the upper aqueous layer was transferred to a clean tube. The RNA was then precipitated by adding 675 μl of isopropanol followed by centrifugation at 12,000$g$ for 15 min at 4 °C. The pellet was washed with 75% ethanol for 3 times by centrifugation at 12,000$g$ for 5 min, and was dissolved in 30 μl of DEPC-treated water. The concentration of RNA was determined using a NanoDrop 2000 spectrophotometer (Thermo Scientific). A total of 4 μg of RNA was diluted with DEPC-treated water to a final volume of 10 μl at 65 °C for 5 min, and immediately chilled on ice. Random Primer Mix, Enzyme Mix and 5× RT buffer (all from the ReverTra Ace qPCR RT kit) were then added to the RNA solution, followed by incubation at 37 °C for 15 min, and then at 98 °C for 5 min on a thermocycler. The reverse-transcribed cDNA was quantified with Maxima SYBR Green/ROX qPCR master mix on a LightCycler 96 system (Roche) with the following programmes: pre-denaturing at 95 °C for 10 min; denaturing at 95 °C for 10 s, then annealing and extending at 60 °C for 30 s in each cycle; cycle number: 40. The following primer pairs were used for qPCR: mouse *Actin*, 5′-TTTGTGACCACAGCTGAGAGA-3′ and 5′-TGCCCATCAGGCAACTCG-3′; mouse *G6pc1*, 5′-TTGTGGCTTCCTTGGTCCTC-3′ and 5′-CAAAGG

GAACTGTTGCGCTC-3'; mouse *Pck1*, 5'-CTCCTCAGCTGCATAACGGT-3' and 5'-GTGGATA TACTCCGGCTGGC-3'. Data were analysed using Light-Cycler 96 software (v.1.1, Roche).

## Identification of metformin-binding proteins

Lysosomes purified from sixty 10-cm dishes of MEFs (see below, 'Purification of lysosomes' section, for a detailed procedure) were lysed with 500 µl of ice-cold HEPES lysis buffer (50 mM HEPES, pH 8.0, 150 mM NaCl, 1% (v/v) NP40 with protease inhibitor cocktail), followed by sonication. After centrifugation, supernatants were incubated with 10 µM synthesized metformin probes (Met-Ps) at 4 °C for 2 h, then exposed to 365-nm wavelength UV (CX-2000, UVP) for 10 min. Supernatants were then adjusted to final concentrations of 1 mM TCEP, 0.1 mM TBTA, 1 mM $CuSO_4$ and 1 mM biotin-$N_3$ linker, and were incubated at 4 °C for 1 h. Protein aggregates were cleared by centrifugation at 20,000*g* for 15 min, and NeutrAvidin beads (1:100) were then added to the lysates for 2 h with gentle rotation. Beads were then washed with 100× volume of HEPES lysis buffer for 3 times at 4 °C, and then mixed with an equal volume of 2× SDS sample buffer, followed by SDS gel electrophoresis. The gels were stained with staining solution (1% (m/v) Coomassie Brilliant Blue R-250 dye dissolved in 45% (v/v) methanol and 10% (v/v) acetic acid in water) for 30 min, followed by decolouring with staining solution without R-250 dye before subjecting to MS analysis.

For validation of metformin binding by pull down, proteins were prepared from a 10-cm dish of HEK293T cells (grown to 80% confluence) transfected with 10 µg of indicated plasmids per dish for 24 h. Samples were similarly prepared as those using lysosomes, and were subjected to immunoblotting using the indicated antibodies.

For identification of metformin-binding sites on PEN2 by MS, 10 µg of FLAG-tagged PEN2 (expressed and purified from HEK293T cells) was used. Samples were similarly prepared as those using lysosomes except sonication.

## Confocal microscopy

For determining the lysosomal localization of AXIN, cells grown to 80% confluence on coverslips in 6-well dishes were fixed for 20 min with 4% (v/v) formaldehyde in PBS at room temperature. The coverslips were rinsed twice with PBS and permeabilized with 0.1% (v/v) Triton X-100 in PBS for 5 min at 4 °C. After rinsing twice with PBS, the coverslips were incubated with anti-AXIN and anti-LAMP2 antibodies (both at 1:100, diluted in PBS) overnight at 4 °C. The cells were then rinsed 3 times with 1 ml of PBS, and then incubated with secondary antibody for 8 h at room temperature in the dark. Cells were washed for another 4 times with 1 ml of PBS, and then mounted on slides using ProLong Diamond antifade mountant. Confocal microscopy images were taken on a Zeiss laser scanning microscope (LSM) 780 with a ×63, 1.4 NA oil objective.

For detecting the pH of lysosomes, cells were grown on 35-mm glass-bottom dishes, and were cultured to 60–80% confluence. Cells were treated with 1 µM (final concentration) LysoSensor Green DND-189 for 1 h, then washed twice with PBS and incubated in fresh medium for another 30 min. In the meantime, 2 µg ml$^{-1}$ (final concentration) Hoechst, along with ProLong Live antifade reagent, was added into the medium for staining the nucleus before taking images. For determining the mitochondrial membrane potential, cells were incubated with 10 µM JC-1 dye for 20 min at 37 °C, washed twice with PBS and then incubated in fresh medium before taking images. Mitochondrial membrane potentials were analysed by the red (emitted at 590 nm):green (emitted at 529 nm) fluorescence intensity ratio of JC-1 dye (excited at 488 nm). For determining ROS in MEFs, cells were incubated with 5 µM CellROX Deep Red for 30 min at 37 °C, washed twice with PBS and then incubated in fresh medium before taking images. During imaging, live cells were kept at 37 °C, 5% $CO_2$ in a humidified incubation chamber (Zeiss, Incubator PM S1). Images were taken using a Zeiss LSM 780 with a ×63, 1.4 NA oil objective.

For detecting ROS in *C. elegans*, synchronized worms cultured to L4 stage were treated with metformin for 24 h. Approximately 50 worms were soaked in 100 µl of M9 buffer containing 0.05% Triton X-100 and 10 µM CM-H$_2$DCFDA for 30 min at 20 °C, followed by washing with M9 buffer containing 0.05% Triton X-100 three times. Worms were then placed on an injection pad prepared as described in the '*C. elegans* strains' section, except that the 4% (w/v) agarose was used. Images were taken with a Zeiss LSM 900 with a ×20, 0.8 NA plan-Apochromat air objective.

For imaging PEN2, the Semi-intact IF protocol[47] was used. MEFs were grown on a 35 mm dish (cat. D35-20-10-N, In Vitro Scientific) to 50–60% confluence. Cells were rinsed with PBS once, and treated with buffer I (25 mM HEPES, pH 7.2, 125 mM potassium acetate, 5 mM magnesium acetate, 1 mM DTT, 1 mg l$^{-1}$ glucose and 25 µg ml$^{-1}$ digitonin) for 2 min on ice, and then buffer II (25 mM HEPES, pH 7.2, 125 mM potassium acetate, 5 mM magnesium acetate, 1 mM DTT and 1 mg l$^{-1}$ glucose) for another 15 min on ice. The cells were then treated with 4% (v/v) formaldehyde in PBS at room temperature for 10 min. The slides were rinsed twice with PBS and then permeabilized with 0.05% (v/v) Triton X-100 in PBS for 5 min at 4 °C. After rinsing twice with PBS, the slides were blocked in block buffer (10% FBS (v/v) in PBS, with 0.1% (m/v) saponin) for 30 min. The slides were washed twice with PBS and incubated with primary antibodies diluted in block buffer overnight at 4 °C. The cells were then rinsed three times with PBS, and then incubated with a secondary antibody for another 8 h at 4 °C in the dark, followed by washing for four times with PBS and then mounted on slides using ProLong Diamond antifade mountant. Images were taken on a Zeiss LSM 780 with a ×63, 1.4 NA oil objective.

For taking images using the Zeiss LSM 780 system, the following lasers were used: Hoechst dye was excited with a Diode laser set at 405 nm; Lysosensor, JC-1, Alexa 488 and CM-H$_2$DCFDA were visualized with an Ar gas laser (laser module LGK 7812) at 488 nm; Alexa 594 was visualized with a HeNe gas laser (LGK 7512 PF) at 594 nm; and CellROX Deep Red with a HeNe gas laser (LGK 7634) at 633 nm. When images were taken using the Zeiss LSM 900, CM-H$_2$DCFDA was excited with laser module URGB (cat. 400102-9301-000, Toptica) using a 10-mW laser line. The parameters, including 'PMT voltage', 'Offset', 'Pinhole' and 'Gain', were kept unchanged between each picture taken. The resolution of images was 1,024 × 1,024 pixels. Images were processed using Zen 2012 software (for Zeiss LSM 780) or Zen 3.1 (for Zeiss LSM 900), and formatted using Photoshop 2021 software (Adobe). The intensities of CM-H$_2$DCFDA in nematode intestines were quantified using ImageJ software (v.1.8.0, National Institutes of Health Freeware).

## STORM imaging

MEFs stably expressing HA–PEN2 were cultured in a Lab-Tek II chambered no. 1.5 German coverglass system (155409, 8 Chamber, Nunc) to 50% confluence, and were treated following the Semi-intact IF protocol as described above, except that the cells were incubated in rabbit anti-HA tag primary antibody and rat anti-LAMP2 primary antibody, and then with the Atto 488 goat anti-rabbit IgG and Alexa-Fluor 647 donkey anti-mouse IgG secondary antibodies. The slides were then fixed with 4% (v/v) formaldehyde for another 10 min, and washed twice with PBS. The STORM imaging buffer with MEA was then prepared according to the manufacturer's instructions. In brief, 7 µl of GLOX (14 mg of glucose oxidase, 50 µl catalase (17 mg ml$^{-1}$), 200 µl buffer A (10 mM Tris, pH 8.0 and 50 mM NaCl), vortexed to dissolve and cooled on ice) and 70 µl 1 M MEA (77 mg MEA dissolved in 1.0 ml of 0.25 M HCl) were added to 620 µl buffer B (50 mM Tris, pH 8.0, 10 mM NaCl and 10% (m/v) glucose) in a 1.5-ml Eppendorf tube, followed by a brief vortex. The mixture was then added to each well, and images were taken on an N-STORM (Nikon). The imaging was performed using an inverted microscope system (Ti-E Perfect Focus; Nikon) equipped with a monolithic laser combiner (MLC400, Agilent) containing solid-state lasers of wavelengths 405 nm, 488 nm and 561 nm at 50 mW (maximum

fibre output power) and a 647-nm laser at 125 mW. After locating a suitable field, a diffraction-limited TIRF image was acquired for reference, followed by a STORM acquisition. The 647-nm laser was then sequentially fired at 100% power to excite all possible fluorophore molecules and photoswitch them into a non-emitting dark state, and then the 488-nm laser. The emitted wavelengths from Alexa Fluor 647 and Atto 488 fluorophores were then sequentially collected by the plan-Apochromat ×100/1.49 TIRF objective (Nikon), filtered by an emission filter set (Nikon TIRF Cube consisting of a TRF89902-EM filter set, Chroma Technology), and detected on an electron-multiplying charge-coupled device camera (Ixon DU-897, Andor Technology). During imaging, 20,000 sequential frames of each channel were acquired. The image acquisition, lateral drift correction and data processing were performed using NIS Elements software with STORM package (v.4.30 build 1053, Nikon) as previously described[48,49].

## Measurement of oxygen consumption rates

Cells were plated on a 96-well Seahorse XF cell culture microplate (Agilent). For primary hepatocytes, the microplates were pre-coated with collagen. Primary hepatocytes were plated at a density of 1,000 cells per well, whereas MEFs and HEK293T cells were plated at 10,000 cells per well. Cells were incubated in full medium (Williams' medium E containing 1% (w/v) BSA for hepatocytes, and DMEM containing 10% FBS for MEFs and HEK293T cells) overnight before experiments. Medium was then changed to Seahorse XF base medium (Agilent) supplemented with glucose (25 mM for MEFs and HEK293T cells, and 10 mM for hepatocytes), glutamine (4 mM) and pyruvate (1 mM), 1 h before the experiment. Cells were then placed in a $CO_2$-free, XF96 Extracellular Flux Analyzer Prep Station (Agilent) at 37 °C for 1 h. The oxygen consumption rate was then measured at 37 °C in an XF96 Extracellular Flux Analyzer (Agilent), with a Seahorse XFe96 sensor cartridge (Agilent) pre-equilibrated in Seahorse XF Calibrant solution (Seahorse Bioscience, Agilent) in a $CO_2$-free incubator at 37 °C overnight. Respiratory chain inhibitors (10 µM oligomycin, 10 µM FCCP, and a mixture of 1 µM antimycin A and 1 µM rotenone; all final concentrations) were then sequentially added to cells during the assay. Data were collected using Wave 2.6.1 Desktop software (Agilent) and exported to Prism 9 (GraphPad) for further analysis according to the manufacturer's instructions.

## APEX-based TEM imaging

APEX2 imaging was performed as previously described[50], but with minor modifications. In brief, HEK293T cells stably expressing APEX2-tagged proteins were grown on a 3.5-cm dish containing 1 ml of DMEM to approximately 70% confluence. Cells were fixed by gently adding 1 ml of 4% (v/v) glutaraldehyde solution (diluted in 1× phosphate buffer (0.1 M $Na_2HPO_4$:0.1 M $NaH_2PO_4$ = 81:19, in water, pH 7.4), freshly prepared) pre-warmed to 37 °C. Cells were incubated in glutaraldehyde solution for 10 min, then substituted to 1 ml of ice-cold 2% (v/v) glutaraldehyde solution (freshly prepared) on ice for another 1 h, followed by 0.67% (v/v) glutaraldehyde (freshly prepared) solution at 4 °C overnight. Cells were then washed with 1 ml of ice-cold 1× phosphate Buffer for 5 times, 2 min each, followed by the addition of 1 ml of ice-cold 20 mM glycine solution (in 1× phosphate buffer) for 10 min on ice, and were then washed with 1× phosphate buffer for 5 times, 2 min each. Cells were then incubated in 1 ml of freshly prepared 1× DAB solution (0.5 mg ml$^{-1}$) supplemented with 10 mM $H_2O_2$ (all dissolved in 1× phosphate buffer) for 40 min, followed by washing with 1× phosphate buffer for 5 times, 2 min each. Cells were then stained with 2% (w/v) $OsO_4$ solution in 1× phosphate buffer on ice for 30 min, followed by washing for 5 times, 2 min each with ice-cold water. Cells were then stained in ice-cold 2% (w/v) uranyl acetate solution overnight at 4 °C in the dark, and were then washed for 5 times, 2 min each, with ice-cold water. Dehydration was then performed by sequentially incubating cells in the following ice-cold solutions: 20, 50, 70, 90 and 100% (v/v) ethanol (in water), each for 5 min, followed by incubation in

anhydrous ethanol at room temperature for another 5 min. Cells were then quickly submerged ethanol/Spon 812 resin (3:1) mixture at room temperature for a 45-min incubation, and then in ethanol/resin (1:1) mixture at room temperature for 2 h, followed by ethanol/resin (1:3) at room temperature for 2 h, and finally 100% resin at room temperature for two rounds: first round overnight, and next round for 4 h. Resin was then completely drained, and the cells were spread with a thin layer of resin (with a total volume of approximately 400 µl, and below the thickness of 1 mm), followed by baking at 60 °C in a hot-wind drying oven for 48 h. The embedded cells were then sectioned into 75 nm slices after cooling down to room temperature. Images were taken on an electron microscope (Hitachi, HT-7800).

## Purification of lysosomes

Lysosomes were purified using a lysosome isolation kit according to the manufacturer's instructions, but with minor modifications. In brief, MEFs from sixty 10-cm dishes (60–80% confluence), or 200 mg of mouse livers, were collected by directly scrapping at room temperature, followed by centrifugation for 5 min at 500$g$ at 37 °C. Cells were resuspended in 7 ml of 1× extraction buffer containing protease inhibitor cocktail at room temperature, and were dounced in a 7-ml Dounce homogenizer (Sigma, cat. P0610) for 120 strokes on ice followed by centrifugation for 10 min at 1,000$g$, 4 °C, yielding post-nuclear supernatant (PNS). The PNS was then centrifuged for 20 min at 20,000$g$ and the pellet was suspended in 1× extraction buffer by gentle pipetting, generating the crude lysosomal fraction (CLF). The volume of CLF was adjusted to 2.4 ml and then equally divided into six 1.5-ml Eppendorf tubes (400 µl per tube). A volume of 253 µl of OptiPrep and 137 µl of 1× OptiPrep dilution buffer were added to each CLF, and mixed by gentle pipetting. The mixture is defined as the diluted OptiPrep fraction (DOF). Each DOF (0.8 ml) was loaded into an 11 × 60 mm centrifuge tube at the top of 27% (0.4 ml) and 22.5% (0.5 ml) OptiPrep solution cushions, and then overlaid with 16% (1 ml), 12% (0.9 ml) and 8% (0.3 ml) OptiPrep solutions. The tube was then centrifuged on an SW60 Ti rotor (Beckman) at 150,000$g$ for 4 h at 4 °C, and the fraction at the top of 12% OptiPrep solution was collected as the CLF. The fraction was diluted with two volumes of PBS, followed by centrifugation at 20,000$g$ for 20 min. The supernatant was then aspirated, and the sediment was the lysosome fraction.

## Measurement of v-ATPase activity in vitro

For each assay, lysosomes purified from two 10-cm dishes of MEFs were used. ATP hydrolysis activity was measured using a coupled spectrophotometric method as previously described[51], but with some modifications. In brief, lysosomes were suspended in ATPase assay buffer (50 mM NaCl, 30 mM KCl, 20 mM HEPES-NaOH, pH 7.0, 10% (v/v) glycerol, 1 mM $MgCl_2$, 1.5 mM phosphoenolpyruvate, 0.35 mM NADH, 20 U ml$^{-1}$ pyruvate kinase and 10 U ml$^{-1}$ lactate dehydrogenase) with 5 µM concanamycin A (for calculating v-ATPase-specific ATP hydrolysis activity) or DMSO, and pre-warmed at 37 °C for 10 min. The assay was initiated by the addition of 5 mM ATP, and the $OD_{341}$ was continuously recorded by a SpectraMax M5 microplate reader.

ATP-dependent proton transport activity was assessed by measuring the initial velocity (early reaction periods during the assay, before levelling off due to depletion of substrate) of ATP-dependent fluorescent quenching of FITC–dextran, as previously described[52,53]. In brief, lysosomes were loaded with FITC–dextran by incubating cells in DMEM supplemented with 2 mg ml$^{-1}$ FITC–dextran (final concentration) on ice for 5 min, then transferred to a 37 °C incubator for 30 min. Cells were washed with DMEM for 3 times and incubated with DMEM for another 30 min at 37 °C to allow transport of FITC–dextran to lysosomes. Cells were collected and lysosomes were purified as described above. The lysosomes were resuspended in assay buffer (125 mM KCl, 1 mM EDTA, 20 mM HEPES, pH 7.5, with KOH) and were balanced on ice for 1 h, then mixed with 5 µM concanamycin A (for calculating the v-ATPase-specific

proton transport activity) or DMSO, then warmed at 37 °C for 10 min. Fluorescence of FITC was recorded with excitation at 490 nm and emission at 520 nm using a SpectraMax M5 microplate reader. The initial slope of fluorescence quenching was measured after the addition of 5 mM Mg-ATP (final concentration).

### Purification of mitochondria

Mitochondria were purified as previously described[54], but with minor modifications[27]. In brief, forty 10-cm dishes of metformin-treated MEFs (60–80% confluence) were collected by direct scrapping at room temperature, followed by centrifugation for 5 min at 500$g$ at 37 °C. Cells were then resuspended in 20 ml of ice-cold IB$_{cells}$-1 buffer (225 mM mannitol, 75 mM sucrose, 0.1 mM EGTA and 30 mM Tris-HCl, pH 7.4), and dounced for 100 strokes in a 40-ml Dounce homogenizer (Sigma, cat. D9188), followed by two centrifugation rounds of 5 min at 600$g$ at 4 °C. The supernatants were then collected and centrifuged for 10 min at 7,000$g$ at 4 °C. The pellets were then washed twice with 20 ml of ice-cold IB$_{cells}$-2 buffer (225 mM mannitol, 75 mM sucrose and 30 mM Tris-HCl pH 7.4). The suspensions were centrifuged at 7,000$g$, and again at 10,000$g$, both for 10 min at 4 °C. The pellets were then resuspended in 2 ml of ice-cold MRB buffer (250 mM mannitol, 5 mM HEPES pH 7.4 and 0.5 mM EGTA), and were loaded on top of 10 ml of Percoll medium (225 mM mannitol, 25 mM HEPES pH 7.4, 1 mM EGTA and 30% Percoll (v/v)) in 14 × 89-mm centrifuge tubes (cat. 344059, Beckman). The tubes were then centrifuged on a SW41 rotor (Beckman) at 95,000$g$ for 0.5 h at 4 °C, and the dense band located approximately at the bottom of each tube was collected. The collected fractions were diluted with 10 volumes of MRB buffer, followed by centrifugation at 6,300$g$ for 10 min at 4 °C; the pellets were resuspended and washed with 2 ml of MRB buffer, followed with centrifugation at 6,300$g$ for 10 min at 4 °C. The pellets contained pure mitochondria.

### Purification of cytosol

Cytosol was purified as previously described[55]. In brief, ten 10-cm dishes of cells were homogenized in 800 μl of the homogenization buffer (HB) containing 250 mM sucrose, 3 mM imidazole, pH 7.4. Homogenates were then passed through a 22-gauge needle attached to a 1-ml syringe for 6 times, and were then centrifuged at 2,000$g$ for 10 min to yield PNS. PNS samples were then loaded on to the top of 11 × 60-mm centrifuge tubes that had been sequentially loaded with 1 ml of 40.6% sucrose (dissolved in HB), 1 ml of 35% sucrose (dissolved in HB), and 1 ml of 25% sucrose (dissolved in HB). Tubes were then centrifuged on an SW60 Ti rotor (Beckman) at 35,000 r.p.m. for 1 h at 4 °C, and the top fractions (about 200 μl) were collected as cytosolic fraction.

### Protein expression

FLAG-tagged PEN2, as well as Strep-tagged ATP6AP1, was expressed in suspension HEK293T cells. A total of 200 ml cells at $5 × 10^6$ per ml (with viability higher than 90%, as determined by Trypan Blue staining) were transfected with 200 μg of each plasmid dissolved in in 2 ml of SMM 293-TII medium containing 1,500 μg of PEI. After 48 h, cells were collected by centrifugation at 2,000$g$, and then lysed with 100 ml of ice-cold lysis buffer. The lysates were then sonicated and centrifuged at 20,000$g$ for another 30 min at 4 °C. Anti-FLAG M2 Affinity Gel (1:100, balanced in lysis buffer), or streptavidin agarose (1:100, balanced in lysis buffer) was added into the supernatant and mixed for 4 h at 4 °C. The beads were then washed with 200 times volume of lysis buffer for 3 times at 4 °C. Proteins were then eluted with 1 ml of FLAG peptide (400 μg ml$^{-1}$ final concentration) or desthiobiotin (2.5 mM final concentration) for another 1 h at 4 °C. Eluent was further diluted with 15 ml of PBS buffer, then concentrated to 1 ml using an Amicon Ultra-15 centrifugal filter unit with Ultracel-10 regenerated cellulose membrane. Such a process for buffer exchanging was performed for another two times before experiments. Of note, experiments involving proteins expressed and purified described above should be performed on the same day after the purification, and any freeze–thaw cycle should be avoided.

### Differential scanning calorimetry

Differential scanning calorimetry assays were performed on a VP-DSC (GE Healthcare). The VP-DSC was run on a mode without feedback, and 15 min of equilibration at 20 °C was performed before and between each scan. The scanning range was set from 20 to 80 °C, and heating rate at 90 °C h$^{-1}$. The instrument was pre-equilibrated by running for five heating–cooling cycles with both the sample cell and the reference cell loaded with PBS. The sample cell was then loaded with 380 μl of FLAG-tagged PEN2 protein at 1 mg ml$^{-1}$ in 20 μM metformin (in PBS) or PBS, and curves of heat capacity (Cp) versus temperature were recorded. Data were collected using VPViewer 2000 software (v2.66.7, GE Healthcare), and were then corrected for PBS baselines and normalized for scan rate and protein concentration[56] using Origin 7 software (v.7.0552, OriginLab).

### ITC

ITC was performed using a MicroCal iTC200 isothermal titration calorimeter (GE Healthcare), with the sample cell and the reference cell maintained at 25 °C. The instrument was pre-equilibrated before the experiment. A total of 280 μl of FLAG-tagged PEN2 (at 60 μM) and 40 μl of metformin (2 mM stock solution), all in PBS buffer, were loaded into the sample cell and the injector, respectively, and PBS to the reference cell. Metformin was then titrated stepwise (0.4 μl during 0.8 s for the first injection, and 2 μl during 4 s for the rest of 19 injections, all injected at constant velocity, and with an interval of 150 s between each injection) into the PEN2 solution, or PBS as a control. During titration, the sample cell was continuously stirred at 1,000 r.p.m. Data of heat changes acquired during metformin–PEN2 titration were collected using ITC200 software (v.1.26.1, GE Healthcare), and were subtracted by those of metformin–PBS, then analysed using Origin 7 software (v.7.0552, OriginLab), which fits a 'Two Set of Sites' model.

### SPR

Experiments were performed in triplicate at 25 °C on a BIAcore T200 using CM5 sensor chips, and data were analysed using BIAcore T200 Evaluation software (GE Healthcare) following the manufacturer's instruction. In brief, a cell on the CM5 sensor chip was activated with a mixture of 200 μM 1-ethyl-3-(3-dimethylaminopropyl)carbodiimide (EDC) and 50 μM N-hydroxysuccinimide (NHS) at 10 μl min$^{-1}$ for 10 min. A total of 20 μl of FLAG-tagged PEN2 or PEN2-2A protein (1 mg ml$^{-1}$) purified as described above and adjusted to pH 4.0 (by mixing with 180 μl of 10 mM sodium acetate solution, pH 4.0) was then immobilized on the surface of the cell at 10 μl min$^{-1}$ for 5 min for two repetitive runs. The cell was then blocked with 1 M ethanolamine (10 μl min$^{-1}$ for 10 min). A neighbouring cell that served as a reference was similarly activated and blocked, except that PBS adjusted to pH 4.0 was used for immobilization. Both of the cells were then equilibrated with PBS. Metformin stock solution (2 mM) was diluted to a series of concentrations (6.25, 3.125, 1.5625, 0.78125 and 0.05 μM (all in PBS)), and was flowed at 30 μl min$^{-1}$ for 150 s in each run. At the end of each flow, cells were regenerated for 5 min with 10 mM glycine-HCl (pH 2.0) solution at 10 μl min$^{-1}$. Data from the sample cell were collected using BIAcore T200 Control software (v. 2.0, GE Healthcare), and were subtracted by those from the reference cell. Association and dissociation constants were obtained by global fitting of the data to a 1:1 Langmuir binding model using BIAcore T200 Evaluation software (v.2.0, GE Healthcare). Data were exported to Origin 7 software (v.7.0552, OriginLab) for generating the final figures.

### Synthesis of metformin probes

Compounds were purified using a preparative HPLC (Sail 1000, Welch Materials) equipped with an XB-C18 column (30 × 250 mm, 5 μm, Welch Materials). Mass spectra for compound characterization were collected

using an Autopurification LC Prep system equipped with an ACQUITY QDa detector (Waters) with the ESI mode. High-resolution mass spectra were obtained on a Q Exactive Orbitrap mass spectrometer (Thermo Scientific). NMR spectra were measured on an Avance III 600 MHz NMR spectrometer (Bruker) using tetramethylsilane as the internal standard, and the chemical shift was reported in δ (ppm), multiplicities (s = singlet, d = doublet, t = triplet, q = quartet, p = pentet, m = multiplet and br = broad), integration and coupling constants ($J$ in Hz). $^1$H and $^{13}$C chemical shifts are relative to the solvent: $δ_H$ 2.50 and $δ_C$ 39.5 for DMSO-$d_6$.

The biotin-$N_3$ linker was synthesized as previously described[57], and photoactive metformin probes as previously described[58]. In brief, metformin (converted from its hydrochloride form as previously described[59]; 50 mg, 0.387 mM, 1.0 eq), 3-(but-3-yn-1-yl)-3-(2-iodoethyl)-3$H$-diazirine (189.8 mg, 0.120 ml, 0.765 mM, 1.98 eq) and anhydrous acetone (1.0 ml) were stirred at 23 °C for 18 h. The mixture was then evaporated under reduced pressure at room temperature, followed by purification on a preparative HPLC. Mobile phase buffer A was $H_2O$, and mobile phase buffer B was methanol. The gradients were as follows: $t$ = 0 min, 0% B; $t$ = 3 min, 0% B; $t$ = 18 min, 100% B; $t$ = 30 min, 100% B with a constant flow rate at 20 ml min$^{-1}$. Type I (eluted at 80% phase B, referred to as Met-P1) and type II (eluted at 93% phase B, Met-P2) forms of metformin probes were obtained as a white solid with yields of 12.7% (18.5 mg) and 3.2% (4.7 mg), respectively.

Type I (Met-P1), $^1$H-NMR (600 MHz, DMSO-$d_6$): δ 7.52 (t, $J$ = 5.4 Hz, 1H), 6.57 (s, 4H), 2.97 (dt, $J$ = 7.5, 5.4 Hz, 2H), 2.90 (s, 6H), 2.85 (t, $J$ = 2.7 Hz, 1H), 2.00 (dt, $J$ = 7.3, 2.7 Hz, 2H), 1.62 (t, $J$ = 7.4 Hz, 2H), 1.58 (t, $J$ = 7.3 Hz, 2H); $^{13}$C-NMR (150 MHz, DMSO-$d_6$): δ 159.45, 156.58, 83.14, 71.90, 37.95, 37.45, 32.18, 31.25, 27.05, 12.71; HRMS ($m/z$): [M-I]$^+$ calculated for $C_{11}H_{20}N_7{}^+$, 250.1775; found, 250.1772.

Type II (Met-P2), $^1$H-NMR (600 MHz, DMSO-$d_6$): δ 7.22 (s, 2H), 6.96–6.26 (br m, 3H), 2.97-2.93 (m, 2H), 2.92 (s, 6H), 2.85 (t, $J$ = 2.7 Hz, 1H), 2.00 (dt, $J$ = 7.4, 2.7 Hz, 2H), 1.60 (dt, $J$ = 7.4, 2.2 Hz, 4H); $^{13}$C-NMR (150 MHz, DMSO-$d_6$): δ 154.20, 109.53, 83.14, 71.82, 37.48, 31.99, 31.41, 27.13, 12.70; HRMS ($m/z$): [M-I]$^+$ calculated for $C_{11}H_{20}N_7{}^+$, 250.1775; found, 250.1770.

## Protein and peptide MS

To identify metformin-interacting proteins, the pulled down, biotinylated proteins were subjected to SDS–PAGE. After staining with Coomassie Brilliant Blue R-250 dye, gels were decoloured and the excised gel segments were subjected to in-gel trypsin digestion and dried. Samples were analysed on a nanoElute (Bruker) coupled to a timsTOF Pro (Bruker) equipped with a CaptiveSpray source, or a NanoLC 425 System (SCIEX) coupled to a TripleTOF 5600+ mass spectrometer (SCIEX). Peptides were dissolved in 10 µl 0.1% formic acid (v/v) and were loaded onto a homemade C18 column (35 cm × 75 µm, ID of 1.9 µm, 100 Å). Samples were then eluted for 60 min with linear gradients of 3–35% acetonitrile (v/v, in 0.1% formic acid) at a flow rate of 0.3 µl min$^{-1}$. MS data were acquired with a timsTOF Pro mass spectrometer (Bruker) operated in PASEF mode, and were analysed using Peaks Studio software (X$^+$, Bioinformatics Solutions), or a TripleTOF 5600+ mass spectrometer, and were analysed using ProteinPilot software (v.5.0, SCIEX). The mouse UniProt Reference Proteome database was used during data analysis.

To determine the PEN2-interacting proteins, the HA-tagged PEN2 immunoprecipitants (immunoprecipitated from twenty 10-cm dishes of MEFs stably expressing HA-tagged PEN2) were subjected to SDS–PAGE, and were processed as described above. Data acquisition was performed as described above, except that an EASY-nLC 1200 System (Thermo Scientific) coupled to an Orbitrap Fusion Lumos Tribrid spectrometer (Thermo Scientific) equipped with an EASY-Spray Nanosource was used. Data were analysed using Proteome Discoverer (v.2.2, Thermo Scientific) against the mouse UniProt Reference Proteome database.

For identifying metformin-binding sites on PEN2, the Met-P1-conjugated FLAG-tagged PEN2 (purified from suspension HEK293T cells) were subjected to SDS–PAGE, and were processed as described above. MS analysis was performed as that for timsTOF Pro, except that the following parameters of Peaks Studio X$^+$ software were set: (1) precursor ion mass tolerance: 15 ppm; (2) fragment ion mass tolerance (error tolerance): 0.05 Da; (3) tryptic enzyme specificity with two missed cleavages: allowed; (4) mode of monoisotopic precursor mass and fragment ion mass was chosen; (5) mode of a fixed modification of cysteine carbamidomethylation was chosen; and (6) variable modifications including $N$-acetylation of proteins, oxidation@M and 222.17@X.

## Measurement of adenylates

ATP, ADP and AMP from cells or tissues were analysed by capillary electrophoresis-based MS as previously described[6]. In brief, each measurement required cells collected from a 10-cm dish (60–70% confluence) or 100 mg of liver tissue dissected by freeze clamp. For analysis of metabolites, cells were rinsed with 20 ml of 5% (m/v) mannitol solution (dissolved in water) and instantly frozen in liquid nitrogen. Cells were then lysed with 1 ml of methanol containing Internal Standards 1 (IS1 (Human Metabolome Technologies, H3304-1002, 1:200), used to standardize the metabolite intensity and to adjust the migration time), and were scraped from the dish. For analysis of metabolites in liver, mice were anaesthetized after indicated treatments. The tissue was excised by freeze-clamping, then ground in 1 ml of methanol with 50 µM IS1. The lysate was then mixed with 1 ml of chloroform and 400 µl of water by 20 s of vortexing. After centrifugation at 15,000$g$ for 15 min at 4 °C, 450 µl of aqueous phase was collected and was then filtrated through a 5-kDa cut-off filter (Millipore, cat. UFC3LCCNB-HMT) by centrifuging at 12,000$g$ for 3 h at 4 °C. In parallel, quality control (QC) samples were prepared by combining 100 µl of the aqueous phase from each sample and then similarly filtered. The filtered aqueous phase was then freeze-dried in a vacuum concentrator (Labconco, CentriVap Benchtop centrifugal vacuum concentrator, equipped with a CentriVap −84 °C Cold Trap and a Scroll vacuum pump) at 4 °C, and then dissolved in 100 µl of water containing Internal Standards 3 (IS3 (Human Metabolome Technologies, H3304-1104, 1:200), to adjust the migration time). A total of 20 µl of redissolved solution was then loaded into an injection vial with a conical insert for CE-QTOF MS (Agilent Technologies 7100, equipped with 6545 mass spectrometer) analysis. Data were collected using MassHunter LC/MS acquisition 10.1.48 (Agilent), and were processed using Qualitative Analysis B.06.00 (Agilent). Levels of AMP, ADP and ATP were measured using full scan mode with $m/z$ values of 346.0558, 426.0221, and 505.9885, respectively. Note that a portion of ADP and ATP could lose one phosphate group during in-source fragmentation, thus leaving the same $m/z$ ratios as AMP and ADP, and should be corrected according to their different retention times in the capillary. Therefore, the total amount of ADP is the sum of the latter peak of the $m/z$ 346.0558 spectrogramme and the former peak of the $m/z$ 426.0221 spectrogramme, and the same is applied for ATP.

To analyse ATP, ADP and AMP in nematodes, HPLC–MS was performed. In brief, 150 nematodes maintained on NGM or siRNA plates (with or without 50 mM metformin) for 48 h were washed with ice-cold M9 buffer containing Triton X-100. Bacteria were removed by quickly spinning down the slurry at 100$g$ for 5 s. Nematodes were then instantly lysed in 1 ml of methanol, then mixed with 1 ml of chloroform and 400 µl of water (containing 4 µg ml$^{-1}$ [U-$^{13}$C]-glutamine), followed by 20 s of vortexing. After centrifugation at 15,000$g$ for another 15 min at 4 °C, 800 µl of aqueous phase was collected, lyophilized in a vacuum concentrator at 4 °C, and then dissolved in 30 µl of 50% (v/v, in water) acetonitrile. Measurement of AMP and ATP level was based on ref. [60] using a QTRAP MS (SCIEX, QTRAP 5500) interfaced with a UPLC system (SCIEX, ExionLC AD). A total of 2 µl of each sample was loaded onto a HILIC column (ZIC-pHILIC, 5 µm, 2.1 × 100 mm, PN: 1.50462.0001, Millipore). The mobile phase consisted of 15 mM ammonium acetate containing 3 ml l$^{-1}$ ammonium hydroxide (>28%, v/v) in the LC-MS-grade water (mobile phase A) and LC–MS-grade 90% (v/v) acetonitrile in LC-MS-grade water

(mobile phase B) run at a flow rate of 0.2 ml min⁻¹. AMP, ADP and ATP were separated with the following HPLC gradient elution programme: 95% B held for 2 min, then to 45% B in 13 min, held for 3 min, and then back to 95% B for 4 min. The mass spectrometer was run on a Turbo V ion source in negative mode with a spray voltage of −4,500 V, source temperature at 550 °C, gas no.1 at 50 psi, gas no.2 at 55 psi, and curtain gas at 40 psi. Metabolites were measured using the multiple reactions monitoring mode, and declustering potentials and collision energies were optimized using analytical standards. The following transitions were used for monitoring each compound: 505.9/158.9 and 505.9/408.0 for ATP; 425.9/133.9, 425.9/158.8 and 425.9/328.0 for ADP; 345.9/79.9, 345.9/96.9 and 345.9/133.9 for AMP; and 149.9/114 for [U-¹³C]-glutamine. Data were collected using Analyst 1.7.1 software (SCIEX), and the relative amounts of metabolites were analysed using MultiQuant 3.0.3 software (SCIEX). Note that a portion of ADP and ATP could lose one or two phosphate groups during in-source fragmentation, thus leaving same $m/z$ ratios as AMP and ADP, which was corrected according to their different retention times in the column.

For quantification of AMP, ADP and ATP in cells, tissues or nematodes, [U-¹³C, ¹⁵N]AMP, [U-¹³C, ¹⁵N]ADP and [U-¹³C, ¹⁵N]ATP dissolved in individual lysates were used to generate corresponding standard curves by plotting the ratios of detected labelled AMP, ADP or ATP (areas) to the products of IS1 and IS3 (for CE-MS), or [U-¹³C]-glutamine (for HPLC−MS), against the added concentrations of labelled AMP, ADP or ATP. The amounts of AMP, ADP and ATP were estimated according to standard curves, and were then divided by protein wet weight. The protein wet weight of each sample was determined by Bradford assay after dissolving the naturally dried protein sediment with 0.2 M KOH at room temperature.

## Determination of metformin concentration

To measure metformin concentrations in serum, 50 μl of serum collected from each mouse or human participant was mixed with 80% methanol (v/v) in water using buformin at 100 μg l⁻¹ as an internal standard. The aqueous phase was then collected after centrifugation at 15,000g for 15 min at 4 °C. Cells, liver tissues or intestinal tissues were prepared as in CE-MS measurement of adenylates, except that liver and intestinal tissues (50 mg each) were collected from anaesthetized, blood-drained mice, and no ultrafiltration was required. In addition, cells or liver tissues were rinsed with PBS before homogenization. Nematode samples were prepared as in for the HPLC−MS measurement of adenylates. Measurement was performed on a QTRAP MS (SCIEX, QTRAP 6500+) connected to a UPLC system (SCIEX, ExionLC AD). A total of 2 μl of each sample was loaded onto a pHILIC column (ZIC-pHILIC, 5 μm, 2.1 × 100 mm, PN: 1.50462.0001, Millipore). The mobile phase consisted of 10 mM ammonium formate containing 0.1% formate (v/v) in the LC−MS-grade water (mobile phase A) and LC−MS-grade acetonitrile containing 0.1% (v/v) formate in LC−MS-grade water (mobile phase B) run at a flow rate of 0.3 ml min⁻¹. The HPLC gradient was as follows: 95% B held for 1 min, then to 40% B in 6 min, held for 1 min, then to 95% B within 7.5 min, and held for 2.5 min. The QTRAP mass spectrometer was run on a Turbo V ion source and running in negative mode run in a spray voltage of −5,500 V, with source temperature at 500 °C, gas no.1 at 50 psi, gas no.2 at 55 psi, and curtain gas at 40 psi. Compounds were measured using the multiple reactions monitoring mode, and declustering potentials and collision energies were optimized using analytical standards. The following transitions were used for monitoring each compound: 130/71 for metformin and 158.1/60 for buformin. A standard curve was generated in each experiment for quantification. Data were collected using Analyst 1.6.3 software (SCIEX), and the relative amounts of metabolites were analysed using MultiQuant 3.0.2 software (SCIEX). The average cell volume of MEFs was estimated to be 2,263 μm³, HEK293T cells 4,240 μm³ and primary hepatocytes 17,062 μm³ using Imaris 7.4.0 software (Bitplane) from the axial image stacks of CDFA-SE labelled cells taken using a Zeiss LSM780.

## Determination of TAG synthesis

TAG synthesis rates were determined by analysing the contents of labelled TAG in cells treated with [U-¹³C] glucose or [U-¹³C] palmitic acid (PA). Glucose was dissolved in PBS, and PA was conjugated to BSA before use. To conjugate PA, 200 mg of PA was first dissolved in 20 ml of ethanol in a conical flask by stirring, followed by dropwise mixing with 156 μl of 5 M NaOH. The slurry was constantly stirred for 12 h, which leads to a complete evaporation of ethanol. The dried sediment was then dissolved with 10% fatty-acid-free BSA to a final concentration of 2 mM.

Primary hepatocytes were isolated and cultured in DMEM containing 1% BSA for 12 h before the experiment. Cells were then incubated in glucose-free DMEM supplemented with 25 mM [U-¹³C] glucose and 1% BSA, or DMEM containing 100 μM [U-¹³C] PA and 1% BSA for another 12 h. Cells on a 10-cm dish were rinsed with 25 ml of PBS twice, and instantly frozen in liquid nitrogen. Cells were then lysed with 1 ml of methanol containing TAG (15:0/15:0/15:0) as an internal standard, and were scraped from the dish. The lysate was then quickly mixed with 1 ml of chloroform and 400 μl of water by 20 s of vortexing. After centrifugation at 15,000g for 15 min at 4 °C, 700 μl of organic phase was collected, followed by lyophilization with nitrogen blow on a pressured gas blowing concentrator (MGS-2200, EYELA) at room temperature. Analysis of TAG was performed on a Shimadzu Prominence UPLC system (Nexera UHPLC LC-30A) interfaced with a TripleTOF 5600+ system (SCIEX) equipped with an ESI source. Lyophilized samples were dissolved in 20 μl of dichloromethane/methanol solution (2/1, v/v), and was diluted with 380 μl of methanol/isopropanol/H₂O solution (65/30/5, v/v/v). The injection volume was 5 μl. TAGs were separated through a C8 column (2.1 × 100 mm with 1.7 μm particle size, cat. 186002878, Waters) with column temperature maintained at 55 °C. Mobile phases consisted of 10 mM ammonium formate in acetonitrile/H₂O (60/40, v/v) (mobile phase A) and 10 mM ammonium formate in isopropanol/acetonitrile (90/10, v/v) (mobile phase B) and was run at a flow rate of 0.26 ml min⁻¹. The gradient was as follows: 32% B for 1.5 min, then increased to 97% B within 19.5 min and held for 4 min, then back to 32% B and held for another 5 min. The flow rate for mobile phases was set at 0.26 ml min⁻¹. The mass spectrometer was run in positive, information-dependent acquisition (IDA) mode, with the source temperature of 550 °C, the ion source gas 1 and 2 at 55 psi, the curtain gas at 35 psi, the collision energy at 40 eV, the ion spray voltage floating at 5.5 kV, and the mass range at 500−1,250 $m/z$. The accumulation time for full scan was set at 150 ms, and the accumulation time for each IDA scan was 45 ms. Peaks of metabolites with intensities larger than 100 c.p.s. after adding up the signal from 10 rounds of IDA scans were chosen for further analysis. Data were collected using Analyst TF 1.6 software (SCIEX), and were analysed using MS-DIAL 4.7 software (RIKEN), through which the deconvolution and streamline criteria were used for peak/TAG identification.

## Determination of β-oxidation rates

The rates of β-oxidation were determined through the labelled intermediates of the TCA cycle in cells treated with [U-¹³C] PA for a certain time duration. Primary hepatocytes were cultured and treated as those used for the determination of TAG synthesis. Cells were rinsed with PBS twice, froze in liquid nitrogen and then lysed with 1 ml of 80% methanol (v/v) in water containing 10 μg ml⁻¹ myristic-d27 acid as an internal standard, followed by 20 s of vortexing. After centrifugation at 15,000g for 1 min at 4 °C, 600 μl of supernatant (aqueous phase) was freeze-dried at 4 °C. The lyophilized sample was then vortexed for 1 min after mixing with 50 μl of freshly prepared methoxyamine hydrochloride (20 mg ml⁻¹ in pyridine), followed by incubating at 4 °C for 1 h. The mixture was sonicated at 0 °C by bathing in an ice slurry for 10 min, and was then incubated at 37 °C for 1.5 h, followed by mixing with 50 μl of MTBSTFA and incubated at 55 °C for 1 h. Before subjecting to GC−MS, samples were centrifuged at 15,000g for 10 min, and 60 μl

of supernatant was loaded into an injection vial. GC was performed on a HP-5MS column (30 m × 0.25 mm i.d., 0.25 μm film thickness) using a GC/MSD instrument (7890-5977B, Agilent). The injector temperature was 260 °C. The column oven temperature was first held at 70 °C for 2 min, then increased to 180 °C at the rate of 7 °C min$^{-1}$, then to 250 °C at the rate of 5 °C min$^{-1}$, then to 310 °C at the rate of 25 °C min$^{-1}$, where it was held for 15 min. The MSD transfer temperature was 280 °C. The MS quadrupole and source temperature were maintained at 150 °C and 230 °C, respectively. Data were collected using MassHunter GC/MS Acquisition software (B.07.04.2260, Agilent), and were analysed using GC-MS MassHunter Workstation Qualitative Analysis software (v.B.07.01SP1, Agilent).

## Statistical and reproducibility

Statistical analyses were performed using Prism 9 (GraphPad software), except for the survival curves, which were analysed using SPSS 27.0 (IBM). Each group of data was subjected to Kolmogorov–Smirnov test, Anderson–Darling test, D'Agostino–Pearson omnibus test or Shapiro–Wilk test for normal distribution where applicable. Unpaired two-tailed Student's $t$-test was used to determine significance between two groups of normally distributed data. Welch's correction was used for groups with unequal variances. Unpaired two-tailed Mann–Whitney test was used to determine significance between data without a normal distribution. For comparisons between multiple groups, an ordinary one-way or two-way analysis of variance (ANOVA) was used, followed by Tukey's, Sidak's, Dunnett's or Dunn's test as specified in the figure legends. The assumptions of homogeneity of error variances were tested using $F$-test ($P > 0.05$). For comparison between multiple groups with two fixed factors, an ordinary two-way ANOVA or two-way repeated measures ANOVA (for GTT, ITT and PTT data) was used, followed by Tukey's or Sidak's multiple comparisons test as specified in the legends. Geisser–Greenhouse's correction was used where applicable. The adjusted means and s.e.m. or s.d. were recorded when the analysis met the above standards. Differences were considered significant when $P < 0.05$, or $P > 0.05$ with large differences of observed effects (as suggested in refs. [61,62]). All specific statistical details can be found in the figure captions and source data. All images shown without biological replicates are representative of a minimum of three independent experiments.

## Reporting summary

Further information on research design is available in the Nature Research Reporting Summary linked to this paper.

## Data availability

The data supporting the findings of this study are available within the paper and its Supplementary Information files. The MS proteomics data have been deposited to the ProteomeXchange Consortium (http://proteomecentral.proteomexchange.org) through the iProX partner repository[63] with the dataset identifier PXD030090. Materials, reagents or other experimental data are available upon request. Full immunoblots are provided as Supplementary Information Fig. 1. Source data are provided with this paper.

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

**Acknowledgements** We thank D. Julius for providing *Trpv*$^{-/-}$ mice (The Jackson Laboratory, 003770); B. De Strooper for the *Aph1a*$^{F/F}$*Aph1b*$^{-/-}$*Aph1c*$^{F/F}$ mice (The Jackson Laboratory, 030985); N. Mizushima for the *Atg5*$^{F/F}$ mice (RIKEN, RBRC02975); B. Viollet for the *Ampka1/2*-DKO MEFs; S. Morrison for the *Ampka1*$^{F/F}$ (The Jackson Laboratory, 014141) and *Ampka2*$^{F/F}$ mice (The Jackson Laboratory, 014142); J. Taylor for providing the pLiv-Le6 vector; Z. Wang for providing the suspension HEK293T cell line; the *Caenorhabditis* Genetics Center for supplying nematode strains; S.-Q. Wu for mouse in vitro fertilization; all the other members of the S.-C.L. laboratory for technical assistance; G. Hardie for critical reading and comments; Z.-X. Wang for instructions on ITC assays; and M. Hu for the discussions and suggestions on electrophysiological properties of lysosomes. The artworks shown in Fig. 2a and Extended Data Fig. 13m were modified from elements created by Servier Medical Art (https://smart.servier.com/) licenced under a Creative Commons Attribution 3.0 Unported Licence (https://creativecommons.org/licenses/by/3.0/). This work was supported by grants from the National Natural Science Foundation of China (82088102, 91854208, 31922034, 32070753, 31730058, 22025702 and 91853203), the Natural Science Foundation of Fujian Province of China (2020J02003), the Fundamental Research Funds for the Central Universities (20720200069, 20720200014 and 20720190101), the Programme of Introducing Talents of Discipline to Universities (BP2018017), and the XMU Training Programme of Innovation and Entrepreneurship for Undergraduates (202010384190, 2020X901 and 2020X895).

**Author contributions** T.M., X.T., B.Z., X.D., C.-S.Z. and S.-C.L. conceived the study and designed the experiments. T.M. and X.T. identified lysosomal metformin-binding proteins and the

PEN2-interacting proteins, generated the cell strains and performed the IP, confocal imaging acquisition, in vitro reconstitution and the associated western blot analyses (with assistance from M.L., X.W., Q.Q., S.L., J.-W.F. and C.L.). B.Z. and X.D. designed, synthesized and purified the photoactive metformin probe and performed in silico modelling assays. Y. Wang and Yaxin Yu performed the nematode experiments (under the guidance of Yong Yu), and T.M. and X.T. performed the mouse experiments. M.L. performed STORM imaging acquisition, and X.W. and L.Y. performed APEX-based TEM imaging acquisition. C.Y. and J.C. determined the binding affinity of metformin to PEN2. J.W. generated the *Pen2*-floxed and *Atp6ap1*-floxed mouse strains. C.Z. analysed the pharmacokinetics of metformin and performed the HPLC–MS-based analysis of adenylates. X.T., Z.-C.W. and W.W. performed the CE–MS-based analysis of adenylates (under the guidance of H.-L.P.). C.X., Y. Wu and Z.X. performed protein-related mass spectrometry analysis. X.H. generated the antibody for immunoprecipitating endogenous ATP6AP1. Y.H. generated the liver-specific Tg-ALDOA and Tg-ALDOA-D34S mouse strains. L.Z. analysed the rates of TAG synthesis by HPLC–MS, and M. Zhu analysed the rates of β-oxidation by GC–MS. M. Zhang, Y.B. and W.J. enrolled human participants and analysed metformin concentrations in human serum. S.-Y.L. and Z.Y. helped supervise the project. C.-S.Z. and S.-C.L. wrote the manuscript.

**Competing interests** The authors declare no competing interests.

**Additional information**

**Correspondence and requests for materials** should be addressed to Xianming Deng, Chen-Song Zhang or Sheng-Cai Lin.

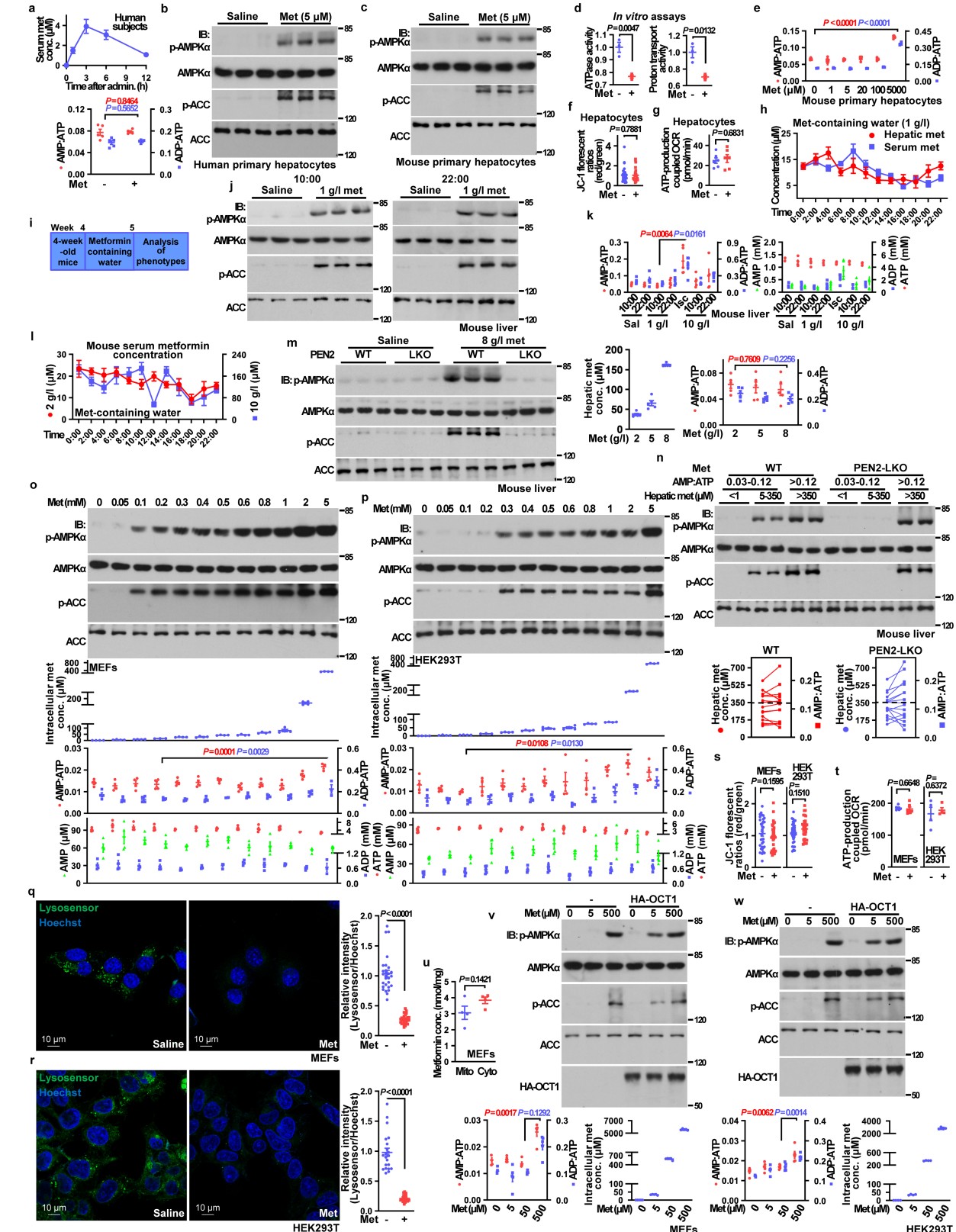

**Extended Data Fig. 1** | See next page for caption.

**Extended Data Fig. 1 | Low metformin can activate AMPK without altering energy levels. a**, Serum metformin concentrations in human subjects. Serum samples were collected at indicated time points from subjects after taking 0.5 g of Metformin Hydrochloride Extended-release Tablets. Data are shown as mean ± s.e.m.; $n$ = 6. **b, c, e**, Low metformin activates AMPK in primary hepatocytes without elevating AMP. Human (**b**) or mouse (**c, e**) primary hepatocytes were treated with metformin (Met), or PBS (Saline) for 2 h, and the levels of p-AMPKα and p-ACC (**b, c**), as well as the AMP:ATP and ADP:ATP ratios (**b, e**) were determined [shown as mean ± s.e.m.; $n$ = 5 (**b**) or 4 (**e**) cells for each condition, and $P$ value by two-sided Student's $t$-test (**b**), or one-way ANOVA followed by Sidak (**e**)]. **d**, Metformin inhibits v-ATPase in purified lysosomes. Lysosomes purified from mouse livers were incubated with 5 µM metformin for 1 h. The activity of v-ATPase was determined by the rates to hydrolyse ATP (left panel) and to transport protons (right panel). Data are shown as mean ± s.e.m.; $n$ = 3; $P$ value by two-sided Student's $t$-test. **f, s**, Low metformin does not affect mitochondrial membrane potential. Mouse primary hepatocytes (**f**), MEFs (**s**, left panel), or HEK293T cells (**s**, right panel) were treated with 5 µM metformin for 2 h (**f**), 200 µM metformin for 12 h (**s**, left panel) or 300 µM metformin for 12 h (**s**, right panel) (higher concentrations and longer treatment times were used in MEFs and HEK293T cells because of lack of OCTs), and were loaded with JC-1 dye for another 30 min. After normalisation to the group without metformin treatment, the data are shown as mean ± s.e.m.; $n$ = 37 (control) and 34 (metformin-treated) with mouse primary hepatocytes, $n$ = 36 (control) and 40 (metformin-treated) for MEFs, and $n$ = 35 (control) and 36 (metformin-treated) with HEK293T cells; and $P$ value by two-sided Mann-Whitney (**f** and **s**, left panel) or two-sided Student's $t$-test (**s**, right panel). **g, t**, Low metformin does not affect mitochondrial respiration. Mouse primary hepatocytes (**g**, approximately 3,000 cells in total), MEFs (**t**, left panel, approximately 10,000 cells in total), or HEK293T cells (**t**, right panel, approximately 10,000 cells in total) were treated as in **f**, left panel of **s**, and right panel of **s**. ATP production-coupled OCR was determined by subtracting basal OCR from that treated with 10 µM oligomycin. Data are mean ± s.e.m.; $n$ = 6 with hepatocytes and MEFs, and $n$ = 4 with HEK293T cells; and $P$ value by two-sided Student's $t$-test. **h, i, l**, Mice taking 1 g/l metformin from drinking water resembles the situation of human patients taking standard clinical doses of metformin. As depicted in **i**, mice at 4-week old were treated with metformin in drinking water for 7 days. At day 8, mice were sacrificed at indicated times of the day. The mice were then divided into two groups, one for sacrifice to collect serum, and the others for the liver tissue. Results are mean ± s.e.m.; $n$ = 5 for each time point, except $n$ = 4 for the 2 g/l group at 0:00, 4:00 and 18:00. Note that perhaps owing to the bitterness of metformin at higher doses (10 g/l), some of the mice showed a decreased water intake (hence metformin), and larger variations of the serum metformin concentrations than those of 1 g/l and 2 g/l were observed. **j, k, m, n**, High doses of metformin leads to increased AMP/ADP levels, and bypasses the requirement of PEN2 for AMPK activation. Mice were treated as in **i**, followed by analysis of p-AMPKα and p-ACC (**j, m, n**) and hepatic AMP:ATP and ADP:ATP ratios, the absolute concentrations of AMP, ADP and ATP, and the hepatic metformin concentrations. Results are mean ± s.e.m.; $n$ = 5 (**k, m**) and $n$ = 16 (**n**) for each treatment, and $P$ value by one-way ANOVA followed by Dunn (**k**) or Tukey (**m**). Isc; hepatic ischemia (for 5 sec). Note that in **m, n**, readouts were determined in the liver from the mice that did not undergo the step of blood draining (different from **h**), because ischemia will increase AMP and ADP, and will cause AMPK activation unrelated to the lysosomal pathway[27]. The legitimacy for skipping the step of blood draining was based on the observation that hepatic metformin concentration is similar to that in the serum in our animal setting, as shown in **h** - the residual blood would not significantly interfere with the readout of the hepatic metformin concentration. **o, p, v, w**, AMPK can be activated in MEFs and HEK293T cells in AMP/ADP-independent manner in low metformin. MEFs (**o**), HEK293T cells (**p**), and the OCT1-expressing MEFs (**v**) and HEK293T cells (**w**) were treated with metformin at indicated concentrations for 12 h (**o, p**) or 2 h (**v, w**), followed by analysis of intracellular metformin concentrations [shown as mean ± s.e.m.; $n$ = 4 (for each metformin concentration in **o**, **p** and **w**, except $n$ = 3 for the 0.2 mM metformin in **o** and **p**) or 5 (**v**)], p-AMPKα and p-ACC, and AMP:ATP, and ADP:ATP ratios [shown as mean ± s.e.m.; $n$ = 4 (for each metformin concentration in **o** and **p**, except $n$ = 3 for the ratios at 5 mM metformin in **o**) or 5 (**v, w**); and $P$ values by one-way ANOVA, followed by Sidak (**o**, and AMP:ATP of **p**), Tukey (ADP:ATP of **p**, and **w**), or Dunn (**v**)], as well as the absolute concentrations of AMP, ADP and ATP. **q, r**, Low metformin deacidifies lysosomes. MEFs (**q**) and HEK293T cells (**r**) pre-labelled with LysoSensor Green DND-189 and Hoechst were treated as in **s**. Representative images are shown (left panel); the relative fluorescent intensities of Lysosensor (normalised to the intensity of Hoechst) are shown on the right. Results are mean ± s.e.m.; $n$ = 28 (control) and 27 (metformin-treated) from 3 dishes/experiments for MEFs, and $n$ = 21 (control) and 20 (metformin-treated) from 3 dishes/experiments for HEK293T cells; and $P$ value by two-sided Mann-Whitney test. **u**, Metformin is not accumulated in mitochondria. MEFs were treated as in **s**, and metformin concentrations in mitochondria and cytosol fractions (normalised to protein concentration) are shown as mean ± s.e.m.; $n$ = 4, and $P$ value by two-sided Student's $t$-test. Experiments in this figure were performed three times, except **c, h, j, k** and **o** four times.

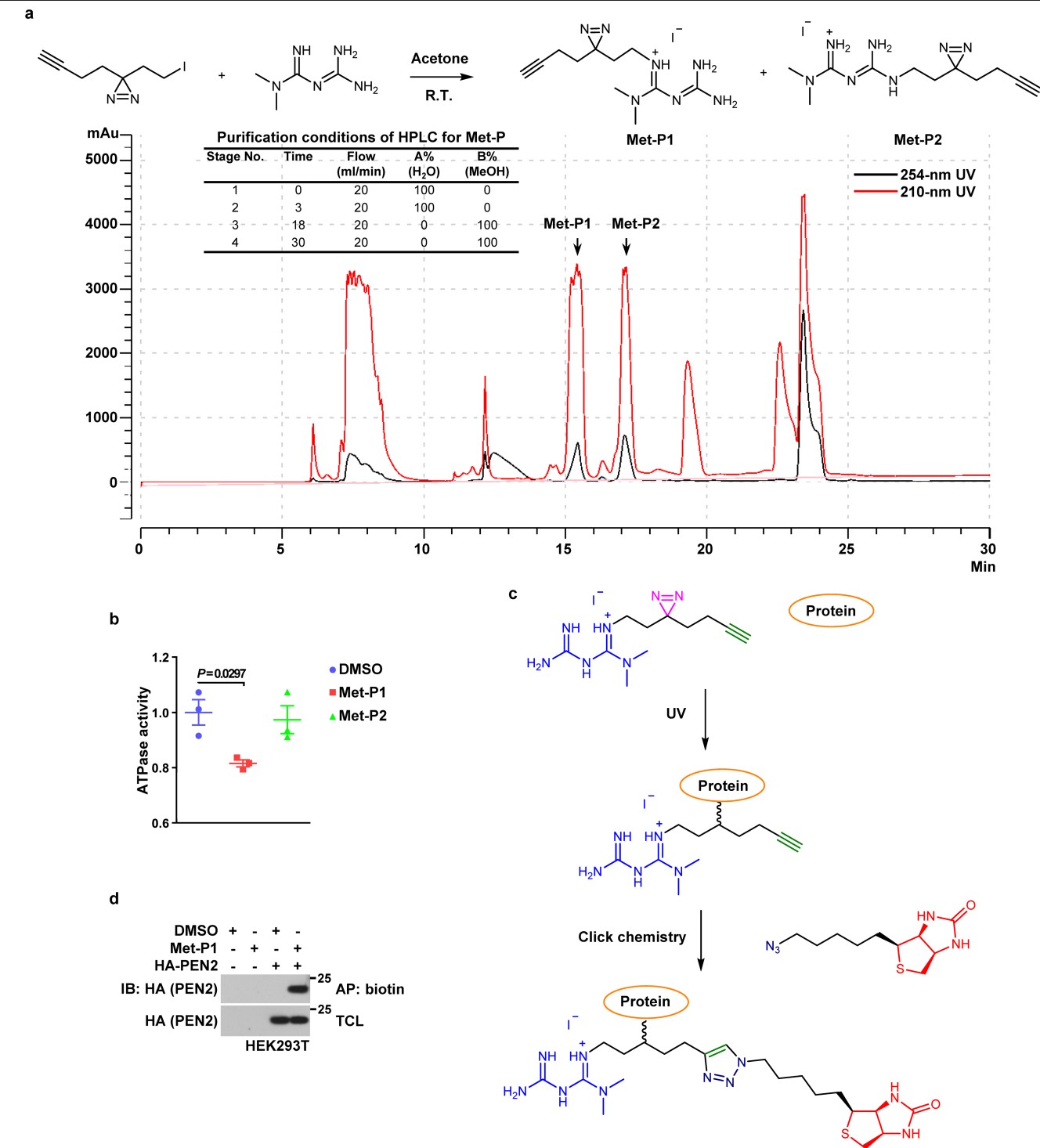

**Extended Data Fig. 2 | Identification of metformin-interacting proteins by the metformin probe. a**, Synthesis and purification of photoactive metformin probes (Met-Ps). Reactions for conjugating 3-(but-3-yn-1-yl)-3-(2-iodoethyl)-3H-diazirine to metformin that introduced a diazirine with a terminal alkyne moiety at either the N4 or N1/N2 position of metformin yields two types of Met-P (Met-P1 and Met-P2) products (upper panel). The two products were further separated on a preparative HPLC (lower panel). See detailed procedures, HSMS data, and NMR data in Methods section and Supplementary Fig. 2. **b**, Met-P1 is able to inhibit v-ATPase. Lysosomes purified from MEFs were incubated with the two Met-Ps at 10 μM for 1 h. The activity of v-ATPase was determined by its rate to hydrolyse ATP as in Extended Data Fig. 1d. After normalisation to the group without Met-P added, the data are shown as mean ± s.e.m.; $n = 3$ for each condition, and $P$ value by one-way ANOVA, followed by Dunnett. **c**, Reactions

taking place to form the Met-P1 and proteins conjugates. First, proteins were incubated with Met-P1. The metformin probe-protein mixture was exposed to UV light, followed by addition of Cu(II) salt, which catalyses a [3 + 2] azide-alkyne cycloaddition with biotin-azide, thus biotinylating probe-target complexes, allowing for the pull down of such complexes with NeutrAvidin beads. **d**, Interaction between PEN2 and metformin probe. HEK293T cells transfected with HA-tagged PEN2 were lysed. Total cell lysates (TCL) were incubated with 10 μM Met-P1, and subsequent exposure to UV, and were then mixed with 1 mM biotin-$N_3$ linker. The biotinylated proteins were then affinity-pulldown (AP) by NeutrAvidin beads, followed by immunoblotting with antibody against HA tag. Experiments in this figure were performed three times, except **a** seven times.

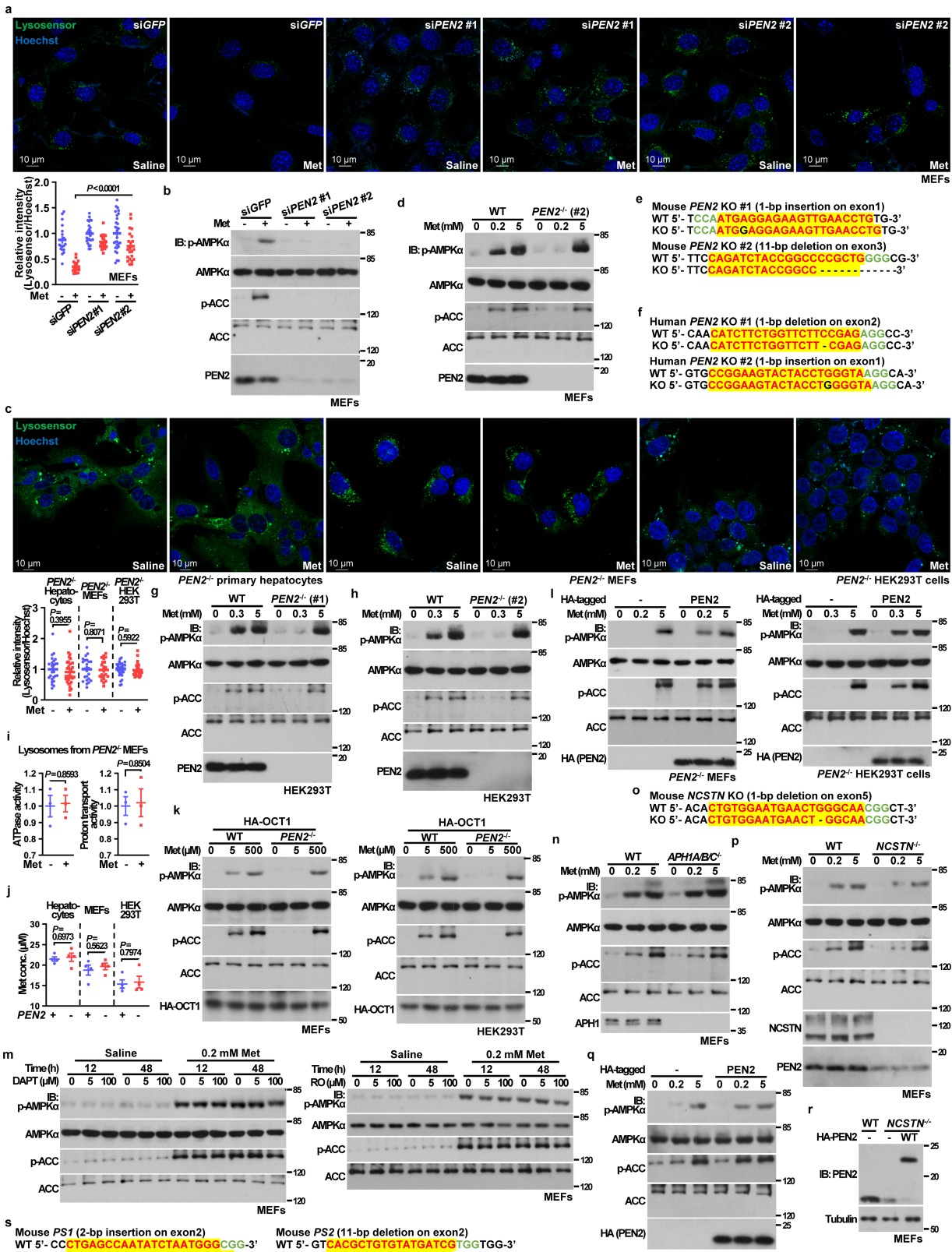

**Extended Data Fig. 3** | See next page for caption.

**Extended Data Fig. 3 | PEN2 is required for AMPK activation by low metformin. a**, **b**, Knockdown of *PEN2* impairs the activation of AMPK, and inhibition of v-ATPase by metformin. MEFs infected with lentivirus carrying two distinct siRNAs (#1 or 2#) against *PEN2*, or *GFP* as a control, were treated with 200 μM metformin for 12 h, representative images of the experiments shown in **a** upper, followed by analysis of the lysosomal pH [**a** lower, shown as mean ± s.e.m., *n* = 20 (control) and 21 (metformin-treated) cells for si*GFP*, *n* = 25 cells for si*PEN2*#1, and *n* = 29 (control) and 22 (metformin-treated) cells for si*PEN2*#2, all from 2 dishes/experiments; and *P* value by two-way ANOVA, followed by Tukey] and the determination of p-AMPKα and p-ACC (**b**). **c**, Knockout of *PEN2* abrogates the inhibition of v-ATPase by metformin in mouse primary hepatocytes (left panel), MEFs (middle panel), and HEK293T cells (right panel). MEFs, HEK293T cells were treated with 200, 300 μM metformin for 12 h, mouse primary hepatocytes were treated with 5 μM metformin for 2 h, and then labelled with Lysosensor, along with Hoechst. The lysosomal pH was determined as in Fig. 1a. Data are shown as mean ± s.e.m., *n* = 26 (control) and 31 (metformin-treated) from 6 dishes/experiments for primary hepatocytes, *n* = 25 (control) and 22 (metformin-treated) from 4 dishes/experiments for MEFs, and *n* = 30 (control) and 29 (metformin-treated) from 6 dishes/experiments for HEK293T cells; *P* value within each cell type was determined by two-sided Student's *t*-test. **d**, **g**, **h**, **k**, Knockout of *PEN2* blocks AMPK activation by low metformin. Clone #2 of *PEN2*[−/−] MEFs (**d**) treated with 200 μM (low concentration) metformin, or clone #1 and clone #2 of *PEN2*[−/−] HEK293T cells (**g** and **h**), treated with 300 μM (low concentration for the cell line) metformin, or OCT1-expressing *PEN2*[−/−] MEFs and HEK293T cells treated with 5 μM (low concentration) metformin (**k**) or 5 mM (high concentration, as a control for **d**, **g** and **h**), 500 μM metformin (high concentration, for **k**), for 12 h (**d**, **g** and **h**) or 2 h (**k**), were subjected to immunoblotting for the analysis of p-AMPKα and p-ACC. See also results with clone #1 of *PEN2*[−/−] MEFs in Fig. 2c. **e**, **f**, Strategies to generate MEFs (**e**) and HEK293T cells (**f**) with knockout of *PEN2*. Two distinct sets of sgRNAs for each cell line, whose sequences are listed in Methods section, were applied to generate *PEN2*[−/−] cells. Two clones (#1 and #2) for each cell line type were established. **i**, Knockout of *PEN2* blocks the inhibition of v-ATPase by metformin in purified lysosomes. Lysosomes purified from *PEN2*[−/−] MEFs were incubated with 5 μM metformin for 1 h. The activity of v-ATPase was determined as in Extended Data Fig. 1d. Data are shown as mean ± s.e.m.; *n* = 3 for each condition, and *P* value by two-sided Student's *t*-test. **j**, Knockout of *PEN2* does not affect metformin uptake. Mouse primary hepatocytes, MEFs and HEK293T cells were treated as in **c**, followed by determining intracellular metformin concentrations. Data are shown as mean ± s.e.m., *n* = 4 for each genotype, and *P* value within each cell type by two-sided Student's *t*-test. **l**, Re-introduction of PEN2 into *PEN2*[−/−] MEFs or HEK293T cells restores AMPK activation. *PEN2*[−/−] MEFs (left panel) or HEK293T cells (right panel) were infected with lentiviruses expressing HA-tagged PEN2 (all expressed at close-to-endogenous levels driven by pBOBI vector). Cells were treated with 200 or 300 μM (low concentration), or 5 mM (high concentration, as a control) metformin for 12 h, followed by analysis of p-AMPKα and p-ACC. **m**, Activity of the γ-secretase holoenzyme is dispensable for metformin-induced AMPK activation. MEFs were treated with DAPT (left panel) or RO4929097 (RO, right panel) at indicated concentrations for 12 h or 48 h. Twelve hours before lysis, cells were treated with 200 μM metformin, then lysed for analysis of p-AMPKα and p-ACC. **n**, Loss of APH1 does not affect metformin-induced activation of AMPK. MEFs with *APH1A*, *APH1B* and *APH1C* triple knockout were treated with 200 μM or 5 mM metformin for 12 h, followed by analysis of p-AMPKα and p-ACC. **o**, Strategies to generate MEFs with knockout of nicastrin. sgRNAs against *NCSTN*, whose sequences are listed in Methods section, were applied to generate *NCSTN*[−/−] MEFs. **p**, **q**, Knockout of *NCSTN*, through decreasing the protein levels of PEN2, impairs metformin-induced activation of AMPK. *NCSTN*[−/−] MEFs (**p**) or *NCSTN*[−/−] MEFs with HA-tagged PEN2 expressed (**q**, expressed at close-to-endogenous levels driven by the lentiviral system using pBOBI vector, as validated in **r**) were treated with 200 μM (low concentration) or 5 mM (high concentration, as a control) metformin for 12 h, followed by analysis of p-AMPKα and p-ACC. **r**, Protein levels of PEN2 in MEFs with knockout of *NCSTN*. Cells were lysed for analysis of PEN2 protein levels by immunoblotting, followed by densitometry analysis. **s**, Strategies to generate MEFs with knockout of presenilins. sgRNAs against *PS1* (left panel) and *PS2* (right panel), whose sequences are listed in Methods section, were applied to generate *PS1*- or *PS2*-KO MEFs. Experiments in this figure were performed three times, except **b**, **h**, **l**, four times.

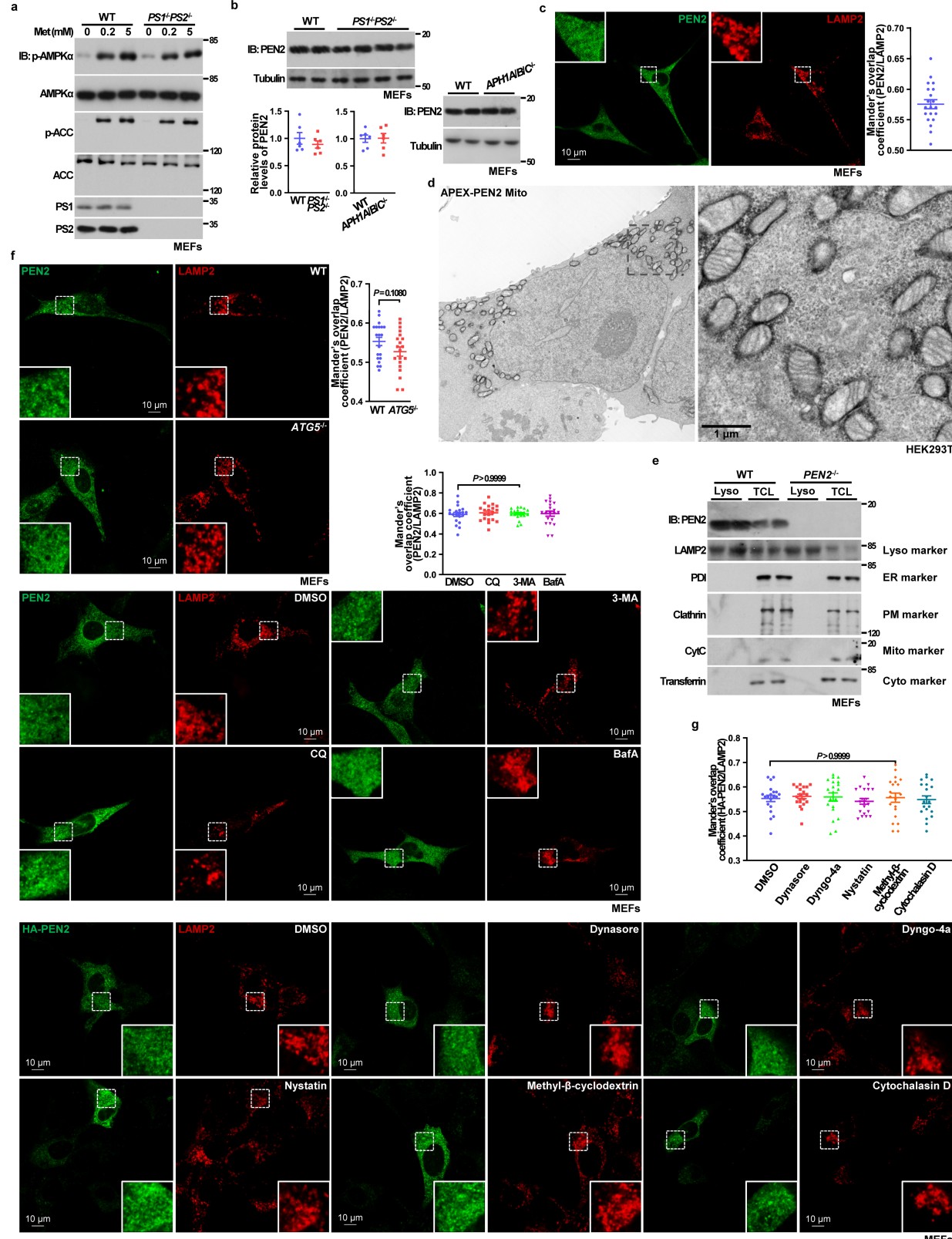

**Extended Data Fig. 4** | See next page for caption.

**Extended Data Fig. 4 | Other subunits of γ-secretase are not required for AMPK activation by low metformin. a**, Loss of presenilins does not affect metformin-induced activation of AMPK. MEFs with *PS1* and *PS2* double knocked out were treated with 200 µM or 5 mM metformin for 12 h, followed by analysis of p-AMPKα and p-ACC. **b**, Protein levels of PEN2 in MEFs with knockout of presenilins (left panel) or *APH1* (right panel). Cells were lysed for analysis of PEN2 protein levels by immunoblotting, followed by densitometry analysis. Statistical analysis results were shown as mean ± s.e.m.; *n* = 6. **c**, A portion of PEN2 is localised on the surface of lysosome. MEFs were stained with rabbit anti-PEN2 antibody and rat anti-LAMP2 antibody, followed by staining with Alexa Fluor 488 goat anti-rabbit IgG and Alexa Fluor 594 donkey anti-rat IgG secondary antibodies. The areas defined by dashed boxes on each representative image are enlarged as insets. Mander's overlap coefficients are plotted as mean ± s.e.m., *n* = 20. **d**, As a control for Fig. 2e, APEX-tagged TOMM20-PEN2 chimeric construct shows mitochondrial localisation. HEK293T cells stably expressing APEX-tagged TOMM20-PEN2 fused protein (APEX-PEN2 Mito, expressed at close-to-endogenous levels) were imaged under a transmission electron microscope. **e**, Subcellular fractionation assays show that the lysosomal fraction contains PEN2. The lysosome fractions (purified as described in Methods section), along with total cell lysates of MEFs, or *PEN2*<sup>-/-</sup> MEFs as a control, were subjected to immunoblotting using the indicated antibodies (Lyso, lysosome; ER, endoplasmic reticulum; PM, plasma membrane; Mito, mitochondrion; Cyto, cytosol). **f**, The PEN2 lysosomal localisation is not altered when autophagy is blocked. The lysosomal localisation of PEN2 was determined by staining PEN2 and LAMP2 in WT MEFs and *ATG5*<sup>-/-</sup> MEFs, or WT MEFs treated with 20 µM chloroquine (CQ), 4 mM 3-MA, or 0.5 µM bafilomycin A (bafA) for 12 h. Mander's overlap coefficients are plotted as mean ± s.e.m., *n* = 20 for each genotype/treatment, with *P* values calculated by one-way ANOVA, followed by Dunn (for autophagy inhibitors) or by two-sided Student's *t*-test (for *ATG5*<sup>-/-</sup> MEFs). **g**, Inhibition of endocytosis does not alter the lysosomal PEN2 localisation. MEFs were treated with 60 µM Dynasore for 0.5 h, 10 µM Dyngo-4a for 30 h, 20 µM Nystatin for 1 h, 2 mM methyl-β-cyclodextrin for 6 h, or 1 µM cytochalasin D for 0.5 h. Co-localisation of HA-tagged PEN2 and LAMP2 was then determined by immunofluorescent staining, and the Mander's overlap coefficients are plotted as mean ± s.e.m., *n* = 20 for each genotype/treatment, with *P* values calculated by one-way ANOVA, followed by Dunnett. Experiments in this figure were performed three times.

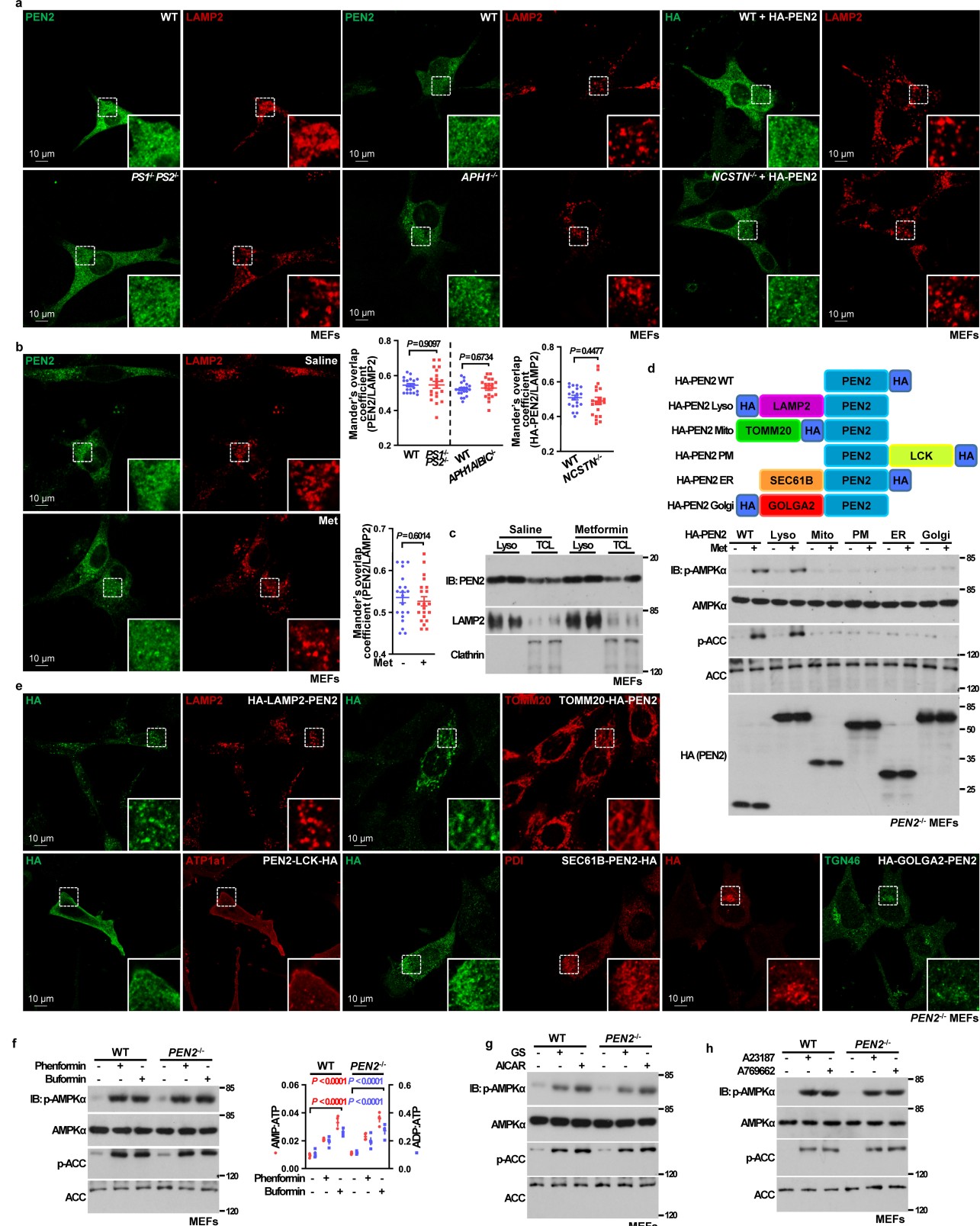

**Extended Data Fig. 5** | See next page for caption.

**Extended Data Fig. 5 | Lysosomal localisation of PEN2 is required for AMPK activation by low metformin. a**, Other γ-secretase subunits play no role in regulating lysosomal PEN2 localisation. MEFs with *PS1* and *PS2* double knockout or *APH1A/B/C* triple knockout were stained with antibodies against PEN2 and LAMP2. MEFs with *NCSTN* knockout were infected with lentivirus expressing HA-tagged PEN2 (at a close-to-endogenous level), and were stained with HA-tag and LAMP2 antibodies. Mander's overlap coefficients are plotted as mean ± s.e.m., *n* = 20 for each genotype/treatment, with *P* values calculated by two-sided Student's *t*-test (for *APH1A/B/C* triple knockout MEFs) or by two-sided Student's *t*-test with Welch's correction (*PS1* and *PS2* double knockout MEFs and *NCSTN* knockout MEFs). **b**, **c**, Metformin does not alter the PEN2 lysosomal localisation. MEFs were treated with 200 µM metformin for 12 h, followed by determination of PEN2 and LAMP2 co-localisation (**b**) or PEN2 protein levels on the lysosomal fractions (**c**). In **b**, Mander's overlap coefficients are plotted as mean ± s.e.m., *n* = 20 for control and 21 for metformin treatment, with *P* values calculated by two-sided Student's *t*-test. **d**, **e**, Disruption of lysosomal localisation of PEN2 impairs AMPK activation by metformin. *PEN2^-/-* MEFs were infected with lentiviruses carrying PEN2 constructs fused to N-terminus to TOMM20 (for tethering to the mitochondrial outer membrane),

SEC61B (for tethering to the endoplasmic reticulum), LAMP2 (for tethering to the cytoplasmic face of lysosome) or GOLGA2 (for tethering to the cytoplasmic face of cis-Golgi complex), or at their C-terminus to LCK (for tethering to the cytoplasmic face of plasma membrane) (diagrammed on upper panel of **d**, and validated in **e**). Cells were treated with 200 µM metformin for 12 h, followed by analysis of p-AMPKα and p-ACC (lower panel of **d**). **f**, Phenformin and buformin, two biguanides that increase AMP levels, activate AMPK independently of PEN2. *PEN2^-/-* MEFs were treated with 1 mM of phenformin or buformin for 2 h, followed by analysis of p-AMPKα and p-ACC (left panel), as well as the AMP:ATP and ADP:ATP ratios (right panel, results are shown as mean ± s.e.m.; *n* = 4 for each genotype/treatment, and *P* value by two-way ANOVA followed by Tukey). **g**, **h**, PEN2 plays a specific role in the metformin-induced AMPK activation. *PEN2^-/-* MEFs were treated with glucose-free DMEM (GS, shown in **g**), 1 mM AICAR (as an AMP mimetic, **g**), 5 µM A23187 (to release calcium for AMPK activation via CaMKK2, **h**), or 200 µM A769662 (acting downstream by directly binding to AMPK, **h**) for 2 h, followed by analysis of p-AMPKα and p-ACC. Experiments in this figure were performed three times, except **f**, **g** and **h** four times.

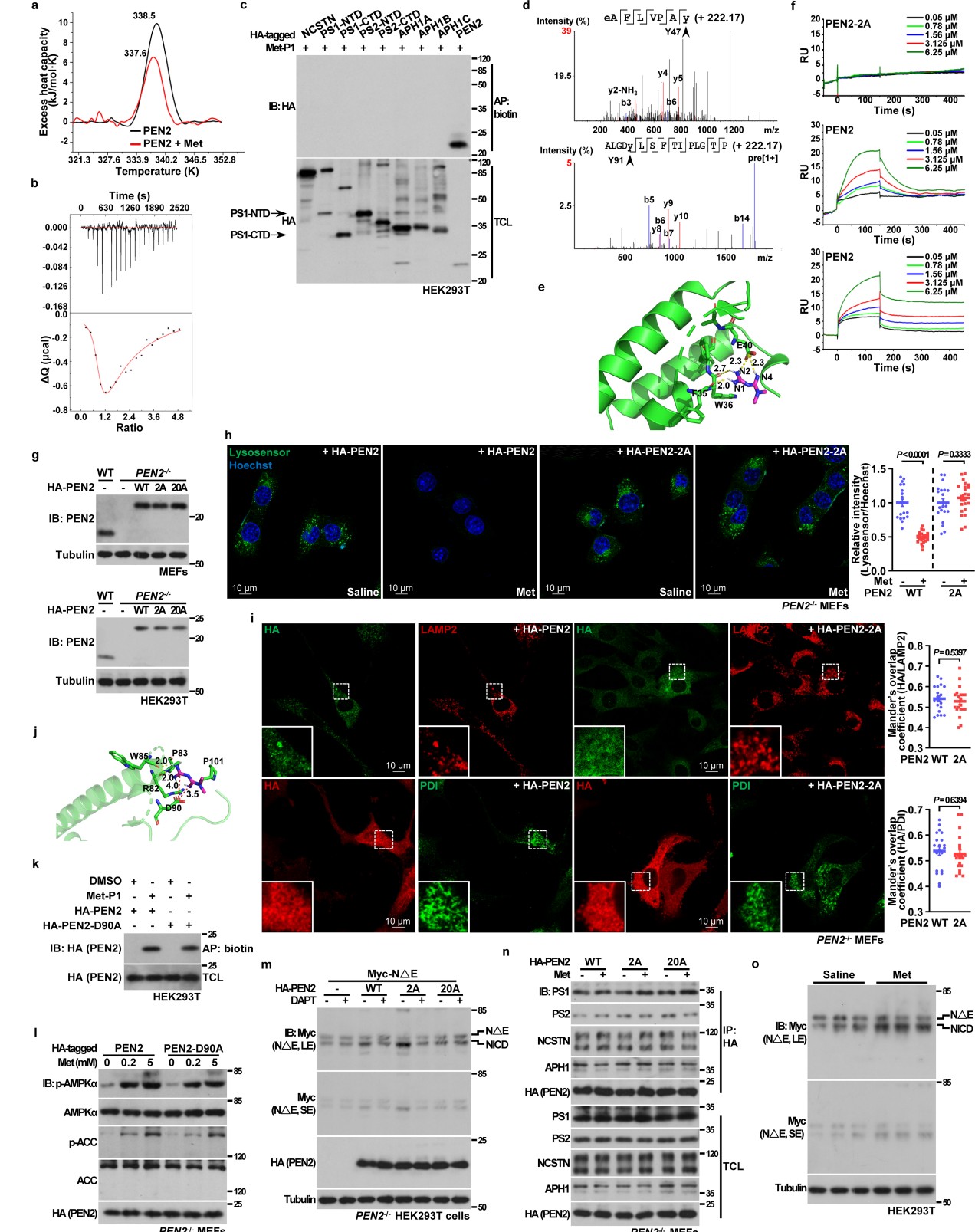

**Extended Data Fig. 6** | See next page for caption.

**Extended Data Fig. 6 | PEN2 binds to metformin. a**, Incubation with metformin decreases the thermal transition midpoint (Tm) of PEN2. FLAG-tagged PEN2 was ectopically expressed in HEK293T cells and purified. Some 10 μM of the purified PEN2 was then incubated with 10 μM metformin in PBS buffer, followed by determining the Tm on a differential scanning calorimetre. Enthalpy changes of PEN2, and PEN2 incubated with metformin at indicated temperatures are shown. **b**, ITC assay showing that PEN2 is able to bind metformin. Metformin (2 mM stock concentration) was loaded stepwise to 60 μM PEN2 (purified as in **a**) in PBS buffer. Integrated data (lower panel) were obtained by fitting raw data (upper panel) with the two sets of sites model. **c**, Unlike PEN2, other γ-secretase subunits do not bind metformin. HEK293T cells transfected with HA-tagged PS1 (either its NTD or CTD), PS2 (either its NTD or CTD), NCSTN, APH1A, APH1B, APH1C, or PEN2 as a control, were lysed, followed by incubation with 10 μM Met-P1, exposure to UV, and were then mixed with 1 mM biotin-N$_3$ linker. The biotinylated proteins were then pulled down by NeutrAvidin beads, followed by immunoblotting with antibody against HA tag. **d**, Determination of the binding sites of PEN2 for metformin by mass spectrometry. HEK-293T cells expressing HA-tagged PEN2 were lysed, followed by incubation with 10 μM Met-P1, exposure to UV, and the potential modified residues (conjugated with biotinylated Met-P1, with an increase of m/z by 222.17) were determined by mass spectrometry, revealing two Met-P1-conjugated residues, Y47 and Y91, as shown by the typical spectrograms, with Y91 conjugated at a much lower efficiency. **e**, *In silico* modelling of metformin bound to the N-terminal, cytosolic face of PEN2. Modelling was performed according to the reported cryo-electron microscopy structure (PDB ID: 6IYC). As shown in this figure, metformin could be protonated by carboxyl groups from residue E40 of PEN2, and then be docked onto PEN2 via salt bridges formed between N2-E40 (position of metformin to the residue of PEN2, and the same below) (2.3 Å) and N4-E40 (2.3 Å). Furthermore, two potential hydrogen bonds formed between N1-F35 (2.0 Å), N2-W36 (2.7 Å) may further strengthen the interaction between metformin and PEN2. **f**, PEN2-2A fails to bind metformin. PEN2-2A (F35A and E40A, purified as in **a**) was immobilised on a BIAcore CM5 sensor chip, followed by analysing its interaction with metformin by an SPR assay as in Fig. 2f. Sensorgrams of each measurement are shown. See also sensorgrams from another two repeats of Fig. 2f below. **g**, Validation data showing that re-introduced PEN2 and its mutants are expressed at a close-to-endogenous level in *PEN2$^{-/-}$* MEFs and HEK293T cells. **h**, PEN2-2A fails to mediate the effect of metformin on v-ATPase inhibition. MEFs were treated as in Extended Data Fig. 1q, followed by analysis of lysosomal pH as in Fig. 1a. After normalisation to the group without metformin treatment within each genotype (same hereafter), results are shown as mean ± s.e.m.; $n$ = 20 (control) and 23 (metformin-treated) from 4 dishes/experiments for PEN2-WT, and $n$ = 20 (normal) and 23 (metformin-treated) from 6 dishes/experiments for

PEN2-2A; and $P$ value within each genotype was determined by two-sided Student's $t$ test with Welch's correction (for re-introduction of wild type PEN2) or by two-sided Student's $t$ test (for re-introduction of PEN2-2A). **i**, PEN2-2A mutant retains proper subcellular localisation as wildtype PEN2. *PEN2$^{-/-}$* MEFs were infected with lentivirus expressing HA-tagged PEN2-2A or wildtype PEN2. Cells were then stained with mouse anti-HA antibody and rat anti-LAMP2 antibody, followed by incubation with Alexa Fluor 488 goat anti-mouse IgG and Alexa Fluor 594 donkey anti-rat IgG secondary antibodies (upper panel), or mouse anti-HA antibody and Alexa Fluor 594 goat anti-mouse IgG secondary antibody, followed with Alexa Fluor 488-conjugated rabbit anti-PDI antibody (lower panel). Mander's overlap coefficients are plotted as mean ± s.e.m., $n$ = 20 for each genotype, with $P$ values calculated by two-sided Student's $t$-test. The areas defined by dashed boxes on each representative image are enlarged as insets. **j**, *In silico* modelling of metformin bound to the C-terminal of PEN2. Residues D90 with the top-ranked score, as well as salt bridges and hydrogen bonds formed, are shown. **k**, PEN2-D90A displays full affinity for metformin. HEK293T cells transfected with HA-tagged PEN2-D90A or wild type PEN2 were lysed, followed by incubation with 10 μM Met-P1, exposure to UV, and were then mixed with 1 mM biotin-N$_3$ linker. The biotinylated proteins were then pulled down by NeutrAvidin beads, followed by immunoblotting with antibody against HA tag. **l**, Residue D90 in PEN2 is not involved in metformin-induced AMPK activation. *PEN2$^{-/-}$* MEFs re-introduced with HA-tagged PEN2-D90A or wild type PEN2 (both were expressed at close-to-endogenous levels) were treated with 200 μM (low concentration) or 5 mM (high concentration, as a control) metformin for 12 h, followed by analysis of p-AMPKα and p-ACC. **m, o**, Effects of PEN2 mutant and metformin on the activity of γ-secretase. HEK293T cells (**o**) or *PEN2$^{-/-}$* HEK293T cells (**m**) were infected with lentivirus expressing Myc-tagged NotchΔE (NΔE). In **m**, cells were also infected with lentivirus expressing HA-tagged PEN2-2A, PEN2-20A or wildtype PEN2, in addition to the Myc-tagged NotchΔE (NΔE). Cells were then treated with 300 μM metformin for 12 h (**o**), or 100 μM DAPT for 12 h, the inhibitor to γ-secretase as a control (**m**). The cleavage of NotchΔE was determined by the protein levels of its NICD domain by immunoblotting. LE, long exposure; SE, short exposure. **n**, The PEN2 mutations or metformin do not affect the complex formation of γ-secretase. *PEN2$^{-/-}$* MEFs infected with lentivirus expressing HA-tagged PEN2-2A, PEN2-20A (residues 27, 28, 30, 31, 34, 38, 42, 43, 57, 58, 60, 63 to 65, 67, 68, 71, 72, 74 and 75 of PEN2 mutated to alanine, see Extended Data Fig. 7e) or wildtype PEN2, were treated with 200 μM metformin for 12 h and lysed, followed by immunoprecipitation (IP) with antibodies against HA. Immunoprecipitants were than subjected to immunoblotting with antibodies against PS1, PS2, NCSTN, APH1A/B/C, as well as HA (PEN2). Experiments in this figure were performed three times, except **g**, **k** and **l** four times.

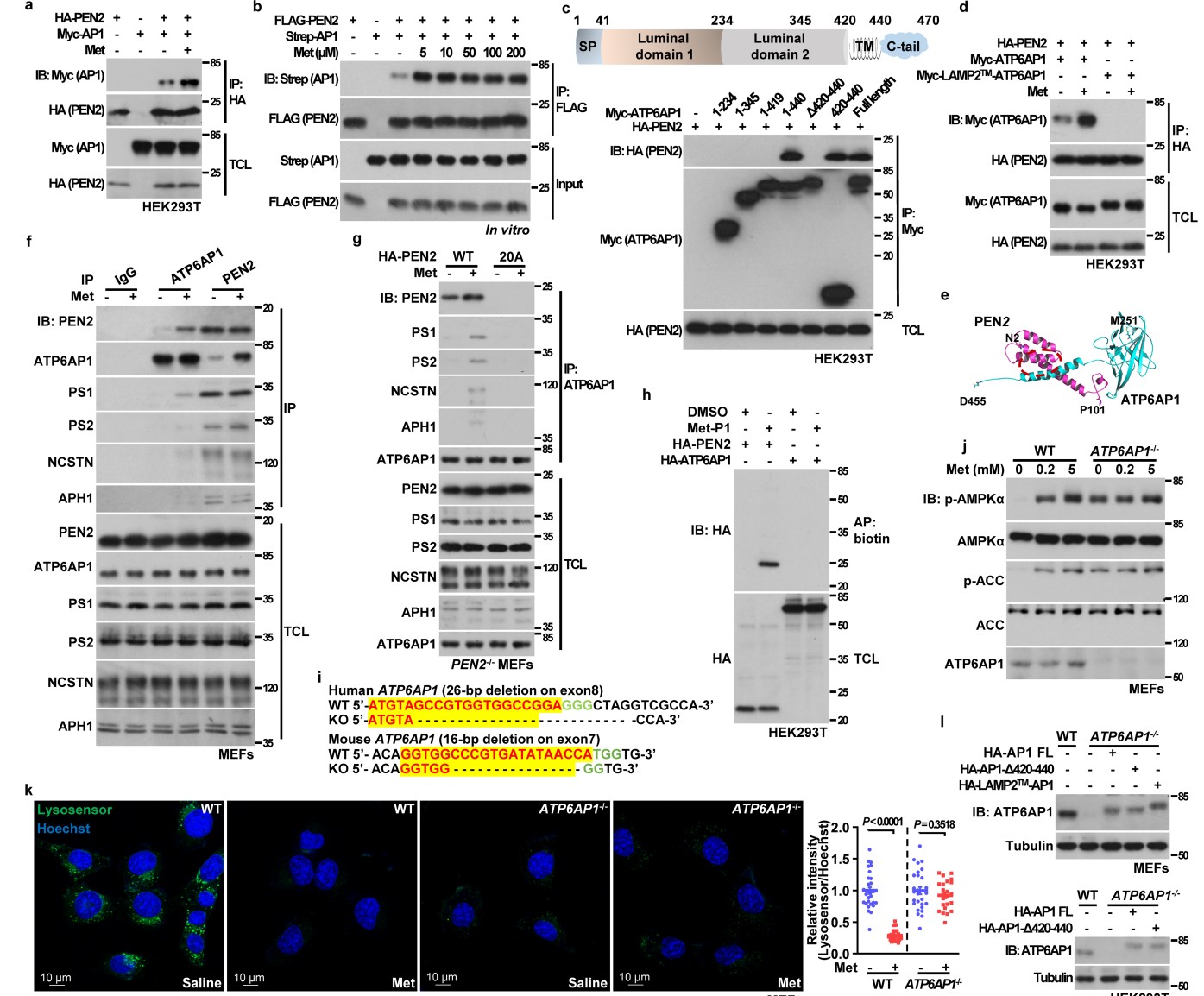

**Extended Data Fig. 7** | See next page for caption.

**Extended Data Fig. 7 | ATP6AP1 tethers PEN2 to v-ATPase. a**, PEN2 interacts with ATP6AP1. HEK293T cells were transfected with HA-tagged PEN2 and Myc-tagged ATP6AP1 (AP1). Cells were lysed, and 10 μM metformin (final concentration) or PBS was added to the lysates. Immunoprecipitation (IP) was performed using antibodies against HA, followed by immunoblotting with antibodies indicated. **b**, PEN2 interacts with ATP6AP1 *in vitro*. Some 1 μg of FLAG-tagged PEN2 (expressed in HEK293T cells, and purified through eluting with FLAG® peptide) were incubated with 1 μg of Strep-tagged ATP6AP1 (expressed in HEK293T cells, and purified through eluting with desthiobiotin) (input) in lysis buffer, then with metformin at indicated concentrations for 2 h. Immunoprecipitation was performed using ANTI-FLAG® M2 Affinity Gel, followed by immunoblotting with antibodies indicated. **c**, Domain mapping for the region on ATP6AP1 responsible for PEN2-binding. HA-tagged PEN2 was co-transfected with Myc-tagged ATP6AP1, or its deletion mutants into HEK293T cells. Immunoprecipitation was performed using antibody against Myc-tag, followed by immunoblotting with antibodies indicated. **d**, Replacement of ATP6AP1 transmembrane domain with that of LAMP2, blocks its interaction with PEN2. HEK293T cells transfected with HA-tagged PEN2-D90A, along with Myc-tagged LAMP2™-ATP6AP1 or wildtype ATP6AP1, were lysed, and 10 μM metformin was added to the lysates, followed by immunoprecipitation with antibody against HA, and immunoblotting with antibodies indicated. **e**, *In silico* modelling of ATP6AP1 (cyan) bound to PEN2 (magenta). Circled area indicates the predicted interface, in which residues 27, 28, 30, 31, 34, 38, 42, 43, 57, 58, 60, 63 to 65, 67, 68, 71, 72, 74 and 75, within the transmembrane domain of PEN2, are involved. **f**, ATP6AP1 shows much weaker interaction with other γ-secretase subunits than PEN2. MEFs were lysed and incubated with metformin as in **a**, followed by immunoprecipitation with antibodies against ATP6AP1, or PEN2 as a control. Immunoprecipitants were than subjected to immunoblotting with antibodies against PS1, PS2, NCSTN, APH1A/B/C, as well as PEN2 and ATP6AP1.

**g**, Metformin, through promoting the association between PEN2 and ATP6AP1, enhances association of ATP6AP1 and γ-secretase. *PEN2^{-/-}* MEFs infected with lentivirus expressing HA-tagged PEN2 or its 20A mutant (lacking the interface for ATP6AP1) were lysed and incubated with metformin as in **a**, followed by immunoprecipitation with antibodies against ATP6AP1. Immunoprecipitants were than subjected to immunoblotting with antibodies against PS1, PS2, NCSTN, APH1A/B/C, as well as PEN2 and ATP6AP1. **h**, ATP6AP1 does not interact with metformin. HEK293T cells transfected with HA-tagged ATP6AP1, or HA-tagged PEN2 as a control, were lysed, followed by analysing the interaction between ATP6AP1 or PEN2 with Met-P1 as in Extended Data Fig. 2d. **i**, Strategies to generate MEFs (lower panel) or HEK293T cells (upper panel) with knockout of *ATP6AP1*. sgRNAs against *ATP6AP1*, whose sequences are listed in Methods section, were applied to generate *ATP6AP1^{-/-}* MEFs and HEK293T cells. **j**, Knockout of ATP6AP1 leads to constitutive activation of AMPK. MEFs with *ATP6AP1* knocked out, along with its wildtype control, were incubated with metformin at indicated concentrations for 12 h, followed by analysing p-AMPK and p-ACC. **k**, Knockout of ATP6AP1 renders v-ATPase inactive. *ATP6AP1^{-/-}* MEFs were treated with 200 μM metformin for 12 h, followed by analysis of lysosomal pH with the Lysosensor dye. Data (relative intensity of Lysosensor, processed as in Fig. 1a) were graphed as mean ± s.e.m., $n = 29$ (control) and 26 (metformin-treated) cells from 6 dishes/experiment for WT MEFs, and 28 (control) and 23 (metformin-treated) cells from 4 dishes/experiments for *ATP6AP1^{-/-}* MEFs, *P* value within each genotype was determined by two-sided Mann-Whitney test (for WT MEFs), or by two-sided Student's *t*-test (for *ATP6AP1^{-/-}* MEFs). **l**, Validation data showing that the re-introduced ATP6AP1 and its mutants are expressed at a close-to-endogenous level in *ATP6AP1^{-/-}* MEFs (upper panel) and HEK293T cells (lower panel). Experiments in this figure were performed three times, except **b**, **f**, **h**, four times and **l** five times.

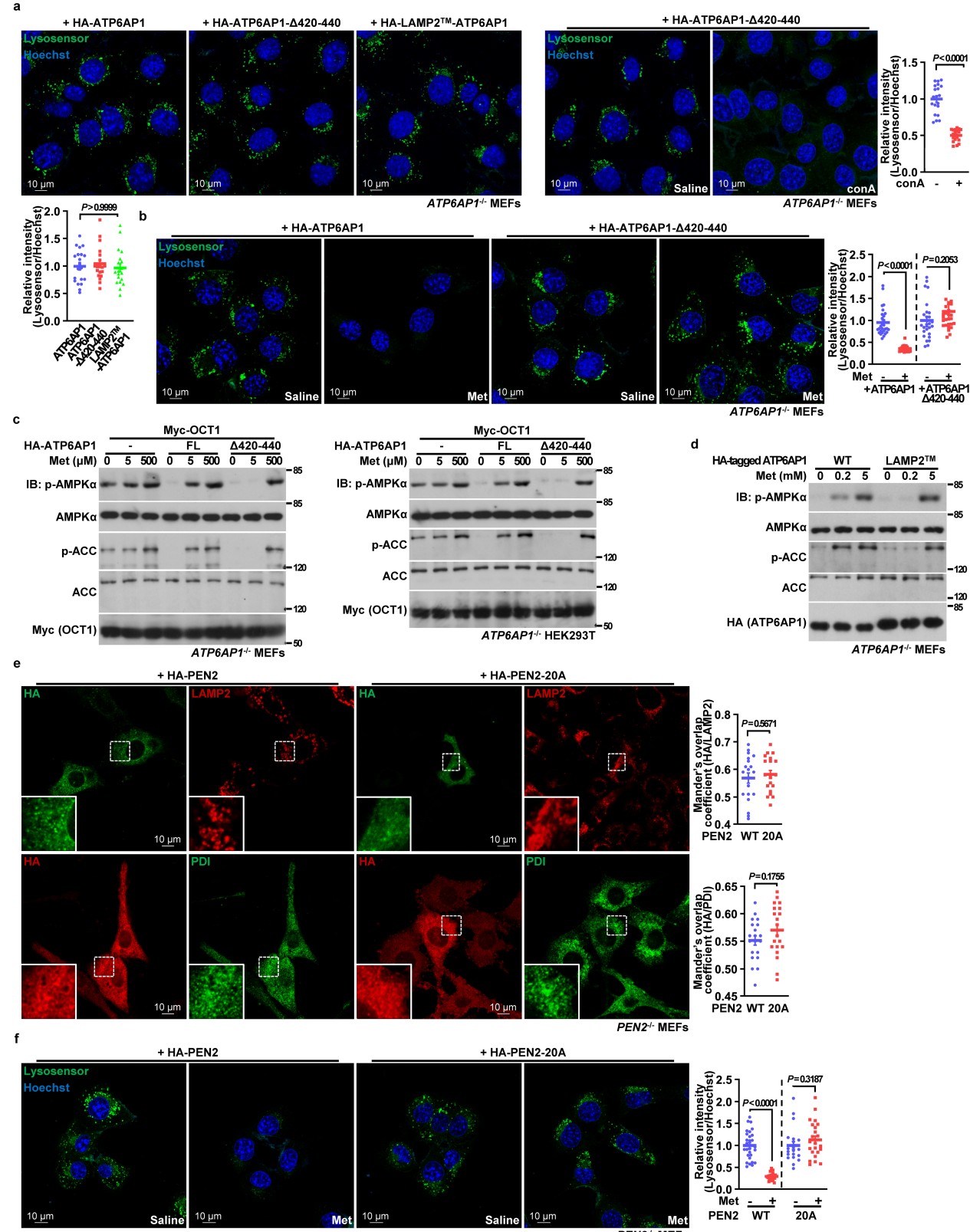

**Extended Data Fig. 8** | See next page for caption.

**Extended Data Fig. 8 | The interaction between PEN2 and ATP6AP1 is required for metformin-induced AMPK activation. a**, **b**, The truncated ATP6AP1 mutant (ATP6AP1$^{\Delta420\text{-}440}$) or the chimeric construct LAMP2$^{TM}$-ATP6AP1, which are able to maintain the basal activity of v-ATPase, fails to mediate metformin to inhibit v-ATPase. *ATP6AP1$^{-/-}$* MEFs re-introduced with full length (FL) ATP6AP1, ATP6AP1$^{\Delta420\text{-}440}$ or LAMP2$^{TM}$-ATP6AP1 (all expressed at close-to-endogenous levels) were treated with 200 μM metformin for 12 h (or 5 μM conA for 2 h as a control, right panel of **a**), followed by analysis of lysosomal pH by labelling cells with Lysosensor, along with Hoechst. Data (relative intensity of Lysosensor, processed as in Fig. 1a) were graphed as mean ± s.e.m., *n* = 20 cells from 3 dishes/experiment (**a**) and *n* = 28 (control) and 23 (metformin-treated) for *ATP6AP1$^{-/-}$* MEFs re-introduced with full length ATP6AP1, and *n* = 26 (control) and *n* = 25 (metformin-treated) for *ATP6AP1$^{-/-}$* MEFs re-introduced with ATP6AP1$^{\Delta420\text{-}440}$, from 4 dishes/experiment (**b**) for each genotype/treatment. Results are mean ± s.e.m.; *P* value was determined by one-way ANOVA followed by Dunn (**a**, left panel) or by two-sided Student's *t* test with Welch's correction (**a**, right panel) or by two-sided Mann-Whitney test (**b**). **c**, ATP6AP1$^{\Delta420\text{-}440}$ mutant fails to mediate the effects of metformin on AMPK activation. *ATP6AP1$^{-/-}$* MEFs (left panel) or HEK293T cells (right panel) stably expressing Myc-tagged OCT1 were re-introduced with full length (FL) ATP6AP1 or its Δ420-440 mutant (expressed at close-to-endogenous levels). Cells were treated with 5 μM metformin or 500 μM metformin (high concentration, as a control) for 2 h, followed by analysis of p-AMPK and p-ACC.

**d**, ATP6AP1 mutant that cannot interact with PEN2 fails to mediate AMPK activation by metformin. *ATP6AP1$^{-/-}$* MEFs re-introduced with full length ATP6AP1 or LAMP2$^{TM}$-ATP6AP1 (expressed at close-to-endogenous levels) were treated with 200 μM metformin for 12 h, followed by analysis of p-AMPKα and p-ACC by immunoblotting. **e**, PEN2-20A mutant retains proper subcellular localisation. *PEN2$^{-/-}$* MEFs were infected with lentivirus expressing HA-tagged PEN2-20A or wildtype PEN2. Cells were then stained with mouse anti-HA antibody and rat anti-LAMP2 antibody, followed by incubation with Alexa Fluor 488 goat anti-mouse IgG and Alexa Fluor 594 donkey anti-rat IgG secondary antibodies (upper panel), or mouse anti-HA antibody and Alexa Fluor 594 goat anti-mouse IgG secondary antibody, and subsequently with Alexa Fluor 488-conjugated rabbit anti-PDI antibody (lower panel). Mander's overlap coefficients are plotted as mean ± s.e.m., *n* = 20 for each genotype, with *P* values calculated by two-sided Student's *t*-test. The areas defined by dashed boxes on each representative image are enlarged as insets. **f**, *PEN2$^{-/-}$* MEFs re-introduced with wildtype PEN2 or PEN2-20A mutant were treated like Extended Data Fig. 6h, followed by analysis of the lysosomal pH [shown as mean ± s.e.m., *n* = 29 (control) and 27 (metformin-treated) from 6 dishes/experiments for *PEN2$^{-/-}$* MEFs re-introduced with wildtype PEN2, *n* = 20 (control) and 23 (metformin-treated) from 4 dishes/experiments for *PEN2$^{-/-}$* MEFs re-introduced with PEN2-20A; and *P* value within each genotype by two-sided Student's *t* test with Welch's correction (wildtype PEN2) or two-sided Student's *t* test (PEN2-20A)]. Experiments in this figure were performed three times, except **d** four times.

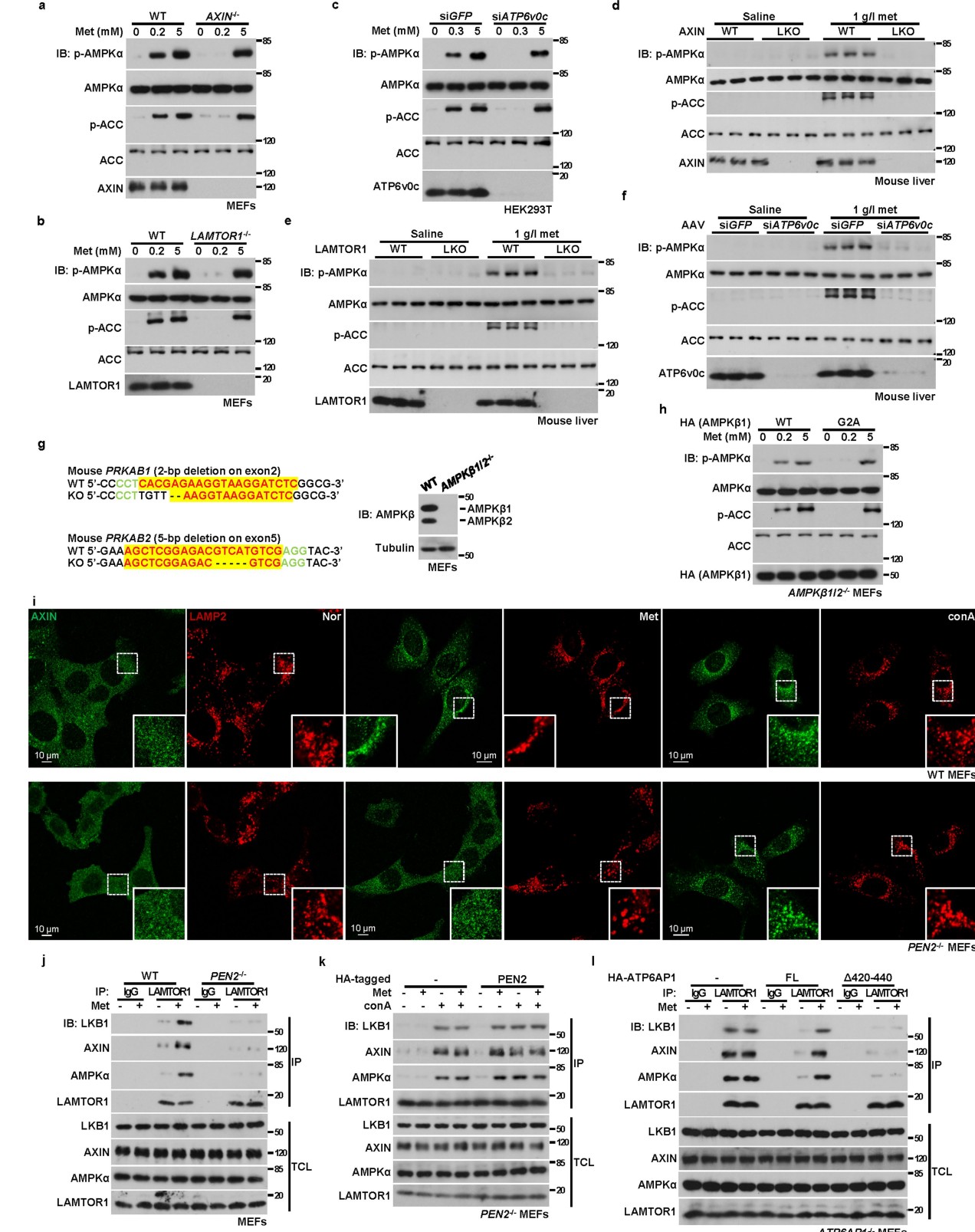

**Extended Data Fig. 9** | See next page for caption.

**Extended Data Fig. 9 | PEN2 and ATP6AP1 act as upstream factors of v-ATPase. a-c**, v-ATPase and its downstream factors AXIN and Ragulator, are required for AMPK activation by low metformin in cells. MEFs with *AXIN* (**a**) or *LAMTOR1* (**b**) knocked out, or HEK293T cells with *ATP6v0c* knocked down (**c**), were treated with metformin at indicated concentrations for 12 h, followed by analysis of p-AMPKα and p-ACC by immunoblotting. **d**–**f**, v-ATPase, AXIN and Ragulator are required for AMPK activation by low metformin in mouse liver. Mice at 6 weeks old with hepatic *AXIN* knocked out (**d**, AXIN-LKO, by crossing *AXIN*-floxed mice with mice carrying *albumin*-Cre, and those not carrying Cre as controls), hepatic *LAMTOR1* knocked out (**e**, LAMTOR1-LKO, generated same as in **d**, except that *LAMTOR1*-floxed mice were used), or hepatic *ATP6v0c* knocked down (**f**, si*ATP6v0c*, by intravenously injected with AAVs carrying siRNA against *ATP6v0c*, or *GFP* as a control, two weeks before experiments), were treated with metformin in drinking water (1 g/l) for 7 days. At the day 8, mice livers were quickly dissected from euthanised mice, and p-AMPKα and p-ACC levels in livers were determined by immunoblotting. **g**, Strategies to generate MEFs with knockout of *PRKAB1* and *PRKAB2* (*AMPKβ1* and *AMPKβ2*). sgRNAs against these two genes, whose sequences are listed in Methods section, were applied to generate *AMPKβ1/2⁻/⁻* MEFs. See also knockout efficiency of AMPKβ1 and AMPKβ2, as determined by immunoblotting using antibodies against pan-AMPKβ. **h**, Membrane localisation of AMPK is required for AMPK activation by low metformin. *AMPKβ1/2⁻/⁻* MEFs were infected with lentivirus carrying AMPKβ1 or its G2A mutant (defective in N-myristoylation and hence preventing membrane/lysosomal localization of AMPK). Cells were treated with 200 μM metformin or 5 mM for 12 h, followed by determining p-AMPKα and p-ACC levels by immunoblotting. **i**, Representative images of the experiments shown in Fig. 3g, indicate that PEN2 is required for the lysosomal translocation of AXIN. *PEN2⁻/⁻* MEFs and its wildtype control were treated with 200 μM metformin for 12 h, or with the v-ATPase inhibitor concanamycin A (conA, 5 μM) for 2 h as a control. AXIN and the lysosomal marker LAMP2 were stained with goat anti-AXIN antibody (green) and rat anti-LAMP2 antibody (red), respectively. Images were taken by confocal microscopy after incubating cells with Alexa Fluor 488 donkey anti-goat IgG and Alexa Fluor 594 donkey anti-rat IgG. The areas defined by dashed boxes on each representative image are enlarged as insets. **j**, **k**, **l**, Loss of PEN2, hence PEN2-ATP6AP1 interaction, abolishes the formation of the AXIN-based lysosomal complex by metformin. *PEN2⁻/⁻* MEFs (**j**), or *PEN2⁻/⁻* MEFs re-introduced with PEN2 (**k**), or *ATP6AP1⁻/⁻* MEFs re-introduced with full length (FL) ATP6AP1 or its Δ420-440 mutant (**l**, all expressed at close-to-endogenous levels) were treated with 200 μM metformin for 12 h, or treated with 5 μM concanamycin A (**k**, conA) for 2 h as a control. Endogenous LAMTOR1 was immunoprecipitated with rabbit anti-LAMTOR1 antibody, and IgG as control, followed by immunoblotting using the indicated antibodies. Experiments in this figure were performed three times, except **a-c** four times.

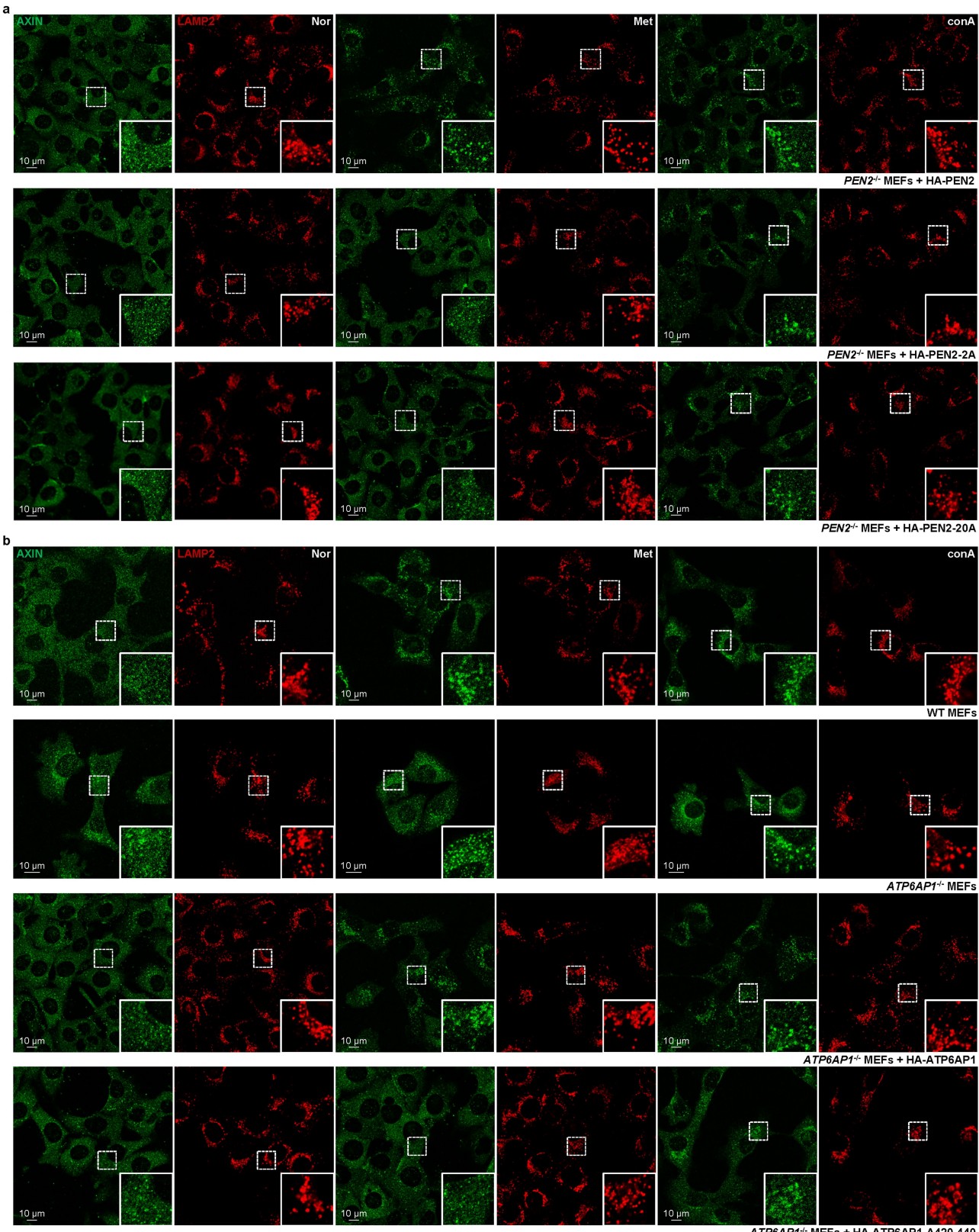

**Extended Data Fig. 10 | PEN2 and ATP6AP1 control the lysosomal translocation of AXIN. a**, **b**, Representative images of the experiments shown in Fig. 3g, indicate that PEN2, along with its interaction with ATP6AP1, is required for the lysosomal translocation of AXIN. *PEN2*[-/-] MEFs, along with *PEN2*[-/-] MEFs re-introduced with wildtype PEN2, its mutant 2A or mutant 20A (**a**), and *ATP6AP1*[-/-] MEFs, along with its wildtype control, or those re-introduced with full length (FL) ATP6AP1, or ATP6AP1[Δ420-440] (**b**, Re-introduced proteins were all expressed at close-to-endogenous levels), were treated with 200 μM metformin for 12 h, or with the v-ATPase inhibitor concanamycin A (conA, 5 μM) for 2 h as a control. AXIN and the lysosomal marker LAMP2 were stained with goat anti-AXIN antibody (green) and rat anti-LAMP2 antibody (red), respectively. Images were taken by confocal microscopy after incubating cells with Alexa Fluor 488 donkey anti-goat IgG and Alexa Fluor 594 donkey anti-rat IgG. The areas defined by dashed boxes on each representative image are enlarged as insets. Experiments in this figure were performed three times.

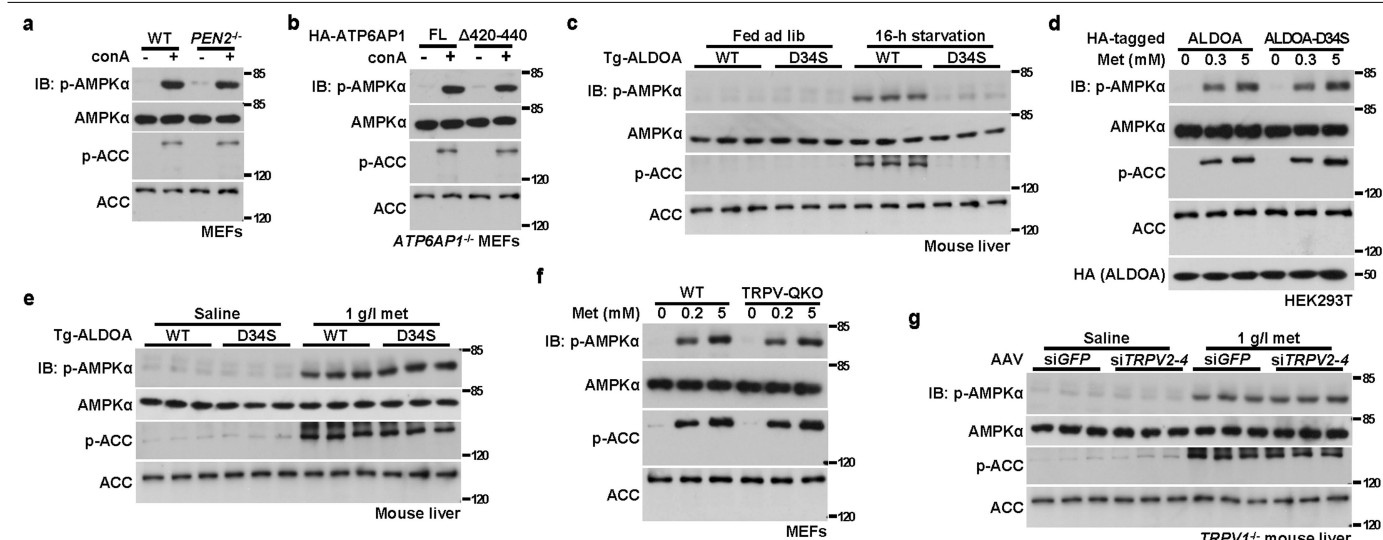

**Extended Data Fig. 11 | Aldolase-TRPV act as a separate route from PEN2-ATP6AP1. a**, **b**, Inhibition of v-ATPase by conA activates AMPK in cells lacking the PEN2 and ATP6AP1 axis. *PEN2*[-/-] MEFs (**a**) or *ATP6AP1*[-/-] MEFs re-introduced with full length (FL) ATP6AP1 or its Δ420-440 mutant (**b**; expressed at close-to-endogenous levels) were treated with 5 μM conA for 2 h, followed by determining p-AMPKα and p-ACC levels by immunoblotting. **c**, Liver-specific ALDOA-D34S transgenic (Tg) mice renders hepatic AMPK inactive under starvation. Tg mice (6-week-old) expressing ALDOA-D34S (expressed under an *ApoE* promoter and its hepatic control region included in the pLiv-Le6 vector), which can still bind FBP in low glucose, and therefore mimics a high glucose state in which AMPK is inactivated (see cell-based data in ref. [6]), were starved for 16 h. P-AMPKα and p-ACC levels in livers were then determined by immunoblotting. **d**, Aldolase is dispensable for low metformin-induced AMPK activation in HEK293T cells. Cells stably expressing HA-tagged ALDOA-D34S mutant, or wildtype ALDOA were treated with 300 μM metformin (low concentration) or 5 mM (high concentration, as a control) for

12 h, followed by analysis of p-AMPKα and p-ACC levels by immunoblotting. **e**, Aldolase is dispensable for low metformin-induced AMPK activation in mouse liver. Mice with Tg-ALDOA-D34S at 6 weeks old were treated with metformin in drinking water (1 g/l) for another 7 days. At the day 8, mice were sacrificed, and hepatic p-AMPKα and p-ACC levels were determined by immunoblotting. **f**, **g**, TRPV is dispensable for low metformin-induced AMPK activation. Experiments pertaining to TRPV dependency were performed in MEFs (**f**) as well as in mouse livers (**g**) as described in **d** (except 200 μM metformin was used) and **e**, respectively, except that MEFs with quadruple knockout of *TRPV1-4* (**f**, TRPV-QKO), or the *TRPV1*[-/-] mice injected with a combination of AAV-carried siRNAs against *TRPV2*, *TRPV3* and *TRPV4* (**g**, si*TRPV2-4*, at 8 weeks old, which had been validated to show little expression of TRPVs, see ref. [25]), were used (viruses were injected at 4 weeks old, and metformin was supplied at 8 weeks old). Experiments in this figure were performed three times, except **d** five times.

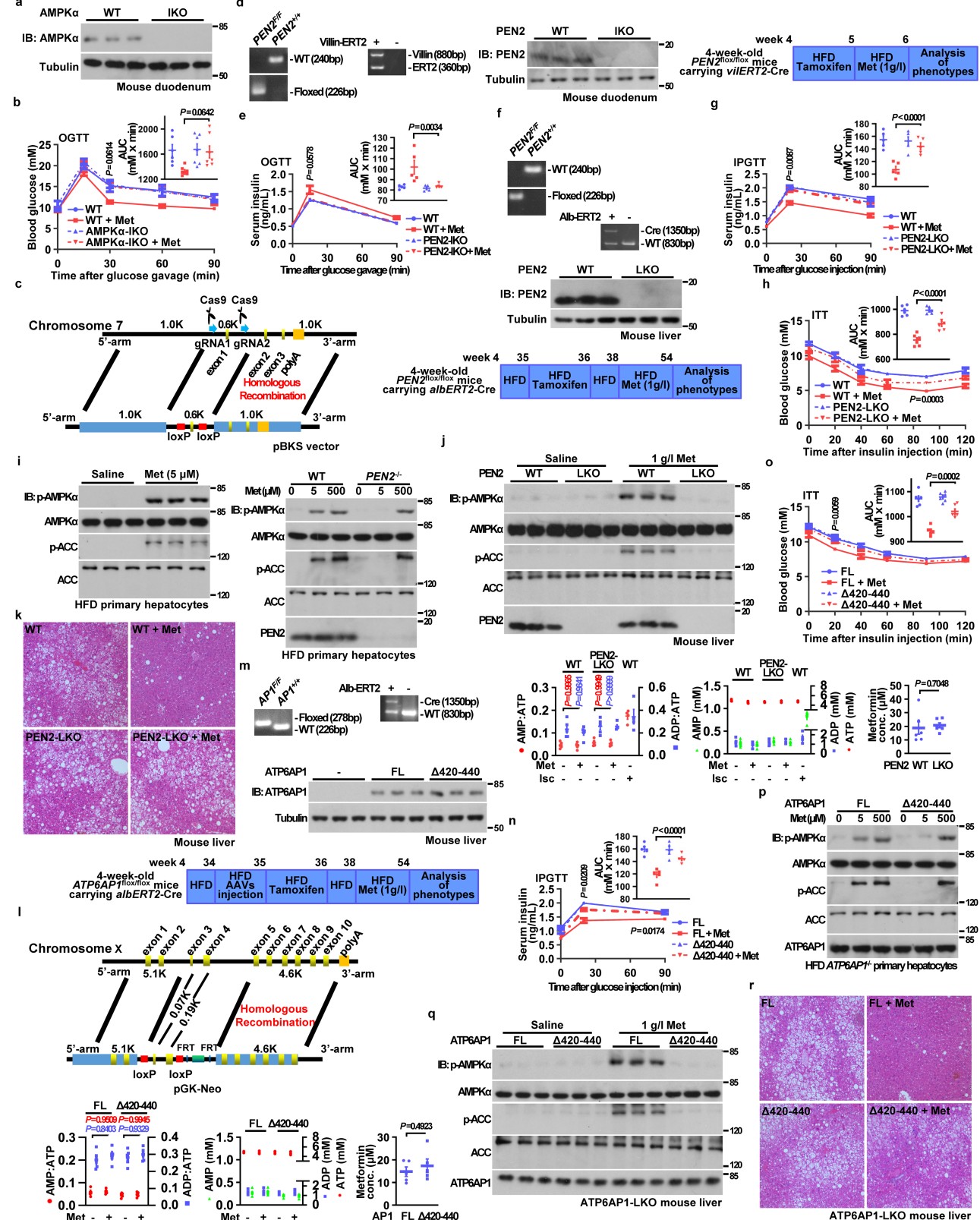

**Extended Data Fig. 12** | See next page for caption.

**Extended Data Fig. 12 | PEN2-ATP6AP1 axis is required for metformin-mediated glucose absorption and hepatic fat reduction. a**, Verification of AMPKα knockout efficiency in the duodenum of AMPKα intestine-specific knockout (IKO) mice. Mouse offsprings carrying floxed *AMPKα1* and *AMPKα2*, as well as *vilERT2*-Cre (tamoxifen-sensitive, ERT2-fused Cre recombinase expressed under the control of the *villin* promoter, for deletion of *AMPKα* in intestine), along with their wildtype littermates (carrying floxed *AMPKα1* and *AMPKα2*, but not *vilERT2*-Cre) were injected with tamoxifen (as described in Method section) for knockout of *AMPKα*. Protein levels of AMPKα in duodenum were then analysed by immunoblotting. **b**, Intestinal AMPK is required for metformin to induce glucose lowering. Mice at 5 weeks old with intestinal *AMPKα1/2* double knockout (AMPKα1/2-IKO), along with its wildtype littermates, were treated with metformin in drinking water (+Met, 1 g/l) for 7 days (tamoxifen was injected at 4 weeks old). At day 8, mice were starved for 6 h, followed by oral glucose tolerance test (OGTT, results are shown as mean ± s.e.m., $n$ = 7 for each genotype/treatment, except for WT mice treated with drinking water without metformin, $n$ = 6; and $P$ value by two-way RM ANOVA, followed by Tukey, compared blood glucose levels between WT + Met group and AMPKα1/2-IKO + Met group at the same time point; see also inset for AUC values, $P$ value by two-way ANOVA, followed by Tukey). **c, l**, Strategies to generate *PEN2*-floxed (**c**) or *ATP6AP1*-floxed (**l**) allele. **d, f**, Depletion of *PEN2* in the intestine or liver in *PEN2*-floxed mice. PCR analysis results of mouse offsprings carrying floxed *PEN2*, as well as *vilERT2*-Cre (**d**) or *albERT2*-Cre (**f**, same as in **d**, except Cre under *albumin* promoter, for deletion of *PEN2* in liver) allele are shown on the upper panels. Protein levels of PEN2 in duodenal (**d**), or hepatic (**f**) tissues in *PEN2* intestine- or liver-specific knockout (IKO or LKO) mice are also shown. See also panels of **d** and **f** for the experimental timeline of the analysis of phenotypes of PEN2-IKO (**d**) and PEN2-LKO (**f**) mice. **e**, Intestinal PEN2 is required for metformin-induced glucose-lowering effect. Mice were fed with HFD as in Fig. 4a. Serum insulin levels are shown as mean ± s.e.m., $n$ = 5 for each genotype/treatment, and $P$ value by two-way RM ANOVA, followed by Tukey, compared insulin levels between WT + Met group and PEN2-IKO + Met group at the same time point; see also inset for AUC values, $P$ value by two-way ANOVA, followed by Tukey. **g, h**, PEN2 is required for metformin-induced reduction of hepatic fat. As depicted in **f**, mice at 38 weeks old with hepatic *PEN2* knockout (PEN2-LKO), along with its wildtype littermates, were treated with metformin in drinking water (+Met, 1 g/l) for 16 weeks. At week 54, mice were starved for 6 h, followed by intraperitoneal glucose tolerance test (**g**, IPGTT; serum insulin levels are shown as mean ± s.e.m., $n$ = 5 for each genotype/treatment, and $P$ values by two-way RM ANOVA, followed by Tukey, compared blood glucose or insulin levels between WT + Met group and PEN2-LKO + Met group at the same time point; see also inset for AUC values, $P$ values by two-way ANOVA, followed by Tukey), insulin tolerance test (**h**, ITT, results are shown as

mean ± s.e.m., $n$ = 6 for each genotype/treatment, and $P$ value by two-way RM ANOVA, followed by Tukey, compared as in **c**; see also inset for AUC values, $P$ value by two-way ANOVA, followed by Tukey). **i, p**, The PEN2-ATP6AP1 axis is required for AMPK activation by metformin in primary hepatocytes from HFD-fed mice. *PEN2*-LKO mice were generated as in Fig. 4c (**i**), and ATP6AP1 or ATP6AP1$^{Δ420\text{-}440}$ mutant was reintroduced to in the liver of mice lacking hepatic *ATP6AP1* as in Fig. 4e (**p**), and the resultant mice were fed with HFD for 35 weeks. Primary hepatocytes were then isolated, and treated with 5 μM metformin or 500 μM metformin for 2 h, and then subjected to analysis p-AMPKα and p-ACC levels by immunoblotting. **j, q**, The PEN2-ATP6AP1 axis is required for AMPK activation in the liver of HFD-fed mice. Mice were treated as in Fig. 4c (**j**) or 4e (**q**). Hepatic AMPK activation (immunoblots), AMP:ATP and ADP:ATP ratios (scatter plots, left panel), as well as the absolute concentrations of AMP, ADP and ATP [scatter plots, middle panel, shown as mean ± s.e.m., $n$ = 5 (**j**) and $n$ = 5 (**q**) for each genotype/treatment, except for ischemic treatment, n = 4; $P$ value by one-way ANOVA, followed by Tukey], and metformin concentrations [scatter plots, right panel, shown as mean ± s.e.m. on right panel, $n$ = 6 (**j**) and $n$ = 5 (**q**) for each genotype, and $P$ value by two-sided Student's $t$-test] in mice after 1-week treatment of metformin (1 g/l in drinking water) are shown. **k, r**, The PEN2-ATP6AP1 axis is required for the reduction of hepatic fat by metformin. Mice were fed with HFD as in Fig. 4c (**k**) or 4e (**r**), and images from H&E staining of the liver in mice after 16-week treatment of metformin are shown. **m**, Depletion of ATP6AP1, and re-introduction of ATP6AP1 or ATP6AP1$^{Δ420\text{-}440}$ in liver in ATP6AP1-LKO mice. PCR analysis results of mouse offsprings carrying floxed *ATP6AP1*, as well as *albERT2*-Cre are shown on the upper panel. Protein levels of endogenous ATP6AP1, as well as full length (FL) ATP6AP1, or ATP6AP1$^{Δ420\text{-}440}$ mutant expressed via AAVs, are shown on the middle panel. Experimental timeline of the analysis of the phenotypes of liver-specific ATP6AP1 (FL) and ATP6AP1$^{Δ420\text{-}440}$-expressing mice were shown on the lower panel. **n, o**, ATP6AP1 is required for metformin-induced reduction of hepatic fat. As depicted in **m**, mice at 38 weeks old with hepatic depletion of ATP6AP1, and re-introduction of ATP6AP1 or ATP6AP1$^{Δ420\text{-}440}$ (FL or Δ420-440) were treated with metformin in drinking water (+Met, 1 g/l) for 16 weeks. At week 54, mice were starved for 6 h, followed by intraperitoneal glucose tolerance test (**n**, serum insulin levels are shown as mean ± s.e.m., $n$ = 5 for each genotype/treatment, and $P$ value by two-way RM ANOVA, followed by Tukey, compared insulin levels between ATP6AP1 (FL) + Met group and ATP6AP1$^{Δ420\text{-}440}$ + Met group at the same time point; see also inset for AUC values, $P$ value by two-way ANOVA, followed by Tukey), insulin tolerance test (**o**, shown as mean ± s.e.m., $n$ = 6 for each genotype/treatment, and $P$ value by two-way RM ANOVA, followed by Tukey, compared as in **n**; see also inset for AUC values, and $P$ value by two-way ANOVA, followed by Tukey). Experiments in this figure were performed three times, except **i** four times.

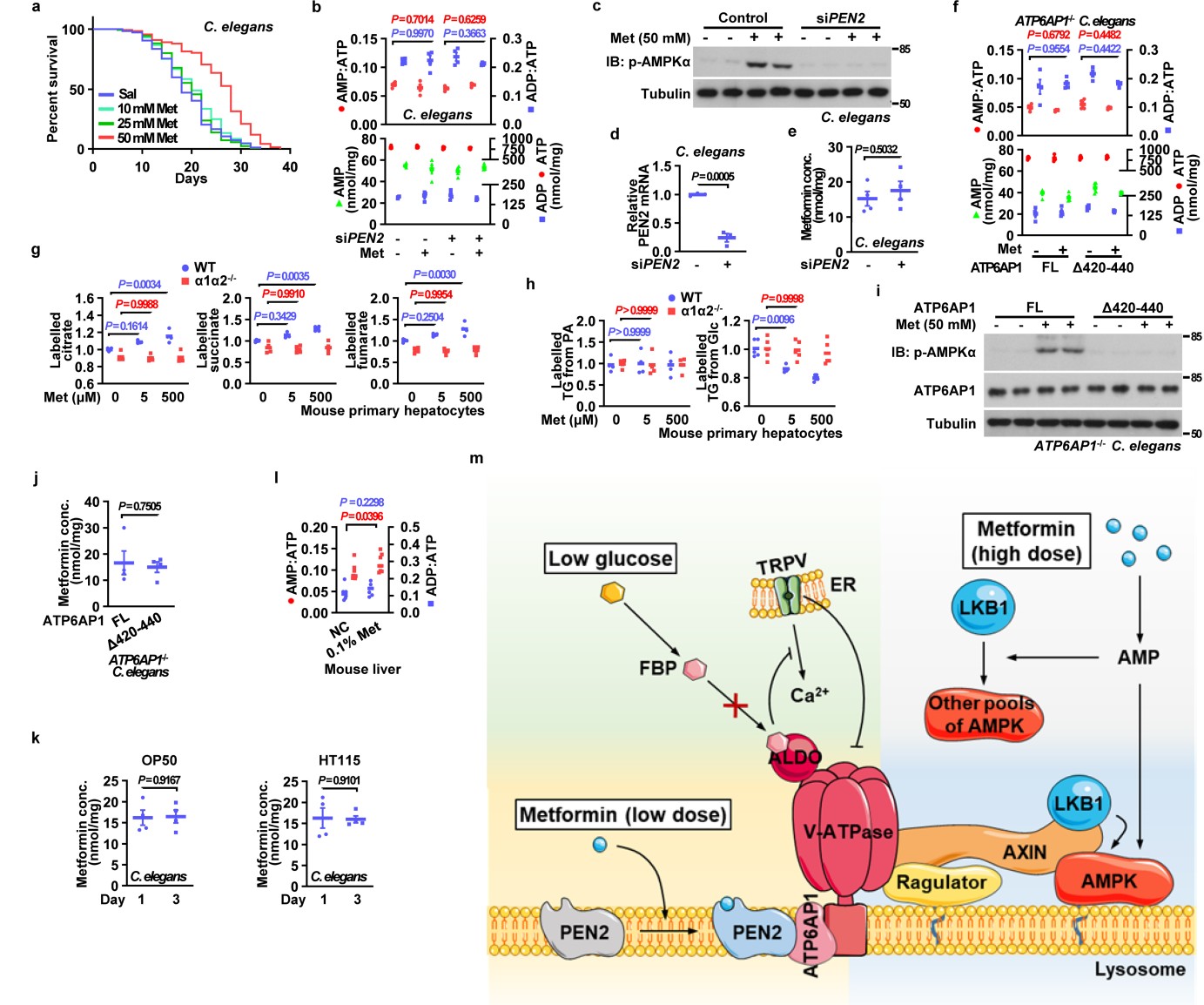

**Extended Data Fig. 13** | See next page for caption.

**Extended Data Fig. 13 | PEN2-ATP6AP1 axis is required for metformin-mediated lifespan extension in nematodes. a**, Metformin extends lifespan of *C. elegans*. Wildtype nematodes (N2) were cultured on NGM plates containing metformin at different concentrations. Lifespans were determined, and results are shown as Kaplan-Meier curves, see also statistical analysis data on Supplementary Table 3. **b**, **d**, Metformin at 50 mM does not elevate AMP/ADP levels in *C. elegans* with knockdown of *PEN2*. Nematodes were knocked down of *PEN2* (**b**, generated as in Fig. 4g by culturing on RNAi plates containing *E. coli* expressing siRNA against *PEN2*; the knockdown efficiency was assessed by determining mRNA levels of *PEN2* as shown in **d**; results are mean ± s.e.m., *n* = 3 for each genotype; and *P* value by two-sided Student's *t*-test), and were maintained on RNAi plates containing 50 mM metformin for 2 more days. Worms were then subjected to the analysis of AMP:ATP and ADP:ATP ratios. Data are shown as mean ± s.e.m., *n* = 4 for each genotype/treatment, and *P* value by two-way ANOVA, followed by Tukey. **c**, PEN2 is required for metformin-induced lifespan extension in C. elegans. Nematodes were generated as in Fig. 4g and AMPK activation after 2-day treatment of metformin, were determined. **e**, **j**, Metformin uptake is not altered in nematodes lacking PEN2 or ATP6AP1. L4 larvae of nematodes with *PEN2* knockdown (**e**, generated as in Fig. 4g), or of nematodes with knockout of *ATP6AP1* and stable expression of full length ATP6AP1 or ATP6AP1$^{\Delta420\text{-}440}$ mutant (**j**, generated as in Fig. 4h), were cultured on RNAi plates (**e**) or NGM plates (**j**) containing 50 mM metformin, for 2 days. Metformin concentrations were determined by HPLC-MS, and normalised to protein concentrations. Data are shown as mean ± s.e.m., *n* = 4 for each genotype; and *P* value by two-sided Student's *t*-test. **f**, Metformin at 50 mM does not elevate AMP/ADP levels in *ATP6AP1$^{-/-}$ C. elegans* re-introduced with ATP6AP1$^{\Delta420\text{-}440}$. Nematodes were knocked out of *ATP6AP1* (*vha-19*), then re-introduced with ATP6AP1$^{\Delta420\text{-}440}$ or ATP6AP1 as a control, as shown in Fig. 4h, and then cultured on NGM plates containing 50 mM metformin for 2 more days before subjecting to analysis of AMP:ATP and ADP:ATP ratios. Data are shown as mean ± s.e.m., *n* = 4 for each genotype/treatment; and *P* value by two-way ANOVA, followed by Tukey. **g**, Low metformin fails to promote fatty acid β-oxidation. *AMPKα1/2$^{-/-}$* mouse primary hepatocytes were treated with 100 μM BSA-conjugated [U-$^{13}$C]-palmitic acid and 5 μM or 500 μM metformin for 4 h. Relative contents of labelled citrate, succinate and fumarate were determined by GC-MS. Data are shown as mean ± s.e.m., *n* = 4 for each genotype/treatment;

and *P* value by two-way ANOVA, followed by Tukey. **h**, Low metformin inhibits de novo lipogenesis, but does not affect TAG synthesis from fatty acids. Mouse primary hepatocytes were isolated as in **g**, and were treated with 100 μM BSA-conjugated [U-$^{13}$C]-palmitic acid (left panel) and 25 mM [U-$^{13}$C]-glucose (right panel) for 12 h for determining the TAG synthesis from fatty acids and de novo lipogenesis, respectively. Metformin at 5 μM or 500 μM was added to the culture medium along with labelled palmitic acid or glucose. Relative contents of labelled TAG were determined by LC-MS. Data are shown as mean ± s.e.m., *n* = 4 for the determination of TAG synthesis and *n* = 5 for the determination of de novo lipogenesis; and *P* value by two-way ANOVA, followed by Tukey. **i**, ATP6AP1 is required for metformin-induced lifespan extension in *C. elegans*. Nematodes were treated as in Fig. 4h, AMPK activation after 2-day treatment of metformin, were determined. **k**, The presence of living bacteria on the culture plates does not affect metformin uptake by nematodes. The L4 larvae of N2 were cultured on NGM plates (left panel) or RNAi plates (right panel) containing 50 mM metformin. Metformin concentrations of nematodes at day 1 and day 3 (before culture plate refreshing) were measured by HPLC-MS. Data are shown as mean ± s.e.m., *n* = 4 for each treatment, and *P* value by two-sided Student's *t*-test. **l**, No changes in AMP:ATP and ADP:ATP ratios in the liver from mice fed with 0.1% metformin in diet. Mice at 4 weeks old were fed with normal chow (NC) diet, or normal chow diet containing 0.1% metformin (This way of giving metformin has been shown to be effective in extending the lifespan of mice[10]) for a week. Hepatic AMP:ATP and ADP:ATP ratios were then determined. Results are shown as mean ± s.e.m., *n* = 9 for normal chow or *n* = 7 for 0.1% metformin; and *P* value by two-sided Student's *t*-test. **m**, Schematic diagramme showing that the FBP-unoccupied aldolase-triggered glucose sensing and the PEN2-mediated metformin signalling axes converge at the v-ATPase complex to trigger AMPK activation. In response to lowering glucose levels, aldolase inhibits the cation channel TRPV, the latter of which then disrupts the association of the former with, and inhibits, v-ATPase. Metformin at low concentrations binds to PEN2, and metformin-bound PEN2 is recruited to v-ATPase via interacting with ATP6AP1, thereby inhibiting the activity of v-ATPase. Also shown is that high concentrations of metformin activates AMPK via elevating cellular AMP levels. Experiments in this figure were performed three times, except **l** five times.

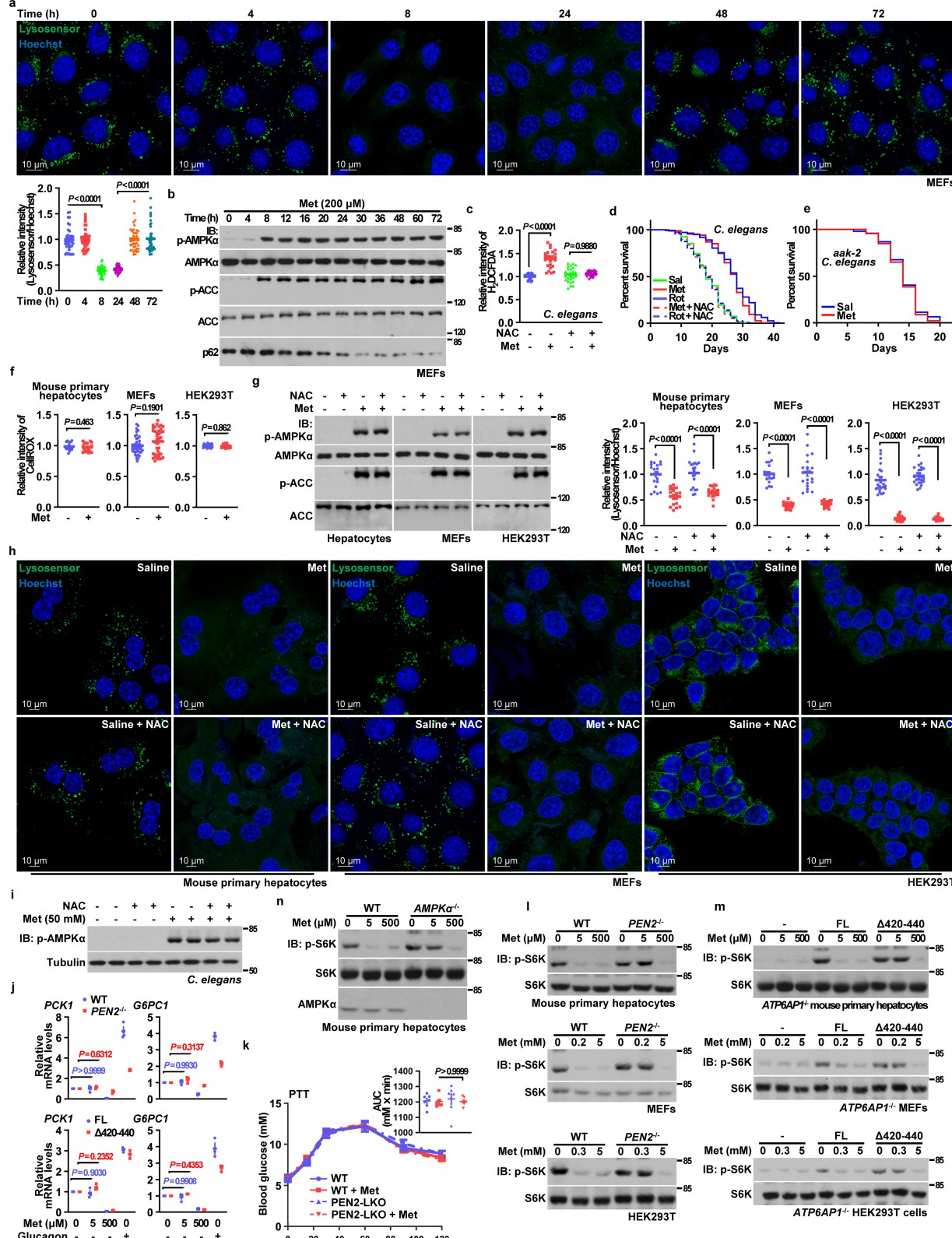

**Extended Data Fig. 14 |** See next page for caption.

**Extended Data Fig. 14 | Autophagy and ROS elevation are downstream events of AMPK, and are not involved in metformin-induced AMPK activation. a**, **b**, Autophagy occurs in MEFs treated with metformin. MEFs were treated with 200 μM metformin for indicated durations. Protein levels of the autophagy reporter p62 and the phosphorylation levels of AMPK and ACC (**b**), as well as the lysosomal pH (**a**), were determined by immunoblotting using the indicated antibodies (**b**) or by visualisation with Lysosensor dye (**a**). Results of **a** (relative intensities of Lysosensor, processed as in Fig. 1a) are shown as mean ± s.e.m., $n = 40$ cells from 4 dishes/experiment for 0 h, and $n = 40$ cells for 4 h, 24 h and 48 h, $n = 41$ cells for 8 h, and $n = 42$ cells for 72 h, all from 6 dishes/experiments; and $P$ value by one-way ANOVA, followed by Dunn. **c**, **f**, Metformin elevates ROS levels in *C. elegans*, but not in MEFs, HEK293T cells and mouse primary hepatocytes. In **c**, L4 larvae of N2 were cultured on NGM plates containing 50 mM metformin, 5 mM NAC for one day, and were then stained with 10 μM CM-H$_2$DCFDA followed by determination of ROS by imaging. Data are shown as mean ± s.e.m., $n = 30$ worms for control (untreated) condition or $n = 25$ worms for other conditions; and $P$ value by two-way ANOVA, followed by Tukey (**c**). In **f**, MEFs and HEK293T cells were treated with 200 or 300 μM metformin for 12 h, and mouse primary hepatocytes were treated with 5 μM metformin for 2 h. Cells were stained with 5 μM CellROX Deep Red, followed by determination of ROS by imaging. Data are shown as mean ± s.e.m., $n = 20$ (for hepatocytes and HEK293T cells) or 40 (for MEFs) cells from 4 dishes/experiment; and $P$ value by two-sided Student's *t*-test (hepatocytes) or two-sided Mann-Whitney test (MEFs and HEK293T cells). **d**, **e**, ROS and AMPK are both required for the metformin-mediated lifespan extension of *C. elegans*. L4 larvae of N2 or AMPK-deficient (*aak-2*) strain were cultured on NGM plates containing 50 mM metformin, 5 mM NAC (in **d**, for inhibiting ROS), or 0.1 μM rotenone (in **d**, as a positive control for ROS induction). Lifespans were determined, and results are shown as Kaplan-Meier curves, see also statistical analysis data on Supplementary Table 3. **g**, **i**, NAC does not affect metformin-induced AMPK activation. Mouse primary hepatocytes, MEFs, HEK293T cells (**g**), or nematodes (**i**) were pre-treated with 5 mM NAC for 24 h, followed by treating with 5 μM metformin for 2 h, 200 μM metformin for 12 h, 300 μM metformin for 12 h, or 50 mM metformin for 2 days. Levels of p-AMPKα and p-ACC were then determined by immunoblotting. **h**, NAC does not affect metformin-induced v-ATPase inhibition. Primary hepatocytes, MEFs, or

HEK293T cells were treated as in **g**, followed by analysis of lysosomal pH with the Lysosensor dye. Data (relative intensity of Lysosensor, processed as in Fig. 1a) were graphed as mean ± s.e.m., $n = 20$ cells from 3 dishes/experiment (for hepatocytes), $n = 20$ cells from 3 dishes/experiment (for MEFs), or $n = 24$ cells for normal condition and normal condition with NAC, and $n = 25$ for metformin treatment and metformin plus NAC treatment, all from 4 dishes/experiment (for HEK293T cells). $P$ value within each treatment was determined by two-way ANOVA, followed by Tukey. **j**, Low metformin does not alter the expression of gluconeogenic genes. Mouse primary hepatocytes isolated from PEN2-LKO mice or its wildtype littermates, or ATP6AP1-LKO mice re-introduced full length ATP6AP1 or ATP6AP1$^{Δ420-440}$ mutant were treated with 5 μM metformin, 500 μM metformin (high dose, as a control), or 100 nM of glucagon (as a positive control for elevation of the expression of gluconeogenic genes) for 2 h, followed by analysis of mRNA levels of *PCK1* and *G6PC1* by real-time PCR. Data are shown as mean ± s.e.m.; $n = 3$ for each genotype/treatment, and $P$ value by two-way ANOVA, followed by Tukey. **k**, Hepatic PEN2 is not required for the inhibition of gluconeogenesis by metformin. Mice at 5 weeks old with hepatic *PEN2* knocked out, along with its wildtype littermates, were treated with metformin in drinking water (1 g/l) for another 7 days (tamoxifen was injected at 4 weeks old). At the day 8, mice were starved for 16 h, followed by pyruvate tolerance test. Results are shown as mean ± s.e.m., $n = 7$ for each genotype/treatment, and $P$ value by two-way RM ANOVA followed by Tukey; see also inset for AUC values, $P$ value by two-way ANOVA followed by Tukey. **l**, **m**, The PEN2-ATP6AP1 axis is required for the inhibition of mTORC1 by low metformin. MEFs, HEK293T cells and mouse primary hepatocytes with *PEN2* knockout (**l**), *ATP6AP1* knockout and with re-introduced full length ATP6AP1 or ATP6AP1$^{Δ420-440}$ (**m**), were treated with 200 μM and 5 mM metformin (for MEFs), 300 μM and 5 mM metformin (for HEK293T cells) for 12 h, 5 μM and 500 μM metformin (for mouse primary hepatocytes). Cells were then lysed, and the activity of mTORC1 was determined by immunoblotting the phosphorylation levels of its substrate S6K (p-S6K). **n**, AMPK is required for the inhibition of mTORC1 by low metformin. *AMPKα1/2*$^{-/-}$ mouse primary hepatocytes were treated with 5 μM and 500 μM metformin for 2 h. Cells were then lysed, and the activity of mTORC1 was determined by immunoblotting the phosphorylation levels of its substrate S6K (p-S6K). Experiments in this figure were performed three times, except **a**, **j**, **l**, **m** four times.

# Reporting Summary

## Statistics

For all statistical analyses, confirm that the following items are present in the figure legend, table legend, main text, or Methods section.

| n/a | Confirmed | |
|---|---|---|
| ☐ | ☒ | The exact sample size (*n*) for each experimental group/condition, given as a discrete number and unit of measurement |
| ☐ | ☒ | A statement on whether measurements were taken from distinct samples or whether the same sample was measured repeatedly |
| ☐ | ☒ | The statistical test(s) used AND whether they are one- or two-sided<br>*Only common tests should be described solely by name; describe more complex techniques in the Methods section.* |
| ☐ | ☒ | A description of all covariates tested |
| ☐ | ☒ | A description of any assumptions or corrections, such as tests of normality and adjustment for multiple comparisons |
| ☐ | ☒ | A full description of the statistical parameters including central tendency (e.g. means) or other basic estimates (e.g. regression coefficient) AND variation (e.g. standard deviation) or associated estimates of uncertainty (e.g. confidence intervals) |
| ☐ | ☒ | For null hypothesis testing, the test statistic (e.g. *F*, *t*, *r*) with confidence intervals, effect sizes, degrees of freedom and *P* value noted<br>*Give P values as exact values whenever suitable.* |
| ☒ | ☐ | For Bayesian analysis, information on the choice of priors and Markov chain Monte Carlo settings |
| ☒ | ☐ | For hierarchical and complex designs, identification of the appropriate level for tests and full reporting of outcomes |
| ☒ | ☐ | Estimates of effect sizes (e.g. Cohen's *d*, Pearson's *r*), indicating how they were calculated |

*Our web collection on statistics for biologists contains articles on many of the points above.*

## Software and code

Policy information about availability of computer code

**Data collection**

EPSON Scan 3.9.3.4 was used to scan blots from X-ray films.
Zen 2012, LAS X (version 3.0.2.16120), NIS Elements software with STORM package (version 4.30 build 1053), and Hitachi TEM system control 01.20 were used to collect microscopic images, as described in related method sections.
VPViewer 2000 (version 2.66.7) was used to collect DSC data, ITC200 (version 1.26.1) to ITC, and BIAcore T200 Control software (version 2.0) to BIAcore.
OCR results were collected by Wave 2.6.1.
MassLynx 4.1 was used to monitor and collect data from preparative HPLC.
TopSpin (version 3.2) was used to collect NMR data, and Tune (version 2.1) to HRMS.
MassHunter LC/MS acquisition 10.1.48 was used to collect data of adenylate concentrations on CE-MS.
Analyst TF 1.6 was used to collect data of TAG levels on HPLC-MS, Analyst 1.6.3 for metformin, and Analyst 1.7.1 for adenylate concentrations on HPLC-MS.
MassHunter GC/MS acquisition B.07.04.2260 was used to collect data from GC-MS.

**Data analysis**

The band intensities of immunoblots and the intensity of CM-H2DCFDA in nematodes intestine were quantified using Image J software (version 1.8.0).
Plots were generated by Prism v9.
Statistical analysis was performed by Prism v9 and SPSS 27.0.
Microscopic images were analysed and processed by Zen 2012, Imaris 7.4.0, NIS Elements software with STORM package version 4.30 build 1053, and Hitachi TEM system control 01.20 as described in related Methods section, and were formatted on Photoshop 2021.
Results of metabolites measured on CE-MS were analysed on Qualitative Analysis B.06.00, and HPLC-MS on MultiQuant 3.0.3 (for adenylates) or MultiQuant 3.0.3 (for metformin).
Protein and peptide mass spectrometry data were analysed using Peaks Studio (version X+, for timsTOF Pro), ProteinPilot software (version

5.0, for TripleTOF 5600+), or Proteome Discoverer (version 2.2, for Orbitrap Fusion Lumos Tribrid).
RT-PCR results were analysed by Bio-Rad CFX Manager 3.1 (for C. elegans) or Roche LightCycler 96 1.1 (for hepatocytes) .
Preparative HPLC data were analysed by EasyChrom 2.0.
NMR data were analysed on MestReNova 9.0, and on HRMS by Xcalibur 2.2.
DSC and ITC results were analysed by Origin 7 (version 7.0552), and SPR results on BIAcore T200 Evaluation software (version 2.0).
OCR results were analysed by Wave 2.6.1.
Results of TAG measured on HPLC-MS were analysed by MS-DIAL 4.7, and TCA cycle intermediates by GC-MS MassHunter Workstation Software (version B.07.01SP1, Qualitative Analysis).

For manuscripts utilizing custom algorithms or software that are central to the research but not yet described in published literature, software must be made available to editors and reviewers. We strongly encourage code deposition in a community repository (e.g. GitHub). See the Nature Portfolio guidelines for submitting code & software for further information.

## Data

Policy information about availability of data

All manuscripts must include a data availability statement. This statement should provide the following information, where applicable:

- Accession codes, unique identifiers, or web links for publicly available datasets
- A description of any restrictions on data availability
- For clinical datasets or third party data, please ensure that the statement adheres to our policy

The data supporting the findings of this study are available within the paper and its Supplementary Information files. The proteomics data have been deposited in the iProX integrated proteome resources (IPX0003776000). Materials, reagents or other experimental data are available upon reasonable request from the corresponding author. Full immunoblots are provided as Supplementary Information Fig. 1. Source data for all graphs are provided with this paper.
The analysis was performed using standard protocols with previously described computational tools. No custom code was used in this study.

# Field-specific reporting

Please select the one below that is the best fit for your research. If you are not sure, read the appropriate sections before making your selection.

[×] Life sciences      [ ] Behavioural & social sciences      [ ] Ecological, evolutionary & environmental sciences

For a reference copy of the document with all sections, see nature.com/documents/nr-reporting-summary-flat.pdf

# Life sciences study design

All studies must disclose on these points even when the disclosure is negative.

| | |
|---|---|
| Sample size | The chosen sample sizes were similar to those previously used by us and others in this field: n = 4-6 human participants or mice were used to determine the pharmacokinetics of metformin (ref. 16,31,32); n = 5-7 mice were usually used to determine the effects of metformin on blood glucose (ref. 33,34) and fatty liver (ref. 20,35); n = 100-300 worms were used to determine lifespan (ref. 36,37); n = 20-42 cells from 2 - 6 experiments/dishes were included when conclusions were based on immunofluorescent staining (ref. 6,25); n = 3-5 samples were used for evaluation of the levels of AMP, ADP and ATP in cells and tissues (ref. 6,15,25,27); n = 3 samples to determine the expression levels and phosphorylation levels of a specific protein (ref. 24,38); n = 3 samples to determine the mRNA levels of a specific gene (ref. 20,24,39); and n = 3 samples to determine the activity of v-ATPase in vitro (ref. 25). No statistical methods were used to predetermine sample size. |
| Data exclusions | No data was excluded. |
| Replication | All experimental findings were repeated at least three times as stated in figure legends. |
| Randomization | Randomisation was applied wherever possible. For example, during MS analyses, samples were processed and subjected to the mass spectrometer in random orders. In animal experiments, sex-matched (only for mice), age-matched litter-mate mice in each genotype were randomly assigned to pharmacological or diet treatments. In cell experiments, cells of each genotype were parallel seeded and randomly assigned to different treatments. Otherwise, randomisation was not performed. For example, when performing immunoblotting, samples needed to be loaded in a specific order to generate the final figures. |
| Blinding | Blinding was applied wherever possible. For example, samples, cages or agar plates during sample collection and processing were labelled as code names that were later revealed by the individual who picked and treated animals or cells, but did not participate in sample collection and processing, until assessing outcome. Similarly, during microscopy data collection and statistical analyses, the fields of view were chosen on a random basis, and are often performed by different operators, preventing potentially biased selection for desired phenotypes. Otherwise, blinding was not performed, such as the measurement of v-ATPase activity, and the determination of metformin binding to PEN2 in vitro, where operators had to know the conditions of each well and added reagents to the well accordingly during the measurement. |

# Reporting for specific materials, systems and methods

We require information from authors about some types of materials, experimental systems and methods used in many studies. Here, indicate whether each material, system or method listed is relevant to your study. If you are not sure if a list item applies to your research, read the appropriate section before selecting a response.

## Materials & experimental systems

| n/a | Involved in the study |
|---|---|
| ☐ | ☒ Antibodies |
| ☐ | ☒ Eukaryotic cell lines |
| ☒ | ☐ Palaeontology and archaeology |
| ☐ | ☒ Animals and other organisms |
| ☐ | ☒ Human research participants |
| ☐ | ☒ Clinical data |
| ☒ | ☐ Dual use research of concern |

## Methods

| n/a | Involved in the study |
|---|---|
| ☒ | ☐ ChIP-seq |
| ☒ | ☐ Flow cytometry |
| ☒ | ☐ MRI-based neuroimaging |

# Antibodies

**Antibodies used**

Rabbit polyclonal antibody against LAMTOR1 was raised and validated as described previously (ref. 24), and was diluted 1:100 for immunoprecipitation (IP) or 1:500 for immunoblotting (IB). Rabbit polyclonal antibody against ATP6AP1 was raised with bacterially expressed and purified ATP6AP1 (aa 440-470, GST-tagged), and was diluted 1:100 for IP. Rabbit anti-phospho-AMPKα-T172 (cat. #2535, 1:1,000 for IB), anti-AMPKα (cat. #2532, 1:1,000 for IB), anti-AMPKβ1/2 (cat. #4150, 1:1,000 for IB), anti-phospho-ACC-Ser79 (cat. #3661, 1:1,000 for IB), anti-ACC (cat. #3662, 1:1,000 for IB), anti-phospho-p70 S6K-S389 (cat. #9234, 1:1000 for IB), anti-p70 S6K (cat. #2708, 1:1000 for IB), anti-LKB1 (cat. #3047, 1:1,000 for IB), anti-AXIN1 (cat. #2074, 1:1,000 for IB), anti-presenilin 1 (cat. #3622, 1:1,000 for IB), anti-presenilin 2 (cat. #2192, 1:1,000 for IB), anti-nicastrin (cat. #3632, 1:1,000 for IB), anti-β-tubulin (cat. #2128, 1:1,000 for IB), anti-HA-tag [(cat. #3724, 1:1,000 for IB or 1:120 for immunofluorescent staining (IF)], anti-PDI (cat. #3501, 1:1,000 for IB), anti- cytochrome c (cat. #4280, 1:1,000 for IB), anti-clathrin (cat. #4796, 1:1,000 for IB), anti-p62 (cat. #23214, 1:1,000 for IB), anti-ATG5 (cat. #12994, 1:1,000 for IB), anti-PDI (Alexa Fluor 488-conjugated, cat. #5051, 1:60 for IF), HRP-conjugated mouse anti-rabbit IgG (conformation specific, cat. #5127, 1:2,000 for IB), HRP-conjugated goat anti-rat IgG (conformation specific, cat. #98164, 1:2,000 for IB) and mouse anti-Myc-tag (cat. #2276, 1:500 for IB) antibodies were purchased from Cell Signaling Technology. Rabbit anti-ATP6v0c (cat. NBP1-59654, 1:1,000 for IB or 1:100 for IP) antibody was purchased from Novus Biologicals. Mouse ANTI-FLAG® M2 (cat. F1804, 1:1,000 for IB), goat anti-rabbit IgG antibody and ANTI-FLAG® M2 Affinity Gel (cat. A2220, 1:500 for IP) were purchased from Sigma. Rabbit anti-PEN2 (cat. ab154830, 1:1,000 for IB or 1:100 for IP and IF), anti-ATP6AP1 (cat. ab176609, 1:500 for IB), anti-transferrin (cat. ab1223, 1:500 for IB), anti-TGN46 (cat. ab76282, 1:60 for IF), and rat anti-LAMP2 (cat. ab13524; 1:1000 for IB or 1:120 IF) antibodies were purchased from Abcam. Goat anti-AXIN (cat. sc-8567, 1:120 for IF), mouse anti-HA (cat. sc-7392, 1:1,000 for IB, 1:500 for IP or 1:120 for IF), mouse anti-goat IgG-HRP antibody were purchased from Santa Cruz Biotechnology. Normal rabbit control IgG (cat. CR1, 1:100 for IP) was purchased from Sino Biological. Goat anti-mouse IgG (cat. 115-035-003, 1:1,000 for IB) and anti-rabbit (cat. 111-035-003, 1:1,000 for IB) antibodies were purchased from Jackson ImmunoResearch. Donkey anti-goat IgG (cat. #A-11055, 1:1000 for IB), anti-rabbit IgG (cat. #A-21206, 1:1000 for IB), anti-rat IgG (cat. #A21209, 1:1,000 for IB), goat anti-rat IgG (cat. #A-21247, 1:1,000 for IB), rabbit anti-APH1 (cat. #PA1-2010, 1:1,000 for IB), mouse anti-Strep-tag (cat. #MA5-17283, 1:1,000 for IB) antibodies were purchased from Thermo Scientific. Rabbit anti-ATP1A1 (cat. #14418-1-AP, 1:60 for IF) and anti-TOMM20 (cat. #11802-1-AP, 1:60 for IF) antibodies were purchased from Proteintech.

**Validation**

Rabbit polyclonal antibody against LAMTOR1 was validated as described previously (ref. 6). The in-house generated rabbit polyclonal antibody against ATP6AP1 (for IP), as well as the rabbit anti-ATP6AP1 (cat. ab176609, validated by the manufacturer: https://www.abcam.com/atp6ap1atp6s1-antibody-ab176609.html) purchased from Abcam was validated in this study using ATP6AP1 knockout cell lines/mouse strains. Rabbit anti-PEN2 (cat. ab154830) was validated by the manufacturer: https://www.abcam.com/pen2-antibody-epr9200-ab154830.html, and also in this study using PEN2 knockout cell lines/mouse strains.

The following commercially available antibodies were validated by the company, as well as other researchers (as the information collected by the RRID database):
Rabbit anti-phospho-AMPKα-T172 (cat. #2535, RRID: AB_331250), anti-AMPKα (cat. #2532, RRID: AB_330331), anti-AMPKβ1/2 (cat. #4150, RRID: AB_10828832), anti-phospho-ACC-Ser79 (cat. #3661, RRID: AB_330337), anti-ACC (cat. #3662, RRID: AB_2219400), anti-phospho-p70 S6K-S389 (cat. #9234, AB_2269803), anti-p70 S6K (cat. #2708, RRID:AB_390722), anti-LKB1 (cat. #3047, RRID: AB_2198327), anti-AXIN1 (cat. #2074, RRID: AB_2062419), anti-Presenilin 1 (cat. #3622, RRID: AB_2172895), anti-Presenilin 2 (cat. #2192, RRID: AB_2170609), anti-Nicastrin (cat. #3632, RRID: AB_2149581), anti-β-tubulin (cat. #2128, RRID: AB_823664), anti-HA-tag (cat. #3724, RRID: AB_1549585), anti-PDI (cat. #3501, RRID: AB_2156433), anti- Cytochrome c (cat. #4280, RRID: AB_10695410), anti-Clathrin (cat. #4796, RRID: AB_10828486), anti-p62 (cat. #23214, RRID:AB_2798858), anti-ATG5 (cat. #12994, RRID:AB_2630393), anti-PDI (Alexa Fluor 488-conjugated, cat. #5051, RRID:AB_10950503),and mouse anti-Myc-tag (cat. #2276, RRID: AB_331783) by Cell Signaling Technology; rabbit anti-ATP6v0c (cat. NBP1-59654, RRID: AB_11004830) antibody by Novus Biologicals; mouse ANTI-FLAG® M2 (cat. F1804, RRID: AB_262044), ANTI-FLAG® M2 Affinity Gel (cat. A2220, RRID: AB_10063035), and Atto 488 goat anti-rabbit IgG (cat. 18772, RRID: AB_1137637) antibodies by Sigma; rabbit anti-Transferrin (cat. Ab1223, RRID: AB_298951), anti-TGN46 (cat. ab76282, RRID:AB_1524486), and rat anti-LAMP2 (cat. ab13524; RRID: AB_369111) antibodies by Abcam; goat anti-AXIN (cat. sc-8567, RRID: AB_2227789), mouse anti-HA (cat. sc-7392, RRID: AB_627809) by Santa Cruz Biotechnology; Alexa Fluor 488 donkey anti-goat IgG (cat. #A-11055, RRID: AB_2534102), Alexa Fluor 488 donkey anti-rabbit IgG (cat. #A-21206, RRID: AB_2535792), Alexa Fluor 594 donkey anti-rat IgG (cat. #A21209, RRID: AB_2535795), Alexa-Fluor 647 goat anti-rat IgG (cat. #A-21247, RRID: AB_141778), rabbit anti-APH1 (cat. #PA1-2010, RRID: AB_2227105), mouse anti-Strep-tag (cat. #MA5-17283, RRID: AB_2538749) antibodies by Thermo; Rabbit anti-ATP1A1 (cat. #14418-1-AP, RRID:AB_2227873) and anti-TOMM20 (cat. #11802-1-AP, RRID:AB_2207530) antibodies by Proteintech.

# Eukaryotic cell lines

Policy information about cell lines

**Cell line source(s)**

HEK293T, and L929 (NCTC clone 929) cells were purchased from ATCC. AD293 (Adeno-X 293) cells were purchased from

Clontech. MEFs and primary hepatocytes were obtained from the indicated mouse strains. Suspension HEK293T cell line was established from the HEK293T cells and a generous gift from Dr. Zhen Wang from Xiamen Immocell Biotechnology Co., Ltd.

| Authentication | HEK293T cells were obtained from and pre-authenticated by ATCC by STR sequencing and used at low passages. L929 cells and AD293 (Adeno-X 293) cells were obtained from and pre-authenticated by ATCC and Clontech, respectively. Although no detailed authentication information was provided by the suppliers, both strains were used at very low passages in this study (i.e., packaging virus for infecting other cell lines/mice, and examining the efficiencies of sgRNAs used for generating KO mice), and no phenotype was analysed from these two cell lines. |
|---|---|
| Mycoplasma contamination | The cell lines were routinely tested negative for mycoplasma contamination in our lab. |
| Commonly misidentified lines (See ICLAC register) | No commonly misidentified lines were used. |

## Animals and other organisms

Policy information about studies involving animals; ARRIVE guidelines recommended for reporting animal research

| Laboratory animals | Mice were housed with free access to water and standard diet (65% carbohydrate, 11% fat, 24% protein) under specific pathogen-free (SPF) conditions. The light was on from 8 a.m. to 8 p.m., with temperature kept at 21-24 °C, and humidity at 40-70%. Male littermate controls were used throughout the study. Metformin was supplied either in drinking water at desired concentrations, or in standard diet at 0.1% (w/w) for 1 week. For creating the diabetic mouse model, mice were fed with HFD (high fat diet, with 60% calories from fat, D12492, Research Diets) for desired time periods starting at 4 weeks old.

AXIN-floxed, LAMTOR1-floxed, AXIN-LKO, and LAMTOR1-LKO mice were generated and maintained as described previously (ref. 24). Wildtype C57BL/6 mice (#000664), DBA2 mice (#000671), and ROSA26-FLPe mice (#016226) were obtained from The Jackson Laboratory. ICR mice (#N000294) were obtained from GemPharmatech. TRPV1-KO mice were obtained from The Jackson Laboratory provided by Dr. David Julius (#003770). TRPV1-KO mice with knockdown of TRPV2-4 or GFP were generated as described previously (ref. 25). APH1A-floxed/APH1B-KO/APH1C-floxed mice were obtained from The Jackson Laboratory provided by Dr. Bart De Strooper (#030985). ATG5-floxed mice were obtained from RIKEN, provided by Dr. Noboru Mizushima (BRC No. RBRC02975). AMPKα1-floxed (Jackson Laboratory, # 014141) and AMPKα2-floxed mice (Jackson Laboratory, # 014142) were obtained from Jackson Laboratory, provided by Dr. Sean Morrison. TG-ALDOA, Tg-ALDOA-D34S, PEN2-floxed and ATP6AP1-floxed mice were generated in this study.

The PEN2-floxed mice were the crossed with Alb-CreERT2 or Villin-CreERT2 mice to generate inducible liver-specific PEN2 knockout mice (PEN2-LKO) or inducible intestine-specific PEN2 knockout mice (PEN2-IKO). The ATP6AP1-floxed mice were crossed with Alb-CreERT2 to generate inducible liver-specific ATP6AP1 knockout mice (ATP6AP1-LKO). AMPKα1/2-floxed mice were crossed with Villin-CreERT2 mice to generate inducible intestine-specific AMPKα knockout mice (AMPKα-IKO). PEN2, ATP6AP1 and AMPKα were deleted by injecting intraperitoneally the mice with tamoxifen (dissolved in corn oil) at 200 mg/kg, 3 times a week. Knockout efficiencies were analysed at 1 week after the last injection by western blotting. ATP6AP1-LKO mice expressing wildtype ATP6AP1 or ATP6AP1Δ420-440 were generated by injection via the tail vein of the different adeno-associated viruses (AAV) carrying indicated inserts before knockout of the endogenous ATP6AP1 by tamoxifen. Levels of the re-introduced ATP6AP1 proteins were analysed at 4 weeks after the virus injection.

The ages of mice used were as follows: a) for isolating primary hepatocytes, normal chow diet-fed wildtype and AMPKα-floxed mice of 4 weeks old (Fig. 1a, b, Extended Data Fig. 1c, e-g, 13g, h, 14f-h, j, n), normal chow diet-fed PEN2-LKO mice of 6 weeks old (Fig. 2b, Extended Data Fig. 3c, j, 14l; into which tamoxifen was injected at 4 weeks old), HFD-fed wildtype mice of 38 weeks old (Extended Data Fig. 12i; fed with HFD for 34 weeks starting from 4 weeks old), HFD-fed PEN2-LKO mice of 38 weeks old (Extended Data Fig. 12i; into which tamoxifen was injected at 35 weeks old after 31 weeks of HFD-treatment starting from 4 weeks old), normal chow diet-fed ATP6AP1-LKO mice expressing full-length ATP6AP1 or ATP6AP1Δ420-440 of 8 weeks old (Extended Data Fig. 14m; into which AAV carrying ATP6AP1 was injected at 4 weeks old, and tamoxifen at 5 weeks old), and HFD-fed ATP6AP1-LKO mice expressing full-length ATP6AP1 or ATP6AP1Δ420-440 of 38 weeks old (Extended Data Fig. 12p; into which AAV was injected at 34 weeks old, and tamoxifen at 35 weeks old, after 34 weeks of HFD-treatment starting from 4 weeks old) were used; b) for GTT, ITT, and measurements metformin, GLP-1, insulin and TAG contents, PEN2-IKO and AMPKα1/2-IKO mice of 6 weeks old (Fig. 4a, b, Extended Data Fig. 12a, b, d, e; mice at 4 weeks old were fed with HFD for a week, and then treated with metformin for another week), HFD-fed PEN2-LKO mice of 54 weeks old (Fig. 4c, d, Extended Data Fig. 12g, h; mice at 4 weeks old were fed with HFD for 31 weeks, and then injected with tamoxifen. At 38 weeks old, mice were treated with metformin for 16 weeks), HFD-fed ATP6AP1-LKO mice expressing full-length ATP6AP1 or ATP6AP1Δ420-440 of 54 weeks old (Fig. 4e, f, Extended Data Fig. 12n, o; mice at 4 weeks old were fed with HFD for 30 weeks, and were injected with AAV at 34 weeks old. At 35 weeks old, the mice were injected with tamoxifen, and then treated with metformin for 16 weeks); c) for PTT, PEN2-LKO mice of 6 weeks old (Extended Data Fig. 14k; mice at 4 weeks old were injected with tamoxifen, and were treated with metformin for a week starting from 5 weeks old); d) for immunoblotting and the measurement of adenylates, wildtype mice of 5 weeks old (Extended Data Fig. 1h-n, 13l; treated with metformin for a week starting from 4 weeks old), PEN2-LKO mice of 6 weeks old (Fig. 4i, Extended Data Fig. 12f; mice at 4 weeks old were injected with tamoxifen, and were treated with metformin for a week starting from 5 weeks old), ATP6AP1-LKO mice expressing full-length ATP6AP1 or ATP6AP1Δ420-440 of 9 weeks old (Fig. 4j, Extended Data Fig. 12m; mice at 4 weeks old were injected with AAV. At 5 weeks old, the mice were injected with tamoxifen, and then treated with metformin for a week starting from 8 weeks old), HFD-fed PEN2-LKO mice of 39 weeks old (Extended Data Fig. 12j; mice at 4 weeks old were fed with HFD for 31 weeks, and then injected with tamoxifen. At 38 weeks old, mice were treated with metformin for a week), HFD-fed ATP6AP1-LKO mice expressing full-length ATP6AP1 or ATP6AP1Δ420-440 39 weeks old (Extended Data Fig. 12q; mice at 4 weeks old were fed with HFD for 30 weeks, and were injected with AAV at 34 weeks old. At 35 weeks old, the mice were injected with tamoxifen, and then treated with metformin for a weeks starting from 38 weeks old), AXIN-LKO mice, LAMTOR1-LKO mice and Tg-ALDOA-D34S of 7 weeks old (Extended Data Fig. 9d, e, 11c, e; treated with metformin for a week starting from 6 weeks old), hepatic ATP6v0c-KD mice of 7 weeks old (Extended Data Fig. 9f; treated with metformin 1-week starting from 6 weeks old, into which AAV carrying siRNA against ATP6v0c was intravenously injected at 4 weeks old), and TRPV1-KO and hepatic TRPV2/3/4-KD mice of 8 weeks old (Extended Data Fig. 11g; treated with metformin 1-week starting from 7 weeks old, into which AAV carrying siRNAs against TRPV2/3/4 was intravenously injected at 5 weeks old); e) for hepatic H&E staining, HFD-fed PEN2-LKO mice of 54 weeks old (Extended Data Fig. 12k; mice at 4 weeks old were fed with HFD for 31 |

weeks, and then injected with tamoxifen. At 38 weeks old, mice were treated with metformin for 16 weeks), HFD-fed ATP6AP1-LKO mice expressing full-length ATP6AP1 or ATP6AP1Δ420-440 of 54 weeks old (Extended Data Fig. 12r; mice at 4 weeks old were fed with HFD for 30 weeks, and were injected with AAV at 34 weeks old. At 35 weeks old, the mice were injected with tamoxifen, and then treated with metformin for 16 weeks); and f) for all the other experiments, mice of 4 weeks old were used.

Worms were maintained on nematode growth medium (NGM) plates [1.7% (w/v) agar, 0.3% (w/v) NaCl, 0.25% (w/v) bacteriological peptone, 1 mM CaCl2, 1 mM MgSO4, 25 mM KH2PO4-K2HPO4, pH 6.0, 0.02% (w/v) streptomycin, and 5 μg/ml cholesterol] spread with E. coli OP50 as standard food. Metformin of desired concentrations was added to the autoclaved NGM (cooled down to 50 °C) before pouring onto pates. All worms were cultured at 20 °C. Wildtype (N2 Bristol), aak-2(ok524), and unc-76(e911) strains were obtained from Caenorhabditis Genetics Center. The ATP6AP1-KO C. elegans strains expressing ATP6AP1 or its Δ420-440 mutant were established in this study.

The ages of nematodes used in this study were as follows: a) for lifespan assays, worms at L4 stage were used (Fig.4 g, h, Extended Data Fig. 13a, 14d, e; treated with metformin, NAC, etc. until death); b) for analysis of adenylates and pharmacokinetics of metformin, phospho-AMPKα, and ROS levels (Extended Data Fig. 13b, c, e, f, i-k, 14c, i), worms at L4 stage (after treatment of metformin for a day) were used; and c) for the experiments using PEN2-KD worms, worms at L1 stage were used to knocked down of PEN2 (Extended Data Fig. 13d; fed with HT115 E.coli strain containing RNAi against PEN2).

| | |
|---|---|
| Wild animals | The study did not involve wild animals |
| Field-collected samples | The study did not involve samples collected from the field. |
| Ethics oversight | Protocols for all mouse experiments were approved by the Institutional Animal Care and the Animal Committee of Xiamen University (XMULAC20180028). |

Note that full information on the approval of the study protocol must also be provided in the manuscript.

# Human research participants

Policy information about studies involving human research participants

| | |
|---|---|
| Population characteristics | Participants were administered with diane-35 (Bayer, Germany), one tablet per day, plus metformin (Bristol-Myers Squibb) at 2000 mg/day. Such a combined treatment lasted 6 months, which was then switched to metformin-alone treatment for another 6 months. At the endpoint of the treatment, six subjects with the lowest BMI were selected for determining pharmacokinetics. A potential selection bias may be introduced. However, they were characterised as follows (mean ± SD): ages 25.5 ± 6.2 years, weight 69.0 ± 6.4 kg, BMI 26.5 ± 1.1 kg/m2, waist circumference 89.8 ± 5.8 cm, hip circumference 102.2 ± 11.4 cm, W/H 0.89 ± 0.12, fasting plasma glucose 5.1 ± 0.7 mM, fasting serum insulin 28.2 ± 15.3 μU/ml, HbA1c 5.1 ± 0.2 %, total cholesterol 5.1 ± 1.0 mM, triglycerides 1.4 ± 0.7 mM, HDL-C 1.4 ± 0.3 mM, LDL-C 3.0 ± 0.6 mM, and were therefore able to represent the general population. |
| Recruitment | Obese women diagnosed with polycystic ovary syndrome (PCOS) between September 2017 and July 2020 at Shanghai Jiao Tong University Affiliated Sixth People's Hospital were recruited. Subjects included fulfilled the following criteria: a) ages 18 to 40 (inclusive); b) BMI higher than 27.5 kg/m2; and c) meeting the diagnostic criteria of PCOS. The diagnostic criteria of PCOS were as follows: a) irregular period over a year; b) hyperandrogenism; and/or c) ultrasound examination with polycystic ovaries. Subjects with following conditions were excluded: a) after hysterectomy; b) with congenital adrenal hyperplasia, Cushing's syndrome, or androgen secreting tumours within five years; c) mothers pregnant or lactating; d) with thrombosis related history or risk factors, such as deep vein thrombosis, pulmonary embolism, myocardial infarction (angina), valvular heart disease, atrial fibrillation, and cerebrovascular accident (TIA); e) with abnormal liver function (e.g., caused by viral hepatitis); f) with a history of liver malignancy or adenoma, or a history of genital or breast malignancies; g) history of severe or frequent migraine attacks; h) with renal insufficiency; and i) with other factors that may affect the efficacy of the drug or cause complications by the drug. |
| Ethics oversight | The study was approved by the Ethics Committee of Shanghai Sixth People's hospital and was in accordance with the Helsinki Declaration and Good Clinical Practice. All participants signed an informed consent form before enrollment. This study was registered on ClinicalTrials.gov (ChiCTR-IOR-17013169). |

Note that full information on the approval of the study protocol must also be provided in the manuscript.

# Clinical data

Policy information about clinical studies
All manuscripts should comply with the ICMJE guidelines for publication of clinical research and a completed CONSORT checklist must be included with all submissions.

| | |
|---|---|
| Clinical trial registration | The trial was registered with the Chinese Clinical Trial Registry (ChiCTR), with the URL link: http://www.chictr.org.cn/showproj.aspx?proj=21769. |
| Study protocol | Participants were fasted for 12 h before experiment (started from 10:00 p.m. of previous day), followed by taking orally 0.5 g of Metformin Hydrochloride Extended-release Tablets (Bristol-Myers Squibb) per person. Blood samples were taken at 1, 3, 6 and 12 h after metformin intake, followed by serum preparation.<br>Some 50 μl of serum collected from each mouse or human subject was mixed with 80% methanol (v/v) in water using buformin at 100 μg/l as an internal standard. Aqueous phase was then collected after centrifugation at 15,000g for 15 min at 4 °C. The merformin concentration was then determined by HPLC-MS. |
| Data collection | Measurement was performed on a QTRAP MS (SCIEX, QTRAP 6500+) connected to a UPLC system (SCIEX, ExionLC AD). Some 2 μl of |

each sample was loaded onto a pHILIC column (ZIC-pHILIC, 5 μm, 2.1× 100 mm, PN: 1.50462.0001, Millipore). The mobile phase consisted of 10 mM ammonium formate containing 0.1% formate (v/v) in the LC-MS grade water (mobile phase A) and LC-MS grade acetonitrile containing 0.1% (v/v) formate in LC-MS grade water (mobile phase B) run at a flow rate of 0.3 ml/min. The HPLC gradient was as follows: 95% B held for 1 min, then to 40% B in 6 min, held for 1 min, then to 95% B within 7.5 min, and held for 2.5 min. The QTRAP mass spectrometer was run on a Turbo V ion source and running in negative mode run in a spray voltage of -5,500 V, with source temperature at 500 °C, Gas No.1 at 50 psi, Gas No.2 at 55 psi, and curtain gas at 40 psi. Compounds were measured using the multiple reactions monitoring mode (MRM), and declustering potentials and collision energies were optimised through using analytical standards. The following transitions were used for monitoring each compound: 130/71 for metformin and 158.1/60 for buformin.

Outcomes

The concentration of metformin in each sample was calculated according to the standard curves generated using desired concentrations of metformin.

