## [Peer Review File · Nature]

Manuscript Title: Low-dose metformin targets the lysosome-AMPK pathway through PEN2

Reviewer Comments & Author Rebuttals

Reviewer Reports on the Initial Version:

Referee #1 (Remarks to the Author):

The paper by Ma et al. describes a diverse set of experiments looking at the mechanism of AMPK activation in response to low metformin concentrations and how this might impact on metformin-induced therapeutic benefits. Metformin is commonly used to treat type-2 diabetes and there is increasing interest in understanding its physiological effects because of its pleiotropic action (e.g., in cancer and aging). However, the exact mechanism through which it acts has yielded conflicting results. Several mechanisms by which complex I inhibition has been implicated for the metformin's glucose-lowering effect have been proposed including altered cellular energy charge, inhibition of adenylate cyclase, inhibition of fructose 1,6-bisphosphatase and activation of AMPK. However, the physiological relevance of these mechanisms has been contested due to the use of supra-pharmacological (millimolar) concentrations. These studies have been challenged by conflicting data showing that a clinically relevant metformin dose (micromolar) suppresses glucose production and gluconeogenic gene expression in primary hepatocytes via an AMPK-dependent mechanism but independently of changes in adenine nucleotides. To resolve this paradox, the authors interrogated a mechanism by which low metformin concentrations can activate AMPK. Based on their previous study showing that metformin can directly act on v-ATPase and promote the translocation of AXIN/LKB1 onto the surface of lysosome to activate AMPK (Cell Metab 2016), the authors attempted to identify the direct metformin lysosomal targets. The authors demonstrated that metformin inhibits the lysosomal proton pump v-ATPase by targeting the PEN2-ATP6AP1 axis leading to the activation of AMPK without changes in AMP:ATP ratio. Consistent with a direct role of PEN2 in metformin-induced activation of AMPK, deletion/ knockdown of PEN2 abrogated the increase of AMPK and ACC phosphorylation after metformin treatment. The authors further showed that mice lacking hepatic and intestinal PEN2 were resistant to metformin-mediated reduction in fatty liver and glucose-lowering effects, respectively. Furthermore, knockdown of PEN2 in *C. elegans* was shown to abrogate metformin-induced extension of lifespan, extending previous data showing that v-ATPase-mediated AMPK activation contribute to the lifespan extension effect of metformin in worms (PMID 29027899).

In general, this is a well written paper and the experiments generally are well performed. The authors revealed a potentially interesting mechanism for metformin action through the lysosomal pathway. I have a few concerns that should be addressed before publication, as listed below:

- Title is misleading and should be changed. Title indicates that PEN2 is the molecular target of metformin but does not mention this is linked to the activation of AMPK.

- Following on previous comment, it remains unclear if metformin action through the PEN2-ATP6AP1 axis requires AMPK. Metformin exhibits pleiotropic effects which can be AMPK-dependent or -independent. In addition to AMPK activation at the surface of the lysosome, metformin also inhibits mTORC1 through the lysosomal pathway (v-ATPase-Ragulator complex) independently of AMPK (PMID 29027899). The authors do not show data on the effect of low metformin concentrations on mTOR signaling and have not evaluated the role played by PEN2-ATP6AP1 axis in this regulation. These data are required to support the main conclusion and to rule out AMPK-independent effects in the metformin alleviation of fatty liver and improvement of glucose tolerance. Are these metformin beneficial effects blunted in the absence of AMPK?

- I am most surprised that the authors investigated the effect of metformin on fatty liver because there is scarce evidence for efficacy of metformin in the treatment of NAFLD/MAFLD. Systematic reviews and meta-analysis evaluating the effectiveness of metformin for the treatment of NAFLD/MAFLD concluded that metformin did not substantially impact liver disease (PMID: 31783920, 24648894). What is the clinical relevance for such findings?

- Many studies have shown that metformin accumulates at high concentration within mitochondria. When measuring the intracellular metformin content, does the calculated concentration reflect free or mitochondria-trapped metformin?

- Consistent with previous studies (PMID: 31974165), accumulation of intracellular metformin in primary hepatocytes corresponds to an intracellular/ extracellular ratio of 5 (Figure 1d). However, it appears this is not the case in vivo where the metformin concentrations in serum and liver are very similar (Figure 1e). How explain these discrepancies? For in vivo analysis, what is the rationale for using 4-week-old mice? Are young mice more sensitive/ tolerant to metformin than older/ adult mice?

- In contrast with in vitro data, while mice fed a 0.1% metformin diet display significant

changes in AMP:ATP ratio (Extended Figure 9b, $P=0.0396$), activation of AMPK was blunted in both PEN2 and ATP6AP1 LKO mice. It would be interesting to define a dose-response in vivo and determine the maximum effective dose of metformin unable to activate AMPK in the liver of PEN2 LKO mice. This will help to characterize the clinical relevant doses (without alteration of hepatic AMP:ATP ratio) that can be used in preclinical models. As ischemia induces significant alteration in AMP:ATP and ADP:ATP ratios, this condition could be used as a positive control. In addition, it is unclear in Figure 1g if metformin treatment (1 g/l and 10 g/l) induces alteration in AMP:ATP and ADP:ATP ratios as no saline control is provided. To confirm that AMP:ATP and ADP:ATP ratios are unchanged in the liver of mice taking 10 g/l metformin, activation of AMPK should be monitored in the liver of PEN2 LKO mice.

- The improvement in glucose tolerance and elevated GLP-1 secretion in metformin-treated WT mice is suggestive of an improved incretin response. What is the impact on plasma insulin levels during the glucose tolerance test? While PEN2 deletion in intestine blunted the metformin effect on glucose tolerance, it remains unclear if this is related to impaired AMPK activation or an AMPK-independent effect of PEN2 deletion. Recent studies have reported that deletion of AMPK in intestinal cells was dispensable for metformin-induced improvement of glucose tolerance (PMID 33548500).

- To fully interpret the data presented in extended figure 9c showing the effect of PEN2 deletion on the reduction of gluconeogenesis in response to metformin, control mice with no metformin treatment should be shown.

Referee #2 (Remarks to the Author):

This is an interesting and comprehensive study which potentially reveals a novel mechanism of action for metformin. Metformin action attracts a lot of attention and many mechanisms have been proposed in the literature as referenced in the manuscript. The study will have immediate interest to researchers in energy metabolism and type 2 diabetes and accordingly may be more suitable for a specialist journal.

Lin and colleagues demonstrate that metformin, at doses insufficient to disturb AMP/ATP ratios, directly interacts with the lysosomal protein PEN2, leading to interaction with ATP6AP1 and inhibition of the v-ATPase subunit. This triggers AMPK signalling via AXIN mediated recruitment of the upstream kinase LKB1, a mechanism previously explored by

the team. The manuscript is well written and on the whole data are presented clearly.

1. SPR experiment Fig 3c needs optimising to ensure accurate kinetics. The curves show signs of aggregating analyte, non-specific binding and do not reach plateau. A longer injection may be needed. For a K_d of 0.49 μM the conc series should be from no binding to saturation i.e. 0.05 μM to 5 μM . According to Methods, association and dissociation constants were obtained, but only K_d is reported. There are no overlays of replicates and no SDs reported.

2. Fig 4n - is there an explanation why met does not induce significant AXIN/LAMP2 co-localisation in the WT MEF control (next to ATP6AP1-/-)? Images in ED6l suggest otherwise.

3. PEN2 interaction with ATP6AP1 is the least convincing section of the study, relying on IP, in vitro reconstitution and in silico modelling to predict interacting residues. Details of the in vitro reconstitution are thin – what protein concentrations/incubation times were used, detergents? It is little surprise two isolated membrane proteins interact in the tube. The HA-PEN2 signal with myc IP (Fig. 4b) is not convincing.

4. To increase relevance of the lysosomal compartment, the authors should test whether 5 μM metformin induces AMPK signalling in cells expressing just the non-myristoylated beta-G2A mutated complex.

5. The PEN2-20A mutant is shown to adopt similar cell localisation to WT, but does it perform its normal functions as part of the γ -secretase complex? In a similar vein, does met-recruitment of PEN2 to ATP6AP1/v-ATPase impinge on γ -secretase functions?

Minor:

6. line 279: change "metformin" to "Met-P1"

7. line 295: I could not find the evidence to show LAMP2-ATP6AP1 restores v-ATPase activity.

8. Fig 5c, h, o, s – nucleotides – it's not immediately clear what P value comparisons refer to.

Referee #3 (Remarks to the Author):

A. Summary of the key results

Ma et al. provide an extensive data set in complementary model systems to establish a novel working mechanism for metformin at physiologically relevant concentrations, which involves the interaction of PEN2 with ATP6AP1, leading to V-ATPase inhibition and AMPK activation. This pathway does not alter AMP/ATP levels, indicating that it operates independently of ETC complex 1 inhibition, which has been reported before, but only takes place at higher, less relevant, metformin concentrations. The PEN2/ATP6AP1 interaction inhibits V-ATPase activity, leading to activation of AMPK through interaction with the Ragulator complex and AXIN, recruiting the latter to the lysosomes. In mouse models and *C. elegans*, the absence of PEN2 disturbs the beneficial effects of metformin treatment, such as amelioration of fatty liver, glucose lowering, and lifespan extension.

B. Originality and significance: if not novel, please include reference

Metformin is a widely prescribed drug for diabetes with multiple beneficial effects that have been extensively studied. While metformin may affect different pathways, the underlying mechanisms, mainly at clinically relevant concentrations, remain unclear. Building on their previous observations that metformin activates AMPK via V-type ATPase inhibition and promotes assembly of the Ragulator/AXIN/LKB1/AMPK super-complex, the authors now establish the upstream molecular mechanism that leads to AMPK activation. Unexpectedly, metformin binds to PEN2, a molecular component of the gamma-secretase complex implicated in Alzheimer's disease, which leads to ATP6AP1 interaction and V-type inhibition. This represents a novel mode of action of metformin at clinically relevant concentrations, revealing a new AMP-independent mechanism of AMPK activation.

C. Data & methodology: validity of approach, quality of data, quality of presentation

A combination of methods and experimental models are used to support the main conclusions. The experiments appear to be well executed and properly interpreted. Molecular level insights are established by characterizing a series of mutated and tagged proteins. The main conclusions were cross-validated *in vivo*, which strengthens the conclusions.

1) Many experiments are performed only twice, whereas a minimum of 3 experiments should be reported. For some of the imaging experiments, the authors note that images were acquired from e.g. '2 dishes/experiments'. This should be a minimum of 3 instead of 2. Also, it is not entirely clear what the authors mean by 'dishes/experiments'. The cells should ideally be plated on different days to count as independent experiments.

- 2) None of the western blots show molecular weights, these should be included.
- 3) Figure 5 is very dense in information and the order of panels makes it difficult to find the proper panel easily. The lay-out of this figure should be improved.

D. Appropriate use of statistics and treatment of uncertainties

Statistical tests are appropriate. The only concern is the low number of experiments that were performed, see comment in point C.

E. Conclusions: robustness, validity, reliability

In general, the main conclusions (metformin binding to PEN2 and stimulating interaction with ATP6AP1) are valid and robust, and are based on molecular level insights that were cross-validated in complementary models. These main conclusions are novel and clearly of interest for the field. The large number of experiments and techniques, however, also requires an extensive set of controls, which are sometimes lacking (see comments below). But the confirmation of results in complementary model systems provides a solid basis for the overall conclusions.

The authors state that the "lysosomal PEN2 pool" is responsible for the reported mechanism, but this conclusion remains premature (see comments below).

F. Suggested improvements: experiments, data for possible revision

- 1) The authors focused on the effects of metformin on primary hepatocytes at low concentration (5 μM), but then they turned to MEF and HEK293T cells to identify and characterize the metformin/PEN2/ATP6AP1 interaction. As the authors already indicate, these cells don't express OCT's to take up metformin in the cells, and higher metformin concentrations (200-300 μM) had to be used to observe the same phenotype observed in hepatocytes. In a study focusing on low concentration effects of metformin, this change to higher concentrations represents a weak point, as it may lead to other side effects.
 - a. These implications and limitations should be clearly discussed. Also the concentrations of the Met-P1/2 probes used in the screening experiment (not mentioned) and metformin

concentrations and exposure time elsewhere should be carefully reported in the main text, materials & methods and/or in the legends.

b. What is the uptake mechanism of metformin in OCT deficient cells? If uptake in the absence of OCT relies on endocytosis, this may lead to higher subcellular metformin concentrations at the level of the lysosome than in OCT expressing cells. This could lead to binding of metformin to the extra luminal C-terminal binding site on PEN2.

c. Would MEF or HEK293 cells with induced OCT expression lead to a similar induction of the PEN2/ATP6AP1 system at 5 μ M metformin concentration?

2) While PEN2 was established as a new molecular target, other genes that play a role may have been easily missed as a consequence of the screening strategy. The authors started with the metformin probe on protein extracts of lysosomal preparations (of MEF cells?? - specify in main text/fig2). This yielded over 3000 proteins (only 219 were lysosomal), which is surprisingly high since lysosomal protein extracts were used, questioning the specificity of the probe and purity of the lysosomal preparation. Conversely, cytosolic regulatory proteins regulating lysosomal membrane proteins may have been missed with this approach focusing on lysosomes. Moreover, 90 additional proteins from 'the list of lysosomal proteins' were added, but this list may be incomplete and cell type specific.

a. Despite the successful identification of PEN2, the screening cascade presents several flaws and other relevant targets may have been easily missed. How do the authors exclude that besides PEN2 other genetic factors may play a role that converge at the level of the lysosome?

b. It will be important to check whether the PEN2/ATP6AP1 pathway is the major mechanism of metformin action in respect to other already reported pathways. Is the expression of PCK1 and G6PC1 altered with the use of low metformin vs. high metformin doses?

3) The authors claim that the "lysosomal PEN2 pool" is responsible for activating the lysosomal AMPK pathway, which offers "beneficial effects". This conclusion needs to be carefully validated.

a) Decreasing V-ATPase activity and thereby affecting lysosomal pH is expected to disturb lysosomal function, which may reduce degradative capacity and impairing autophagy. How would this contribute to cell protection?

o The lysosomal acidity was determined with lysosensor, but what is the size of the pH effect? Does this affect lysosomal degradative capacity and/or autophagy?

b) The authors generated a number of constructs to alter the subcellular PEN2 location. Data should be presented to show that these fusion proteins are indeed localized to the expected subcellular locations. Overexpression of PEN2 is challenging, since it often mislocalizes, and fusion proteins may get trapped in the ER. This would lead to the false conclusion that only lysosomal PEN2 is involved in the described mechanism. For the HA-PEN2-2A and HA-PEN2-20A, the images are not very convincing and better images should be presented that also show ER markers.

c) What would be the % of PEN2 that is localized to lysosomes, and is there more PEN2 recruited to the lysosomes upon metformin treatment? This can be studied via (i) a Western blot in which PEN2 levels are determined on lysosomes purified before and after metformin treatment; or (ii) a time-series with immunofluorescence to see if PEN2 relocates to the lysosomes upon metformin treatment?

d) What is the mechanism of PEN2 targeting to lysosomes? Is the targeting of PEN2 to lysosomes a consequence of autophagy induction, or can lysosomal targeting motifs be recognized in the protein sequence? Can lysosomal targeting be prevented by endocytosis or autophagy blockers?

e) It is expected that the entire cellular pool of PEN2 binds metformin. Does this lead to other complex formations in other organelles?

4) The interaction between PEN2 and ATP6AP1 and its regulation by metformin is elegantly proven in this manuscript. The authors call the PEN2/ATP6AP1 effect on AMPK a 'serendipitous' intersection modulated by metformin. I would refrain from such wording, since it is equally possible that this mechanism has been evolved to support a clear physiological function, as it is conserved from *C. elegans* to mammals. Are there cell conditions known that lead to PEN2/ATP6AP1 interaction in the absence of Metformin?

5) Several controls for the various constructs/mutants are lacking. The authors state that the HA-tagged PEN2 variants are expressed at close-to-endogenous levels, which is more physiological. Can this point be demonstrated on blots? It is not always shown whether the protein mutants/fusions remain functional, e.g. does mutated PEN2 maintain γ -secretase activity? For the ATP6AP1 mutant: does it affect V-ATPase function? Loss of PEN2/ATP6AP1 interaction by expressing the delta420-440 mutant prevents the decrease in V-ATPase activity, but the expression of this mutant might result in a constitutively active V-ATPase. A control experiment showing pharmacological inhibition of the V-ATPase after expression of mutant ATP6AP1 would make this conclusion more robust. The confocal imaging showing the lysosomal localization of HA-PEN2, HA-PEN2-2A and HA-PEN2-20A is

unconvincing. HA-PEN2 appears to have a stronger lysosomal localization compared to HA-PEN2-2A and HA-PEN2-20A. Quantification of LAMP2 colocalization would more clearly demonstrate if they are localized equally to the lysosomes.

6) It is shown that inhibiting the V-ATPase leads to AXIN recruitment to the lysosomes, however it is not shown if this also activates AMPK.

7) Would metformin/PEN2/ATP6AP1 play a role in mouse primary hepatocytes from a diabetic mouse model (fed with high fat diet), which is a relevant model to study metformin effects?

8) The activity of the γ -secretase complex seems not required for the metformin effect on V-type ATPase and AMPK.

a. However, does metformin binding to PEN2 change the γ -secretase activity? Additionally, it is not shown whether the mutant forms of PEN2 still form functional γ -secretase complexes.

b. While an effect of NCT knockout on PEN2 levels is shown here, the loss of either APH1 or PSEN appears to have no effect on PEN2 mediated AMPK activation. However, knockout of PSEN is reported to destabilize PEN2 and cause ER-retention (Sannerud et al., 2016; Wang et al., 2004). The authors do not show PEN2 levels or subcellular localization in their MEF APH tripleKO or PSEN doubleKO cells. In addition, they do not confirm the lysosomal localization of HA-PEN2 expressed in NCT KO cells. As PEN2 levels and localization are known to be disturbed when γ -secretase complex formation is prevented, it is necessary to confirm lysosomal PEN2 expression in all γ -secretase subunit knockout cell lines. Further exploring the role of γ -secretase activity through inhibitors could strengthen this novel γ -secretase-independent function of PEN2.

c. Both NCT and PSEN have been shown already to interact with the V-ATPase (Nylandsted et al., 2011; Wakabayashi et al., 2009). Additionally, NCT is also reported to possibly interact with ALDOA. The pulldown of ATP6AP1 in supplemental figure 5c, does show coIP of PSEN1, PSEN2 and NCT, although less pronounced than that of PEN2. Are these interactions that are reported in literature without metformin treatment, relevant for the stronger interaction between PEN2 and ATP6AP1 upon metformin administration?

d. γ -secretase has been linked previously to lysosomal acidification, with PSEN deficient cells having an increased lysosomal pH. PEN2 knockdown seems to also have a modest effect on lysosomal pH (Fig. 1c). Is this significant? And if so, how would this contribute to the interpretation of the results?

e. Metformin has been reported as a neuroprotective agent. Would the interaction of metformin with PEN2 possibly play a role in a neuronal model or in models of Alzheimer's disease (such implications can be highlighted in the discussion)?

G. References: appropriate credit to previous work?

Yes.

H. Clarity and context: lucidity of abstract/summary, appropriateness of abstract, introduction and conclusions

Clear and well-written. The abstract gives a good overview of the study, and the introduction shows the need for their investigation. Moreover, the conclusions are clearly written and well structured. The schematic diagram, shown in supplemental figure 9, nicely sums up the conclusion and would be good to have in the main figures, not just in supplementals. Indicate the lysosomes on figure 2n.

Referee #4 (Remarks to the Author):

The authors show that PEN2 bind metformin at low doses which is then recruited to the ATP6AP1 subunit of v-ATPase, to inhibit the latter. By employing Ragulator and Axin, this complex then tethers upstream LKB1 to AMP-dependent kinase to activate the latter. The data presented provide a novel and potentially AMP-independent model for the mode of action of metformin at low/physiological doses.

(i) Given the downstream read-out, namely activation of AMPK, it appears necessary to exclude any impact of such AMPK activation on the supposedly upstream phenotypes. For example (Figs. 1a, j, k): is the deacidification of lysosomes impacted by activation or inhibition of AMPK? To answer this, a corresponding experiment using AICAR to activate AMPK, or pre-treatment with Compound C, a competitive AMPK inhibitor, before addition of metformin should be tested.

If either AICAR or Compound C show an effect on (de)acidification, all subsequently corresponding experiments should be expanded accordingly.

(ii) Likewise and as acknowledged by the authors, metformin is a known inhibitor of complex I of the electron transport chain (PubMedID 15047621 and others), and by this produces reactive oxygen species (ROS). Hence, the authors may wish to pre-incubate their respective cell lines with an antioxidant, e.g. N-acetyl cysteine, for at least 16 hrs, and repeat the experiments as depicted in Figs. 1a, j, k under such conditions.

(iii) According to Fig. 1d the intracellular concentration of metformin is 8fold higher than in the media, in primary hepatocytes. By contrast, (Fig. 1h and i), in MEFs and HEK293s, the intracellular concentration is 12 times *lower* than in the media. Despite the differences in cell types this is difficult to understand, however highly relevant given the proposed distinction between low- versus high-dose metformin action.

(iv) The authors throughout show AMP, ADP, and ATP as ratios only. While if find this acceptable for the main figures, the absolute values for each of the three should be provided in the extended data section, preferably normalized to protein.

This applies to Figs. 1d, h, i, e, g, Figs. 5c, f, h, o, s, t.

(v) Fig. 5a, i: Please provide corresponding plasma insulin values, including AUCs.

(vi) In the *C. elegans* data set, a concentration of 50 *milli*M of metformin was used. While based on the literature re. lifespan extension in nematodes, it is highly questionable whether this concentration qualifies as "low-dose". The data provided in Fig. 5p indicate no *relative* change in intra-organismal metformin concentration. However and unlike for the cell lines, the *absolute* concentrations for metformin is missing. Please provide these (as for cell lines), also normalized for protein.

(vii) In line with the previous, the reviewer anticipates that in *C. elegans*, the intra-organismal concentration at 50 mM supplementation is much higher than in cells. If so, the authors will have to identify the lowest concentration of metformin that still suffices to extend lifespan (and hopefully also qualifies as "low dose").

In this regards, the authors may wish to consider using heat-inactivated OP50, to prevent degradation of metformin by the bacteria.

Such heat-inactivation, however, prevents usage of HT115 (as in Fig. 5m), so in this case PEN2 needs to be inactivated by CRISPR/Cas9 in the nematode.

(viii) Given the rather high concentration of metformin, it is particularly important to exclude a role of metformin-mediated ROS formation through inhibition of complex I. To exclude this, N-acetyl cystein can be used again, and the complex I inhibitor rotenone at 100 nM can be used as a positive control (PubMedID 24199155).

(ix) Extended Data Table 1: Please provided not only number of nematodes scored, but rather total number, as well as number of nematodes *censored*.

(x) How many times were the nematodes used in Fig. 5q backcrossed to N2, to exclude presence of additional random mutations?

Author Rebuttals to Initial Comments:

Responses to Reviewer #1

We are very grateful to the reviewer for the constructive criticisms, which have helped substantiate the conclusions. We believe that we have now addressed all of them, either by providing clarification, by presenting additional results, or by performing new experiments. In particular, we have determined the involvement of the PEN2-ATP6AP1 axis in the inhibition of mTORC1, measured the mitochondrial metformin concentrations, and determined the range of metformin that acts via PEN2 to activate AMPK independent of AMP changes. Point-by-point responses to the reviewer's comments are given below, with the reviewer's comments given verbatim in plain italic type, and our responses in bold:

Referee #1 (Remarks to the Author):

The paper by Ma et al. describes a diverse set of experiments looking at the mechanism of AMPK activation in response to low metformin concentrations and how this might impact on metformin-induced therapeutic benefits. Metformin is commonly used to treat type-2 diabetes and there is increasing interest in understanding its physiological effects because of its pleiotropic action (e.g., in cancer and aging). However, the exact mechanism through which it acts has yielded conflicting results. Several mechanisms by which complex I inhibition has been implicated for the metformin's glucose-lowering effect have been proposed including altered cellular energy charge, inhibition of adenylate cyclase, inhibition of fructose 1,6-bisphosphatase and activation of AMPK. However, the physiological relevance of these mechanisms has been contested due to the use of supra-pharmacological (millimolar) concentrations. These studies have been challenged by conflicting data showing that a clinically relevant metformin dose (micromolar) suppresses glucose production and gluconeogenic gene expression in primary hepatocytes via an AMPK-dependent mechanism but independently of changes in adenine nucleotides. To resolve this paradox, the authors interrogated a mechanism by which low metformin concentrations can activate AMPK. Based on their previous study showing that metformin can directly act on v-ATPase and promote the translocation of AXIN/LKB1 onto the surface of lysosome to activate AMPK (Cell Metab 2016), the authors attempted to identify the direct metformin lysosomal targets. The authors demonstrated that metformin inhibits the lysosomal proton pump v-ATPase by targeting the PEN2-ATP6AP1 axis leading to the activation of AMPK without changes in AMP:ATP ratio. Consistent with a direct role of PEN2 in metformin-induced activation of AMPK, deletion/knockdown of PEN2 abrogated the increase of AMPK and ACC phosphorylation after metformin treatment. The authors further showed

that mice lacking hepatic and intestinal PEN2 were resistant to metformin-mediated reduction in fatty liver and glucose-lowering effects, respectively. Furthermore, knockdown of PEN2 in C. elegans was shown to abrogate metformin-induced extension of lifespan, extending previous data showing that v-ATPase-mediated AMPK activation contribute to the lifespan extension effect of metformin in worms (PMID 29027899).

In general, this is a well written paper and the experiments generally are well performed. The authors revealed a potentially interesting mechanism for metformin action through the lysosomal pathway.

We thank the reviewer for the positive comments.

I have a few concerns that should be addressed before publication, as listed below:

- Title is misleading and should be changed. Title indicates that PEN2 is the molecular target of metformin but does not mention this is linked to the activation of AMPK.

We thank the reviewer for the comment. We have now changed the title to 'Metformin targets PEN2 to intersect the lysosomal AMPK pathway'.

- Following on previous comment, it remains unclear if metformin action through the PEN2-ATP6AP1 axis requires AMPK. Metformin exhibits pleiotropic effects which can be AMPK-dependent or -independent. In addition to AMPK activation at the surface of the lysosome, metformin also inhibits mTORC1 through the lysosomal pathway (v-ATPase-Ragulator complex) independently of AMPK (PMID 29027899). The authors do not show data on the effect of low metformin concentrations on mTOR signaling and have not evaluated the role played by PEN2-ATP6AP1 axis in this regulation. These data are required to support the main conclusion and to rule out AMPK-independent effects in the metformin alleviation of fatty liver and improvement of glucose tolerance. Are these metformin beneficial effects blunted in the absence of AMPK?

We thank the reviewer for raising the complicated but interesting issue. We have now examined mTORC1 signalling in various settings, by determining the

phosphorylation of T389 residue on S6K, a substrate of mTORC1, in PEN2-KO MEFs/HEK293T cells, and ATP6AP1-KO MEFs/HEK293T cells re-introduced with ATP6AP1 or its $\Delta 420-440$ (the truncate mutant was to maintain v-ATPase activity after KO of ATP6AP1). As shown below [current Extended Data (ED) Fig. 9s, t], while metformin treatment reduced levels of phosphorylated S6K (p-S6K), the deficiency of the PEN2-ATP6AP1 axis rendered low metformin unable to dampen mTORC1, indicating that the PEN2-ATP6AP1 axis is required for metformin to inhibit mTORC1, through inhibiting the v-ATPase. The v-ATPase is thus the converging point for metformin and low glucose, to regulate both AMPK and mTORC1, with the former targeting ATP6AP1 and the latter changing the conformation of aldolase, both of which are integral components of the v-ATPase.

ED Fig. 9s

ED Fig. 9t

As for the beneficial effects of metformin in promoting post-prandial glucose absorption, it has been shown that AMPK alone is sufficient, as it has been shown that the rats with the dominant-negative form of AMPK expressed in duodena failed to exhibit metformin-mediated glucose lowering (PMID 25849133). As for role of mTORC1 in glucose lowering, however, controversial results have been shown. Although the dominant-negative form of mTOR expressed in duodena is able to promote glucose absorption (PMID 30755615), many other studies have shown that inhibition of mTORC1 by short-term rapamycin treatment even impairs glucose tolerance (e.g., PMID 23473038). Following the suggestion made by the reviewer, we tested the dependency of glucose tolerance improvement on duodenal AMPK, showing that intestinal knockout of AMPK (DKO of *AMPK α 1* and *AMPK α 2*) blocked low-dose metformin-improved glucose tolerance in mice (shown below, current ED Fig. 8b), confirming that AMPK is required for metformin to stimulate at least post-prandial glucose absorption.

We also appreciate the reviewer for raising the concern as to if hepatic TAG reduction caused by metformin treatment depends on AMPK. It has been shown that double knock-in of ACC-1/-2 (ACC1-S79A, ACC2-S212A, which lack their respective phosphorylation site for AMPK), impairs the effects of metformin on reducing hepatic fat accumulation (PMID 24185692). It has also been shown that ACC1 and ACC2 are phosphorylated exclusively by AMPK (PMID 24093679). Hence, these published studies have nicely indicated that the hepatic AMPK is responsible for metformin-mediated reduction of TAG in the liver. In addition, we have shown that deficiency of the PEN2-ATP6AP1 axis in the liver blocked metformin-induced AMPK activation, and impaired the effects of metformin on reducing hepatic TAG (Fig. 5e, h, and ED Fig. 8h, m). As for mTORC1, it has been shown that hepatic depletion of *Raptor*, which leads to a constitutive inactivation of mTORC1, did not affect HFD-induced fatty liver in mice (PMID 24910242). Although long-term rapamycin treatment can alleviate fatty liver (PMID 19496779), such an effect was later attributed to the inhibition of mTORC2, given that liver-specific knockout of *Rictor*, which renders mTORC2 inactivated blocks the HFD-induced hepatic TAG accumulation (PMID 22773877 and 22461615). Therefore, inhibition of mTORC1 is unlikely to play a significant role in the fatty liver alleviation.

Pertaining to the dependency of AMPK for metformin to reduce hepatic fat, we would have to first breed the AMPK α 1/ α 2-DKO mice to a sufficient population, then feed them with HFD for at least three months, followed by treating them with metformin for another four months. This would be a long duration. Based on all the existing data suggesting that it is AMPK that mediates metformin effects on liver fat contents, we have sought the advice from the editor Dr. Gebala if it is fine

for us not to examine the effects of metformin in liver-specific AMPK-DKO mice. Dr. Gebala agreed that the experiments could be waived, provided that we carefully discuss the published studies in the reply letter, and have performed other experiments requested by the reviewer to consolidate that the metformin action through the PEN2-ATP6AP1 axis requires AMPK. We sincerely hope that the reviewer would agree with our appeal not to do that particular experiment.

We also thank the reviewer for referring to the interesting study showing that metformin can inhibit mTORC1 in an AMPK (*aak-2*)-independent manner in *C. elegans* (PMID 29027899). Indeed, that observation can now be explained by the existence of PEN2-ATP6AP1 axis, in addition to our published (PMID 33349646) and unpublished data that mTORC1 inhibition actually only depends on AXIN translocation to the lysosomal surface (performed in cells lacking AMPK α 1/2). As for the dependency on AMPK and/or mTORC1 for lifespan extension in *C. elegans* by metformin, while metformin can still extend lifespan in the *Raptor* (*daf-15*) deficient nematodes (inactive mTORC1; PMID 29027899), it fails to extend lifespan in AMPK (*aak-2*) deficient nematodes (PMID 20090912); knockdown of *Raptor*, or treatment of rapamycin has also shown to extend lifespan in *C. elegans* (PMID 24889782). These data have demonstrated that AMPK activation and mTORC1 inhibition each can lead to lifespan extension, with AMPK activation playing a dominant role.

- I am most surprised that the authors investigated the effect of metformin on fatty liver because there is scarce evidence for efficacy of metformin in the treatment of NAFLD/MAFLD. Systematic reviews and meta-analysis evaluating the effectiveness of metformin for the treatment of NAFLD/MAFLD concluded that metformin did not substantially impact liver disease (PMID: 31783920, 24648894). What is the clinical relevance for such findings?

We totally agree with the reviewer on the role of metformin in fatty liver. Indeed, the beneficial effects of metformin on NAFLD have been under dispute, especially by many clinical reports. The discrepancies between different studies could have been caused by many reasons. We found that metformin can reduce severity of fatty liver only after treating HFD-treated mice for 4 months. As for the ineffectiveness observed in human patients, the difficulty to draw a conclusion could be caused by the nature of the development of fatty liver among different

individuals. If the fatty liver is not caused by over-eating, but caused by other reasons such as different genetic backgrounds, metformin may not have an effect. We would like to add that the chemical compound we have identified to inhibit aldolase from binding to FBP and hence activates AMPK, can quickly alleviate fatty liver and make them lose weight in HFD-treated mice, but not in slowly/spontaneously developed obese monkeys (*Macaca fascicularis*), or mice with normal sizes. Similarly, it was reported that metformin alleviates fatty liver in mice depending strictly on the AMPK-ACCs axis (PMID 24185692), as discussed in our response to the preceding comment. Therefore, the reason for our observed effect of metformin to reduce the severity of fatty liver is that we used HFD-fed obese mice. Nevertheless, we agree with the reviewer's caution and have now changed phrases like 'alleviate fatty liver' to 'reduce hepatic fat', to better reflect the accepted view that inhibition of lipid synthesis is a fundamental function of AMPK.

- Many studies have shown that metformin accumulates at high concentration within mitochondria. When measuring the intracellular metformin content, does the calculated concentration reflect free or mitochondria-trapped metformin?

We have now determined the mitochondrial metformin contents. To avoid interference from other organelles such as ER (MAMs) and lysosomes, we applied a subcellular fractionation method that allows separation of mitochondria from MAMs and lysosomes (PMID 19816421; modified and validated by us, PMID 30948787). As shown below (current ED Fig. 1k), we found similar contents of metformin between the cytosolic and mitochondrial pools.

- Consistent with previous studies (PMID: 31974165), accumulation of intracellular metformin in primary hepatocytes corresponds to an intracellular/ extracellular ratio of 5 (Figure 1d). However, it appears this is not the case in vivo where the metformin concentrations in serum and liver are very similar (Figure 1e). How explain these discrepancies?

Indeed, as reported previously, compounds like metformin should have liver-to-plasma water partition coefficient (K_{pu}) values over 5 (PMID 15858854), like those observed in cultured hepatocytes in our study. However, such values were derived solely based on the affinity of metformin to intracellular and extracellular proteins/phospholipids when using cell models. It does not take into account the high secretion/elimination rates of hepatic metformin inside animals, driven by relatively low concentrations of metformin in hepatic vein and high capability of renal OCT2 transporter in mouse models, with the clearance rate 3 folds higher than the maximal hepatic metformin uptake rate (PMID 22407892). This situation might be pronounced in our animal model to which low metformin was administered: the hepatic metformin's secretion/elimination rates were high and far from saturation after a stable (at least 1 week), low metformin (1 g/l in drinking water) treatment. Such a high capacity of metformin clearance was also supported by some pharmacokinetic studies, e.g., PMID 16937336, showing that in short time durations (within 120 min) after intravenous injection (i.v.) of metformin, the plasma metformin concentrations are not dose-proportional, but within a narrow change, between the 50 mg/kg (mpk) and the 100 mpk groups (shown below, Fig 1 of PMID 16937336; solid circle: 50 mpk, open square: 100 mpk), and was increased only at 200 mpk (solid triangle). Given that the plasma metformin achieved in our model is far below that of i.v. at 50 mpk within 120 min, it is reasonable to suggest that metformin is rapidly eliminated in the liver and may therefore be equally distributed between liver and plasma. We have discussed this point in the text (Supplementary Note 1).

For in vivo analysis, what is the rationale for using 4-week-old mice? Are young mice more sensitive/ tolerant to metformin than older/ adult mice?

We used young mice merely because it could save time, given that mice at 4-week-old are weaned and share similar metabolism to older mice. For experiments involving HFD treatment, the ages of mice are older because it takes at least three months to reach obesity, and at least one month for genetic manipulation by injecting AAVs together with tamoxifen, so the mice used for metformin administration were usually older than six months. The sensitivity of metformin between young and ‘old’ mice seems to be of no difference because the AMPK activation and intraorganismic/serous metformin contents are similar for given doses (compare Fig. 1e with ED Fig. 8g, k).

*- In contrast with in vitro data, while mice fed a 0.1% metformin diet display significant changes in AMP:ATP ratio (Extended Figure 9b, $P=0.0396$), activation of AMPK was blunted in both *PEN2* and *ATP6AP1* LKO mice. It would be interesting to define a dose-response in vivo and determine the maximum effective dose of metformin unable to activate AMPK in the liver of *PEN2* LKO mice. This will help to characterize the clinical relevant doses (without alteration of hepatic AMP:ATP ratio) that can be used in preclinical models.*

We thank the reviewer for bringing up this interesting issue. Indeed, as determined and shown in ED Fig. 9b, the 0.1% metformin diet caused a significant increase of

the ADP:ATP ratio (the *P* value is 0.0396), although the increase of AMP:ATP ratio is not significant (*P* value is 0.2298). Following the suggestion made by the reviewer, we re-titrated the doses of metformin by increasing the concentrations of metformin in drinking water and determined AMPK activation in the mouse liver of wildtype mice and the PEN-KO mice. Metformin concentrations in drinking water were increased from 2 g/l (not yet sufficient for increasing hepatic AMP:ATP, as shown below and in current ED Fig. 1g) to 5 g/l, 8 g/l, then to 10 g/l, and the hepatic AMP:ATP and ADP:ATP ratios, metformin concentrations, as well as AMPK activation were determined. We found that metformin at 8 g/l did not increase hepatic AMP:ATP and ADP:ATP ratios or bypass the requirement of PEN2 for AMPK activation (as shown below, current ED Fig. 1g).

However, when metformin reached 10 g/l, some of the mice showed decreased water/metformin intake (perhaps owing to the unpleasant taste of metformin, as described in the figure legends of original/current ED Fig. 1f). At this concentration, some mice already exhibited sufficient increase of hepatic AMP:ATP (>0.12, and the hepatic concentration of metformin >350 µM) to bypass the PEN2 requirement for activating AMPK, whereas others with lower intake of metformin (and hence lower hepatic metformin concentrations) did not (shown below, current ED Fig. 1h). Together, 8 g/l metformin in drinking water may be defined as the maximum

effective dose of metformin that is still unable to activate AMPK in the liver of PEN2-LKO mice.

Of note, because ischemia will cause AMPK activation unrelated to the lysosomal pathway in the liver likely due to the calcium-mediated CaMKK2 activation and the increase of AMP or ADP, we technically had to determine the above readouts in the liver from the mice that did not undergo the step of blood draining. The legitimacy for skipping the step of blood draining is also based on the observation that hepatic metformin concentration is similar to that in the serum in our animal setting (metformin administered in drinking water, as described above) – the residual blood would not significantly interfere with the readout of the hepatic metformin concentration. We hope the reviewer would agree with our way of preparing the liver samples.

The data shown above (current Fig. 1g, h) are also consistent with our previous finding that at least one-fold increase of AMP:ATP ratio, e.g., from 0.006 to 0.015 in MEFs, is required for bypassing the lysosomal pathway for AMPK activation (PMID 30948787). Lower basal AMP in MEFs than that in the liver may be attributed to higher amounts of AMP occupied or sequestered by different levels of other AMP-binding proteins in MEFs. Therefore, since mass spectrometry

determines the total amount of AMP, the 'basal' levels of free AMP may vary from cell to cell, so are the levels of free AMP required to bypass the AMP(increase)-independent lysosomal pathway. Together, in original ED Fig. 9b (current ED Fig. 9h), the changes of the AMP:ATP ratio under 0.1% metformin treatment (from 0.047 to 0.055) is not sufficient for bypassing the requirement of the PEN2/ATP6AP1 axis.

As ischemia induces significant alteration in AMP:ATP and ADP:ATP ratios, this condition could be used as a positive control. In addition, it is unclear in Figure 1g if metformin treatment (1 g/l and 10 g/l) induces alteration in AMP:ATP and ADP:ATP ratios as no saline control is provided. To confirm that AMP:ATP and ADP:ATP ratios are unchanged in the liver of mice taking 10 g/l metformin, activation of AMPK should be monitored in the liver of PEN2 LKO mice.

According to the suggestion made by the reviewer, we included the saline group in the revised Fig. 1g. The statistical results obtained indicated that changes of AMP:ATP and ADP:ATP ratios between saline and 1 g/l groups, or saline and 10 g/l groups, are not significant. However, as described above, the lack of statistical significance between saline and 10 g/l groups may be attributed to the large variations of hepatic metformin concentrations between individual mice under this concentration. In the livers that have metformin concentrations higher than 350 μ M, AMPK activation can be seen in a PEN2-independent manner, in which the AMP:ATP ratios are significantly higher than the saline group ($n = 8$ and $n = 5$ for 10 g/l group and saline group, respectively; $P = 0.0016$ by Mann-Whitney U test).

- The improvement in glucose tolerance and elevated GLP-1 secretion in metformin-treated WT mice is suggestive of an improved incretin response. What is the impact on plasma insulin levels during the glucose tolerance test?

We thank the reviewer for this constructive suggestion (also raised by Reviewer #4). We have now determined the insulin levels during the oral glucose tolerance test. As shown below (current Fig. 5a), higher insulin levels were detected in the metformin-treated groups, and the intestinal-specific knockout of *PEN2* blocked such an effect.

Intrigued by such a suggestion, we performed additional experiments to determine the levels of insulin in HFD-fed mice treated with metformin for 4 months which showed significantly improved insulin sensitivity. As shown below (current Fig. 5c and 5f), decreased insulin levels were detected in the metformin-treated wildtype mice due to alleviated hepatic fat, but not in mice with hepatic knockout of PEN2, or with re-expressed ATP6AP1- Δ 420-440.

While PEN2 deletion in intestine blunted the metformin effect on glucose tolerance, it remains unclear if this is related to impaired AMPK activation or an AMPK-

independent effect of PEN2 deletion. Recent studies have reported that deletion of AMPK in intestinal cells was dispensable for metformin-induced improvement of glucose tolerance (PMID 33548500).

As described above, we have now shown that intestinal-specific knockout of AMPK (DKO of *AMPKα1* and *AMPKα2*) blocked low metformin-improved glucose tolerance in mice (current ED Fig. 8b). We thank the reviewer for referring to the study of PMID 33548500. The different conclusions obtained between that study and our work might result from the different ways and dosages metformin was administered. In PMID 33548500, metformin at 300 mpk was acutely orally gavaged rather than added in drinking water at 1 g/l. According to the results obtained during testing serum concentrations of metformin after different ways of administration, serum concentrations of metformin were found to exceed 30 μM after 30 min of p.o. metformin at 50 mpk, higher than those of 1 g/l in drinking water (approx. 15 μM). The higher concentration of metformin may cause AMP elevation and promote many other pathways such as glycolysis (allosteric activation of phosphofruktokinase) to enhance glucose catabolism, which may or may not be attributed to AMPK activation. However, we really do not know how to further comment on the physiological meaning of the results derived from high doses. We have also mentioned this point and cited this study in our revised manuscript (line 464, page 20).

- To fully interpret the data presented in extended figure 9c showing the effect of PEN2 deletion on the reduction of gluconeogenesis in response to metformin, control mice with no metformin treatment should be shown.

We thank the reviewer for pointing out the missing controls in ED Fig. 9c. We have now performed the pyruvate tolerance test using PEN2-LKO mice and its wildtype littermates treated with metformin (added in drinking water at 1 g/l for 7 days) or saline. As shown below (current ED Fig. 9r), liver-specific knockout of PEN2 did not affect gluconeogenesis under both normal and metformin-treated conditions.

Responses to Reviewer #2

We are very grateful to the reviewer for all the constructive criticisms, which have helped substantiate the conclusions. We have now addressed all of them, either by providing clarification, by presenting additional results, or by performing new experiments. In particular, we have now re-performed the determination of the Kd of PEN2 for metformin by SPR, and confirmed the requirement of membrane-localisation of AMPK for its activation by metformin. Point-by-point responses to the reviewer's points are given below, with the reviewer's comments given verbatim in plain italic type, and our responses in bold:

Referee #2 (Remarks to the Author):

This is an interesting and comprehensive study which potentially reveals a novel mechanism of action for metformin. Metformin action attracts a lot of attention and many mechanisms have been proposed in the literature as referenced in the manuscript. The study will have immediate interest to researchers in energy metabolism and type 2 diabetes and accordingly may be more suitable for a specialist journal.

Lin and colleagues demonstrate that metformin, at doses insufficient to disturb AMP/ATP ratios, directly interacts with the lysosomal protein PEN2, leading to interaction with AT6AP1 and inhibition of the v-ATPase subunit. This triggers AMPK signalling via AXIN mediated recruitment of the upstream kinase LKB1, a mechanism previously explored by the team. The manuscript is well written and on the whole data are presented clearly.

We thank the reviewer for the kind comments.

1. SPR experiment Fig 3c needs optimising to ensure accurate kinetics. The curves show signs of aggregating analyte, non-specific binding and do not reach plateau. A longer injection may be needed. For a Kd of 0.49 uM the conc series should be from no binding to saturation i.e. 0.05 uM to 5 uM. According to Methods, association and dissociation constants were obtained, but only Kd is reported. There are no overlays

of replicates and no SDs reported.

We thank the reviewer for the instructions. We have now re-performed the SPR experiment using a series of concentrations of metformin starting from 0.05 μM as suggested. Please note that PEN2 is a protein of relatively low molecular weight and hence exhibits a small change of response unit (RU, lower than 20 during the measurement) on Biacore T200 after binding to metformin whose molecular weight is also small. We have to apply metformin up to relatively high concentrations during the assay to maximise the signal-to-noise ratio. The metformin gradient was therefore set as follows: 0.05 μM , 0.78125 μM , 1.5625 μM , 3.125 μM and 6.25 μM . In addition, we found that 150 s of injection duration seemed to be sufficient for the metformin binding curve to reach plateau, so the injection duration was set as such. As shown below (also listed as current Fig. 3c, with 3 replicates overlaid), the newly performed SPR assay gave a $K_D = 0.15 \pm 0.06$ μM (mean \pm s.e.m.), and $k_a = 2815 \pm 2084$ Ms^{-1} for PEN2 (mean \pm s.e.m.), similar to the values obtained previously.

2. Fig 4n - is there an explanation why met does not induce significant AXIN/LAMP2 co-localisation in the WT MEF control (next to $ATP6AP1^{-/-}$)? Images in ED6I suggest otherwise.

The P value calculated in WT MEFs (as a control to $ATP6AP1^{-/-}$ MEFs) shown in Fig. 4n was wrongly labelled. It was mistakenly copied/pasted. We apologise for the mistake. It has been corrected in our revised manuscript.

3. *PEN2 interaction with ATP6AP1 is the least convincing section of the study, relying on IP, in vitro reconstitution and in silico modelling to predict interacting residues. Details of the in vitro reconstitution are thin – what protein concentrations/incubation times were used, detergents? It is little surprise two isolated membrane proteins interact in the tube. The HA-PEN2 signal with myc IP (Fig. 4b) is not convincing.*

The amount of PEN2 and ATP6AP1 used was 1 μg each in the in vitro reconstitution assays, with 1% Triton-X 100 as detergent (in the lysis buffer), and the incubation time was 2 h. This information was initially provided in the Methods section due to space limitation of the main text and legends. We have now moved it to the legend of Fig. 4e.

To enhance the clarity of Fig. 4b, we have now included statistical analysis data showing the ratios of immunoprecipitated PEN2:total PEN2. As shown below (also on the right panel of current Fig. 4b), metformin significantly increased the interaction between PEN2 and ATP6AP1.

4. *To increase relevance of the lysosomal compartment, the authors should test whether 5 μM metformin induces AMPK signalling in cells expressing just the non-myristoylated beta-G2A mutated complex.*

We thank the reviewer for pointing this out. We have now determined the phosphorylation of AMPK in response to 200 μM metformin using our existing AMPK β -KO MEFs [both AMPK β 1 and AMPK β 2; the knockout strategies were

shown in current Extended Data (ED) Fig. 6g] expressing AMPK β 1-G2A. As shown below (current ED Fig. 6h), metformin failed to induce levels of p-AMPK in those cells, indicating that the membrane-/lysosomal-localisation is required for AMPK activation in response to low metformin treatment, as for glucose starvation (PMID 20974912).

5. The PEN2-20A mutant is shown to adopt similar cell localisation to WT, but does it perform its normal functions as part of the γ -secretase complex? In a similar vein, does met-recruitment of PEN2 to ATP6AP1/v-ATPase impinge on γ -secretase functions?

Following the instructive suggestions by the reviewer, we have now determined the interaction between PEN2-20A (and also PEN2-2A) and other subunits of the γ -secretase. The results showed that the mutations on PEN2 did not affect its interaction with other γ -secretase subunits (shown below, current ED Fig. 4i). However, we found that PEN2-2A increased the activity of γ -secretase (by 20% compared with that by wildtype PEN2), as determined by the cleavage of Notch (PMID 10206645), while PEN2-20A inhibited the activity of γ -secretase (shown below, current ED Fig. 4h; the γ -secretase inhibitor, DAPT was used as a control) (LE, long exposure; SE, short exposure).

ED Fig. 4i

ED Fig. 4h

We also measured the activities of γ -secretase under metformin treatment, and found that metformin could increase the activity of γ -secretase, by approximately 50% (shown below, current ED Fig 4j).

Minor:

6. line 279: change “metformin” to “Met-P1”

We thank the reviewer for the correction. It has been changed.

7. line 295: I could not find the evidence to show LAMP2-ATP6AP1 restores v-ATPase activity.

In our original manuscript, the restoration of v-ATPase activity in *ATP6AP1*^{-/-} MEFs by expressing LAMP2-ATP6AP1 was assessed by the extent of the suppression of AMPK activation in *ATP6AP1*^{-/-} MEFs in which the v-ATPase was inactive. We have now also determined the pH of lysosomes in these cells in our revised manuscript (shown below, current ED Fig. 5h).

8. Fig 5c, h, o, s – nucleotides – it's not immediately clear what P value comparisons refer to.

We thank the reviewer for pointing this out and have re-organised those panels.

Responses to Reviewer #3

We are very grateful to the reviewer for the constructive criticisms, which have helped substantiate the conclusions. We have now addressed all of them, either by providing clarification, by presenting additional results, or by performing new experiments. In particular, we have analysed/discussed the relationship between v-ATPase inhibition and autophagy induction under metformin treatments, and the results indicate that inhibition of v-ATPase at early stage of metformin treatment (8 h to 24 h) did not induce autophagy, prolonged metformin treatment (after 30 h, up to 72 h) induced autophagy. We have also determined the effects of metformin/PEN2-ATP6AP1 on γ -secretase. Moreover, we also performed a series of validation assays concerning the localisation of PEN2 and PEN2 mutants/chimaeras. Point-by-point responses to the reviewer's points are given below, with the reviewer's comments given verbatim in plain italic type, and our responses in bold:

Referee #3 (Remarks to the Author):

A. Summary of the key results

*Ma et al. provide an extensive data set in complementary model systems to establish a novel working mechanism for metformin at physiologically relevant concentrations, which involves the interaction of PEN2 with ATP6AP1, leading to V-ATPase inhibition and AMPK activation. This pathway does not alter AMP/ATP levels, indicating that it operates independently of ETC complex 1 inhibition, which has been reported before, but only takes place at higher, less relevant, metformin concentrations. The PEN2/ATP6AP1 interaction inhibits V-ATPase activity, leading to activation of AMPK through interaction with the Ragulator complex and AXIN, recruiting the latter to the lysosomes. In mouse models and *C. elegans*, the absence of PEN2 disturbs the beneficial effects of metformin treatment, such as amelioration of fatty liver, glucose lowering, and lifespan extension.*

B. Originality and significance: if not novel, please include reference

Metformin is a widely prescribed drug for diabetes with multiple beneficial effects that have been extensively studied. While metformin may affect different pathways, the underlying mechanisms, mainly at clinically relevant concentrations, remain

unclear. Building on their previous observations that metformin activates AMPK via V-type ATPase inhibition and promotes assembly of the Ragulator/AXIN/LKB1/AMPK super-complex, the authors now establish the upstream molecular mechanism that leads to AMPK activation. Unexpectedly, metformin binds to PEN2, a molecular component of the gamma-secretase complex implicated in Alzheimer's disease, which leads to ATP6AP1 interaction and V-type inhibition. This represents a novel mode of action of metformin at clinically relevant concentrations, revealing a new AMP-independent mechanism of AMPK activation.

C. Data & methodology: validity of approach, quality of data, quality of presentation

A combination of methods and experimental models are used to support the main conclusions. The experiments appear to be well executed and properly interpreted. Molecular level insights are established by characterizing a series of mutated and tagged proteins. The main conclusions were cross-validated in vivo, which strengthens the conclusions.

We thank the reviewer for the constructive comments.

1) Many experiments are performed only twice, whereas a minimum of 3 experiments should be reported. For some of the imaging experiments, the authors note that images were acquired from e.g. '2 dishes/experiments'. This should be a minimum of 3 instead of 2. Also, it is not entirely clear what the authors mean by 'dishes/experiments'. The cells should ideally be plated on different days to count as independent experiments.

We thank the reviewer for giving us the opportunity to clarify on the experimental repetition. We normally have performed many rounds of 'preliminary' experiments before repeating the shown experiments by omission of some of conditions/treatments to avoid overly large sizes of individual panels. For example, experiments related to Fig. 1d were originally performed and repeated for three times by treating primary hepatocytes with 5 μ M metformin at the following time points: 0, 0.5, 1, 2, 4, 6, 8, 10, 12, 16, 20, 24, 30, 36, 42, 48, 60 and 72 hours. We attached herein some of the representative data of the earlier experiments (shown below, upper panel). We also learned that the accumulation of metformin inside cells had reached the plateau after 12 h of incubation (lower panel). Owing to the

space limitation, we then repeated the same experiments at time points of 0, 2, 12, 24 and 48 h. Therefore, only the 0, 2, 12 h, 24 h and 48 h time points as shown in Fig. 1d were included, which we repeated for additional two more times.

Other examples include Fig. 1e and 1g (initially performed with more time points), 1h and 1i (more metformin concentrations), 2g (metformin concentration gradients, rather than a single concentration), 2m (with metformin-treated groups), 2n (APEX-PEN2 was repeatedly paired with controls such as the APEX-PEN2 ER and APEX-PEN2 lyso for imaging, and only the pair APEX-PEN2/APEX-PEN2 mito was shown), 4e (re-organised from two experiments, one determining the PEN2-ATP6AP1 interaction, and the other interaction under a series of metformin concentrations), 4h (with metformin-treated groups as additional controls), 4o (re-organised from PEN2-WT Vs PEN2-2A, and PEN2-WT Vs PEN2-20A), 4p (re-organised from vector Vs ATP6AP1-FL, vector Vs ATP6AP1-Δ420-440), Extended Data (ED) Fig. 1a (more time points), ED 1b, c (more metformin concentrations), ED 1d, e, g, h (high metformin as a control), ED 1f (more time points), ED 2a (a scale-up experiment, before which 5 rounds of pilot trials were performed), ED 3k (more DAPT and RO concentrations), ED 3o (re-organised from different PEN2 chimeras Vs wildtype PEN2), ED 3p (more concentrations of phenformin and buformin), ED 5c (re-organised from IgG Vs ATP6AP1 and IgG Vs PEN2), ED 6a-h, 6j, 6k, and ED 7 (serial metformin concentrations in cultured cells, and phenformin in mice).

However, two experiments were indeed performed twice. One is the ED Fig. 5a mapping the interacting domain(s) required for ATP6AP1 to bind PEN2. We have now performed one more round experiment, to a total of three repeats, and similar results were obtained. In addition, the conclusion that $\Delta 420-440$ of ATP6AP1 was required for PEN2-binding to transmit metformin signal has been cross-validated by many other experiments in our study (e.g., Fig. 4k, n). The other is the animal experiments shown in original Fig. 5 and ED Fig. 8. Owing to the relatively large numbers of animals of the same genotypes are needed for such animal studies, and in many experiments they have to be fed with HFD for at least four months, and the metformin treatment lasts for four additional months, we had not finished the third repeat at the time of the initial manuscript submission. We sincerely apologise for this. We have now finished these experiments, and similar results were obtained. The statistical data support the conclusion that PEN2-ATP6AP1 axis is required for low metformin-promoted glucose absorption and fatty liver/insulin sensitivity. Of note, in this round experiment, the observed effect of hepatic ATP6AP1- $\Delta 420-440$ re-introduction on insulin sensitivity (Fig. 5g) seemed to be more pronounced than the earlier experiment (shown below).

On the basis that we have repeated all the experiments for at least three times and to comply with the guideline, we have now indicated 'three times' or 'at least three times' to better reflect the real situations. This has been an educational experience.

The '2 dishes/experiments' means that the data (plots) shown were obtained from cells cultured on two separate dishes for each round of experiment. They were then repeated for three times on different days. Data obtained in each round were piled for statistical analysis. We mentioned dishes because to image cells, the LSM 780 microscope was run at a 'tile scanning' mode, which collected images at multiple positions of the same dish to ensure that the n value (cell number) of each experiment was larger than 20. As indicated in corresponding legends, all imaging experiments have been performed for three times and were indeed performed on three different days.

2) *None of the western blots show molecular weights, these should be included.*

We thank the reviewer for this suggestion. We in fact tried to label the protein molecular weights next to our cropped blots. However, due to space limit, most of the blots have to be cropped to such an extent that molecular weights cannot be labelled on the final figures. As required by the guidelines of *Nature*, we have already provided the uncropped gel scans (the file 'Full scans' in Supplementary Information), on which we have labelled the molecular weights to each blot, as on the example gel shown below. We, therefore, hope the reviewer would allow us not to label the molecular weights on the composite figures. We have discussed about this point with the editor.

3) *Figure 5 is very dense in information and the order of panels makes it difficult to find the proper panel easily. The lay-out of this figure should be improved.*

We thank the reviewer for the instructive comment. We have now re-organised the order of panels shown in Fig. 5 in our revised manuscript. We also moved some panels (original Fig. 5c, g, h, l, o, p, s, t) to ED Figures (listed as ED Fig. 8g, h, l, m and 9b, d, e, f). We have also moved the summary model (re-drawn version) to Fig. 5 (listed as Fig. 5o) – we initially did not do so because of space limit.

D. Appropriate use of statistics and treatment of uncertainties

Statistical tests are appropriate. The only concern is the low number of experiments that were performed, see comment in point C.

Again, we thank the reviewer for this comment. As described above, after providing additional repetitions, we have now assured that all experiments have been repeated at least three times.

E. Conclusions: robustness, validity, reliability

In general, the main conclusions (metformin binding to PEN2 and stimulating interaction with ATP6AP1) are valid and robust, and are based on molecular level insights that were cross-validated in complementary models. These main conclusions are novel and clearly of interest for the field. The large number of experiments and techniques, however, also requires an extensive set of controls, which are sometimes lacking (see comments below). But the confirmation of results in complementary model systems provides a solid basis for the overall conclusions.

The authors state that the “lysosomal PEN2 pool” is responsible for the reported mechanism, but this conclusion remains premature (see comments below).

We thank the reviewer for the comments. We have now carefully validated the localisation of PEN2 and its mutants on lysosome. We also explored how localisation of PEN2 may be regulated according to the suggestions made by the reviewer. We found that about 50% PEN2 was co-localised with the lysosomal marker LAMP2, and the mutants of PEN2 (2A and 20A) showed intact lysosomal localisation (ED Fig. 4d and 5l). In addition, autophagy/endocytosis did not affect PEN2 lysosomal localisation (ED Fig. 3q, r).

F. Suggested improvements: experiments, data for possible revision

1) The authors focused on the effects of metformin on primary hepatocytes at low concentration (5 μ M), but then they turned to MEF and HEK293T cells to identify and characterize the metformin/PEN2/ATP6AP1 interaction. As the authors already indicate, these cells don't express OCT's to take up metformin in the cells, and higher metformin concentrations (200-300 μ M) had to be used to observe the same phenotype observed in hepatocytes. In a study focusing on low concentration effects of metformin, this change to higher concentrations represents a weak point, as it may lead to other side effects.

We thank the reviewer for the insightful comments concerning the different concentrations of metformin for different cell lines. Throughout the manuscript, we have used primary hepatocytes wherever possible. On the other hand, we also used MEFs and HEK293T cells as a system for the analysis of molecular interactions/mechanisms because they are easy to be infected with lentiviruses and are therefore convenient to establish stable cell lines. Indeed, the metformin uptake between MEFs/HEK293T cells are different from that by primary hepatocytes. That is why we not only titrated the concentrations of metformin used in the culture medium, but also determined intracellular metformin concentrations, to make sure that intracellular metformin concentrations achieved in MEFs/HEK293T cells are similar to those of hepatocytes (Fig. 1h and i). We have now established MEFs and HEK293T cells with OCT1 stably expressed. As shown below (current Fig. 1l, m), 2 h of metformin (5 μ M) treatment led to 34 μ M and 35 μ M metformin in MEFs and HEK293T cells, respectively, which are similar to those in primary hepatocytes. The OCT1-expressing MEFs and HEK293T cells have yielded similar results on AMPK activation without elevating AMP:ATP ratios in low metformin.

We also determined PEN2 and ATP6AP1 dependency in those cells. As shown below (current ED Fig. 3h), the knockout of *PEN2* in both MEFs and HEK293T cells with OCT1 stably expressed blocked the 5 μ M metformin-induced AMPK activation. Similarly, re-introduction of ATP6AP1- Δ 420-440 (able to maintain v-ATPase activity while lacking the site for binding to PEN2), rather than wildtype ATP6AP1, into *ATP6AP1*^{-/-} MEFs or HEK293T cells with OCT1 stably expressed blocked AMPK activation under 5 μ M metformin. In comparison, high metformin (500 μ M) could still activate AMPK in PEN2-ATP6AP1 axis-deficient cells (current ED Fig. 5j, also shown below).

a. These implications and limitations should be clearly discussed. Also the concentrations of the Met-P1/2 probes used in the screening experiment (not mentioned) and metformin concentrations and exposure time elsewhere should be carefully reported in the main text, materials & methods and/or in the legends.

Following up on the point above, we have now added discussion on OCT1-deficiency in MEFs and HEK293T cells in our revised manuscript (line 135, page 7).

The concentration of Met-Ps used was 10 μ M, which was decided based on the intracellular concentrations of metformin being 5-20 μ M detected under treatment of low metformin (Fig. 1d, e, h and i), and has been indicated in the revised Method section (line 1326, page 55). We apologise for the omission. Following the instruction by the reviewer, we have now confirmed that all concentrations and the exposure times have been indicated in the revised manuscript, either in the main text or legends/method sections.

b. What is the uptake mechanism of metformin in OCT deficient cells? If uptake in the absence of OCT relies on endocytosis, this may lead to higher subcellular metformin concentrations at the level of the lysosome than in OCT expressing cells. This could lead to binding of metformin to the extra luminal C-terminal binding site on PEN2.

We really do not know through which metformin is uptaken in cells without OCT expression. It seems that metformin could still be transported into cells without OCT1 (e.g., in HEK293T cells, PMID 17476361; and in our current study), or into the liver in which both *OCT1* and *OCT2* are knocked out (PMID 22407892), albeit in a slower rate. One possible explanation might be that there may be other transporter(s) in those cells/tissues. For example, it has been recently reported that OCT3 (PMID 25920679) and SLC19A3 (PMID 26528626) may also play a part in metformin uptake in the liver.

We fully agree with the reviewer that metformin may be transported via endocytosis and we mentioned this possibility in the text (line 272, page 12). As for the binding of metformin to the luminal, C-terminus of PEN2, we showed that mutation of D90 residue of PEN2 which abolishes the binding of metformin on the C-terminus, had no effect on AMPK activation in low metformin (ED Fig. 4g). Therefore, the C-terminus of PEN2 (at least) did not participate in metformin-mediated AMPK activation.

c. Would MEF or HEK293 cells with induced OCT expression lead to a similar induction of the PEN2/ATP6AP1 system at 5 μ M metformin concentration?

We thank the reviewer for the constructive suggestion. As described above, we have now utilised OCT1-expressing MEFs and HEK293T cells and have shown that 5 μ M metformin could activate AMPK via the PEN2-ATP6AP1 axis in these cells.

2) While PEN2 was established as a new molecular target, other genes that play a role may have been easily missed as a consequence of the screening strategy. The authors started with the metformin probe on protein extracts of lysosomal preparations (of MEF cells?? - specify in main text/fig2). This yielded over 3000 proteins (only 219 were lysosomal), which is surprisingly high since lysosomal protein extracts were used, questioning the specificity of the probe and purity of the lysosomal preparation. Conversely, cytosolic regulatory proteins regulating lysosomal membrane proteins may have been missed with this approach focusing on lysosomes. Moreover, 90 additional proteins from 'the list of lysosomal proteins' were added, but this list may be incomplete and cell type specific.

The lysosome extracts were prepared from MEFs as indicated on 'Identification of metformin-binding proteins' section of Methods section of our initial manuscript, and this information has now also been indicated in the legend of Fig. 2a. We chose lysosomal, rather than total/cytosolic, protein extracts to identify the targets of metformin, because that we found that metformin is able to inhibit v-ATPase in purified lysosomes (e.g., Fig. 1c), which suggested that those lysosomal proteins contain mediators that can signal to inhibit v-ATPase to prime AMPK activation. It was also because we had previously found that knockout of *AXIN* or *LAMTOR1*, factors of the lysosomal pathway, abolished metformin-induced AMPK activation (PMID 27732831), which have now been recapitulated (ED Fig. 6a, b). These data all point to the lysosome as being responsible for the signalling of low metformin.

In addition, we used the lysosomal extract as the beginning proteins with the understanding that we would have to validate the screened proteins in the end. We admit that the nomenclature 'lysosomal resident proteins' is a bit arbitrary. According to the published list (PMID 23436907), the 734 'lysosomal resident proteins' were called so because the authors 'validated' them by searching literatures, from a total of 2,385 hit by LTQ-orbitrap mass spectrometer from lysosome extracts. The total number of the proteins chosen as lysosomal proteins

were hence limited by experimental studies available at the time, and is being expanded as more studies are coming out. Although we cannot directly compare the sensitivity between timsTOF Pro and LTQ-orbitrap because we do not have LTQ-orbitrap, we have compared the sensitivity between timsTOF Pro and Orbitrap Fusion Lumos (an upgraded version of LTQ-orbitrap). It was found that timsTOF Pro is more sensitive than the Orbitrap Fusion Lumos as in a 90-min run of 200 ng of HeLa extracts, timsTOF Pro hit more than 6,000 proteins, while Orbitrap Fusion Lumos hit 4,500. That may explain why we identified over 3,000 preys, more than the total of the previously listed 'lysosomal proteins' (PMID 23436907).

In the verification processes, we started with 219 proteins identified from our initial mass spec analysis, which are among the list of 'lysosomal resident proteins'. We also added 58 proteins that had been shown to be related to the lysosome, either according to our mass spectrometry determination of the total proteins in the purified lysosomes or by others in the new literatures. We even went further to include 90 additional proteins in the list of 'lysosomal proteins' that were not among the hits from the MET-P conjugates, with the caution that some proteins, possibly owing to the low expression levels or poor digestion by trypsin, may be missed by mass spectrometry. This resulted in a total of 367 proteins to undergo subsequent verification processes. After gene silencing experiments, we found that knockdown, as well as all the other approaches used in subsequent experiments, of PEN2, but not the others, abolished metformin signalling in MEFs. We have shown that it is PEN2 that physically binds to metformin. As for other regulatory proteins, we fully agree with the reviewer that other cytosolic regulatory proteins may also participate in metformin signalling: many downstream factors of v-ATPase in the lysosomal pathway, such as AXIN, is cytosolic localised and translocate to lysosome in response to metformin. Such a possibility has now been discussed in our revised manuscript (Supplementary Note 5).

a. Despite the successful identification of PEN2, the screening cascade presents several flaws and other relevant targets may have been easily missed. How do the authors exclude that besides PEN2 other genetic factors may play a role that converge at the level of the lysosome?

As explained above, we fully agree that additional regulatory protein(s) may participate in metformin signalling.

b. It will be important to check whether the PEN2/ATP6AP1 pathway is the major mechanism of metformin action in respect to other already reported pathways. Is the expression of PCK1 and G6PC1 altered with the use of low metformin vs. high metformin doses?

As shown in this study, PEN2-ATP6AP1 may be specific for those AMPK-dependent (also related to mTORC1; see our newly obtained results shown in ED Fig. 9s, t, and discussions on Supplementary Note 9) functions. For example, we showed that metformin could still inhibit gluconeogenesis in mice with liver-specific knockout of *PEN2* (current ED Fig. 9r) despite that the hepatic AMPK activation was dampened. Given that metformin could inhibit gluconeogenesis via multiple AMPK-independent pathways, it is possible that *PEN2* may not participate in other, AMPK-independent functions exerted by metformin.

According to the suggestions from the reviewer, we have now determined the mRNA levels of *PCK1* and *G6PC1* in primary hepatocytes isolated from wildtype mice or mice deficient of the *PEN2-ATP6AP1* axis (mice with liver-specific knock out of *PEN2*, or mice with liver-specific knock out of *ATP6AP1* with *ATP6AP1-Δ420-440* re-introduced). As shown below (current ED Fig. 9q), low metformin (5 μM) failed to suppress the expression of *PCK1* and *G6PC1*, consistent with a previous study (PMID 24847880). We also found that blockage of the *PEN2-ATP6AP1* axis did not affect the expression of *PCK1* and *G6PC1* in low metformin.

3) The authors claim that the “lysosomal PEN2 pool” is responsible for activating the lysosomal AMPK pathway, which offers “beneficial effects”. This conclusion needs to be carefully validated.

a) Decreasing V-ATPase activity and thereby affecting lysosomal pH is expected to disturb lysosomal function, which may reduce degradative capacity and impairing autophagy. How would this contribute to cell protection? The lysosomal acidity was determined with lysosensor, but what is the size of the pH effect? Does this affect lysosomal degradative capacity and/or autophagy?

We thank the reviewer for the comments on metformin/AMPK-induced autophagy, which we are also very intrigued by. When we stated that metformin signalling to the lysosomal AMPK pathway has beneficial effects, we mean that it can alleviate fatty liver, improves glucose tolerance, and extends lifespan (in nematodes), similar to that of glucose starvation or calorie restriction (reviewed in PMID 30840912). We also fully agree with the reviewer that autophagy plays critical roles in these beneficial effects (reviewed in PMID 30172870). Following the reviewer’s comments, we determined the autophagy induction, by assessing the protein levels of p62, along with the signals of Lysosensor and p-AMPK, in MEFs treated with low metformin for different periods of time. As shown below (current ED Fig. 9i, j), we found that the decrease of Lysosensor signals and the increase of

p-AMPK signals emerged after 8 h of metformin treatment (especially considering the slow uptake rate of metformin into these cells) and p-AMPK signals remained relatively constant up to 72 h. The decrease of p62, as autophagy indicators, however, only became obvious after 30 h of metformin treatment, which is accompanied with an increase of Lysosensor signal. Therefore, v-ATPase inhibition and AMPK activation seem to occur much earlier than autophagy induction. It remains intriguing as to if the same lysosomes undergo autophagy as those inhibited of the v-ATPase.

One may formally argue that autophagy may in fact happen during early-stages of metformin treatment, and propose that it is the inhibition of early-stage autophagy (caused by the inhibition of v-ATPase) that leads to AMPK activation. To examine this possibility, we treated *ATG5*^{-/-} MEFs with metformin. Although *ATG5* deficiency blocked the induction of autophagy by metformin, it did not affect the activation of AMPK, nor the inhibition of v-ATPase (shown below, left panel). Consistently, cells treated with chloroquine, an inhibitor of autophagy without altering the activity of v-ATPase (as reviewed in PMID 20144757), failed to affect

AMPK activation (shown below, right panel). Together, it is possible that v-ATPase inhibition/AMPK activation act in parallel to autophagy induction.

We determined the lysosomal pH by detecting the fluorescence of LysoSensor™ Green DND-189 dye, because it represents the activity of v-ATPase inside the cells (i.e., short-term exposure of metabolic stresses, like glucose starvation as validated in PMID 31204282). As shown in this study, disruption of metformin-mediated inhibition of v-ATPase, either by mutation on PEN2 or ATP6AP1, all resulted in lower lysosomal pH.

b) The authors generated a number of constructs to alter the subcellular PEN2 location. Data should be presented to show that these fusion proteins are indeed localized to the expected subcellular locations. Overexpression of PEN2 is challenging, since it often mislocalizes, and fusion proteins may get trapped in the ER. This would lead to the false conclusion that only lysosomal PEN2 is involved in the described mechanism. For the HA-PEN2-2A and HA-PEN2-20A, the images are not very convincing and better images should be presented that also show ER markers.

We have now provided the validation data pertaining to the localisation of chimeric PEN2 constructs (current ED Fig. 3w, also shown below). We apologise for the omission.

We agree with the reviewer that overexpression can cause misplaced location of proteins. In this study, the stable, close-to-endogenous expression of PEN2/ATP6A1 was applied wherever applicable, which is ensured by the lentiviral (pBOBI) or adeno-associated viral vectors. The overexpression of PEN2 was applied only for expression and purification of the PEN2 protein from HEK293T cells used for in vitro experiments such as SPR and ITC assays.

We thank the reviewer for mentioning the localisation of PEN2 and its 2A/20A mutants. We have now determined the co-localisation of PEN2 or its mutants with LAMP2 and PDI (ER marker), as assessed by the Mander's overlap coefficients. As shown below (current ED Fig. 4d and 5I), PEN2-2A and PEN2-20A occupy a similar localisation to wildtype PEN2.

c) What would be the % of PEN2 that is localized to lysosomes, and is there more PEN2 recruited to the lysosomes upon metformin treatment? This can be studied via (i) a Western blot in which PEN2 levels are determined on lysosomes purified before and after metformin treatment; or (ii) a time-series with immunofluorescence to see if PEN2 relocates to the lysosomes upon metformin treatment?

We thank the reviewer for the such a constructive suggestion. The localisation of PEN2 before and after metformin treatment has been determined before the initial submission, and is shown below (12-h treatment of metformin on MEFs, which sufficiently activates AMPK, as a representative; also listed as current ED Fig. 3t, u). As assessed by subcellular fractionation, and immunofluorescent staining (with LAMP2 as a lysosomal marker), we found that metformin did not alter the lysosomal localisation of PEN2.

We are afraid that the proportion of lysosome-localised PEN2 could not be precisely determined, because the protein compositions (and hence amount) between lysosome and whole cell extracts are different. Therefore, determining the protein concentrations of lysosomes and cell homogenates could not derive the relative amount of PEN2 localised on lysosomes. We therefore could only estimate the proportion of lysosome localised PEN2 via the Mander's overlap coefficient of PEN2 with LAMP2, and it is 57% on average (current ED Fig. 3o, also shown below).

d) What is the mechanism of PEN2 targeting to lysosomes? Is the targeting of PEN2 to lysosomes a consequence of autophagy induction, or can lysosomal targeting motifs be recognized in the protein sequence? Can lysosomal targeting be prevented by endocytosis or autophagy blockers?

Currently, we really have no idea through which mechanism PEN2 is targeted to the lysosome, because no apparent lysosome-targeting sequence is found on PEN2. According to the PEN2 localisation determined by us and others, it seems that PEN2 is widely distributed across the cell, including lysosome, ER and plasma membrane (PMID 24413617). However, the lysosomal membrane-localised PEN2 can conveniently shift in *cis* to meet the lysosomal membrane-localized v-ATPase,

which constitutes its signalling to the AMPK activation under metformin, while other pools of PEN2 may not be able to encounter v-ATPase geographically.

Following the suggestion made by the reviewer, we have now determined the roles of autophagy/endocytosis in PEN2 localisation. As shown below (current ED Fig. 3q), knockout of *ATG5*, or treatment of chloroquine, 3-MA or bafilomycin A, did not affect the localisation of PEN2. Similarly, Dynasore and Dyngo-4a which inhibit the dynamin-mediated endocytosis (including the clathrin-coated, pit-mediated endocytosis, and the fast endophilin-mediated endocytosis), Nystatin and methyl- β -cyclodextrin which inhibit the clathrin-independent carrier/glycosylphosphatidylinositol-anchored protein-enriched early endocytic compartment endocytosis and the caveolae-dependent endocytosis, all failed to disrupt lysosomal localisation of PEN2. We also used Cytochalasin D which inhibits macropinocytosis and phagocytosis, and found that it did not affect the localisation of PEN2 (current ED Fig. 3r). Therefore, autophagy or endocytosis may not be involved in controlling PEN2 localisation.

e) It is expected that the entire cellular pool of PEN2 binds metformin. Does this lead to other complex formations in other organelles?

As discussed above, it is possible that the physical proximity of PEN2 to v-ATPase would allow PEN2 to trigger its signalling to the AMPK pathway. In other words, metformin may also bind to PEN2 that is localized in other compartments, yet it cannot trigger the activation of lysosomally localized AMPK unless PEN2 is able to translocate across the membrane compartments to the vicinity of v-ATPase. Indeed, as suggested by the reviewer, other PEN2 that is not associated with v-ATPase may participate in the formation/regulation of other complexes. As discussed below (and also point 5 raised by Reviewer #2), we found that metformin, probably through binding to PEN2, could enhance the activity of γ -secretase activity (ED Fig. 4j). While metformin did not promote the γ -secretase complex formation (ED Fig. 5c), it might facilitate the binding of another unknown factors to γ -secretase, thus regulating the activity of γ -secretase. We have now discussed this in our revised manuscript (Supplementary Note 3).

4) The interaction between PEN2 and ATP6AP1 and its regulation by metformin is elegantly proven in this manuscript. The authors call the PEN2/ATP6AP1 effect on AMPK a 'serendipitous' intersection modulated by metformin. I would refrain from such wording, since it is equally possible that this mechanism has been evolved to support a clear physiological function, as it is conserved from *C. elegans* to mammals. Are there cell conditions known that lead to PEN2/ATP6AP1 interaction in the absence of Metformin?

We thank the reviewer for the comment. In our understanding of the English word, ‘serendipity’ refers to a situation where some seemingly unrelated events can take place coincidentally to yield unexpected outcomes; in this case, both metformin and glucose starvation join the same route to AMPK activation to elicit beneficial outcomes. We imply that in hindsight, if metformin does not ‘borrow’ the pathway, it may not have all the benefits without obvious adversary effects. We have now changed it to ‘happen to’ or ‘coincidentally’.

5) Several controls for the various constructs/mutants are lacking. The authors state that the HA-tagged *PEN2* variants are expressed at close-to-endogenous levels, which is more physiological. Can this point be demonstrated on blots?

We apologise for such an omission. The validation data for the HA-tagged *PEN2*, *PEN2*-2A and *PEN2*-20A re-introduced into *PEN2*^{-/-} MEFs and HEK293T cells have now been provided (shown below, current ED Fig. 4c). The validation data for HA-tagged *ATP6AP1*, *ATP6AP1*-Δ420-440, and *LAMP2*TM-*ATP6AP1* re-introduced into *ATP6AP1*^{-/-} MEFs and HEK293T cells, are also provided (shown below, current ED Fig. 5g).

*It is not always shown whether the protein mutants/fusions remain functional, e.g. does mutated *PEN2* maintain γ -secretase activity?*

We thank the reviewer for mentioning the issue of γ -secretase activity (also requested by Reviewer #2). As shown below, it seems that the mutations of PEN2 (2A and 20A) do not affect its interaction with other γ -secretase subunits (shown below, current ED Fig. 4i).

In fact, PEN2-2A was found to increase the activity of γ -secretase, as determined by the cleavage of Notch protein (PMID 10206645), while PEN2-20A inhibited the activity of γ -secretase (shown below, current ED Fig. 4h) (LE, long exposure; SE, short exposure). Given that both 2A and 20A mutants of PEN2 failed to transmit metformin to AMPK activation, it is likely that the activity of γ -secretase is not associated with AMPK activation.

For the ATP6AP1 mutant: does it affect V-ATPase function? Loss of PEN2/ATP6AP1 interaction by expressing the delta420-440 mutant prevents the decrease in V-

ATPase activity, but the expression of this mutant might result in a constitutively active V-ATPase. A control experiment showing pharmacological inhibition of the V-ATPase after expression of mutant ATP6AP1 would make this conclusion more robust.

We thank the reviewer for this constructive suggestion. We have now treated *ATP6AP1*^{-/-} MEFs re-introduced with ATP6AP1-Δ420-440, as well as those with vector control, with concanamycin A (conA). As shown below (current ED Fig. 5i), conA significantly inhibited the acidification of lysosomes in *ATP6AP1*^{-/-} MEFs expressing ATP6AP1-Δ420-440.

The confocal imaging showing the lysosomal localization of HA-PEN2, HA-PEN2-2A and HA-PEN2-20A is unconvincing. HA-PEN2 appears to have a stronger lysosomal localization compared to HA-PEN2-2A and HA-PEN2-20A. Quantification of LAMP2 colocalization would more clearly demonstrate if they are localized equally to the lysosomes.

As discussed above (point 3b), we have now determined the Mander's overlap coefficient of HA-PEN2/LAMP2, HA-PEN2-2A/LAMP2 and HA-PEN2-20A-LAMP2, showing that the two mutants of PEN2 do not show significant differences in lysosomal localisation (current ED Fig. 4d and 5l).

6) It is shown that inhibiting the V-ATPase leads to AXIN recruitment to the lysosomes, however it is not shown if this also activates AMPK.

We thank the reviewer for this question. We have shown inhibition of v-ATPase plays a priori role for AXIN to translocate onto the surface of the lysosome (PMID 25002183) and AMPK activation (PMID 31204282). One recent study reported that a novel v-ATPase inhibitor, archazolid, could recruit AXIN to v-ATPase and activate AMPK (PMID 31604821). We have also performed an additional experiment by treating *PEN2*^{-/-} MEFs or *ATP6AP1*^{-/-} MEFs expressing ATP6AP1-Δ420-440 with the v-ATPase inhibitor concanamycin A (conA), and found that conA was sufficient to facilitate AXIN translocation to the lysosome membrane, and activated AMPK in both genotypes (shown below, current ED Fig. 6o, p), indicating that the step of the inhibition of v-ATPase is a downstream event of the PEN2-ATP6AP1 axis, but upstream of AXIN translocation. For metformin, it binds to PEN2 and relies on ATP6AP1 to inhibit the v-ATPase, leading to AXIN translocation, ultimately leading to AMPK activation by the AXIN-tethered LKB1.

7) Would metformin/PEN2/ATP6AP1 play a role in mouse primary hepatocytes from a diabetic mouse model (fed with high fat diet), which is a relevant model to study metformin effects?

We thank the reviewer for such a constructive suggestion. We have now isolated primary hepatocytes from 4-month-HFD treated mice and treated them with 5 μM metformin. As shown below (current ED Fig. 8f, k), liver-specific knockout of *PEN2*, or re-introduction of ATP6AP1-Δ420-440 into mouse liver with *ATP6AP1* knocked out, blocked metformin-induced AMPK activation.

8) The activity of the γ -secretase complex seems not required for the metformin effect on V-type ATPase and AMPK.

a. However, does metformin binding to PEN2 change the γ -secretase activity? Additionally, it is not shown whether the mutant forms of PEN2 still form functional γ -secretase complexes.

We thank the reviewer for mentioning about the possible effects of metformin on γ -secretase activity, it was also requested by Reviewer #2 (point 5). As shown below (current ED Fig. 4j), we found that metformin increased the activity of γ -secretase (approximately by 50%).

As described above (point 5), the mutants of PEN2 (2A and 20A) may not affect its interaction with other γ -secretase subunits (current ED Fig. 4i).

b. While an effect of NCT knockout on PEN2 levels is shown here, the loss of either APH1 or PSEN appears to have no effect on PEN2 mediated AMPK activation. However, knockout of PSEN is reported to destabilize PEN2 and cause ER-retention (Sannerud et al., 2016; Wang et al., 2004). The authors do not show PEN2 levels or subcellular localization in their MEF APH tripleKO or PSEN doubleKO cells.

We fully agree with the reviewer that the stability of a specific γ -secretase subunit is tightly regulated by others. We have now determined the protein levels of PEN2 in PS1/PS2-DKO MEFs, and observed approximately 15% decrease of PEN2 in those cells (shown below, current ED Fig. 3n). We have also shown that knockout of *NCSTN* severely destabilised PEN2 (decrease of about 70%, shown below, current ED Fig. 3l). However, knockout of APHs in MEFs did not lead to destabilisation of PEN2 (shown below, current ED Fig. 3n). It is possible that the cross-regulated stability of γ -secretase subunits might be cell-type dependent.

As for the PEN2-mediated AMPK activation in low metformin, although knockout of PS1/PS2 led to a 15% decrease of PEN2, it did not affect AMPK activation (Fig.

2l). Therefore, the mild decrease of PEN does not affect metformin-induced AMPK activation.

In addition, they do not confirm the lysosomal localization of HA-PEN2 expressed in NCT KO cells.

We thank the reviewer for asking for such an important control. As discussed below, we have now determined the localisation of HA-tagged PEN2 in *NCSTN*^{-/-} MEFs. Compared with the wildtype control, we found that knockout of *NCSTN* did not affect the localisation of PEN2 (current ED Fig. 3s).

As *PEN2* levels and localization are known to be disturbed when γ -secretase complex formation is prevented, it is necessary to confirm lysosomal *PEN2* expression in all γ -secretase subunit knockout cell lines.

Following the suggestion by the reviewer, we have now determined the co-localisation of *PEN2* with *LAMP2* in *PS1/PS2*-DKO and *APH1a/b/c*-TKO in MEFs by immunofluorescent staining. Since knockout of *NCSTN* largely suppresses the protein levels of *PEN2*, we determined the stably expressed, HA-tagged *PEN2* in

NCSTN-KO MEFs. As shown below (current ED Fig. 3s), we found that the localisation of PEN2 was not changed in cells with knockout of the other γ -secretase subunits. Together, in MEFs, it is the stabilisation, rather than localisation of PEN2, that is regulated by other γ -secretase subunits.

Further exploring the role of γ -secretase activity through inhibitors could strengthen this novel γ -secretase-independent function of PEN2.

We apologise for not able to make it clearer. Such assays were shown in our original submission. As shown in current ED Fig. 3j (original ED Fig. 3k, also shown below), we found that MEFs treated with two γ -secretase inhibitors, DAPT or RO4929097, did not affect metformin-induced AMPK activation. As also described above, the newly obtained data showed that PEN2-2A mutation significantly elevated the activity of γ -secretase, but failed to transmit metformin to AMPK activation. These results again suggest that the activity of γ -secretase is not associated with AMPK activation.

c. Both NCT and PSEN have been shown already to interact with the V-ATPase (Nylandsted et al., 2011; Wakabayashi et al., 2009). Additionally, NCT is also

reported to possibly interact with ALDOA. The pull-down of ATP6AP1 in supplemental figure 5c, does show coIP of PSEN1, PSEN2 and NCT, although less pronounced than that of PEN2. Are these interactions that are reported in literature without metformin treatment, relevant for the stronger interaction between PEN2 and ATP6AP1 upon metformin administration?

We thank the reviewer for reminding us of the two papers. We fully agree that the interactions between NCSTN/PSENs and v-ATPase/ALDOA, and between PEN2 and ATP6AP1, may contribute to the basal interaction between v-ATPase and γ -secretase, and that metformin, through tightening the PEN2-ATP6AP1 interaction, can enhance such an association. We apologise for not discussing these two papers in detail, because their interaction network and biological implications are very complex. In agreement with the previous reports, we detected interactions between ATP6AP1 and all of the γ -secretase subunits, which were enhanced by metformin at low doses (Extended Data Fig 5c). We also found that the interaction between v-ATPase and γ -secretase is abolished in *PEN2*^{-/-} MEFs with HA-tagged PEN2-20A (lacking the interface for ATP6AP1), as shown below, current ED Fig. 5d. These results indicate that it is the PEN2 protein in the holoenzyme of secretase that mediates the interaction with v-ATPase.

As for the localization of PEN2, according to our new data obtained during the revision, we found that PEN2 seems to be able to reside in the lysosomal membrane on its own (in PSEN-KO or APHs-KO MEFs, as described above, ED Fig. 3s). We have

now cited the two references and discussed these possibilities in our revised manuscript.

d. γ -secretase has been linked previously to lysosomal acidification, with PSEN deficient cells having an increased lysosomal pH. PEN2 knockdown seems to also have a modest effect on lysosomal pH (Fig. 1c). Is this significant? And if so, how would this contribute to the interpretation of the results?

Thank you for reminding us of that previous study. However, we have shown that PEN2 knockout did not lead to inhibition of v-ATPase in primary hepatocytes, MEFs or HEK293T cells (Fig. 2c, f and ED Fig. 3a, g). One possible explanation for the discrepancy is that the effects of γ -secretase on lysosomal acidity might be cell-type specific: neuronal cells may regulate lysosomal acidity through mechanisms different from those of hepatocytes, MEFs and HEK293T cells (PMID 20541250). In addition, it has been shown that lysosomes in MEFs with PS1/2-DKO show intact pH (PMID 22753898). We have added the discussion (Supplementary Note 2).

e. Metformin has been reported as a neuroprotective agent. Would the interaction of metformin with PEN2 possibly play a role in a neuronal model or in models of Alzheimer's disease (such implications can be highlighted in the discussion)?

We thank the reviewer for raising this interesting possibility. Indeed, during the revision, we found that metformin, probably through binding to PEN2, could enhance the activity of γ -secretase activity in MEFs (ED Fig. 4j). In addition, one previous study reported that metformin could promote the γ -secretase-mediated production of A β 42, as a result of autophagy induction in Tg6799 AD model mice (PMID 26967226), although (possibly owing to the different mouse models) one previous study showed that metformin instead prevents amyloid plaque deposition and memory impairment in APP/PS1 mice (PMID 29253574). We have now discussed this issue (Supplementary Note 3).

G. References: appropriate credit to previous work?

Yes.

H. Clarity and context: lucidity of abstract/summary, appropriateness of abstract,

introduction and conclusions

Clear and well-written. The abstract gives a good overview of the study, and the introduction shows the need for their investigation. Moreover, the conclusions are clearly written and well structured. The schematic diagram, shown in supplemental figure 9, nicely sums up the conclusion and would be good to have in the main figures, not just in supplementals. Indicate the lysosomes on figure 2n.

We thank the reviewer for such an encourage. We have now slightly modified the schematic diagram and moved it to the main figure (Fig. 5o, also shown below). We have also indicated lysosomes on our TEM image (black triangle).

Responses to Reviewer #4

We sincerely thank the reviewer for the constructive criticisms, which have helped substantiate the conclusions. We have now addressed all of them, either by providing clarification, by presenting additional results, or by performing new experiments. In particular, we have now determined the roles of ROS in low metformin-induced AMPK activation in both cells and nematodes, and explored the relationship between AMPK activation and v-ATPase inhibition. We also determined the absolute concentrations of AMP, ADP and ATP in cultured cells, mouse livers and nematodes in low metformin. Point-by-point responses to the reviewer's points are given below, with the reviewer's comments given verbatim in plain italic type, and our responses in bold:

Referee #4 (Remarks to the Author):

The authors show that PEN2 bind metformin at low doses which is then recruited to the ATP6AP1 subunit of v-ATPase, to inhibit the latter. By employing Ragulator and Axin, this complex then tethers upstream LKB1 to AMP-dependent kinase to activate the latter.

The data presented provide a novel and potentially AMP-independent model for the mode of action of metformin at low/physiological doses.

We thank the reviewer for the kind comments.

(i) Given the downstream read-out, namely activation of AMPK, it appears necessary to exclude any impact of such AMPK activation on the supposedly upstream phenotypes.

For example (Figs. 1a, j, k): is the deacidification of lysosomes impacted by activation or inhibition of AMPK? To answer this, a corresponding experiment using AICAR to activate AMPK, or pre-treatment with Compound C, a competitive AMPK inhibitor, before addition of metformin should be tested. If either AICAR or Compound C show an effect on (de)acidification, all subsequently corresponding experiments should be expanded accordingly.

We thank the reviewer for pointing out this issue. In Fig. 1c, we show that metformin inhibits v-ATPase activity in vitro, indicating that AMPK may not be involved in the v-ATPase inhibition. Following the suggestion made by the reviewer, we have now treated MEFs with compound C or AICAR. As shown below, we found that compound C significantly decreased lysosomal pH, while AICAR significantly increased lysosomal pH. However, neither of the effects was attributed to AMPK, because AICAR and compound C could still modulate lysosomal pH in AMPK α 1/2-DKO MEFs (shown below, $n = 20$ for each treatment, and P value by two-way ANOVA, followed by Tukey). Possible explanations may be that ZMP, a metabolite of AICAR, may compete for ATP binding of v-ATPase and therefore inhibit v-ATPase (as it could similarly compete for ATP binding to glucokinase, see PMID 16567505), and compound C may inhibit PKA (PMID 21740966) to increase v-ATPase activity as reported previously (PMID 32409581).

We therefore performed additional experiments by pre-treating MEFs with the allosteric activator of AMPK A769662, and found that the drug did not alter the lysosomal pH (shown below, upper panel, $n = 20$ for each treatment, and P value

by two-way ANOVA, followed by Tukey). Metformin could still elevate lysosomal pH in MEFs pre-treated with A769662. In addition, we found that metformin could still elevate lysosomal pH in AMPK α 1/2-DKO MEFs (shown below, lower panel, $n = 20$ for each treatment, and P value by two-way ANOVA, followed by Tukey), indicating that inhibition of v-ATPase by metformin does not require AMPK, similar to those observed during glucose starvation (PMID 25002183).

(ii) Likewise and as acknowledged by the authors, metformin is a known inhibitor of complex I of the electron transport chain (PubMedID 15047621 and others), and by this produces reactive oxygen species (ROS). Hence, the authors may wish to pre-incubate their respective cell lines with an antioxidant, e.g. N-acetyl cysteine, for at least 16 hrs, and repeat the experiments as depicted in Figs. 1a, j, k under such conditions.

We thank the reviewer for raising the important issue. It has been shown that metformin can increase ROS production in mammalian cells (e.g., in 3T3-L1 cells treated with 4 mM metformin, PMID 18687824). Following the suggestion, we have now determined the effects of N-acetyl cysteine (NAC)-treatment on low metformin-induced AMPK activation. As shown below [current Extended Data (ED) Fig. 9o], we found that pre-treatment of NAC (16 h and 20 h) did not alter the inhibition of v-ATPase under low metformin, either in primary hepatocytes, MEFs or in HEK293T cells. We also found that NAC did not alter the activation of AMPK (ED Fig. 9n).

ED Fig. 9o

ED Fig. 9n

Intrigued by this reviewer's suggestion, we also performed an additional experiment to determine the levels of mitochondrial ROS (through determining the fluorescence of CellROX dye) in MEFs, because ROS has been reported as an important modulator to AMPK activity (e.g., PMID 20519126). We found that low metformin could not increase ROS (shown below, also listed as ED Fig. 9m). Our results therefore suggest that low metformin (without causing elevation of AMP) could activate AMPK in an ROS-independent manner. Such a conclusion may also be supported by the previous reports that ROS can only activate AMPK in a condition that mitochondria are severely damaged and AMP levels were elevated (PMID 25084564 and 30232152).

(iii) According to Fig. 1d the intracellular concentration of metformin is 8fold higher than in the media, in primary hepatocytes. By contrast, (Fig. 1h and i), in MEFs and HEK293s, the intracellular concentration is 12 times *lower* than in the media. Despite the differences in cell types this is difficult to understand, however highly relevant given the proposed distinction between low- versus high-dose metformin action.

We thank the reviewer for pointing out this critical issue (also by Reviewer #3). The lower rates of metformin accumulation in MEFs/HEK293T cells may be attributed to their lack of metformin transporter, OCTs. We fully agree with the reviewer that such a difference may lead to distinct, even unexpected mechanisms among those cells when treated with different concentrations of metformin. Therefore, we introduced OCT1 (stably expressed) in MEFs and HEK293T cells. We first verified OCT1 expression in these cells. As shown below (current Fig. 1l, m), 2 h of metformin (5 μ M) treatment led to 34 μ M and 35 μ M metformin in MEFs and HEK293T cells, respectively, which were comparable to those in primary hepatocytes. Such a treatment of metformin could readily cause activation of AMPK, without elevation of AMP:ATP ratios, in those cells, similar to those in hepatocytes.

We also determined *PEN2* and *ATP6AP1* dependency in those cells. As shown below (current ED Fig. 3h), the knockout of *PEN2* in both MEFs and HEK293T cells with OCT1 stably expressed blocked the 5 μM metformin-induced AMPK activation. Similarly, re-introduction of *ATP6AP1-Δ420-440* into *ATP6AP1*^{-/-} MEFs or HEK293T cells with OCT1 stably expressed also blocked AMPK activation. By comparison, high metformin (500 μM) could still activate AMPK in cells in which *PEN2-ATP6AP1* axis was deficient (current ED Fig. 5j, also shown below).

(iv) The authors throughout show AMP, ADP, and ATP as ratios only. While if find this acceptable for the main figures, the absolute values for each of the three should be provided in the extended data section, preferably normalized to protein. This applies to Figs. 1d, h, i, e, g, Figs. 5c, f, h, o, s, t.

We thank the reviewer for mentioning the absolute concentrations of AMP, ADP and ATP. We chose to determine the AMP:ATP and ADP:ATP ratios, because such ratios determines whether AMPK could be activated by energy stress. It was initially put forwarded as ‘adenylate energy charge’ (PMID 6027798), and was later simplified as the ratio of AMP:ATP and ADP:ATP in the context of AMPK (PMID 21680840). As told by Prof. Grahame Hardie (personal communication), it has been a convention to use AMP:ATP and ADP:ATP ratios for indicating the energy status. We would also like to note that on some occasions, if the absolute amount of AMP is increased, ATP might be also increased. As shown in a previous study from the Hardie’s group, effects of AMP increase on AMPK activation may be blocked when more ATP was also added to in-vitro AMPK activation assays (PMID 24093679).

Following the suggestion made by the reviewer, we have now determined the absolute concentrations of AMP, ADP and ATP. Since such a quantification process requires standard curves (generated from isotope-labelled AMP, ADP and ATP) to be run alongside samples await being quantified. Therefore, we re-performed all experiments mentioned by the reviewer (original Fig. 1d, g, h, i and Fig. 5c, h, o, s; original Fig. 1e, 5f and 5t did not refer to AMP:ATP) during the revision (shown below, also listed as current Fig. 1d, g, h, l and ED Fig. 8g, 8l, 9b, 9d). Note that to avoid potential matrix effects among different types of samples (i.e., extracts from MEFs, HEK293T cells or primary hepatocytes, or from mouse livers or nematodes),

the standard curves for each type of samples were generated using respective cell/tissue type.

Fig. 1d

Fig. 1g

Fig. 1h

Fig. 1i

ED Fig. 8g

ED Fig. 8I

ED Fig. 9b

ED Fig. 9d

(v) Fig. 5a, i: Please provide corresponding plasma insulin values, including AUCs.

Indeed, as insulin is closely related to the function of GLP-1, we have now measured the values of insulin in intestinal specific *PEN2*-KO mice treated with metformin. As shown below (current Fig. 5a), deficiency of *PEN2* impaired insulin secretion under both basal and glucose-induced conditions.

We determined the levels of insulin in HFD-fed mice treated with low metformin for 4 months, and found that insulin sensitivity has been significantly improved. As shown below (current Fig. 5c and 5f), decreased insulin levels were detected in the metformin-treated groups, but not in mice with hepatic knockout of PEN2, or in mice with re-introduced expression of ATP6AP1- Δ 420-440.

(vi) In the *C. elegans* data set, a concentration of 50 *milli*M of metformin was used. While based on the literature re. lifespan extension in nematodes, it is highly questionable whether this concentration qualifies as "low-dose".

We thank the reviewer for raising the issue on the dosage of metformin used in nematodes. We have in fact determined the lowest concentrations of metformin required for the lifespan extension on nematodes before submission. As shown below (listed as current ED Fig. 9a), we found that 50 mM of metformin is the lowest concentration that is effective to cause lifespan extension according to our experiments, within the previously reported range, 25 to 50 mM (PMID 20090912 and 23540700). Given that such a concentration of metformin could activate AMPK without elevating AMP:ATP and ADP:ATP ratios, we hope the reviewer would agree that the dose used for nematode culture medium could be regarded as 'low dose'.

The data provided in Fig. 5p indicate no *relative* change in intra-organismal metformin concentration. However and unlike for the cell lines, the *absolute* concentrations for metformin is missing. Please provide these (as for cell lines), also normalized for protein.

We showed relative, rather than absolute, concentrations of metformin in *C. elegans*, because we do not know the exact cell volumes of *C. elegans*. We sincerely thank the reviewer for suggesting us to use protein amounts to normalise the results. We have now re-determined the absolute amounts of metformin accumulated in nematodes, expressed with the unit 'nmol/mg protein', as shown below (current ED Fig. 9e, f).

(vii) In line with the previous, the reviewer anticipates that in *C. elegans*, the intra-organismal concentration at 50 mM supplementation is much higher than in cells. If so, the authors will have to identify the lowest concentration of metformin that still suffices to extend lifespan (and hopefully also qualifies as "low dose").

As aforementioned (current ED Fig. 9a), we found that 50 mM of metformin is the lowest concentration effective in lifespan extension of nematodes. Given that such a concentration of metformin could activate AMPK without elevating AMP:ATP and ADP:ATP ratios, we hope the reviewer would agree that the dose of metformin used in the nematode culture medium can be defined as ‘low dose’.

In this regards, the authors may wish to consider using heat-inactivated OP50, to prevent degradation of metformin by the bacteria.

We thank the reviewer for the suggestion. Although heat inactivation indeed blocks the degradation of metformin by bacteria, it has been shown that both the folate metabolism pathway in living bacteria (PMID 23540700) and the AMPK pathway in *C. elegans* (PMID 20090912) are required for the lifespan extension of nematode under metformin treatment. Therefore, heating may not be applicable to the system.

To determine for possible bacterium-mediated degradation of metformin as suggested by the reviewer, we determined the metformin concentrations in nematodes during the lifespan experiment. As shown below (also listed as ED. Fig. 9g; day 1 is the day when experiment starts, and day 3 is the day for plate refreshing), the metformin concentration keeps at relative stable levels during the course of the experiment, indicative of no degradation of metformin by bacteria.

Such heat-inactivation, however, prevents usage of HT115 (as in Fig. 5m), so in this case PEN2 needs to be inactivated by CRISPR/Cas9 in the nematode.

We thank the reviewer for this constructive suggestion. We in fact did try to knock out *PEN2* in nematode during this study. However, knockout of *PEN2* seems to be lethal to *C. elegans*, consistent with a previous report (PMID 12110170). That is why we only knocked down *PEN2* in nematodes. We have now provided related information in our revised manuscript (line 916, page 39).

(viii) Given the rather high concentration of metformin, it is particularly important to exclude a role of metformin-mediated ROS formation through inhibition of complex I. To exclude this, N-acetyl cystein can be used again, and the complex I inhibitor rotenone at 100 nM can be used as a positive control (PubMedID 24199155).

As described for criticism above (Point vi), we hope that the reviewer would agree that the metformin concentration we used in the culture plates for nematodes represents low dose, especially because metformin at this concentration failed to elevate AMP:ATP ratios therein. We also determined mitochondrial ROS levels in nematodes under low metformin treatment, and found that metformin could elevate ROS by 37%, which could be blocked by NAC (shown below, current Fig. ED Fig. 9k). However, NAC failed to alter p-AMPK levels in *C. elegans*, as seen also in mammalian cells (Point ii). These results are consistent with a previous study suggesting that in nematodes, ROS (and subsequently Nrf signalling) may act downstream of AMPK (PMID 20090912).

As for the role of ROS in lifespan, it has been shown that ROS generated under metformin treatment (PMID 24889636) similar to those generated under dietary or glucose restriction (PMID 12559405 and 17908557), has been shown to play a

critical role in lifespan extension in *C. elegans* (PMID 21151885). We have now determined the lifespan of nematodes (treated with metformin) in the presence or absence of NAC, along with rotenone as a positive control as suggested by the reviewer. As shown below (current ED Fig. 9I), NAC could block the lifespan extension mediated by metformin/rotenone. Together, ROS, as a consequence of AMPK activation, plays a role in lifespan extension in nematodes.

(ix) Extended Data Table 1: Please provided not only number of nematodes scored, but rather total number, as well as number of nematodes *censored*.

We thank the reviewer for mentioning about the censored nematodes and apologise for such an omission. We have now provided these data and updated ED Table 1.

(x) How many times were the nematodes used in Fig. 5q backcrossed to N2, to exclude presence of additional random mutations?

The nematodes were backcrossed to N2 six times and was indicated in our Methods section (line 996, page 42), and this has now been indicated in the legend to current Fig. 5k.

Reviewer Reports on the First Revision:

Referee #1 (Remarks to the Author):

In this revised manuscript the authors have added considerable new experimental data to

address the reviewer's concerns and is substantially improved.

Major points

- One intriguing point is the potency of low metformin concentrations (as low as 5 μ M) to activate AMPK in primary hepatocytes at very short times (less than 1 hour treatment). Previous studies failed to report AMPK activation below 40 μ M metformin when primary hepatocytes are incubated for 21 hours (J Biol Chem. 2014 Jul 25; 289(30): 20435–20446, see Fig 5D) or 100 μ M metformin for 7 h (J Biol Chem. 2016 May 13; 291(20): 10562–10570, see Fig 1A). These differences are puzzling. Comment from the authors would be appreciated.

- Previous studies demonstrated that metformin inhibits de novo fatty acid synthesis in mouse primary hepatocytes with IC50 values of 425 μ M (Biochem J. 2015 May 15; 468(1): 125–132). Thus, it remains unclear whether low dose of metformin and lysosomal-dependent AMPK activation is sufficient to alter hepatic lipid accumulation. This reviewer agreed that performing in vivo experiments will request several months and is too long. However, this can be demonstrated in primary hepatocytes from liver-specific AMPK-DKO mice. In addition, as the authors previously reported the existence of various pools of AMPK that are differentially activated leading to the phosphorylation of different sets of targets, it would be of interest to validate in this context that lysosomal-dependent AMPK activation is sufficient to limit hepatic lipid synthesis. Similarly, examination of AMPK-dependency on the inhibition of mTOR signalling by in low metformin concentration could be demonstrated by using primary hepatocytes (these results will give support to the data obtained from MEF and HEK 293 cells).

Minor points

- In Figure 1l and m, it is unclear whether the middle and right panels show data from OCT1-expressing cells. It is surprising that intracellular metformin concentration is not increasing when cells are incubated with 5 μ M metformin when expressing OCT1. Control cells without OCT1 expression should be added.

- line 95: change decrease to decrease.

Referee #2 (Remarks to the Author):

The authors have addressed most of my comments although the quality of the SPR data is still very low and currently unacceptable. The data is critical as it directly addresses the

main argument of the study, and cannot be excluded as the other biophysical analyses with DSC and ITC are themselves unconvincing. The T200 should be more than capable of measuring the PTEN/metformin interaction with greater consistency, which raises concerns to me about experimental design.

Replicates do not overlay at all. It seems that the pink replicates were run first and regeneration of the surface was not complete therefore subsequent replicates demonstrate non-specific binding characterised by incomplete equilibrium, linear association and incomplete dissociation. Another problem could be low immobilisation levels which the authors need to state. Combined, I do not see how meaningful data can be extrapolated from these curves.

I strongly recommend the authors include a negative control protein such as PTEN2-2A and seek expert advice on this technique.

Referee #3 (Remarks to the Author):

The authors have adequately addressed all my comments, included additional controls and conducted experiments that have strengthened their conclusions. I have no further comments.

Referee #4 (Remarks to the Author):

The authors have addressed several of the reviewer's questions and concerns more than adequately.

I still have one key concern (as already raised in my initial comments) which is the high concentration of metformin required for lifespan extension in *C. elegans*.

I do understand that, at least in the authors' hands, 50 mM is the lowest dose capable of extending lifespan, as now experimentally shown.

Nevertheless a previous publication (ref. 113, Fig. S7) has shown lifespan extension at doses as low as 6.25 mM, i.e. 8-fold lower than in the current manuscript.

The authors now show that metformin does **not** increase ROS in MEFs (= low-dose),

however metformin **does** increase ROS in *C. elegans* (= high-dose).

Also they show that this increase in ROS is **required** for extension of lifespan, as previously established for other compounds/interventions (ref. 114).

The authors indicate that this increase in ROS would be a consequence of AMPK activation, given the known activation of OXPHOS via AMPK phosphorylation, and as also previously established in nematodes (ref. 114).

On the other hand and concerningly, metformin at 8-fold lower doses than 50 mM has been shown to induce ROS in *C. elegans* via inhibition of complex I (ref. 113).

The response to reviewers' & manuscript however claims that metformin's primary site of action would be different from such complex I inhibition (as indeed and convincingly shown in the cell culture data set).

Nevertheless and particularly the high concentrations of metformin used in *C. elegans*, plus the beforementioned findings (ref. 113) re. complex I being the primary target of metformin in *C. elegans* and related extension of lifespan do now require one additional experiment:

Does metformin at 50 mM still extend *C. elegans* lifespan in the **absence** of AMPK? If so, this would be a clear indication of complex I inhibition, especially since the newly added data in Extended Data Fig. 9l indicate that such ROS is required for lifespan extension in *C. elegans*.

This can be easily and quickly tested by applying metformin at 50 mM to *aak-2* mutants, as readily obtainable from the CGC (and probably available in the authors' laboratory).

Also and for completeness, could the authors please include HEK193s and primary hepatocytes into the newly added Extended Data Fig. 9m panel (currently MEFs only, while all other experiments were shown for all three cell types)?

Minor comment: Fig. 5f / right panel / Y-axis should read "plasma insulin" rather than "serum insulin".

Author Rebuttals to First Revision:

Responses to Reviewer #1

Point-by-point responses to the reviewer's comments are given below, with the reviewer's comments given verbatim in plain italic type, and our responses in bold:

Referee #1 (Remarks to the Author):

In this revised manuscript the authors have added considerable new experimental data to address the reviewer's concerns and is substantially improved.

We thank the reviewer for the positive evaluation.

Major points

- One intriguing point is the potency of low metformin concentrations (as low as 5 μ M) to activate AMPK in primary hepatocytes at very short times (less than 1 hour treatment). Previous studies failed to report AMPK activation below 40 μ M metformin when primary hepatocytes are incubated for 21 hours (J Biol Chem. 2014 Jul 25; 289(30): 20435–20446, see Fig 5D) or 100 μ M metformin for 7 h (J Biol Chem. 2016 May 13; 291(20): 10562–10570, see Fig 1A). These differences are puzzling. Comment from the authors would be appreciated.

We thank the reviewer for mentioning the robust p-AMPK signal that we can detect under low metformin concentrations. The reason may lie in the fact that the western blotting system in our lab still uses X-ray films for detecting chemiluminescence signals, which is significantly more sensitive than all kinds of CCD cameras (GE, LICOR, etc.) we have tried with. In fact, we have discussed this with Dr. Grahame Hardie, our long-time collaborator, about this discrepancy of sensitivities before, and he agreed that the system LICOR Odyssey XF they have been using is less sensitive. That is why we are still using X-ray films, albeit costlier and more tedious.

- Previous studies demonstrated that metformin inhibits de novo fatty acid synthesis in mouse primary hepatocytes with IC50 values of 425 μ M (Biochem J. 2015 May 15; 468(1): 125–132). Thus, it remains unclear whether low dose of metformin and lysosomal-dependent AMPK activation is sufficient to alter hepatic lipid accumulation. This reviewer agreed that performing in vivo experiments will request several months and is too long. However, this can be demonstrated in primary hepatocytes from liver-specific AMPK-DKO mice. In addition, as the authors previously reported the existence of various pools of AMPK that are differentially

activated leading to the phosphorylation of different sets of targets, it would be of interest to validate in this context that lysosomal-dependent AMPK activation is sufficient to limit hepatic lipid synthesis.

We totally agree on the importance of, and are also intrigued by, the effects of low metformin on lipid metabolism. We deeply appreciate the reviewer for understanding the long duration required for completing the in vivo experiments, and the kind suggestions on using primary hepatocytes as a substitute. We have now determined the content of labelled TAG in primary hepatocytes treated with [U-¹³C]-glucose and metformin for 12 h to analyse the effects of metformin on de novo fatty acid synthesis. For this experiment, time duration was titrated, and was determined using following criteria: a) labelled TAG is detectable on the mass spectrometer; and b) labelled TAG does not reach the content of unlabelled TAG. We found that low metformin (5 μ M) only mildly suppressed de novo lipogenesis (approximately 15%), while high metformin (500 μ M) by 20%, and knockout of AMPK blocked the effects of both high and low metformin on de novo lipogenesis [shown below, also listed as Extended Data (ED) Fig. 9h, right panel], in line with previous reports showing that AMPK is required for metformin-mediated inhibition of de novo lipogenesis (PMID 29343420 and 24185692). Such an observation is also consistent with the view that the lysosomal AMPK is able to phosphorylate ACC1 and inhibit the de novo lipogenesis.

Since under the high-fat diet treatment, hepatic TAG may also be derived from free fatty acids (FAs) or TAG of the diet, we also determined the effects of metformin on TAG synthesis from FAs. We treated primary hepatocytes with [U-¹³C]-palmitate

and metformin for 12 h, and determined the content of labelled TAG. We found that neither low nor high metformin could alter the contents of labelled TAG (shown below, also listed as ED fig. 9h, left panel), indicating that metformin is not involved in regulating TAG synthesis from FAs.

We next determined the effects of metformin on β -oxidation by treating primary hepatocytes with [U- 13 C]-palmitate and metformin for 6 h, and determined the contents of labelled intermediates of the TCA cycle. We found that 5 μ M metformin mildly promoted β -oxidation (less than 10%, not significant), while 500 μ M metformin significantly promoted β -oxidation (approximately 15%) in an AMPK-dependent manner (shown below, also listed as ED fig. 9g), consistent with a previous report (PMID 29343420).

The above findings, particularly that low metformin only mildly promotes β -oxidation, also mirror our observations that a long duration of low metformin

administration time (four months) is needed to reduce the hepatic fat contents in our obese mouse model. It may also explain why many clinical studies show that metformin could not significantly alleviate fatty liver in human patients, as we discussed in our previous response. We have now added these new data/discussion in our revised manuscript (current Supplementary Note 7).

Similarly, examination of AMPK-dependency on the inhibition of mTOR signalling by in low metformin concentration could be demonstrated by using primary hepatocytes (these results will give support to the data obtained from MEF and HEK 293 cells).

We have already shown that in MEFs and HEK 293T cells the PEN2 and ATP6AP1 are required for the inhibition of mTORC1. Here, following the reviewer's suggestion, we have also determined the PEN2 and ATP6AP1 dependency on the inhibition of mTORC1 by low metformin in primary hepatocytes. We have now determined the phosphorylation of S6K in primary hepatocytes isolated from mice of PEN2-LKO, or ATP6AP1-LKO re-introduced with ATP6AP1 or its $\Delta 420-440$ mutants, in low metformin, and the results indicated that PEN2-ATP6AP1 axis is required for metformin to inhibit mTORC1, similar to those observed in MEFs and HEK 293T cells. We have now added these data to ED Fig. 9x, y, next to the MEFs and HEK293T panels (also shown below).

Please note that the dependency of AMPK on mTORC1 inhibition by metformin was reported previously (PMID 28089566), but with higher concentrations. Hence, we decided to performed again in low metformin. We found that metformin at 5 μM sufficiently inhibited mTORC1 in an AMPK-dependent manner, while 500 μM metformin bypass the requirement of AMPK for mTORC1 inhibition (shown below, also listed as ED Fig. 9z), consistent with the conclusion that the inhibition of mTORC1 by very low metformin is dependent on AMPK. Please note that in this study, the concentrations required for AMPK-dependent and independent

inhibition of mTORC1 are different from PMID 28089566 (0.5 mM and 2 mM), such a difference may lie in the experimental conditions.

Minor points

- In Figure 1l and m, it is unclear whether the middle and right panels show data from OCT1-expressing cells. It is surprising that intracellular metformin concentration is not increasing when cells are incubated with 5 μM metformin when expressing OCT1. Control cells without OCT1 expression should be added.

The intracellular metformin concentration was increased when cells were incubated with 5 μM metformin (to approximately 30 μM). However, such an intracellular metformin concentration was much lower compared with those incubated in 50 and 500 μM metformin (intracellular concentrations increased to over 300 and 2000, respectively), and was therefore too low to be distinguished on a linear, single-segment y-axis. We have now replaced the original y-axis with a two-segment y-axis to better illustrate the differences of intracellular metformin concentrations between 0 and 5 μM metformin. We apologise for such an unclarity.

- line 95: change *descrease* to *decrease*.

We thank the reviewer for pointing out the typo error. It has been corrected.

Again, we thank the reviewer for all the constructive comments and suggestions.

Responses to Reviewer #2

Point-by-point responses to the reviewer's comments are given below, with the reviewer's comments given verbatim in plain italic type, and our responses in bold:

Referee #2 (Remarks to the Author):

The authors have addressed most of my comments although the quality of the SPR data is still very low and currently unacceptable. The data is critical as it directly addresses the main argument of the study, and cannot be excluded as the other biophysical analyses with DSC and ITC are themselves unconvincing. The T200 should be more than capable of measuring the PTEN/metformin interaction with greater consistency, which raises concerns to me about experimental design.

Replicates do not overlay at all. It seems that the pink replicates were run first and regeneration of the surface was not complete therefore subsequent replicates demonstrate non-specific binding characterised by incomplete equilibrium, linear association and incomplete dissociation. Another problem could be low immobilisation levels which the authors need to state. Combined, I do not see how meaningful data can be extrapolated from these curves.

We thank the reviewer for the positive evaluations. As for the SPR assay, admittedly the data are less than perfect. However, PEN2 is a typical membrane protein that has to be used immediately after purification. Any freezing and thawing would render the protein aggregated. The PEN2 protein behaves just like that. If we have not had the habit to finish work with membrane proteins in a non-stop manner, we would have never been able to obtain the SPR data (the whole duration for purification and SPR running takes more than 24 hours.). The SPR data were collected from three independent experiments (performed on three different days), in which three batches of PEN2 (purified from HEK293T cells) conjugated separately on three BIAcore CM5 chips were used. In other words, we have to use 'repeats', instead of 'replicates', for determining the Kd of metformin to PEN2. Therefore, the absolute response units (RU) recorded on the BIAcore T200 machine vary among individual repeats, owing to the intrinsically different conjugating efficiencies of each CM5 chip/channel. Even in such an experiments in which the

RU was not overlaid, the K_d values derived from different repeats vary only slightly ($0.15 \pm 0.06 \mu\text{M}$), indicating that PEN2 indeed binds metformin at micromolar levels. For these reasons, we show the results from the three repeats in three different panels to better convey the meaning of the curves (shown below, also listed as Fig. 2f).

Since our data were obtained from different CM5 chips on different days, the regeneration of one repeat on one chip will not affect the results of other repeats on another chip. We have now mentioned this information in our manuscript.

As for the low immobilisation, or low conjugation levels of PEN2 in our SPR experiments, we in fact designed so because we are afraid that high immobilization/conjugation of PEN2 will lead to non-specific bindings. According to the instructions from the manufacturer, high conjugation levels will lead to an increase of non-specific binding, particularly for small molecules (like metformin) binding to a protein with small molecular weights (like PEN2) in which the changes of RU is inherently low. Please note that in our experiment, the maximal RU (R_{max}) is above 20, which is, 600-fold higher than the system noise of BIAcore T200 (approximately 0.03). We therefore sincerely hope that the reviewer can accept that our SPR data sufficiently support that metformin can bind PEN2.

I strongly recommend the authors include a negative control protein such as PTEN2-2A and seek expert advice on this technique.

We thank the reviewer for such a constructive suggestion. Following the instruction, we have now performed the SPR experiments using PEN2-2A as a

negative control, and the results, as shown below (also listed as Extended Data Fig. 3f), indicates that PEN2-2A failed to bind metformin.

Again, we thank the reviewer for all the constructive comments and suggestions.

Responses to Reviewer #3

Point-by-point responses to the reviewer's comments are given below, with the reviewer's comments given verbatim in plain italic type, and our responses in bold:

Referee #3 (Remarks to the Author):

The authors have adequately addressed all my comments, included additional controls and conducted experiments that have strengthened their conclusions. I have no further comments.

We thank the reviewer for the positive comments.

Responses to Reviewer #4

Point-by-point responses to the reviewer's comments are given below, with the reviewer's comments given verbatim in plain italic type, and our responses in bold:

Referee #4 (Remarks to the Author):

The authors have addressed several of the reviewer's questions and concerns more than adequately.

We thank the reviewer for the positive comments.

*I still have one key concern (as already raised in my initial comments) which is the high concentration of metformin required for lifespan extension in *C. elegans*.*

I do understand that, at least in the authors' hands, 50 mM is the lowest dose capable of extending lifespan, as now experimentally shown.

Nevertheless a previous publication (ref. 113, Fig. S7) has shown lifespan extension at doses as low as 6.25 mM, i.e. 8-fold lower than in the current manuscript.

We thank the reviewer for the chance to let us answer this question again. We realised that in ref. 113, the nematodes were cultured in liquid medium, rather than solid medium (agar plates) as we did. The absorption of metformin may therefore be different between them, and that may be the reason for the discrepancy.

*The authors now show that metformin does ***not*** increase ROS in MEFs (= low-dose), however metformin ***does*** increase ROS in *C. elegans* (= high-dose).*

*Also they show that this increase in ROS is ***required*** for extension of lifespan, as previously established for other compounds/interventions (ref. 114).*

The authors indicate that this increase in ROS would be a consequence of AMPK activation, given the known activation of OXPHOS via AMPK phosphorylation, and as also previously established in nematodes (ref. 114).

On the other hand and concerning, metformin at 8-fold lower doses than 50 mM

has been shown to induce ROS in *C. elegans* via inhibition of complex I (ref. 113). The response to reviewers' & manuscript however claims that metformin's primary site of action would be different from such complex I inhibition (as indeed and convincingly shown in the cell culture data set).

Nevertheless and particularly the high concentrations of metformin used in *C. elegans*, plus the beforementioned findings (ref. 113) re. complex I being the primary target of metformin in *C. elegans* and related extension of lifespan do now require one additional experiment:

Does metformin at 50 mM still extend *C. elegans* lifespan in the **absence** of AMPK? If so, this would be a clear indication of complex I inhibition, especially since the newly added data in Extended Data Fig. 9l indicate that such ROS is required for lifespan extension in *C. elegans*.

This can be easily and quickly tested by applying metformin at 50 mM to *aak-2* mutants, as readily obtainable from the CGC (and probably available in the authors' laboratory).

We thank the reviewer for the suggestion. We indeed have the *aak-2* nematode strain, and has been backcrossed to N2 for 6 times. We had not performed this experiment because the similar experiment has been performed by others (PMID 20090912), showing that AMPK is required for 50 mM metformin-induced lifespan extension. Following the suggestion from the reviewer, we have now repeated such an experiment. As shown below [also listed as Extended Data (ED) Fig. 9q], we found that knockout of AMPK indeed blocked the effects of metformin on lifespan extension. Our data therefore support the conclusion made before (PMID 20090912), and also support the view that the increase of ROS in *C. elegans* is attributed to the activation of AMPK.

Also and for completeness, could the authors please include HEK193s and primary hepatocytes into the newly added Extended Data Fig. 9m panel (currently MEFs only, while all other experiments were shown for all three cell types)?

We thank the reviewer for the instruction. We have now determined the ROS levels (through determining the fluorescence of CellROX dye) in HEK293T cells and primary hepatocytes. As shown below (also listed as ED Fig. 9r), low metformin could not induce ROS in HEK293T cells or primary hepatocytes, similar to those observed in MEFs.

Minor comment: Fig. 5f / right panel / Y-axis should read "plasma insulin" rather than "serum insulin".

We thank the reviewer for mentioning this. The insulin levels were indeed determined from serum, rather than plasma, as described in Methods section (line 552, page 24).

Again, we thank the reviewer for all the constructive comments and suggestions.

Reviewer Reports on the Second Revision:

Referee #1 (Remarks to the Author):

Thank you for providing comprehensive responses to the previous comments. Most of my comments have been satisfactorily addressed and this MS is substantially improved. However, after re-reading the revised manuscript, I have the following requests:

- The authors addressed the PEN2- and AMPK-dependent glucoregulatory effect of metformin. But in places I did not share the author's confidence in data interpretation (I'm sorry if I did not make my request clearer before). It appears that the data are at most to demonstrate metformin postprandial glucose-lowering effects in PEN2-IKO mice (Figure 4) and failed to show a significant effect ($p > 0.05$) in AMPK-IKO mice (Extended Figure 8). Thus, raising further questions on the role of targeted lysosome in the blood lowering action of metformin, at least in the intestine. Figure 4a, metformin seems to have a major temporal effect on glucose postprandial glucose-lowering effects, with improved recovery (after glucose excursion at 20 min) and possibly inducing a faster return to baseline. Maybe I missed something but wouldn't plasma insulin levels at 20 min be expected to go up and subsequently reducing blood glucose levels with promotion of GLP-1 secretion (circulating pool of bioactive GLP-1 generally peaks at 15 min following enteral glucose loading)? Extended Figure 8e, I'm not sure AUC is the best way to interpret the data on GLP-1-induced insulin secretion. It also remains unclear if AMPK-IKO mice display impaired GLP-1 secretion by metformin as reported in this study for PEN2-IKO mice. Although previous report from Bahne et al. (PMID 30518693) indicate that an AMPK-dependent effect on GLP-1 secretion seems to contribute to metformin's glucose-lowering effect, note that this conclusion was drawn from experiments using Compound C, a non-recommended AMPK inhibitor, keeping the mechanism unresolved. I think the contribution of GLP-1-secretion in the PEN2/AMPK-dependent glucose lowering action of metformin can be resolved by further clarification and some relatively simple confirmatory experiments (e.g., bypassing GLP-1 secretion by administration of i.p. glucose).

- The end of the discussion may confuse some readers - ref 21 is cited as evidence that metformin action in the intestine promotes post-prandial glucose uptake in an AMPK-dependent manner - yet the experimental design in that study was completely unrelated to the regulation of post-prandial glucose uptake but to the regulation of hepatic glucose production through a gut-brain-liver axis. In addition, in contrast to the statement from the authors, the metformin concentrations used in refs 21 and 26 were similar (200 mg/kg and 250 mg/kg, respectively), and differences in metformin concentration (low vs high) could not explain why the action of metformin is dependent or independent of intestinal AMPK from these 2 publications.

- The authors make efforts to explain the failure of previous reports to show AMPK

activation with low metformin doses. Differences in Western blot signal detection could be one issue. However, measure of AMPK activity also failed to demonstrate robust AMPK activation by metformin lower than 50 μ M unless a 39 hours incubation is achieved (PMID 11602624). I'm a little puzzled by this. It seems advisable to correlate the dosage of metformin within culture medium and AMPK activation at 2 hours in primary hepatocytes.

Minor points:

- line 167: title should be reworded.
- line 243: impairment in metformin glucose lowering effects were shown in PEN2-IKO and AMPK α -IKO but no data on GLP-1 and insulin secretion are provided for AMPK α -IKO mice. Furthermore, citation of ref 21 is not appropriate here.
- Extended Fig. 8e: X-axis legend is lacking in the inset graph

Referee #2 (Remarks to the Author):

The authors have sufficiently addressed my concerns regarding SPR data. The PEN2A data is a convincing negative control.

Referee #4 (Remarks to the Author):

The reviewer has no additional comments to convey.

Author Rebuttals to Second Revision:

Responses to Reviewer #1

Point-by-point responses to the reviewer's comments are given below, with the reviewer's comments given verbatim in plain italic type, and our responses in bold:

Referee #1 (Remarks to the Author):

Thank you for providing comprehensive responses to the previous comments. Most of my comments have been satisfactorily addressed and this MS is substantially improved.

We thank the reviewer for the positive evaluations.

*However, after re-reading the revised manuscript, I have the following requests:
- The authors addressed the PEN2- and AMPK-dependent glucoregulatory effect of metformin. But in places I did not share the author's confidence in data interpretation (I'm sorry if I did not make my request clearer before). It appears that the data are at most to demonstrate metformin postprandial glucose-lowering effects in PEN2-IKO mice (Figure 4) and failed to show a significant effect ($p > 0.05$) in AMPK-IKO mice (Extended Figure 8). Thus, raising further questions on the role of targeted lysosome in the blood lowering action of metformin, at least in the intestine.*

We thank the reviewer for his/her carefully reading our manuscript. We admit that the p value for the blood glucose levels compared between metformin-administered wildtype (WT) and metformin-administered AMPK-IKO mice is slightly larger than 0.05 [$p = 0.064$, Extended Data (ED) Fig. 8b]. However, please take note that p value is not a sole indicator for judging outcome. Instead, the current view on statistical significance dictates that a meaningful conclusion should not be dichotomously/categorically made solely according to the p values (e.g., PMID 30894741 and 26961635). Rather, the observed average effects can be given a priority for making conclusions. Here, we showed that metformin decreased blood glucose from 15.2 ± 3.5 (mean \pm SEM) to 11.2 ± 0.9 in WT mice, while 15.4 ± 2.1 to 14.8 ± 2.8 in AMPK-IKO mice (at 30 min after glucose gavaging). Such observed effects (or point estimates) were similar between PEN2-IKO mice and their WT littermates, with 14.5 ± 2.4 to 10.2 ± 2.7 in WT mice, and 14.6 ± 1.9 to 13.9 ± 1.9 in PEN2-IKO mice, in which the p values are below 0.05 (Fig. 4a). It is therefore reasonable to conclude that lysosomal AMPK plays a role in metformin-mediated post-prandial glucose absorption.

Figure 4a, metformin seems to have a major temporal effect on glucose postprandial glucose-lowering effects, with improved recovery (after glucose excursion at 20 min) and possibly inducing a faster return to baseline. Maybe I missed something but wouldn't plasma insulin levels at 20 min be expected to go up and subsequently reducing blood glucose levels with promotion of GLP-1 secretion (circulating pool of

bioactive GLP-1 generally peaks at 15 min following enteral glucose loading)?
Extended Figure 8e, I'm not sure AUC is the best way to interpret the data on GLP-1-induced insulin secretion.

We thank the reviewer for raising such an interesting point. It is known that GLP-1 is secreted earlier (5-10 min to peak concentration) than insulin (approx. 15 min to peak concentration) after glucose gavaging, because intestines encounter glucose earlier than the pancreatic islets, and insulin is mainly stimulated by GLP-1 (rather than the increase of blood glucose per se) under this condition (reviewed in PMID 31767182). In comparison, blood glucose only started to decrease after insulin reaching the peak concentration (lagging 15 min), which is a result of glucose uptake by peripheral tissues after insulin stimulation (e.g., PMID 7750665 and PMID 25827219). In Fig. 4a, we showed that blood glucose levels descended only after 15 min of glucose gavaging. Importantly, the prominent differences seen in glucose levels between PEN2-IKO and WT mice (as well as between AMPK-IKO and WT mice shown in ED Fig. 8b) appeared at those time points after 15 min of glucose gavaging, exactly matching the physiology of GLP-1 and insulin. We therefore have to stand with the conclusion that dynamics of insulin, GLP-1 and blood glucose fit our model/conclusion that PEN2 is required for the glucose lowering effects of metformin by promoting the secretion of GLP1/insulin.

During this round of revision, we found an error in the legend of Fig. 4b, in which the levels of GLP-1 were in fact measured at 15 min, rather than 1 h after the glucose gavaging, based on the protocols from previous reports (e.g. PMID 25827219 and PMID 28669086).

Again, we thank the reviewer for this insightful point, through which we have learned a lot.

It also remains unclear if AMPK-IKO mice display impaired GLP-1 secretion by metformin as reported in this study for PEN2-IKO mice. Although previous report from Bahne et al. (PMID 30518693) indicate that an AMPK-dependent effect on GLP-1 secretion seems to contribute to metformin's glucose-lowering effect, note that this conclusion was drawn from experiments using Compound C, a non-recommended

AMPK inhibitor, keeping the mechanism unresolved. I think the contribution of GLP-1-secretion in the PEN2/AMPK-dependent glucose lowering action of metformin can be resolved by further clarification and some relatively simple confirmatory experiments (e.g., bypassing GLP-1 secretion by administration of i.p. glucose).

We agree that i.p. glucose to bypass GLP-1 secretion is a good way to determine the roles of GLP-1 in PEN2/AMPK-dependent glucose lowering effects of metformin. We would like to point out that such experiments have in fact been performed previously (PMID 30150719 and 24185692), and the results showed that metformin could not decrease blood glucose, even in WT mice, when glucose was i.p., unless high metformin was administered (through which the hepatic glucose output will be suppressed after the inhibition of FBP1 by AMP). We have also performed such an ipGTT experiment before, and our results [shown below, in which 50 mpk metformin (orally gavaged) represented low dose as described previously (PMID 8165821), and 250 mpk high dose (PMID 30150719)] supported the conclusion above. The inability of regular doses of metformin to reduce blood levels of glucose implies that metformin may have to depend on intestinal secretion of GLP-1 to exert its glucose-lowering effects.

- The end of the discussion may confuse some readers - ref 21 is cited as evidence that metformin action in the intestine promotes post-prandial glucose uptake in an AMPK-dependent manner - yet the experimental design in that study was completely unrelated to the regulation of post-prandial glucose uptake but to the regulation of hepatic glucose production through a gut-brain-liver axis.

We thank the reviewer for pointing out our misinterpretation of the experiments performed in ref 21. Indeed, although both metformin and glucose were injected, and the blood glucose was monitored in ref 21, the euglycemic clamping was applied. Therefore, it is the resting, rather than the post-prandial glucose maintenance was measured. We have now corrected the description by deleting the “post-prandial”.

In addition, in contrast to the statement from the authors, the metformin concentrations used in refs 21 and 26 were similar (200 mg/kg and 250 mg/kg, respectively), and differences in metformin concentration (low vs high) could not explain why the action of metformin is dependent or independent of intestinal AMPK from these 2 publications.

We have in fact never said that metformin administered in ref 21 is low, because no pharmacokinetics (PK) of metformin was measured/shown in that study. We just described the conclusion of ref 21, quoting that AMPK is required for GLP-1 secretion under the combined treatment of metformin and glucose. As for the dosages of metformin administered in ref 21 and 26, we agree that the concentrations of them (200 mpk and 250 mpk) are similar. However, we would like to point out that the ways of metformin administration between these two studies are different: ref 21 applied two ways, intraduodenal infusion and bolus gastric feeding, to administer 200 mpk metformin, while ref 26 orally gavaged 250 mpk metformin. Although neither of the studies showed PK data, according to the descriptions on the method section of ref 21, metformin administered was not completely absorbed at the time of sample collection. In comparison, many other studies have shown that orally gavaged metformin was rapidly absorbed, and reached its peak serum concentrations within 30 min (exactly the time point of sample collection in ref 26). Therefore, we may deduce that the fraction of bioavailability of metformin in ref 21 may be lower than that of ref 26. Such an effect may also be seen in another data in ref 21 showing that 50 mpk metformin (intraduodenal infused) exerted similar effects to those of 200 mpk (implying that 200 mpk metformin was indeed not completely absorbed). Together, we propose that the PK values of ref 21 and ref 26 may be different. As for stating the metformin administered in ref 26 as high dose, it was based on published works. For example, it was shown that 250 mpk metformin (orally gavaged) could elevate hepatic AMP levels (PMID 30150719). Given that the PK of metformin in gut is

usually four folds higher than that in the liver (PMID 8165821), we conclude that metformin administered in ref 26 is a high dose.

- The authors make efforts to explain the failure of previous reports to show AMPK activation with low metformin doses. Differences in Western blot signal detection could be one issue. However, measure of AMPK activity also failed to demonstrate robust AMPK activation by metformin lower than 50 μ M unless a 39 hours incubation is achieved (PMID 11602624). I'm a little puzzled by this. It seems advisable to correlate the dosage of metformin within culture medium and AMPK activation at 2 hours in primary hepatocytes.

With the sensitivity issue on the robust activation of AMPK, we have numerous data from western blots throughout the entire figure set, showing that p-AMPK as well as p-ACC levels are readily detected 60 min after applying 5 μ M metformin to cells like hepatocytes, and other low concentrations of metformin were carefully titrated based on PK data with various other cells/cell lines: MEFs (ED Fig. 1o), HEK293T cells (ED Fig. 1p), OCT1-expressing MEFs (ED Fig. 1v), OCT1-expressing HEK293T (ED Fig. 1w), and liver tissues (ED Fig. 1j, see PK data in ED Fig. 1h). We just faithfully reported the data. As we explained in details in our previous response, we used traditional way of detecting the signal of luminescence by using X-ray films. In addition, there may be other reasons from all the steps of cell preparation, protein extract preparation, and western blotting protocols, which may have variations among different laboratories. Similarly, for the measurement of AMPK enzymatic activity, results obtained in different laboratories also varied a lot, depending on the experimental systems (protocols, reagents, and equipment used). For example, in PMID 21947382, it was observed that metformin as high as 100 μ M failed to activate AMPK in primary hepatocytes, the same cells used as in PMID 11602624. Variations of such thresholds, pertaining to the measurement of AMPK enzymatic activity, have also been seen in our lab: there are two machines in our lab that can be used for in vitro enzymatic assay, one is SpectraMax M5 (Molecular Devices), and the other is Lambda 365 (PerkinElmer) which is much more sensitive. Lambda 365 always gives much lower threshold of metformin concentration during titration because much smaller magnitudes of OD changes can be recorded.

In fact, for the PEN2-dependency of AMPK activation, the concentrations of metformin can be as low as 5 μ M (in hepatocytes, could be even lower) to as high as 350 μ M under which the intracellular AMP/ATP ratio was not increased to 0.12 (ED Fig. 1m, n). Therefore, for the community to repeat our work in the future under situations when their detection systems are relatively less sensitive (e.g., using CCD cameras), other concentrations (i.e., not higher than 350 μ M) of metformin can be used. However, we would like to point out again that we chose to use 5 μ M based the PK values in blood samples of patients taking clinical doses of metformin.

To further enhance the reproducibility, we have now provided additional experimental details/conditions, including those for isolating and harvesting hepatocytes and performing immunoblotting in the Methods section of our manuscript, which is, also suggested by the editor Dr. Caputa. We have also checked and have now made sure that all buffer/medium ingredients, timing/temperature of sampling, mouse strain information for each experiment are listed.

Minor points:

- line 167: title should be reworded.

It was wrongly copy-pasted during the manuscript shortening, and has now been corrected. We apologise for this mistake.

- line 243: impairment in metformin glucose lowering effects were shown in PEN2-IKO and AMPK α -IKO but no data on GLP-1 and insulin secretion are provided for AMPK α -IKO mice. Furthermore, citation of ref 21 is not appropriate here.

We thank the reviewer for his/her carefully reading of our manuscript. The description and figure citation have now been corrected, and ref 21 deleted.

- Extended Fig. 8e: X-axis legend is lacking in the inset graph

Again, we thank the reviewer for the carefully reading. The (in fact Y-axis) legends has been labelled (shown below).

Responses to Reviewer #2

Point-by-point responses to the reviewer's comments are given below, with the reviewer's comments given verbatim in plain italic type, and our responses in bold:

Referee #2 (Remarks to the Author):

The authors have sufficiently addressed my concerns regarding SPR data. The PEN2A data is a convincing negative control.

We thank the reviewer for his/her positive evaluation on our SPR data, and all the constructive comments and suggestions during the reviewing process.

Responses to Reviewer #4

Point-by-point responses to the reviewer's comments are given below, with the reviewer's comments given verbatim in plain italic type, and our responses in bold:

Referee #4 (Remarks to the Author):

The reviewer has no additional comments to convey.

We thank the reviewer for all the constructive comments and suggestions.

Reviewer Reports on the Third Revision:

Referee #1 (Remarks to the Author):

The authors have now addressed all the concerns that were mentioned in my previous comments. The manuscript has been revised according to my suggestions. I recommend this manuscript for publication.

Author Rebuttals to Third Revision:

Responses to Reviewer #1

Point-by-point responses to the reviewer's comments are given below, with the reviewer's comments given verbatim in plain italic type, and our responses in bold:

Referee #1 (Remarks to the Author):

The authors have now addressed all the concerns that were mentioned in my previous comments. The manuscript has been revised according to my suggestions. I recommend this manuscript for publication.

We thank the reviewer for the positive evaluations.

nature portfolio